# When Do Neural Networks Outperform Kernel Methods?

**Behrooz Ghorbani,**[*]  **Song Mei,**[†]  **Theodor Misiakiewicz,**[‡]  **Andrea Montanari**[*‡§]

## Abstract

For a certain scaling of the initialization of stochastic gradient descent (SGD), wide neural networks (NN) have been shown to be well approximated by reproducing kernel Hilbert space (RKHS) methods. Recent empirical work showed that, for some classification tasks, RKHS methods can replace NNs without a large loss in performance. On the other hand, two-layers NNs are known to encode richer smoothness classes than RKHS and we know of special examples for which SGD-trained NN provably outperform RKHS. This is true even in the wide network limit, for a different scaling of the initialization.

How can we reconcile the above claims? For which tasks do NNs outperform RKHS? If covariates are nearly isotropic, RKHS methods suffer from the curse of dimensionality, while NNs can overcome it by learning the best low-dimensional representation. Here we show that this curse of dimensionality becomes milder if the covariates display the same low-dimensional structure as the target function, and we precisely characterize this tradeoff. Building on these results, we present the spiked covariates model that can capture in a unified framework both behaviors observed in earlier works.

We hypothesize that such a latent low-dimensional structure is present in image classification. We numerically test this hypothesis by showing that specific perturbations of the training distribution degrade the performances of RKHS methods much more significantly than NNs.

## 1 Introduction

In supervised learning we are given data $\{(y_i, \boldsymbol{x}_i)\}_{i \leq n} \sim_{iid} \mathbb{P} \in \mathscr{P}(\mathbb{R} \times \mathbb{R}^d)$, with $\boldsymbol{x}_i \in \mathbb{R}^d$ a covariate vector and $y_i \in \mathbb{R}$ the corresponding label, and would like to learn a function $f : \mathbb{R}^d \to \mathbb{R}$ to predict future labels. In many applications, state-of-the-art systems use multi-layer neural networks (NN). The simplest such model is provided by two-layers fully-connected networks:

$$\mathcal{F}_{\mathrm{NN}}^N := \left\{ \hat{f}_{\mathrm{NN}}(\boldsymbol{x}; \boldsymbol{b}, \boldsymbol{W}) = \sum_{i=1}^N b_i \sigma(\langle \boldsymbol{w}_i, \boldsymbol{x} \rangle) : \ b_i \in \mathbb{R}, \ \boldsymbol{w}_i \in \mathbb{R}^d, \ \forall i \in [N] \right\}. \tag{1}$$

$\mathcal{F}_{\mathrm{NN}}^N$ is a non-linearly parametrized class of functions: while nonlinearity poses a challenge to theoreticians, it is often claimed to be crucial in order to learn rich representation of the data. Recent efforts to understand NN have put the spotlight on two linearizations of $\mathcal{F}_{\mathrm{NN}}^N$, the random features [27] and the neural tangent [18] classes

$$\mathcal{F}_{\mathrm{RF}}^N(\boldsymbol{W}) := \left\{ \hat{f}_{\mathrm{RF}}(\boldsymbol{x}; \boldsymbol{a}; \boldsymbol{W}) = \sum_{i=1}^N a_i \sigma(\langle \boldsymbol{w}_i, \boldsymbol{x} \rangle) : \ a_i \in \mathbb{R}, \forall i \in [N] \right\}, \tag{2}$$

---

[*]Department of Electrical Engineering, Stanford University

[†]Department of Statistics, University of California, Berkeley

[‡]Department of Statistics, Stanford University

[§]Google Research, Brain Team

$$\mathcal{F}_{\mathrm{NT}}^N(\boldsymbol{W}) := \left\{ \hat{f}_{\mathrm{NT}}(\boldsymbol{x}; \boldsymbol{S}, \boldsymbol{W}) = \sum_{i=1}^N \langle \boldsymbol{s}_i, \boldsymbol{x} \rangle \sigma'(\langle \boldsymbol{w}_i, \boldsymbol{x} \rangle) : \quad \boldsymbol{s}_i \in \mathbb{R}^d, \forall i \in [N] \right\}. \tag{3}$$

$\mathcal{F}_{\mathrm{RF}}^N(\boldsymbol{W})$ and $\mathcal{F}_{\mathrm{NT}}^N(\boldsymbol{W})$ are linear classes of functions, depending on the realization of the input-layer weights $\boldsymbol{W} = (\boldsymbol{w}_i)_{i \leq N}$ (which are chosen randomly). The relation between NN and these two linear classes is given by the first-order Taylor expansion: $\hat{f}_{\mathrm{NN}}(\boldsymbol{x}; \boldsymbol{b} + \varepsilon \boldsymbol{a}, \boldsymbol{W} + \varepsilon \boldsymbol{S}) - \hat{f}_{\mathrm{NN}}(\boldsymbol{x}; \boldsymbol{b}, \boldsymbol{W}) = \varepsilon \hat{f}_{\mathrm{RF}}(\boldsymbol{x}; \boldsymbol{a}; \boldsymbol{W}) + \varepsilon \hat{f}_{\mathrm{NT}}(\boldsymbol{x}; \boldsymbol{S}(\boldsymbol{b}); \boldsymbol{W}) + O(\varepsilon^2)$, where $\boldsymbol{S}(\boldsymbol{b}) = (b_i \boldsymbol{s}_i)_{i \leq N}$. A number of recent papers show that, if weights and SGD updates are suitably scaled, and the network is sufficiently wide ($N$ sufficiently large), then SGD converges to a function $\hat{f}_{\mathrm{NN}}$ that is approximately in $\mathcal{F}_{\mathrm{RF}}^N(\boldsymbol{W}) + \mathcal{F}_{\mathrm{NT}}^N(\boldsymbol{W})$, with $\boldsymbol{W}$ determined by the SGD initialization [18, 13, 12, 3, 34, 25]. This was termed the 'lazy regime' in [9].

Does this linear theory convincingly explain the successes of neural networks? Can the performances of NN be achieved by the simpler NT or RF models? Is there any fundamental difference between the two classes RF and NT? If the weights $(\boldsymbol{w}_i)_{i \leq N}$ are i.i.d. draws from a distribution $\nu$ on $\mathbb{R}^d$, the spaces $\mathcal{F}_{\mathrm{RF}}^N(\boldsymbol{W})$, $\mathcal{F}_{\mathrm{NT}}^N(\boldsymbol{W})$ can be thought as finite-dimensional approximations of a certain RKHS:

$$\mathcal{H}(h) := \mathrm{cl}\left( \left\{ f(\boldsymbol{x}) = \sum_{i=1}^N c_i \, h(\boldsymbol{x}, \boldsymbol{x}_i) : \quad c_i \in \mathbb{R}, \ \boldsymbol{x}_i \in \mathbb{R}^d, \ N \in \mathbb{N} \right\} \right), \tag{4}$$

where $\mathrm{cl}(\cdot)$ denotes closure. From this point of view, RF and NT differ in that they correspond to slightly different choices of the kernel: $h_{\mathrm{RF}}(\boldsymbol{x}_1, \boldsymbol{x}_2) := \int \sigma(\langle \boldsymbol{w}, \boldsymbol{x}_1 \rangle) \sigma(\langle \boldsymbol{w}, \boldsymbol{x}_2 \rangle) \nu(\mathrm{d}\boldsymbol{w})$ versus $h_{\mathrm{NT}}(\boldsymbol{x}_1, \boldsymbol{x}_2) := \langle \boldsymbol{x}_1, \boldsymbol{x}_2 \rangle \int \sigma'(\boldsymbol{w}^\mathsf{T} \boldsymbol{x}_1) \sigma'(\boldsymbol{w}^\mathsf{T} \boldsymbol{x}_2) \nu(\mathrm{d}\boldsymbol{w})$. Multi-layer fully-connected NNs in the lazy regime can be viewed as randomized approximations to RKHS as well, with some changes in the kernel $h$. This motivates analogous questions for $\mathcal{H}(h)$: can the performances of NN be achieved by RKHS methods?

Recent work addressed the separation between NN and RKHS from several points of view, without providing a unified answer. Some empirical studies on various datasets showed that networks can be replaced by suitable kernels with limited drop in performances [5, 21, 20, 24, 19, 10, 14, 29]. At least two studies reported a larger gap for convolutional networks and the corresponding kernels [4, 15]. On the other hand, theoretical analysis provided a number of separation examples, i.e. target functions $f_*$ that can be represented and possibly efficiently learnt using neural networks, but not in the corresponding RKHS [32, 6, 17, 16, 1, 2]. For instance, if the target is a single neuron $f_*(\boldsymbol{x}) = \sigma(\langle \boldsymbol{w}_*, \boldsymbol{x} \rangle)$, then training a neural network with one hidden neuron learns the target efficiently from approximately $d \log d$ samples [22], while the corresponding RKHS has test error bounded away from zero for every sample size polynomial in $d$ [32, 17]. Further even in the infinite width limit, it is known that two-layers neural networks can actually capture a richer class of functions than the associated RKHS, provided SGD training is scaled differently from the lazy regime [23, 7, 28, 30, 8].

Can we reconcile empirical and theoretical results?

## 1.1 Overview

In this paper we introduce a stylized scenario – which we will refer to as the spiked covariates model – that can explain the above seemingly divergent observations in a unified framework. The spiked covariates model is based on two building blocks: (1) Target functions depending on low-dimensional projections; (2) Approximately low-dimensional covariates.

(1) *Target functions depending on low-dimensional projections.* We investigate the hypothesis that NNs are more efficient at learning target functions that depend on low-dimensional projections of the data (the signal covariates). Formally, we consider target functions $f_* : \mathbb{R}^d \to \mathbb{R}$ of the form $f_*(\boldsymbol{x}) = \varphi(\boldsymbol{U}^\mathsf{T} \boldsymbol{x})$, where $\boldsymbol{U} \in \mathbb{R}^{d \times d_0}$ is a semi-orthogonal matrix, $d_0 \ll d$, and $\varphi : \mathbb{R}^{d_0} \to \mathbb{R}$ is a suitably smooth function. This model captures an important property of certain applications. For instance, the labels in an image classification problem do not depend equally on the whole Fourier spectrum of the image, but predominantly on the low-frequency components.

---

The code used to produce our results can be accessed at https://github.com/bGhorbani/linearized_neural_networks.

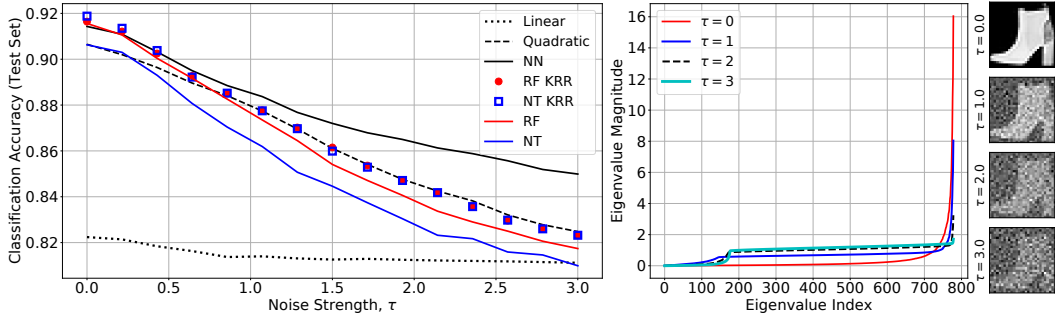

Figure 1: Test accuracy on FMNIST images perturbed by adding noise to the high-frequency Fourier components of the images (see examples on the right). Left: comparison of the accuracy of various methods as a function of the added noise. Center: eigenvalues of the empirical covariance of the images. As the noise increases, the images distribution becomes more isotropic.

As for the example of a single neuron $f_*(\boldsymbol{x}) = \sigma(\langle \boldsymbol{w}_*, \boldsymbol{x} \rangle)$, we expect RKHS to suffer from a curse of dimensionality in learning functions of low-dimensional projections. Indeed, this is well understood in low dimension or for isotropic covariates [6, 17].

(2) *Approximately low-dimensional covariates.* RKHS behave well on certain image classification tasks [4, 21, 24], and this seems to contradict the previous point. However, the example of image classification naturally brings up another important property of real data that helps to clarify this puzzle. Not only we expect the target function $f_*(\boldsymbol{x})$ to depend predominantly on the low-frequency components of image $\boldsymbol{x}$, but the image $\boldsymbol{x}$ itself to have most of its spectrum concentrated on low-frequency components (linear denoising algorithms exploit this very observation).

More specifically, we consider the case in which $\boldsymbol{x} = \boldsymbol{U}\boldsymbol{z}_1 + \boldsymbol{U}^\perp \boldsymbol{z}_2$, where $\boldsymbol{U} \in \mathbb{R}^{d \times d_0}$, $\boldsymbol{U}^\perp \in \mathbb{R}^{d \times (d-d_0)}$, and $[\boldsymbol{U}|\boldsymbol{U}^\perp] \in \mathbb{R}^{d \times d}$ is an orthogonal matrix. Moreover, we assume $\boldsymbol{z}_1 \sim \text{Unif}(\mathbb{S}^{d_0-1}(r_1\sqrt{d_0}))$, $\boldsymbol{z}_2 \sim \text{Unif}(\mathbb{S}^{d-d_0-1}(r_2\sqrt{d-d_0}))$, and $r_1^2 \geq r_2^2$. We find that, if $r_1/r_2$ (which we will denote later as the covariates signal-to-noise ratio) is sufficiently large, then the curse of dimensionality becomes milder for RKHS methods. We characterize precisely how the performance of these methods depend on the covariate signal-to-noise ratio $r_1/r_2$, the signal dimension $d_0$, and the ambient dimension $d$.

Notice that the spiked covariates model is highly stylized. For instance, while we expect real images to have a latent low-dimensional structure, this is best modeled in a nonlinear fashion (e.g. sparsity in wavelet domain [11]). Nevertheless the spiked covariates model captures the two basic mechanisms, and provides useful qualitative predictions. As an illustration, consider adding noise to the high-frequency components of images in a classification task. This will make the distribution of $\boldsymbol{x}$ more isotropic, and –according to our theory– deteriorate the performances of RKHS methods. On the other hand, NN should be less sensitive to this perturbation. (Notice that noise is added both to train and test samples.) In Figure 1 we carry out such an experiment using Fashion MNIST (FMNIST) data ($d = 784$, $n = 60000$, 10 classes). We compare two-layers NN with the RF and NT models. We choose the architectures of NN, NT, RF as to match the number of parameters: namely we used $N = 4096$ for NN and NT and $N = 321126$ for RF. We also fit the corresponding RKHS models (corresponding to $N = \infty$) using kernel ridge regression (KRR), and two simple polynomial models: $f_\ell(\boldsymbol{x}) = \sum_{k=0}^{\ell} \langle \boldsymbol{B}_k, \boldsymbol{x}^{\otimes k} \rangle$, for $\ell \in \{1, 2\}$. In the unperturbed dataset, all of these approaches have comparable accuracies (except the linear fit). As noise is added, RF, NT, and RKHS methods deteriorate rapidly. While the accuracy of NN decreases as well, it significantly outperforms other methods.

## 1.2 Notations and outline

Throughout the paper, we use bold lowercase letters $\{\boldsymbol{x}, \boldsymbol{y}, \boldsymbol{z}, \ldots\}$ to denote vectors and bold uppercase letters $\{\boldsymbol{A}, \boldsymbol{B}, \boldsymbol{C}, \ldots\}$ to denote matrices. We denote by $\mathbb{S}^{d-1}(r) = \{\boldsymbol{x} \in \mathbb{R}^d : \|\boldsymbol{x}\|_2 = r\}$ the set of $d$-dimensional vectors with radius $r$ and $\text{Unif}(\mathbb{S}^{d-1}(r))$ be the uniform probability

distribution on $\mathbb{S}^{d-1}(r)$. Further, we let $\mathsf{N}(\mu, \tau^2)$ be the Gaussian distribution with mean $\mu$ and variance $\tau^2$.

Let $O_d(\,\cdot\,)$ (respectively $o_d(\,\cdot\,)$, $\Omega_d(\,\cdot\,)$, $\omega_d(\,\cdot\,)$) denote the standard big-O (respectively little-o, big-Omega, little-omega) notation, where the subscript $d$ emphasizes the asymptotic variable. We denote by $o_{d,\mathbb{P}}(\,\cdot\,)$ the little-o in probability notation: $h_1(d) = o_{d,\mathbb{P}}(h_2(d))$, if $h_1(d)/h_2(d)$ converges to 0 in probability.

In section 2, we introduce the spiked covariates model and characterize the performance of KRR, RF, NT, and NN models. Section 3 presents numerical experiments with real and synthetic data. Section 4 discusses our results in the context of earlier work.

## 2 Rigorous results for kernel methods and NT, RF NN expansions

### 2.1 The spiked covariates model

Let $d_0 = \lfloor d^\eta \rfloor$ for some $\eta \in (0, 1)$. Let $\boldsymbol{U} \in \mathbb{R}^{d \times d_0}$ and $\boldsymbol{U}^\perp \in \mathbb{R}^{d \times (d-d_0)}$ be such that $[\boldsymbol{U}|\boldsymbol{U}^\perp]$ is an orthogonal matrix. We denote the subspace spanned by the columns of $\boldsymbol{U}$ by $\mathcal{V} \subseteq \mathbb{R}^d$ which we will refer to as the signal subspace, and the subspace spanned by the columns of $\boldsymbol{U}^\perp$ by $\mathcal{V}^\perp \subseteq \mathbb{R}^d$ which we will refer to as the noise subspace. In the case $\eta \in (0, 1)$, the signal dimension $d_0 = \dim(\mathcal{V})$ is much smaller than the ambient dimension $d$. Our model for the covariate vector $\boldsymbol{x}_i$ is

$$\boldsymbol{x}_i = \boldsymbol{U}\boldsymbol{z}_{0,i} + \boldsymbol{U}^\perp \boldsymbol{z}_{1,i}, \quad (\boldsymbol{z}_{0,i}, \boldsymbol{z}_{1,i}) \sim \mathrm{Unif}(\mathbb{S}^{d_0-1}(r\sqrt{d_0})) \otimes \mathrm{Unif}(\mathbb{S}^{d-d_0-1}(\sqrt{d-d_0})).$$

We call $\boldsymbol{z}_{0,i}$ the signal covariates, $\boldsymbol{z}_{1,i}$ the noise covariates, and $r$ the covariates signal-to-noise ratio (or covariates SNR). We will take $r > 1$, so that the variance of the signal covariates $\boldsymbol{z}_{0,i}$ is larger than that of the noise covariates $\boldsymbol{z}_{1,i}$. In high dimension, this model is –for many purposes– similar to an anisotropic Gaussian model $\boldsymbol{x}_i \sim \mathsf{N}(0, (r^2-1)\boldsymbol{U}\boldsymbol{U}^\mathsf{T} + \mathbf{I})$. As shown below, the effect of anisotropy on RKHS methods is significant only if the covariate SNR $r$ is polynomially large in $d$. We shall therefore set $r = d^{\kappa/2}$ for a constant $\kappa > 0$.

We are given i.i.d. pairs $(y_i, \boldsymbol{x}_i)_{1 \le i \le n}$, where $y_i = f_*(\boldsymbol{x}_i) + \varepsilon_i$, and $\varepsilon_i \sim \mathsf{N}(0, \tau^2)$ is independent of $\boldsymbol{x}_i$. The function $f_*$ only depends on the projection of $\boldsymbol{x}_i$ onto the signal subspace $\mathcal{V}$ (i.e. on the signal covariates $\boldsymbol{z}_{0,i}$): $f_*(\boldsymbol{x}_i) = \varphi(\boldsymbol{U}^\mathsf{T}\boldsymbol{x}_i)$, with $\varphi \in L^2(\mathbb{S}^{d_0-1}(r\sqrt{d_0}))$.

For the RF and NT models, we will assume that input layer weights to be i.i.d. $\boldsymbol{w}_i \sim \mathrm{Unif}(\mathbb{S}^{d-1}(1))$. For our purposes, this is essentially the same as $w_{ij} \sim \mathsf{N}(0, 1/d)$ independently, but slightly more convenient technically.

We will consider a more general model in Appendix C, in which the distribution of $\boldsymbol{x}_i$ takes a more general product-of-uniforms form, and we assume a general $f_* \in L^2$.

### 2.2 A sharp characterization of RKHS methods

Given $h : [-1, 1] \to \mathbb{R}$, consider the rotationally invariant kernel $K_d(\boldsymbol{x}_1, \boldsymbol{x}_2) = h(\langle \boldsymbol{x}_1, \boldsymbol{x}_2 \rangle / d)$. This class includes the kernels that are obtained by taking the wide limit of the RF and NT models (here expectation is with respect to $(G_1, G_2) \sim \mathsf{N}(0, \mathbf{I}_2)$)

$$h_{\mathrm{RF}}(t) := \mathbb{E}\{\sigma(G_1)\sigma(tG_1 + \sqrt{1-t^2}G_2)\}, \quad h_{\mathrm{NT}}(t) := t\mathbb{E}\{\sigma'(G_1)\sigma'(tG_1 + \sqrt{1-t^2}G_2)\}.$$

(These formulae correspond to $\boldsymbol{w}_i \sim \mathsf{N}(0, \mathbf{I}_d)$, but similar formulae hold for $\boldsymbol{w}_i \sim \mathrm{Unif}(\mathbb{S}^{d-1}(\sqrt{d}))$.) This correspondence holds beyond two-layers networks: under i.i.d. Gaussian initialization, the NT kernel for an arbitrary number of fully-connected layers is rotationally invariant (see the proof of Proposition 2 of [18]), and hence is covered by the present analysis.

Any RKHS method with kernel $h$ outputs a model of the form $\hat{f}(\boldsymbol{x}; \boldsymbol{a}) = \sum_{i \le n} a_i h(\langle \boldsymbol{x}, \boldsymbol{x}_i \rangle / d)$, with RKHS norm given by $\|\hat{f}(\,\cdot\,; \boldsymbol{a})\|_h^2 = \sum_{i,j \le n} h(\langle \boldsymbol{x}_i, \boldsymbol{x}_j \rangle / d) a_i a_j$. We consider kernel ridge regression (KRR) on the dataset $\{(y_i, \boldsymbol{x}_i)\}_{i \le n}$ with regularization parameter $\lambda$, namely:

$$\hat{\boldsymbol{a}}(\lambda) := \arg \min_{\boldsymbol{a} \in \mathbb{R}^N} \left\{ \sum_{i=1}^n (y_i - \hat{f}(\boldsymbol{x}_i; \boldsymbol{a}))^2 + \lambda\|\hat{f}(\,\cdot\,; \boldsymbol{a})\|_h^2 \right\} = (\boldsymbol{H} + \lambda\mathbf{I}_n)^{-1}\boldsymbol{y},$$

where $\boldsymbol{H} = (H_{ij})_{ij \in [n]}$, with $H_{ij} = h(\langle \boldsymbol{x}_i, \boldsymbol{x}_j \rangle / d)$. We denote the prediction error of KRR by

$$R_{\mathrm{KRR}}(f_*, \lambda) = \mathbb{E}_{\boldsymbol{x}} \left[ \left( f_*(\boldsymbol{x}) - \boldsymbol{y}^{\mathsf{T}} (\boldsymbol{H} + \lambda \mathbf{I}_n)^{-1} \boldsymbol{h}(\boldsymbol{x}) \right)^2 \right],$$

where $\boldsymbol{h}(\boldsymbol{x}) = (h(\langle \boldsymbol{x}, \boldsymbol{x}_1 \rangle / d), \ldots, h(\langle \boldsymbol{x}, \boldsymbol{x}_n \rangle / d))^{\mathsf{T}}$.

Recall that we assume the target function $f_*(\boldsymbol{x}_i) = \varphi(\boldsymbol{U}^{\mathsf{T}} \boldsymbol{x}_i)$. We denote $\mathsf{P}_{\leq k} : L^2 \to L^2$ to be the projection operator onto the space of degree $k$ orthogonal polynomials, and $\mathsf{P}_{>k} = \mathbf{I} - \mathsf{P}_{\leq k}$. Our next theorem shows that the impact of the low-dimensional latent structure on the generalization error of KRR is characterized by a certain *'effective dimension'*, $d_{\mathrm{eff}}$.

**Theorem 1.** *Let $h \in C^{\infty}([-1, 1])$. Let $\ell \in \mathbb{Z}_{\geq 0}$ be a fixed integer. We assume that $h^{(k)}(0) > 0$ for all $k \leq \ell$, and assume that there exists a $k > \ell$ such that $h^{(k)}(0) > 0$. (Recall that $h$ is positive semidefinite whence $h^{(k)}(0) \geq 0$ for all $k$.)*

*Define the effective dimension $d_{\mathrm{eff}} = \max\{d_0, d/r^2\} = d^{\max(1-\kappa, \eta)}$. If $\omega_d(d_{\mathrm{eff}}^{\ell} \log(d_{\mathrm{eff}})) \leq n \leq d_{\mathrm{eff}}^{\ell+1-\delta}$ for some $\delta > 0$, then for any regularization parameter $\lambda = O_d(1)$, the prediction error of KRR with kernel $h$ is*

$$\left| R_{\mathrm{KRR}}(f_*; \lambda) - \|\mathsf{P}_{>\ell} f_*\|_{L^2}^2 \right| \leq o_{d, \mathbb{P}}(1) \cdot \left( \|f_*\|_{L^2}^2 + \tau^2 \right). \tag{5}$$

Remarkably, the effective dimension $d_{\mathrm{eff}} = d^{\max(1-\kappa, \eta)}$ depends both on the signal dimension $\dim(\mathcal{V}) = d^{\eta}$ and on the covariate SNR $r = d^{\kappa/2}$. Sample size $n = d_{\mathrm{eff}}^{\ell}$ is necessary to learn a degree $\ell$ polynomial. If we fix $\eta \in (0, 1)$ and take $\kappa = 0+$, we get $d_{\mathrm{eff}} \approx d$: this corresponds to almost isotropic $\boldsymbol{x}_i$. We thus recover [17, Theorem 4]. If instead $\kappa > 1 - \eta$, then most variance of $\boldsymbol{x}_i$ falls in the signal subspace $\mathcal{V}$, and we get $d_{\mathrm{eff}} = d^{\eta} = \dim(\mathcal{V})$: the test error is effectively the same as if we had oracle knowledge of the signal subspace $\mathcal{V}$ and performed KRR on signal covariates $\boldsymbol{z}_{0,i} = \boldsymbol{U}^{\mathsf{T}} \boldsymbol{x}_i$. Theorem 1 describes the transition between these two regimes.

## 2.3 RF and NT models

How do the results of the previous section generalize to finite-width approximations of the RKHS? In particular, how do the RF and NT models behave at finite $N$? In order to simplify the picture, we focus here on the approximation error. Equivalently, we assume the sample size to be $n = \infty$ and consider the minimum population risk for $\mathsf{M} \in \{\mathrm{RF}, \mathrm{NT}\}$

$$R_{\mathsf{M}, N}(f_*; \boldsymbol{W}) := \inf_{\hat{f} \in \mathcal{F}_{\mathsf{M}}^N(\boldsymbol{W})} \mathbb{E}\left\{ \left[ f_*(\boldsymbol{x}) - \hat{f}(\boldsymbol{x}) \right]^2 \right\}. \tag{6}$$

The next two theorems characterize the asymptotics of the approximation error for RF and NT models. We give generalizations of these statements to other settings and under weaker assumptions in Appendix C.

**Theorem 2** (Approximation error for RF). *Assume $\sigma \in C^{\infty}(\mathbb{R})$, with $k$-th derivative $\sigma^{(k)}(x)^2 \leq c_{0,k} e^{c_{1,k} x^2/2}$ for some $c_{0,k} > 0$, $c_{1,k} < 1$, and all $x \in \mathbb{R}$ and all $k$. Define its $k$-th Hermite coefficient $\mu_k(\sigma) := \mathbb{E}_{G \sim \mathsf{N}(0,1)}[\sigma(G) \mathrm{He}_k(G)]$. Let $\ell \in \mathbb{Z}_{\geq 0}$ be a fixed integer, and assume $\mu_k(\sigma) \neq 0$ for all $k \leq \ell$. Define $d_{\mathrm{eff}} = d^{\max(1-\kappa, \eta)}$. If $d_{\mathrm{eff}}^{\ell+\delta} \leq N \leq d_{\mathrm{eff}}^{\ell+1-\delta}$ for some $\delta > 0$ independent of $N, d$, then*

$$\left| R_{\mathrm{RF}, N}(f_*; \boldsymbol{W}) - \|\mathsf{P}_{>\ell} f_*\|_{L^2}^2 \right| \leq o_{d, \mathbb{P}}(1) \cdot \|\mathsf{P}_{>\ell} f_*\|_{L^2} \|f_*\|_{L^2}. \tag{7}$$

**Theorem 3** (Approximation error for NT). *Assume $\sigma \in C^{\infty}(\mathbb{R})$, with $k$-th derivative $\sigma^{(k)}(x)^2 \leq c_{0,k} e^{c_{1,k} x^2/2}$, for some $c_{0,k} > 0$, $c_{1,k} < 1$, and all $x \in \mathbb{R}$ and all $k$. Let $\ell \in \mathbb{Z}_{\geq 0}$, and assume $\mu_k(\sigma) \neq 0$ for all $k \leq \ell + 1$. Further assume that, for all $L \in \mathbb{Z}_{\geq 0}$, there exist $k_1, k_2$ with $L < k_1 < k_2$, such that $\mu_{k_1}(\sigma') \neq 0$, $\mu_{k_2}(\sigma') \neq 0$, and $\mu_{k_1}(x^2 \sigma')/\mu_{k_1}(\sigma') \neq \mu_{k_2}(x^2 \sigma')/\mu_{k_2}(\sigma')$. Define $d_{\mathrm{eff}} = d^{\max(1-\kappa, \eta)}$. If $d_{\mathrm{eff}}^{\ell+\delta} \leq N \leq d_{\mathrm{eff}}^{\ell+1-\delta}$ for some $\delta > 0$ independent of $N, d$, then*

$$\left| R_{\mathrm{NT}, N}(f_*; \boldsymbol{W}) - \|\mathsf{P}_{>\ell+1} f_*\|_{L^2}^2 \right| \leq o_{d, \mathbb{P}}(1) \cdot \|\mathsf{P}_{>\ell+1} f_*\|_{L^2} \|f_*\|_{L^2}. \tag{8}$$

Here, the definitions of effective dimension $d_{\mathrm{eff}}$ is the same as in Theorem 1. While for the test error of KRR as in Theorem 1, the effective dimension controls the sample complexity $n$ in learning a degree $\ell$ polynomial, in the present case it controls the number of neurons $N$ that is necessary to

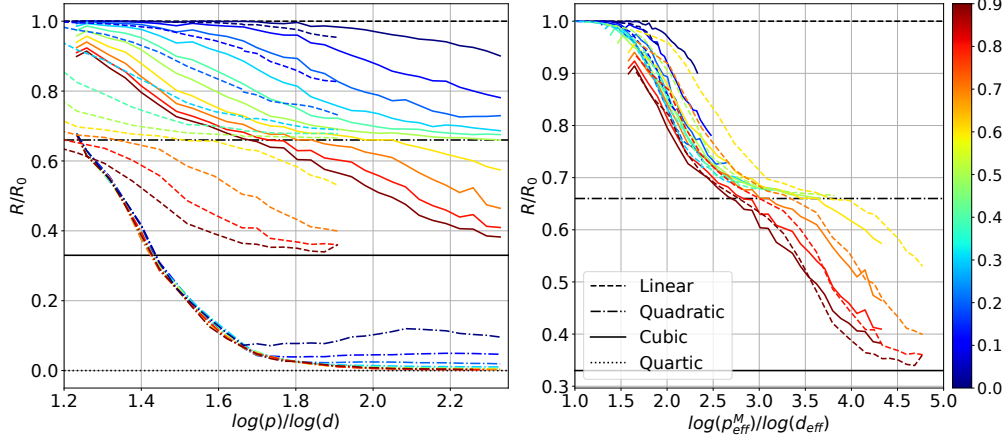

Figure 2: Finite-width two-layers NN and their linearizations RF and NT. Models are trained on $2^{20}$ training observations drawn i.i.d from the distribution of Section 2.1. Continuous lines: NT; dashed lines: RF; dot-dashed: NN. Various curves (colors) refer to values of the exponent $\kappa$ (larger $\kappa$ corresponds to stronger low-dimensional component). Right frame: curves for RF and NT as a function of the rescaled quantity $\log(p_{\text{eff}}^{\text{M}})/\log(d_{\text{eff}})$.

approximate a degree $\ell$ polynomial. In the case of RF, the latter happens as soon as $N \gg d_{\text{eff}}^{\ell}$, while for NT it happens as soon as $N \gg d_{\text{eff}}^{\ell-1}$. If we take $\eta \in (0,1)$ and $\kappa = 0+$, the above theorems, again, recover Theorem 1 and 2 of [17].

Notice that NT has higher approximation power than RF *in terms of the number of neurons*. This is expected, since NT models contain $Nd$ instead of $N$ parameters. On the other hand, NT has less power *in terms of number of parameters*: to fit a degree $\ell + 1$ polynomial, the parameter complexity for NT is $Nd = d_{\text{eff}}^{\ell}d$ while the parameter complexity for RF is $N = d_{\text{eff}}^{\ell+1} \ll d_{\text{eff}}^{\ell}d$. While the NT model has $p = Nd$ parameters, only $p_{\text{eff}}^{\text{NT}} = Nd_{\text{eff}}$ of them appear to matter. We will refer to $p_{\text{eff}}^{\text{NT}} \equiv Nd_{\text{eff}}$ as the *effective number of parameters* of NT models.

Finally, it is natural to ask what are the behaviors of RF and NT models at finite sample size. Denote by $R_{\text{M},N,n}(f_*; \boldsymbol{W})$ the corresponding test error (assuming for instance ridge regression, with the optimal regularization $\lambda$). Of course the minimum population risk provides a lower bound: $R_{\text{M},N,n}(f_*; \boldsymbol{W}) \geq R_{\text{M},N}(f_*; \boldsymbol{W})$. Moreover, we conjecture that the risk is minimized at infinite $N$, $R_{\text{M},N,n}(f_*; \boldsymbol{W}) \gtrsim R_n(f_*; h_{\text{M}})$. Altogether this implies the lower bound $R_{\text{M},N,n}(f_*; \boldsymbol{W}) \gtrsim \max(R_{\text{M},N}(f_*; \boldsymbol{W}), R_n(f_*; h_{\text{M}}))$. We also conjecture that this lower bound is tight, up to terms vanishing as $N, n, d \to \infty$.

Namely (focusing on NT models), if $Nd_{\text{eff}} \lesssim n$, and $d_{\text{eff}}^{\ell_1} \lesssim Nd_{\text{eff}} \lesssim d_{\text{eff}}^{\ell_1+1}$ then the approximation error dominates and $R_{\text{M},N,n}(f_*; \boldsymbol{W}) = \|\mathsf{P}_{>\ell_1}f_*\|_{L^2}^2 + o_{d,\mathbb{P}}(1)\|f_*\|_{L^2}^2$. If on the other hand $Nd_{\text{eff}} \gtrsim n$, and $d_{\text{eff}}^{\ell_2} \lesssim n \lesssim d_{\text{eff}}^{\ell_2+1}$ then the generalization error dominates and $R_{\text{M},N,n}(f_*; \boldsymbol{W}) = \|\mathsf{P}_{>\ell_2}f_*\|_{L^2}^2 + o_{d,\mathbb{P}}(1)\|f_*\|_{L^2}^2$.

## 2.4 Neural network models

Consider the approximation error for NNs

$$R_{\text{NN},N}(f_*) := \inf_{\hat{f} \in \mathcal{F}_{\text{NN}}^N} \mathbb{E}\left\{\left[f_*(\boldsymbol{x}) - \hat{f}(\boldsymbol{x})\right]^2\right\}. \tag{9}$$

Since $\varepsilon^{-1}[\sigma(\langle \boldsymbol{w}_i + \varepsilon\boldsymbol{a}_i, \boldsymbol{x}\rangle) - \sigma(\langle \boldsymbol{w}_i, \boldsymbol{x}\rangle)] \xrightarrow{\varepsilon \to 0} \langle \boldsymbol{a}_i, \boldsymbol{x}\rangle\sigma'(\langle \boldsymbol{w}_i, \boldsymbol{x}\rangle)$, we have $\cup_{\boldsymbol{W}}\mathcal{F}_{\text{NT}}^{N/2}(\boldsymbol{W}) \subseteq \text{cl}(\mathcal{F}_{\text{NN}}^N)$, and $R_{\text{NN},N}(f_*) \leq \inf_{\boldsymbol{W}} R_{\text{NT},N/2}(f_*, \boldsymbol{W})$. By choosing $\overline{\boldsymbol{W}} = (\bar{\boldsymbol{w}}_i)_{i \leq N}$, with $\bar{\boldsymbol{w}}_i = \boldsymbol{U}\bar{\boldsymbol{v}}_i$ (see Section 2.1 for definition of $\boldsymbol{U}$), we obtain that $\mathcal{F}_{\text{NT}}^N(\overline{\boldsymbol{W}})$ contains all functions of the form $\bar{f}(\boldsymbol{U}^\mathsf{T}\boldsymbol{x})$, where $\bar{f}$ is in the class of functions $\mathcal{F}_{\text{NT}}^N(\overline{\boldsymbol{V}})$ on $\mathbb{R}^{d_0}$. Hence if $f_*(\boldsymbol{x}) = \varphi(\boldsymbol{U}^\mathsf{T}\boldsymbol{x})$, $R_{\text{NN},N}(f_*)$ is at most the error of approximating $\varphi(\boldsymbol{z})$ on the small sphere $\boldsymbol{z} \sim \text{Unif}(\mathbb{S}^{d_0-1})$ within

the class $\mathcal{F}_{\mathrm{NT}}^N(\overline{\boldsymbol{V}})$. As a consequence, by Theorem 3, if $d_0^{\ell+\delta} \le N \le d_0^{\ell+1-\delta}$ for some $\delta > 0$, then $R_{\mathrm{NN},N}(f_*) \le R_{\mathrm{NT},N/2}(f_*, \overline{\boldsymbol{W}}) \le (1 + o_{d,\mathbb{P}}(1)) \cdot \|\mathsf{P}_{>\ell+1}f_*\|_{L^2}^2$.

**Theorem 4** (Approximation error for NN). *Assume that $\sigma \in C^\infty(\mathbb{R})$ satisfies the same assumptions as in Theorem 3. Further assume that $\sup_{x \in \mathbb{R}} |\sigma''(x)| < \infty$. If $d_0^{\ell+\delta} \le N \le d_0^{\ell+1-\delta}$ for some $\delta > 0$ independent of $N, d$, then the approximation error of NN models (3) is*

$$R_{\mathrm{NN},N}(f_*) \le (1 + o_d(1)) \cdot \|\mathsf{P}_{>\ell+1}f_*\|_{L^2}^2. \tag{10}$$

*Moreover, the quantity $R_{\mathrm{NN},N}(f_*)$ is independent of $\kappa \ge 0$.*

As a consequence of Theorem 3 and 4, there is a separation between NN and (uniformly sampled) NT models when $d_{\mathrm{eff}} \ne d_0$, i.e., $\kappa < 1 - \eta$. As $\kappa$ increases, the gap between NN and NT becomes smaller and smaller until $\kappa = 1 - \eta$.

## 3   Further numerical experiments

We carried out extensive numerical experiments on synthetic data to check our predictions for RF, NT, RKHS methods at finite sample size $n$, dimension $d$, and width $N$. We simulated two-layers fully-connected NN in the same context in order to compare their behavior to the behavior of the previous models. Finally, we carried out numerical experiments on FMNIST and CIFAR-10 data to test whether our qualitative predictions apply to image datasets. Throughout we use ReLU activations.

In Figure 2 we investigate the approximation error of RF, NT, and NN models. We generate data $(y_i, \boldsymbol{x}_i)_{i \ge 1}$ according to the model of Section 2.1, in $d = 1024$ dimensions, with a latent space dimension $d_0 = 16$, hence $\eta = 2/5$. The per-coordinate variance in the latent space is $r^2 = d^\kappa$, with $\kappa \in \{0.0, \dots, 0.9\}$. Labels are obtained by $y_i = f_*(\boldsymbol{x}_i) = \varphi(\boldsymbol{U}^\mathsf{T}\boldsymbol{x}_i)$ where $\varphi : \mathbb{R}^{d_0} \to \mathbb{R}$ is a degree-4 polynomial, without a linear component. Since we are interested in the minimum population risk, we use a large sample size $n = 2^{20}$: we expect the approximation error to dominate in this regime. (See Appendix A for further details.)

We plot the normalized risk $R_{\mathrm{RF},N}(f_*, \boldsymbol{W})/R_0$, $R_{\mathrm{NT},N}(f_*, \boldsymbol{W})/R_0$, $R_{\mathrm{NN},N}(f_*)/R_0$, $R_0 := \|f_*\|_{L^2}^2$, for various widths $N$. These are compared with the error of the best polynomial approximation of degrees $\ell = 1$ to $3$ (which correspond to $\|\mathsf{P}_{>\ell}f_*\|_{L^2}^2/\|f_*\|_{L^2}^2$). As expected, as the number of parameters increases, the approximation error of each function class decreases. NN provides much better approximations than any of the linear classes, and RF is superior to NT *given the same number of parameters*. This is captured by Theorems 2 and 3: to fit a degree $\ell + 1$ polynomial, the parameter complexity for NT is $Nd = d_{\mathrm{eff}}^\ell d$ while for RF it is $N = d_{\mathrm{eff}}^{\ell+1} \ll d_{\mathrm{eff}}^\ell d$. We denote the effective number of parameters for NT by $p_{\mathrm{eff}}^{\mathrm{NT}} = Nd_{\mathrm{eff}}$ and the effective number of parameter for RF by $p_{\mathrm{eff}}^{\mathrm{RF}} = N$. The right plot reports the same data, but we rescale the x-axis to be $\log(p_{\mathrm{eff}}^{\mathsf{M}})/\log(d_{\mathrm{eff}})$. As predicted by the asymptotic theory of Theorems 2 and 3, various curves for NT and RF tend to collapse on this scale. Finally, the approximation error of RF and NT depends strongly on $\kappa$: larger $\kappa$ leads to smaller effective dimension and hence smaller approximation error. In contrast, the error of NN, besides being smaller in absolute terms, is much less sensitive to $\kappa$.

In Fig. 3 we compare the test error of NN (with $N = 4096$) and KRR for the NT kernel (corresponding to the $N \to \infty$ limit in the lazy regime), for the same data distribution as in the previous figure. We observe that the test error of KRR is substantially larger than the one of NN, and deteriorates rapidly as $\kappa$ gets smaller (the effective dimension gets larger). In the right frame we plot the test error as a function of $\log(n)/\log(d_{\mathrm{eff}})$: we observe that the curves obtained for different $\kappa$ approximately collapse, confirming that $d_{\mathrm{eff}}$ is indeed the right dimension parameter controlling the sample complexity. Notice that also the error of NN deteriorates as $\kappa$ gets smaller, although not so rapidly: this behavior deserves further investigation. Notice also that the KRR error crosses the level of best degree-$\ell$ polynomial approximation roughly at $\log(n)/\log(d_{\mathrm{eff}}) \approx \ell$.

The basic qualitative insight of our work can be summarized as follows. Kernel methods are effective when a low-dimensional structure in the target function is aligned with a low-dimensional structure in the covariates. In image data, both the target function and the covariates are dominated by the low-frequency subspace. In Figure 1 we tested this hypothesis by removing the low-dimensional structure of the covariate vectors: we simply added noise to the high-frequency part of the image. In Figure 4 we try the opposite, by removing the component of the target function that is localized on

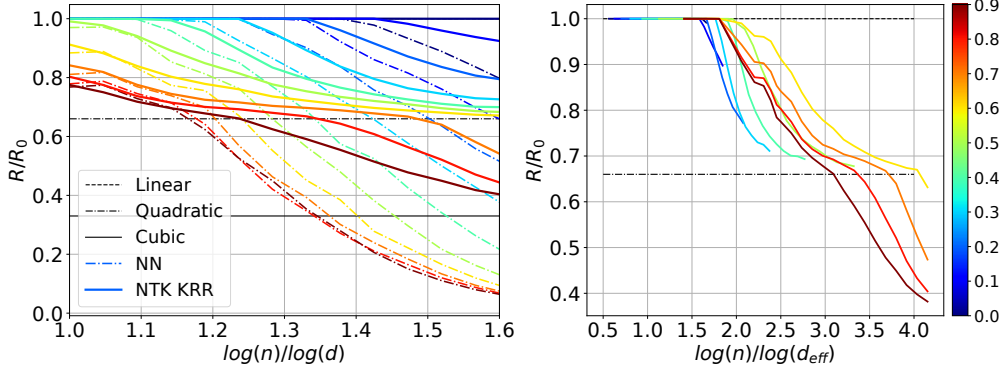

Figure 3: Left: Comparison of the test error of NN (dot-dashed) and NTK KRR (solid) on the distribution of the Section 2.1. Various curves (colors) refer to values of the exponent $\kappa$. Right: KRR test error as a function of the number of observations adjusted by the effective dimension. Horizontal lines correspond to the best polynomial approximation.

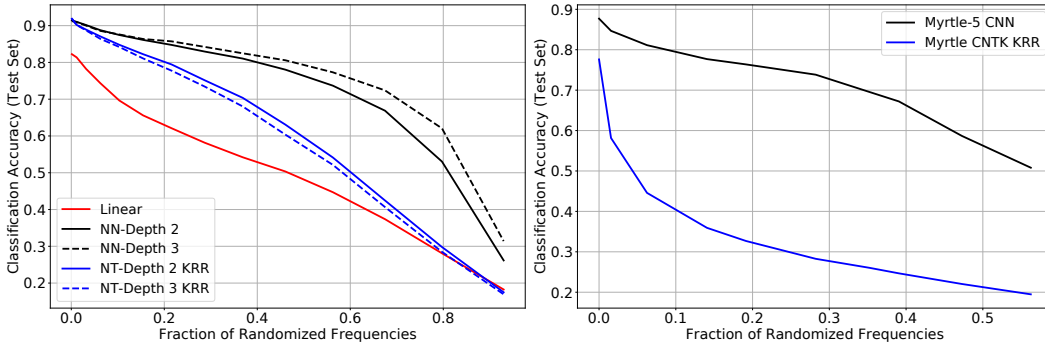

Figure 4: Compartison between multilayer NNs and the corresponding NT models under perturtbations in frequency domain. Left: Fully connected networks on FMNIST data. Right: Comparison of CNN and CNTK KRR classification accuracy on CIFAR-10. We progressively replace the lowest frequencies of each image with Gaussian noise with matching covariance structure. Right: Accuracy for FMNIST.

low-frequency modes. We decompose each images into a low-frequency and a high-frequency part. We leave the high-frequency part unchanged, and replace the low-frequency part by Gaussian noise with the first two moments matching the empirical moments of the data.

In the left frame, we consider FMNIST data and compare fully-connected NNs with 2 or 3 layers (and $N = 4096$ nodes at each hidden layer) with the corresponding NT KRR model (infinite width). In the right frame, we use CIFAR-10 data and compare a Myrtle-5 network (a lightweight convolutional architecture [26, 29]) with the corresponding NT KRR. We observe the same behavior as in Figure 1. While for the original data NT is comparable to NN, as the proportion of perturbed Fourier modes increases, the performance of NT deteriorates much more rapidly than the one of NN.

## 4  Discussion

The limitations of linear methods —such as KRR— in high dimension are well understood in the context of nonparametric function estimation. For instance, a basic result in this area establishes that estimating a Sobolev function $f_*$ in $d$ dimensions with mean square error $\varepsilon$ requires roughly $\varepsilon^{-2-d/\alpha}$ samples, with $\alpha$ the smoothness parameter [31]. This behavior is achieved by kernel smoothing and by KRR: however these methods are not expected to be adaptive when $f_*(\boldsymbol{x})$ only depends on a low-dimensional projection of $\boldsymbol{x}$, i.e. $f_*(\boldsymbol{x}) = \varphi(\boldsymbol{U}^\mathsf{T}\boldsymbol{x})$ for an unknown $\boldsymbol{U} \in \mathbb{R}^{d_0 \times d}$, $d_0 \ll d$. On the contrary, fully-trained NN can overcome this problem [6].

However, these classical statistical results have some limitations. First, they focus on the low-dimensional regime: $d$ is fixed, while the sample size $n$ diverges. This is probably unrealistic for many machine learning applications, in which $d$ is at least of the order of a few hundreds. Second, classical lower bounds are typically established for the minimax risk, and hence they do not necessarily apply to specific functions.

To bridge these gaps, we developed a sharp characterization of the test error in the high-dimensional regime in which both $d$ and $n$ diverge, while being polynomially related. This characterization holds for any target function $f_*$, and expresses the limiting test error in terms of the polynomial decomposition. We also present analogous results for finite-width RF and NT models.

Our analysis is analogous and generalizes the recent results of [17]. However, while [17] assumed the covariates $\boldsymbol{x}_i$ to be uniformly distributed over the sphere $\mathbb{S}^{d-1}(\sqrt{d})$, we introduced and analyzed a more general model in which the covariates mostly lie in the signal subspace with dimension $d_0 \ll d$, and the target function is also dependent on that subspace. In fact our results follow as special cases of a more general model discussed in Appendix C.

Depending on the relation between signal dimension $d_0$, ambient dimension $d$, and the covariate signal-to-noise ratio $r$, the model presents a continuum of different behaviors. At one extreme, the covariates are fully $d$-dimensional, and RKHS methods are highly suboptimal compared to NN. At the other, covariates are close to $d_0$-dimensional and RKHS methods are instead more competitive with NN.

Finally, the Fourier decomposition of images is a simple proxy for the decomposition of the covariate vector $\boldsymbol{x}$ into its low-dimensional dominant component (low frequency) and high-dimensional component (high frequency) [33].

## Broader Impact

This paper focuses on theoretical aspects of modern machine learning. While we expect the results of this paper to be illuminating for the theory community, we do not anticipate any direct societal impact of our work.

## Acknowledgments and Disclosure of Funding

This work was partially supported by the NSF grants CCF-1714305, IIS-1741162, DMS-1418362, DMS-1407813 and by the ONR grant N00014-18-1-2729.

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
