[Supplementary Material]

# Supplementary Material: When Do Neural Networks Outperform Kernel Methods?

**Behrooz Ghorbani,**[*] **Song Mei,**[†] **Theodor Misiakiewicz,**[‡] **Andrea Montanari**[*‡§]

## Contents

[*]Department of Electrical Engineering, Stanford University
[†]Department of Statistics, University of California, Berkeley
[‡]Department of Statistics, Stanford University
[§]Google Research, Brain Team

# A   Details of numerical experiments

## A.1   General training details

All models studied in the paper are trained with squared loss and $\ell_2$ regularization. For multi-class datasets such as FMNIST, one-hot encoded labels are used for training. All models discussed in the paper use ReLU non-linearity. Fully-connected models are initialized according to mean-field parameterization [8, 9, 7]. All neural networks are optimized with SGD with $0.9$ momentum. The learning-rate evolves according to the cosine rule

$$lr_t = lr_0 \max((1 + \cos(\frac{t\pi}{T})), \frac{1}{15}) \tag{1}$$

where $lr_0 = 10^{-3}$ and $T = 750$ is the total number of training epochs. To ensure the stability of the optimization for wide models, we use 15 linear warm-up epochs in the beginning.

When $N \gg 1$, training RF and NT with SGD is unstable (unless extremely small learning-rates are used). This makes the optimization prohibitively slow for large datasets. To avoid this issue, instead of SGD, we use conjugate gradient method (CG) for optimizing RF and NT. Since these two models are strongly convex [5], the optimizer is unique. Hence, using CG will not introduce any artifacts in the results.

In order to use CG, we first implement a function to perform Hessian-vector products in TensorFlow [1]. The function handle is then passed to scipy.sparse.cg for CG. Our Hessian-vector product code uses tensor manipulation utilities implemented by [3].

Unfortunately, scipy.sparse.cg does not support one-hot encoded labels. To avoid running CG for each class separately, when the labels are one-hot encoded, we use Adam optimizer [6] instead. When using Adam, the learning-rate still evolves as (1) with $lr_0 = 10^{-5}$. The batch-size is fixed at $10^4$ to encourage fast convergence to the minimum.

For NN, RF and NT, the training is primary done in TensorFlow (v1.12) [1]. For KRR, we generate the kernel matrix first and directly fit the model in regular python. The kernels associated with two-layer models are calculated analytically. For deeper models, the kernels are computed using neural-tangents library in JAX [2, 10].

## A.2   Synthetic data experiments

The synthetic data follows the distribution outlined in the main text. In particular,

$$\boldsymbol{x}_i = (\boldsymbol{u}_i, \boldsymbol{z}_i), \qquad y_i = \varphi(\boldsymbol{u}_i), \qquad \boldsymbol{u}_i \in \mathbb{R}^{d_0}, \boldsymbol{z}_i \in \mathbb{R}^{d-d_0}, \tag{2}$$

where $\boldsymbol{u}_i$ and $\boldsymbol{z}_i$ are drawn i.i.d from the hyper-spheres with radii $r\sqrt{d_0}$ and $\sqrt{d}$ respectively. We choose

$$r = d^{\kappa/2}, \qquad d_0 = d^\eta, \tag{3}$$

where $d$ is fixed to be 1024 and $\eta = \frac{2}{5}$. We change $\kappa$ in the interval $\{0, \ldots, 0.9\}$. For each value of $\kappa$ we generate $2^{20}$ training and $10^4$ test observations.[6]

The function $\varphi$ is the sum of three orthogonal components $\{\varphi_i\}_{i=1}^3$ with $\|\varphi_i\|_2 = 1$. To be more specific,

$$\varphi_i(\boldsymbol{x}) \propto \sum_{j=1}^{d_0-i} \alpha_j^{(i)} \prod_{k=j}^{j+i} \boldsymbol{x}_k, \qquad \alpha_j^{(i)} \overset{i.i.d}{\sim} \exp(1). \tag{4}$$

This choice of $\varphi_i$ guarantees that each $\varphi_i$ is in the span of degree $i+1$ spherical harmonics.

In the experiments presented in Figure 2, for NN and NT, the number of hidden units $N$ takes 30 geometrically spaced values in the interval $[5, 10^4]$. NN models are trained using SGD with momentum 0.9 (the learning-rate evolution is described above). We use batch-size of 512 for the warm-up epochs and batch-size of 1024 for the rest of the training. For RF, $N$ takes 24 geometrically spaced values in the interval $[100, 711680]$. The limit $N = 711680$ corresponds to the largest model size we are computationally able to train at this scale. All models are trained with $\ell_2$ regularization. The $\ell_2$ regularization grids used for these experiments are presented in Table A.1. In all our experiments, we choose the $\ell_2$ regularization parameter that yields the best test performance.[7] In total, we train approximately 10000 different models just for this subset of experiments.

In Figure 3 of the main text, we compared the generalization performance of NTK KRR with NN. We use the same training and test data as above to perform this analysis. The number of training data points, $n$, takes 24 different values ranging from 50 to $10^5$. The number of test data points is always fixed at $10^4$.

Table A.1: Hyper-parameter details for synthetic data experiments.

| Experiment | Model | $\ell_2$ Regularization grid |
|---|---|---|
| Approximation error (Fig 2) | NN | $\{10^{\alpha_i}\}_{i=1}^{20}$, $\alpha_i$ uniformly spaced in $[-8, -4]$ |
| | NT | $\{10^{\alpha_i}\}_{i=1}^{10}$, $\alpha_i$ uniformly spaced in $[-4, 2]$ |
| | RF | $\{10^{\alpha_i}\}_{i=1}^{10}$, $\alpha_i$ uniformly spaced in $[-5, 2]$ |
| Generalization error (Fig 3) | NN | $\{10^{\alpha_i}\}_{i=1}^{25}$, $\alpha_i$ uniformly spaced in $[-8, -2]$ |
| | NT KRR | $\{10^{\alpha_i}\}_{i=1}^{10}$, $\alpha_i$ uniformly spaced in $[0, 6]$ |

### A.3 High-frequency noise experiment on FMNIST

In effort to make the distribution of the covariates more isotropic, in this experiment, we add high-frequency noise to both the training and test data.

Let $\boldsymbol{x} \in \mathbb{R}^{k \times k}$ be an image. We first remove the global average of the image and then add high-frequency Gaussian noise to $\boldsymbol{x}$ in the following manner:

1. We convert $\boldsymbol{x}$ to frequency domain via Discrete Cosine Transform (DCT II-orthogonal to be precise). We denote the representation of the image in the frequency domain $\tilde{\boldsymbol{x}} \in \mathbb{R}^{k \times k}$.
2. We choose a filter $\boldsymbol{F} \in \{0, 1\}^{k \times k}$. $\boldsymbol{F}$ determines on which frequencies the noise should be added. The noise matrix $\tilde{\boldsymbol{Z}}$ is defined as $\boldsymbol{Z} \odot \boldsymbol{F}$ where $\boldsymbol{Z} \in R^{k \times k}$ has i.i.d $\mathsf{N}(0, 1)$ entries.
3. We define $\tilde{\boldsymbol{x}}_{noisy} = \tilde{\boldsymbol{x}} + \tau(\|\tilde{\boldsymbol{x}}\|/\|\tilde{\boldsymbol{Z}}\|)\tilde{\boldsymbol{Z}}$. The constant $\tau$ controls the noise magnitude.
4. We perform Inverse Discrete Cosine Transform (DCT III-orthogonal) on $\tilde{\boldsymbol{x}}_{noisy}$ to convert the image to pixel domain. We denote the noisy image in the pixel domain as $\boldsymbol{x}_{noisy}$.
5. Finally, we normalize the $\boldsymbol{x}_{noisy}$ so that it has norm $\sqrt{d}$.

In the frequency domain, a grayscale image is represented by a matrix $\tilde{\boldsymbol{x}} \in \mathbb{R}^{k \times k}$. Qualitatively speaking, elements $(\tilde{\boldsymbol{x}})_{i,j}$ with small values of $i$ and $j$ correspond to the low-frequency component of the image and elements with large indices correspond to high-frequency components. The matrix $\boldsymbol{F}$ is chosen such that no noise is added to low frequencies. Specifically, we choose

$$\boldsymbol{F}_{i,j} = \begin{cases} 1 & \text{if } (k-i)^2 + (k-j)^2 \leq (k-1)^2 \\ 0 & \text{otherwise} \end{cases} \tag{5}$$

This choice of $\boldsymbol{F}$ mirrors the average frequency domain representation of FMNIST images (see Figure A.1 for a comparison). Figure A.2 shows the eigenvalues of the empirical covariance of the dataset for various noise levels. As discussed in the main text, the distribution of the covariates becomes more isotropic as more and more high-frequency noise is added to the images.

Figure A.3 shows the normalized squared loss and the classification accuracy of the models as more and more high-frequency noise is added to the data. The normalization factor $R_0 = 0.9$ corresponds to the risk achievable by the (trivial) predictor $\left[\hat{y}_j(\boldsymbol{x})\right]_{1 \leq j \leq 10} = 0.1$.

Figure A.1: Left frame: the pictorial representation of the filter matrix $\boldsymbol{F}$ used for the FMNIST experiments. The matrix entries with value zero are represented by color blue while the entries with value one are represented by red. Coordinates on top left-hand side correspond to lower frequency components while coordinates closer to bottom right-hand side represent the high-frequency directions. Right frame: the absolute value of the frequency components of FMNIST images averaged over the training data. The projection of the dataset into the low-frequency region chosen by the filter retains over $95\%$ of the variation in the data.

Figure A.2: The eigenvalues of the empirical covariance matrix of the FMNIST training data. As the noise intensity increases, the distribution of the eigenvalues becomes more isotropic. Note that due to the conservative choice of the filter $\boldsymbol{F}$, noise is not added to all of the low-variance directions. These left-out directions corresponds to the small eigenvalues appearing in the left-hand side of the plot.

### A.3.1 Experiment hyper-parameters

For NT and NN, the number of hidden units $N = 4096$. For RF, we fix $N = 321126$. These hyper-parameter choices ensure that the models have approximately the same number of trainable

Figure A.3: The normalized test squared error (left) and the test accuracy (right) of the models trained and evaluated on FMNIST data with high-frequency noise.

Figure A.4: Left: FMNIST images with various high-frequency noise levels. Right: CIFAR-2 images with various levels of high-frequency Gaussian noise. The images are converted to grayscale to make the covariate dimension manageable.

parameters. NN is trained with SGD with $0.9$ momentum and learning-rate described by (1). The batch-size for the warm-up epochs is $500$. After the warm-up stage is over, we use batch-size of $1000$ to train the network. Since CG is not available in this setting, NT and RF are optimized using Adam for $T = 750$ epochs with batch-size of $10^4$. The $\ell_2$ regularization grids used for training these models are listed in Table A.2.

## A.4 High-frequency noise experiment on CIFAR-2

We perform a similar experiment on a subset of CIFAR-10. We choose two classes (airplane and cat) from the ten classes of CIFAR-10. This choice provides us with $10^4$ training and $2000$ test data points. Given that the number of training observations is not very large, we reduce the covariate dimension by converting the images to grayscale. This transformation reduces the covariate dimension to $d = 1024$.

Figure A.5 demonstrates the evolution of the model performances as the noise intensity increases. In the noiseless regime ($\tau = 0$), all models have comparable performances. However, as the noise level increases, the performance gap between NN and RKHS methods widens. For reference, the accuracy gap between NN and NT KRR is only $0.6\%$ at $\tau = 0$. However, at $\tau = 3$, this gap increases to $4.5\%$. The normalization factor $R_0 = 0.25$ corresponds to the risk achievable by the trivial estimator $\hat{y}(\boldsymbol{x}) = 0.5$.

Figure A.5: Normalized test squared error (left) and test classification accuracy (right) of the models on noisy CIFAR-2. As the noisy intensity increases, the performance gap between NN and RKHS methods widens. For reference, the accuracy gap between NN and NT KRR is only $0.6\%$ at $\tau = 0$. However, at $\tau = 3$, this gap increases to $4.5\%$. For finite-width models, $N$ is chosen such that the number of trainable parameters is approximately equal across the models. For NN and NT, $N = 4096$ and for RF, $N = 4.2 \times 10^6$. We use the noise filter described in (5).

### A.4.1 Experiment hyper-parameters

For NT and NN, the number of hidden units $N = 4096$. For RF, we fix $N = 4.2 \times 10^6$. These hyper-parameter choices ensure that the models have approximately the same number of trainable parameters. NN is trained with SGD with $0.9$ momentum and learning-rate described by (1). The batch-size is fixed at 250. NT is optimized via CG with 750 maximum iterations. The $\ell_2$ regularization grids used for training these models are listed in Table A.2.

Table A.2: Details of regularization parameters used for high-frequency noise experiments.

| Dataset | Model | $\ell_2$ Regularization grid |
|---|---|---|
| FMNIST | NN | $\{10^{\alpha_i}\}_{i=1}^{20}$, $\alpha_i$ uniformly spaced in $[-6, -2]$ |
| | NT | $\{10^{\alpha_i}\}_{i=1}^{20}$, $\alpha_i$ uniformly spaced in $[-5, 3]$ |
| | RF | $\{10^{\alpha_i}\}_{i=1}^{20}$, $\alpha_i$ uniformly spaced in $[-5, 3]$ |
| | NT KRR | $\{10^{\alpha_i}\}_{i=1}^{20}$, $\alpha_i$ uniformly spaced in $[-1, 5]$ |
| | RF KRR | $\{10^{\alpha_i}\}_{i=1}^{20}$, $\alpha_i$ uniformly spaced in $[-1, 5]$ |
| CIFAR-2 | NN | $\{10^{\alpha_i}\}_{i=1}^{20}$, $\alpha_i$ uniformly spaced in $[-6, -2]$ |
| | NT | $\{10^{\alpha_i}\}_{i=1}^{20}$, $\alpha_i$ uniformly spaced in $[-4, 4]$ |
| | RF | $\{10^{\alpha_i}\}_{i=1}^{40}$, $\alpha_i$ uniformly spaced in $[-2, 10]$ |
| | NT KRR | $\{10^{\alpha_i}\}_{i=1}^{20}$, $\alpha_i$ uniformly spaced in $[-2, 4]$ |
| | RF KRR | $\{10^{\alpha_i}\}_{i=1}^{20}$, $\alpha_i$ uniformly spaced in $[-2, 4]$ |

## A.5 Low-frequency noise experiments on FMNIST

To examine the ability of NN and RKHS methods in learning the information in low-variance components of the covariates, we replace the low-frequency components of the image with Gaussian noise. To be specific, we follow the following steps to generate the noisy datasets:

1. We normalize all images to have mean zero and norm $\sqrt{d}$.

2. Let $\mathcal{D}_{train}$ denote the set of training images in the DCT-frequency domain. We compute the mean $\mu$ and the covariance $\Sigma$ of the elements of $\mathcal{D}_{train}$.

3. We fix a threshold $\alpha \in \mathbb{N}$ where $1 \leq \alpha \leq k$.

4. Let $\boldsymbol{x}$ be an image in the dataset (test or train). We denote the representation of $\boldsymbol{x}$ in the frequency domain with $\tilde{\boldsymbol{x}}$. For each image, we draw a noise matrix $\boldsymbol{z} \sim \mathcal{N}(\mu, \Sigma)$. We have

$$\left[\tilde{\boldsymbol{x}}_{noisy}\right]_{i,j} = \left\{ \begin{array}{ll} (\boldsymbol{z})_{i,j} & \text{if } i,j \leq \alpha \\ \tilde{\boldsymbol{x}}_{i,j} & \text{otherwise} \end{array} \right.$$

5. We perform IDCT on $\tilde{\boldsymbol{x}}_{noisy}$ to get the noisy image $\boldsymbol{x}_{noisy}$.

The fraction of the frequencies replaced by noise is $\alpha^2/k^2$.

Figure A.6: Normalized test squared error (left) and test classification accuracy (right) of the models on FMNIST with low-frequency Gaussian noise.

### A.5.1 Experiment hyper-parameters

For neural networks trained for this experiment, we fix the number of hidden units per-layer to $N = 4096$. This corresponds to approximately $3.2 \times 10^6$ trainable parameters for two-layer networks and $2 \times 10^7$ trainable parameters for three-layer networks. Both models are trained using SGD with momentum with learning rate described by (1) (with $lr_0 = 10^{-3}$). For the warm-up epochs, we use batch-size of 500. We increase the batch-size to 1000 after the warm-up stage. The regularization grids used for training our models are presented in Table A.3.

## A.6 Low-frequency noise experiments on CIFAR-10

To test whether our insights are valid for convolutional models, we repeat the same experiment for CNNs trained on CIFAR-10. The noisy data is generated as follows:

1. Let $\mathcal{D}_{train}$ denote the set of training images in the DCT-frequency domain. Note that CIFAR-10 images have 3 channels. To convert the images to frequency domain, we apply two-dimensional Discrete Cosine Transform (DCT-II orthogonal) to each channel separately. We compute the mean $\mu$ and the covariance $\Sigma$ of the elements of $\mathcal{D}_{train}$.

2. We fix a threshold $\alpha \in \mathbb{N}$ where $1 \leq \alpha \leq 32$.

3. Let $\boldsymbol{x} \in \mathbb{R}^{32 \times 32 \times 3}$ be an image in the dataset (test or train). We denote the representation of $\boldsymbol{x}$ in the DCT-frequency domain with $\tilde{\boldsymbol{x}} \in \mathbb{R}^{32 \times 32 \times 3}$. For each image, we draw a noise

matrix $z \sim \mathcal{N}(\mu, \Sigma)$. We have

$$\left[\tilde{\boldsymbol{x}}_{noisy}\right]_{i,j,k} = \begin{cases} (\boldsymbol{z})_{i,j,k} & \text{if } i, j \leq \alpha \\ \tilde{\boldsymbol{x}}_{i,j,k} & \text{otherwise} \end{cases}$$

4. We perform IDCT on $\tilde{\boldsymbol{x}}_{noisy}$ to get the noisy image $\boldsymbol{x}_{noisy}$.

5. We normalize the noisy data to have zero per-channel mean and unit per-channel standard deviation. The normalization statistics are computed using only the training data.

We use Myrtle-5 architecture for our analysis. The Myrtle family is a collection of simple light-weight high-performance purely convolutional models. The simplicity of these models coupled with their good performance makes them a natural candidate for our analysis. Figure A.7 describes the details of this architecture. We fix the number of channels in all convolutional layer to be $N = 512$. This corresponds to approximately $7 \times 10^6$ parameters. Similar to the fully-connected networks, our convolutional models are also optimized via SGD with $0.9$ momentum (learning rate evolves as (1) with $lr_0 = 0.1$ and $T = 70$). We fix the batch-size to 128. To keep the experimental setting as simple as possible, we do not use any data augmentation for training the network.

Figure A.7: Details of Myrtle-5 architecture. The network only uses convolutions and average pooling. In particular, we do not use any batch-normalization [5] layers in this network. The figure is borrowed from [11].

Figure A.8: Performance of Myrtle-5 and KRR with convolutional neural tangent kernel (CNTK) on noisy CIFAR-10. CNTK is generated from the Myrtle-5 architecture using neural-tangents JAX library. When no noise is present in the data, the CNN achieves $87.7\%$ and the CNTK achieves $77.6\%$ classification accuracy. After randomizing only $1.5\%$ of the frequencies (corresponding to $\alpha = 4$) CNTK classification performance falls to $58.2\%$ while the CNN retains $84.7\%$ accuracy.

Table A.3: Details of regularization parameters used for low-frequency noise experiments.

| Dataset | Model | $\ell_2$ Regularization grid |
|---|---|---|
| FMNIST | NN depth 2 | $\{10^{\alpha_i}\}_{i=1}^{20}$, $\alpha_i$ uniformly spaced in $[-6, -2]$ |
| | NN depth 3 | $\{10^{\alpha_i}\}_{i=1}^{10}$, $\alpha_i$ uniformly spaced in $[-7, -5]$ |
| | NTK KRR depth 2 | $\{10^{\alpha_i}\}_{i=1}^{20}$, $\alpha_i$ uniformly spaced in $[-1, 5]$ |
| | NTK KRR depth 3 | $\{10^{\alpha_i}\}_{i=1}^{20}$, $\alpha_i$ uniformly spaced in $[-4, 3]$ |
| | Linear model | $\{10^{\alpha_i}\}_{i=1}^{30}$, $\alpha_i$ uniformly spaced in $[-1, 5]$ |
| CIFAR-10 | Myrtle-5 | $\{10^{\alpha_i}\}_{i=1}^{10}$, $\alpha_i$ uniformly spaced in $[-5, -2]$ |
| | KRR (Myrtle-5 NTK) | $\{10^{\alpha_i}\}_{i=1}^{20}$, $\alpha_i$ uniformly spaced in $[-6, 1]$ |

Figure A.9: The effect of low-frequency noise for various cut-off thresholds, $\alpha$. The left panel corresponds to the noisy FMNIST images and the right panel corresponds to CIFAR-10 images. In order to plot CIFAR-10 images, we rescale them to the interval $[0, 1]$.

## B  Technical background on function spaces on the sphere

### B.1  Functional spaces over the sphere

For $d \geq 1$, we let $\mathbb{S}^{d-1}(r) = \{\boldsymbol{x} \in \mathbb{R}^d : \|\boldsymbol{x}\|_2 = r\}$ denote the sphere with radius $r$ in $\mathbb{R}^d$. We will mostly work with the sphere of radius $\sqrt{d}$, $\mathbb{S}^{d-1}(\sqrt{d})$ and will denote by $\mu_{d-1}$ the uniform probability measure on $\mathbb{S}^{d-1}(\sqrt{d})$. All functions in the following are assumed to be elements of $L^2(\mathbb{S}^{d-1}(\sqrt{d}), \mu_{d-1})$, with scalar product and norm denoted as $\langle \cdot, \cdot \rangle_{L^2}$ and $\| \cdot \|_{L^2}$:

$$\langle f, g \rangle_{L^2} \equiv \int_{\mathbb{S}^{d-1}(\sqrt{d})} f(\boldsymbol{x}) \, g(\boldsymbol{x}) \, \mu_{d-1}(\mathrm{d}\boldsymbol{x}) \,. \tag{6}$$

For $\ell \in \mathbb{Z}_{\geq 0}$, let $\tilde{V}_{d,\ell}$ be the space of homogeneous harmonic polynomials of degree $\ell$ on $\mathbb{R}^d$ (i.e. homogeneous polynomials $q(\boldsymbol{x})$ satisfying $\Delta q(\boldsymbol{x}) = 0$), and denote by $V_{d,\ell}$ the linear space of functions obtained by restricting the polynomials in $\tilde{V}_{d,\ell}$ to $\mathbb{S}^{d-1}(\sqrt{d})$. With these definitions, we have the following orthogonal decomposition

$$L^2(\mathbb{S}^{d-1}(\sqrt{d}), \mu_{d-1}) = \bigoplus_{\ell=0}^{\infty} V_{d,\ell} \,. \tag{7}$$

The dimension of each subspace is given by

$$\dim(V_{d,\ell}) = B(d, \ell) = \frac{2\ell + d - 2}{\ell} \binom{\ell + d - 3}{\ell - 1} \,. \tag{8}$$

For each $\ell \in \mathbb{Z}_{\geq 0}$, the spherical harmonics $\{Y_{\ell,j}^{(d)}\}_{1 \leq j \in \leq B(d,\ell)}$ form an orthonormal basis of $V_{d,\ell}$:

$$\langle Y_{ki}^{(d)}, Y_{sj}^{(d)} \rangle_{L^2} = \delta_{ij}\delta_{ks}.$$

Note that our convention is different from the more standard one, that defines the spherical harmonics as functions on $\mathbb{S}^{d-1}(1)$. It is immediate to pass from one convention to the other by a simple scaling. We will drop the superscript $d$ and write $Y_{\ell,j} = Y_{\ell,j}^{(d)}$ whenever clear from the context.

We denote by $\mathsf{P}_k$ the orthogonal projections to $V_{d,k}$ in $L^2(\mathbb{S}^{d-1}(\sqrt{d}), \mu_{d-1})$. This can be written in terms of spherical harmonics as

$$\mathsf{P}_k f(\boldsymbol{x}) \equiv \sum_{l=1}^{B(d,k)} \langle f, Y_{kl} \rangle_{L^2} Y_{kl}(\boldsymbol{x}). \tag{9}$$

We also define $\mathsf{P}_{\leq \ell} \equiv \sum_{k=0}^{\ell} \mathsf{P}_k$, $\mathsf{P}_{> \ell} \equiv \mathbf{I} - \mathsf{P}_{\leq \ell} = \sum_{k=\ell+1}^{\infty} \mathsf{P}_k$, and $\mathsf{P}_{< \ell} \equiv \mathsf{P}_{\leq \ell-1}$, $\mathsf{P}_{\geq \ell} \equiv \mathsf{P}_{> \ell-1}$.

## B.2 Gegenbauer polynomials

The $\ell$-th Gegenbauer polynomial $Q_\ell^{(d)}$ is a polynomial of degree $\ell$. Consistently with our convention for spherical harmonics, we view $Q_\ell^{(d)}$ as a function $Q_\ell^{(d)} : [-d, d] \to \mathbb{R}$. The set $\{Q_\ell^{(d)}\}_{\ell \geq 0}$ forms an orthogonal basis on $L^2([-d, d], \tilde{\mu}_{d-1}^1)$, where $\tilde{\mu}_{d-1}^1$ is the distribution of $\sqrt{d}\langle \boldsymbol{x}, \boldsymbol{e}_1 \rangle$ when $\boldsymbol{x} \sim \mu_{d-1}$, satisfying the normalization condition:

$$
\begin{aligned}
\int_{-d}^{d} Q_k^{(d)}(t) \, Q_j^{(d)}(t) \, \mathrm{d}\tilde{\mu}_{d-1}^1 &= \frac{w_{d-2}}{d w_{d-1}} \int_{-d}^{d} Q_k^{(d)}(t) \, Q_j^{(d)}(t) \left(1 - \frac{t^2}{d^2}\right)^{(d-3)/2} dt \\
&= \frac{1}{B(d,k)} \, \delta_{jk} ,
\end{aligned}
\tag{10}
$$

where we denoted $w_{d-1} = \frac{2\pi^{d/2}}{\Gamma(d/2)}$ the surface area of the sphere $\mathbb{S}^{d-1}(1)$. In particular, these polynomials are normalized so that $Q_\ell^{(d)}(d) = 1$.

Gegenbauer polynomials are directly related to spherical harmonics as follows. Fix $\boldsymbol{v} \in \mathbb{S}^{d-1}(\sqrt{d})$ and consider the subspace of $V_\ell$ formed by all functions that are invariant under rotations in $\mathbb{R}^d$ that keep $\boldsymbol{v}$ unchanged. It is not hard to see that this subspace has dimension one, and coincides with the span of the function $Q_\ell^{(d)}(\langle \boldsymbol{v}, \cdot \rangle)$.

We will use the following properties of Gegenbauer polynomials

1. For $\boldsymbol{x}, \boldsymbol{y} \in \mathbb{S}^{d-1}(\sqrt{d})$

$$
\langle Q_j^{(d)}(\langle \boldsymbol{x}, \cdot \rangle), Q_k^{(d)}(\langle \boldsymbol{y}, \cdot \rangle) \rangle_{L^2} = \frac{1}{B(d,k)} \delta_{jk} Q_k^{(d)}(\langle \boldsymbol{x}, \boldsymbol{y} \rangle).
\tag{11}
$$

2. For $\boldsymbol{x}, \boldsymbol{y} \in \mathbb{S}^{d-1}(\sqrt{d})$

$$
Q_k^{(d)}(\langle \boldsymbol{x}, \boldsymbol{y} \rangle) = \frac{1}{B(d,k)} \sum_{i=1}^{B(d,k)} Y_{ki}^{(d)}(\boldsymbol{x}) Y_{ki}^{(d)}(\boldsymbol{y}).
\tag{12}
$$

3. Recurrence formula

$$
\frac{t}{d} Q_k^{(d)}(t) = \frac{k}{2k+d-2} Q_{k-1}^{(d)}(t) + \frac{k+d-2}{2k+d-2} Q_{k+1}^{(d)}(t).
\tag{13}
$$

4. Rodrigues formula

$$
Q_k^{(d)}(t) = (-1/2)^k d^k \frac{\Gamma((d-1)/2)}{\Gamma(k+(d-1)/2)} \left(1 - \frac{t^2}{d^2}\right)^{(3-d)/2} \left(\frac{\mathrm{d}}{\mathrm{d}t}\right)^k \left(1 - \frac{t^2}{d^2}\right)^{k+(d-3)/2}.
\tag{14}
$$

Note in particular that property 2 implies that –up to a constant– $Q_k^{(d)}(\langle \boldsymbol{x}, \boldsymbol{y} \rangle)$ is a representation of the projector onto the subspace of degree -$k$ spherical harmonics

$$
(\mathsf{P}_k f)(\boldsymbol{x}) = B(d,k) \int_{\mathbb{S}^{d-1}(\sqrt{d})} Q_k^{(d)}(\langle \boldsymbol{x}, \boldsymbol{y} \rangle) f(\boldsymbol{y}) \, \mu_{d-1}(\mathrm{d}\boldsymbol{y}) .
\tag{15}
$$

## B.3 Hermite polynomials

The Hermite polynomials $\{\mathrm{He}_k\}_{k \geq 0}$ form an orthogonal basis of $L^2(\mathbb{R}, \gamma)$, where $\gamma(\mathrm{d}x) = e^{-x^2/2}\mathrm{d}x/\sqrt{2\pi}$ is the standard Gaussian measure, and $\mathrm{He}_k$ has degree $k$. We will follow the classical normalization (here and below, expectation is with respect to $G \sim \mathsf{N}(0,1)$):

$$
\mathbb{E}\{\mathrm{He}_j(G) \, \mathrm{He}_k(G)\} = k! \, \delta_{jk} .
\tag{16}
$$

As a consequence, for any function $g \in L^2(\mathbb{R}, \gamma)$, we have the decomposition

$$g(x) = \sum_{k=0}^{\infty} \frac{\mu_k(g)}{k!} \, \text{He}_k(x) \,, \qquad \mu_k(g) \equiv \mathbb{E}\{g(G) \, \text{He}_k(G)\} \,. \qquad (17)$$

Notice that for functions $g$ that are $k$-weakly differentiable with $g^{(k)}$ the $k$-th weak derivative, we have

$$\mu_k(g) = \mathbb{E}_G[g^{(k)}(G)]. \qquad (18)$$

The Hermite polynomials can be obtained as high-dimensional limits of the Gegenbauer polynomials introduced in the previous section. Indeed, the Gegenbauer polynomials are constructed by Gram-Schmidt orthogonalization of the monomials $\{x^k\}_{k \geq 0}$ with respect to the measure $\tilde{\mu}_{d-1}^1$, while Hermite polynomial are obtained by Gram-Schmidt orthogonalization with respect to $\gamma$. Since $\tilde{\mu}_{d-1}^1 \Rightarrow \gamma$ (here $\Rightarrow$ denotes weak convergence), it is immediate to show that, for any fixed integer $k$,

$$\lim_{d \to \infty} \text{Coeff}\{Q_k^{(d)}(\sqrt{d}x) \, B(d,k)^{1/2}\} = \text{Coeff}\left\{ \frac{1}{(k!)^{1/2}} \, \text{He}_k(x) \right\} \,. \qquad (19)$$

Here and below, for $P$ a polynomial, $\text{Coeff}\{P(x)\}$ is the vector of the coefficients of $P$.

### B.4   Tensor product of spherical harmonics

We will consider in this paper the product space

$$\text{PS}^{\boldsymbol{d}} \equiv \prod_{q=1}^{Q} \mathbb{S}^{d_q - 1}\left(\sqrt{d_q}\right), \qquad (20)$$

and the uniform measure on $\text{PS}^{\boldsymbol{d}}$, denoted $\mu_{\boldsymbol{d}} \equiv \mu_{d_1 - 1} \otimes \ldots \otimes \mu_{d_Q - 1} = \bigotimes_{q \in [Q]} \mu_{d_q - 1}$, where we recall $\mu_{d_q - 1} \equiv \text{Unif}(\mathbb{S}^{d_q - 1}(\sqrt{d_q}))$. We consider the functional space of $L^2(\text{PS}^{\boldsymbol{d}}, \mu_{\boldsymbol{d}})$ with scalar product and norm denoted as $\langle \cdot, \cdot \rangle_{L^2}$ and $\| \cdot \|_{L^2}$:

$$\langle f, g \rangle_{L^2} \equiv \int_{\text{PS}^{\boldsymbol{d}}} f(\overline{\boldsymbol{x}}) g(\overline{\boldsymbol{x}}) \, \mu_{\boldsymbol{d}}(\text{d}\overline{\boldsymbol{x}}).$$

For $\boldsymbol{\ell} = (\ell_1, \ldots, \ell_Q) \in \mathbb{Z}_{\geq 0}^Q$, let $\tilde{V}_{\boldsymbol{\ell}}^{\boldsymbol{d}} \equiv \tilde{V}_{d_1, \ell_1} \otimes \ldots \otimes \tilde{V}_{d_Q, \ell_Q}$ be the span of tensor products of $Q$ homogeneous harmonic polynomials, respectively of degree $\ell_q$ on $\mathbb{R}^{d_q}$ in variable $\overline{\boldsymbol{x}}_q$. Denote by $V_{\boldsymbol{\ell}}^{\boldsymbol{d}}$ the linear space of functions obtained by restricting the polynomials in $\tilde{V}_{\boldsymbol{\ell}}^{\boldsymbol{d}}$ to $\text{PS}^{\boldsymbol{d}}$. With these definitions, we have the following orthogonal decomposition

$$L^2(\text{PS}^{\boldsymbol{d}}, \mu_{\boldsymbol{d}}) = \bigoplus_{\boldsymbol{\ell} \in \mathbb{Z}_{\geq 0}^Q} V_{\boldsymbol{\ell}}^{\boldsymbol{d}} \,. \qquad (21)$$

The dimension of each subspace is given by

$$B(\boldsymbol{d}, \boldsymbol{\ell}) \equiv \dim(V_{\boldsymbol{\ell}}^{\boldsymbol{d}}) = \prod_{q=1}^{Q} B(d_q, \ell_q),$$

where we recall

$$B(d, \ell) = \frac{2\ell + d - 2}{\ell} \binom{\ell + d - 3}{\ell - 1} \,.$$

We recall that for each $\ell \in \mathbb{Z}_{\geq 0}$, the spherical harmonics $\{Y_{\ell j}^{(d)}\}_{j \in [B(d, \ell)]}$ form an orthonormal basis of $V_\ell^{(d)}$ on $\mathbb{S}^{d-1}(\sqrt{d})$. Similarly, for each $\boldsymbol{\ell} \in \mathbb{Z}_{\geq 0}^Q$, the tensor product of spherical harmonics $\{Y_{\boldsymbol{\ell}, \boldsymbol{s}}^{\boldsymbol{d}}\}_{\boldsymbol{s} \in [B(\boldsymbol{d}, \boldsymbol{\ell})]}$ form an orthonormal basis of $V_{\boldsymbol{\ell}}^{\boldsymbol{d}}$, where $\boldsymbol{s} = (s_1, \ldots, s_Q) \in [B(\boldsymbol{d}, \boldsymbol{\ell})]$ signify $s_q \in [B(d_q, \ell_q)]$ for $q = 1, \ldots, Q$ and

$$Y_{\boldsymbol{\ell}, \boldsymbol{s}}^{\boldsymbol{d}} \equiv Y_{\ell_1, s_1}^{(d_1)} \otimes Y_{\ell_2, s_2}^{(d_2)} \otimes \ldots \otimes Y_{\ell_Q, s_Q}^{(d_Q)} = \bigotimes_{q=1}^{Q} Y_{\ell_q, s_q}^{(d_q)} \,.$$

We have the following orthonormalization property

$$\langle Y_{\boldsymbol{\ell},\boldsymbol{s}}^{\boldsymbol{d}}, Y_{\boldsymbol{\ell}',\boldsymbol{s}'}^{\boldsymbol{d}}\rangle_{L^2} = \prod_{q=1}^{Q} \left\langle Y_{\ell_q s_q}^{(d_q)}, Y_{\ell_q' s_q'}^{(d_q)} \right\rangle_{L^2\left(\mathbb{S}^{d_q-1}\left(\sqrt{d_q}\right)\right)} = \prod_{q=1}^{Q} \delta_{\ell_q,\ell_q'}\delta_{s_q,s_q'} = \delta_{\boldsymbol{\ell},\boldsymbol{\ell}'}\delta_{\boldsymbol{s},\boldsymbol{s}'}.$$

We denote by $\mathsf{P}_{\boldsymbol{k}}$ the orthogonal projections on $V_{\boldsymbol{k}}^{\boldsymbol{d}}$ in $L^2(\mathrm{PS}^{\boldsymbol{d}}, \mu_{\boldsymbol{d}})$. This can be written in terms of spherical harmonics as

$$\mathsf{P}_{\boldsymbol{k}}f(\overline{\boldsymbol{x}}) \equiv \sum_{\boldsymbol{s}\in[B(\boldsymbol{d},\boldsymbol{k})]} \langle f, Y_{\boldsymbol{k},\boldsymbol{s}}^{\boldsymbol{d}}\rangle_{L^2} Y_{\boldsymbol{k},\boldsymbol{s}}^{\boldsymbol{d}}(\overline{\boldsymbol{x}}). \tag{22}$$

We will denote for any $\mathcal{Q} \subset \mathbb{Z}_{\geq 0}^Q$, $\mathsf{P}_{\mathcal{Q}}$ the orthogonal projection on $\bigoplus_{\boldsymbol{k}\in\mathcal{Q}} V_{\boldsymbol{k}}^{\boldsymbol{d}}$, given by

$$\mathsf{P}_{\mathcal{Q}} = \sum_{\boldsymbol{k}\in\mathcal{Q}} \mathsf{P}_{\boldsymbol{k}}.$$

Similarly, the projection on $\mathcal{Q}^c$, the complementary of the set $\mathcal{Q}$ in $\mathbb{Z}_{\geq 0}^Q$, is given by

$$\mathsf{P}_{\mathcal{Q}^c} = \sum_{\boldsymbol{k}\notin\mathcal{Q}} \mathsf{P}_{\boldsymbol{k}}.$$

### B.5 Tensor product of Gegenbauer polynomials

We recall that $\tilde{\mu}_{d-1}^1$ denotes the distribution of $\sqrt{d}\langle \boldsymbol{x}, \boldsymbol{e}_d\rangle$ when $\boldsymbol{x} \sim \mathrm{Unif}(\mathbb{S}^{d-1}(\sqrt{d}))$. We consider similarly the projection of $\mathrm{PS}^{\boldsymbol{d}}$ on one coordinate per sphere. We define

$$\mathrm{ps}^{\boldsymbol{d}} \equiv \prod_{q=1}^{Q}[-d_q, d_q], \qquad \tilde{\mu}_{\boldsymbol{d}}^1 \equiv \tilde{\mu}_{d_1-1}^1 \otimes \ldots \otimes \tilde{\mu}_{d_Q-1}^1 = \bigotimes_{q=1}^{Q} \tilde{\mu}_{d_q-1}^1, \tag{23}$$

and consider $L^2(\mathrm{ps}^{\boldsymbol{d}}, \tilde{\mu}_{\boldsymbol{d}}^1)$.

Recall that the Gegenbauer polynomials $\{Q_k^{(d)}\}_{k\geq 0}$ form an orthogonal basis of $L^2([-d, d], \tilde{\mu}_{d-1}^1)$.

Define for each $\boldsymbol{k} \in \mathbb{Z}_{\geq 0}^Q$, the tensor product of Gegenbauer polynomials

$$Q_{\boldsymbol{k}}^{\boldsymbol{d}} \equiv Q_{k_1}^{(d_1)} \otimes \ldots \otimes Q_{k_Q}^{(d_Q)} = \bigotimes_{q=1}^{Q} Q_{k_q}^{(d_q)}. \tag{24}$$

We will use the following properties of the tensor product of Gegenbauer polynomials:

**Lemma 1** (Properties of products of Gegenbauer). *Consider the tensor product of Gegenbauer polynomials $\{Q_{\boldsymbol{k}}^{\boldsymbol{d}}\}_{\boldsymbol{k}\in\mathbb{Z}_{\geq 0}^Q}$ defined in Eq. (24). Then*

(a) *The set $\{Q_{\boldsymbol{k}}^{\boldsymbol{d}}\}_{\boldsymbol{k}\in\mathbb{Z}_{\geq 0}^Q}$ forms an orthogonal basis on $L^2(\mathrm{ps}^{\boldsymbol{d}}, \tilde{\mu}_{\boldsymbol{d}}^1)$, satisfying the normalization condition: for any $\boldsymbol{k}, \boldsymbol{k}' \in \mathbb{Z}_{\geq 0}^Q$,*

$$\left\langle Q_{\boldsymbol{k}}^{\boldsymbol{d}}, Q_{\boldsymbol{k}'}^{\boldsymbol{d}} \right\rangle_{L^2(\mathrm{ps}^{\boldsymbol{d}})} = \frac{1}{B(\boldsymbol{d}, \boldsymbol{k})} \delta_{\boldsymbol{k},\boldsymbol{k}'} \tag{25}$$

(b) *For $\overline{\boldsymbol{x}} = (\overline{\boldsymbol{x}}^{(1)}, \ldots, \overline{\boldsymbol{x}}^{(Q)})$ and $\overline{\boldsymbol{y}} = (\overline{\boldsymbol{y}}^{(1)}, \ldots, \overline{\boldsymbol{y}}^{(Q)}) \in \mathrm{PS}^{\boldsymbol{d}}$, and $\boldsymbol{k}, \boldsymbol{k}' \in \mathbb{Z}_{\geq 0}^Q$,*

$$\left\langle Q_{\boldsymbol{k}}^{\boldsymbol{d}}\left(\{\langle\overline{\boldsymbol{x}}^{(q)}, \cdot\rangle\}_{q\in[Q]}\right), Q_{\boldsymbol{k}'}^{\boldsymbol{d}}\left(\{\langle\overline{\boldsymbol{y}}^{(q)}, \cdot\rangle\}_{q\in[Q]}\right) \right\rangle_{L^2(\mathrm{PS}^{\boldsymbol{d}})}$$
$$= \frac{1}{B(\boldsymbol{d}, \boldsymbol{k})} \delta_{\boldsymbol{k},\boldsymbol{k}'} Q_{\boldsymbol{k}}^{\boldsymbol{d}}\left(\{\langle\overline{\boldsymbol{x}}^{(q)}, \overline{\boldsymbol{y}}^{(q)}\rangle\}_{q\in[Q]}\right). \tag{26}$$

(c) *For $\overline{\boldsymbol{x}} = (\overline{\boldsymbol{x}}^{(1)}, \ldots, \overline{\boldsymbol{x}}^{(Q)})$ and $\overline{\boldsymbol{y}} = (\overline{\boldsymbol{y}}^{(1)}, \ldots, \overline{\boldsymbol{y}}^{(Q)}) \in \mathrm{PS}^{\boldsymbol{d}}$, and $\boldsymbol{k} \in \mathbb{Z}_{\geq 0}^Q$,*

$$Q_{\boldsymbol{k}}^{\boldsymbol{d}}\left(\{\langle\overline{\boldsymbol{x}}^{(q)}, \overline{\boldsymbol{y}}^{(q)}\rangle\}_{q\in[Q]}\right) = \frac{1}{B(\boldsymbol{d}, \boldsymbol{k})} \sum_{\boldsymbol{s}\in B(\boldsymbol{d},\boldsymbol{k})} Y_{\boldsymbol{k},\boldsymbol{s}}^{\boldsymbol{d}}(\overline{\boldsymbol{x}}) Y_{\boldsymbol{k},\boldsymbol{s}}^{\boldsymbol{d}}(\overline{\boldsymbol{y}}). \tag{27}$$

Notice that Lemma 1.(c) implies that $Q_{\boldsymbol{k}}^{\boldsymbol{d}}$ is (up to a constant) a representation of the projector onto the subspace $V_{\boldsymbol{k}}^{\boldsymbol{d}}$

$$[\mathsf{P}_{\boldsymbol{k}}f](\overline{\boldsymbol{x}}) = B(\boldsymbol{d},\boldsymbol{k}) \int_{\mathrm{PS}^{\boldsymbol{d}}} Q_{\boldsymbol{k}}^{\boldsymbol{d}}\left(\{\langle \overline{\boldsymbol{x}}^{(q)}, \overline{\boldsymbol{y}}^{(q)}\rangle\}_{q\in[Q]}\right) f(\overline{\boldsymbol{y}})\mu_{\boldsymbol{d}}(\mathrm{d}\overline{\boldsymbol{y}}).$$

*Proof of Lemma 1.* Part $(a)$ comes from the normalization property (10) of Gegenbauer polynomials,

$$\left\langle Q_{\boldsymbol{k}}^{\boldsymbol{d}}, Q_{\boldsymbol{k}'}^{\boldsymbol{d}} \right\rangle_{L^2(\mathrm{ps}^{\boldsymbol{d}})} = \left\langle Q_{\boldsymbol{k}}^{\boldsymbol{d}}\left(\{\sqrt{d_q}\langle \boldsymbol{e}_q, \cdot\rangle\}_{q\in[Q]}\right), Q_{\boldsymbol{k}'}^{\boldsymbol{d}}\left(\{\sqrt{d_q}\langle \boldsymbol{e}_q, \cdot\rangle\}_{q\in[Q]}\right) \right\rangle_{L^2(\mathrm{PS}^{\boldsymbol{d}})}$$

$$= \prod_{q=1}^{Q} \left\langle Q_{k_q}^{(d_q)}\left(\sqrt{d_q}\langle \boldsymbol{e}_q, \cdot\rangle\right), Q_{k_q'}^{(d_q)}\left(\sqrt{d_q}\langle \boldsymbol{e}_q, \cdot\rangle\right) \right\rangle_{L^2(\mathbb{S}^{d_q-1}(\sqrt{d_q}))}$$

$$= \prod_{q=1}^{Q} \frac{1}{B(d_q, k_q)}\delta_{k_q, k_q'}$$

$$= \frac{1}{B(\boldsymbol{d},\boldsymbol{k})}\delta_{\boldsymbol{k},\boldsymbol{k}'},$$

where the $\{\boldsymbol{e}_q\}_{q\in[Q]}$ are unit vectors in $\mathbb{R}^{d_q}$ respectively.

Part $(b)$ comes from Eq. (11),

$$\left\langle Q_{\boldsymbol{k}}^{\boldsymbol{d}}\left(\{\langle \overline{\boldsymbol{x}}^{(q)}, \cdot\rangle\}_{q\in[Q]}\right), Q_{\boldsymbol{k}'}^{\boldsymbol{d}}\left(\{\langle \overline{\boldsymbol{y}}^{(q)}, \cdot\rangle\}_{q\in[Q]}\right) \right\rangle_{L^2(\mathrm{PS}^{\boldsymbol{d}})}$$

$$= \prod_{q=1}^{Q} \left\langle Q_{k_q}^{(d_q)}\left(\langle \overline{\boldsymbol{x}}^{(q)}, \cdot\rangle\right), Q_{k_q'}^{(d_q)}\left(\langle \overline{\boldsymbol{y}}^{(q)}, \cdot\rangle\right) \right\rangle_{L^2(\mathbb{S}^{d_q-1}(\sqrt{d_q}))}$$

$$= \prod_{q=1}^{Q} \frac{1}{B(d_q, k_q)}\delta_{k_q, k_q'} Q_{k_q}^{(d_q)}\left(\langle \overline{\boldsymbol{x}}^{(q)}, \overline{\boldsymbol{y}}^{(q)}\rangle\right)$$

$$= \frac{1}{B(\boldsymbol{d},\boldsymbol{k})}\delta_{\boldsymbol{k},\boldsymbol{k}'} Q_{\boldsymbol{k}}^{\boldsymbol{d}}\left(\{\langle \overline{\boldsymbol{x}}^{(q)}, \overline{\boldsymbol{y}}^{(q)}\rangle\}_{q\in[Q]}\right),$$

while part $(c)$ is a direct consequence of Eq. (12).

$\square$

## B.6 Notations

Throughout the proofs, $O_d(\,\cdot\,)$ (resp. $o_d(\,\cdot\,)$) denotes the standard big-O (resp. little-o) notation, where the subscript $d$ emphasizes the asymptotic variable. We denote $O_{d,\mathbb{P}}(\,\cdot\,)$ (resp. $o_{d,\mathbb{P}}(\,\cdot\,)$) the big-O (resp. little-o) in probability notation: $h_1(d) = O_{d,\mathbb{P}}(h_2(d))$ if for any $\varepsilon > 0$, there exists $C_\varepsilon > 0$ and $d_\varepsilon \in \mathbb{Z}_{>0}$, such that

$$\mathbb{P}(|h_1(d)/h_2(d)| > C_\varepsilon) \le \varepsilon, \qquad \forall d \ge d_\varepsilon,$$

and respectively: $h_1(d) = o_{d,\mathbb{P}}(h_2(d))$, if $h_1(d)/h_2(d)$ converges to 0 in probability.

We will occasionally hide logarithmic factors using the $\tilde{O}_d(\,\cdot\,)$ notation (resp. $\tilde{o}_d(\,\cdot\,)$): $h_1(d) = \tilde{O}_d(h_2(d))$ if there exists a constant $C$ such that $h_1(d) \le C(\log d)^C h_2(d)$. Similarly, we will denote $\tilde{O}_{d,\mathbb{P}}(\,\cdot\,)$ (resp. $\tilde{o}_{d,\mathbb{P}}(\,\cdot\,)$) when considering the big-O in probability notation up to a logarithmic factor.

Furthermore, $f = \omega_d(g)$ will denote $f(d)/g(d) \to \infty$.

## C General framework and main theorems

In this section, define a more general model than the model considered in the main text. In the general model, we will assume the covariate vectors will follow a product of uniform distributions on the sphere, and assume a target function in $L^2$ space. We establish more general versions of Theorems 1, 2, 3 on the two-spheres cases in the main text as Theorems 1, 2, 3. We will prove Theorems 1, 2, 3 in the following sections. At the end of this section, we will show that Theorems 1, 2, 3 will imply Theorems 1, 2, 3 in the main text.

## C.1 Setup on the product of spheres

Assume that the data $\boldsymbol{x}$ lies on the product of $Q$ spheres,

$$\boldsymbol{x} = \left(\boldsymbol{x}^{(1)}, \ldots, \boldsymbol{x}^{(Q)}\right) \in \prod_{q \in [Q]} \mathbb{S}^{d_q - 1}(r_q),$$

where $d_q = d^{\eta_q}$ and $r_q = d^{(\eta_q + \kappa_q)/2}$. Let $\boldsymbol{d} = (d_1, \ldots, d_q) = (d^{\eta_1}, \ldots, d^{\eta_q})$ and $\boldsymbol{\kappa} = (\kappa_1, \ldots, \kappa_Q)$, where $\eta_q > 0$ and $\kappa_q \geq 0$ for $q = 1, \ldots, Q$. We will denote this space

$$\mathrm{PS}_{\boldsymbol{\kappa}}^{\boldsymbol{d}} = \prod_{q \in [Q]} \mathbb{S}^{d_q - 1}(r_q). \tag{28}$$

Furthermore, assume that the data is generated following the uniform distribution on $\mathrm{PS}_{\boldsymbol{\kappa}}^{\boldsymbol{d}}$, i.e.

$$\boldsymbol{x} \overset{i.i.d.}{\sim} \mathrm{Unif}(\mathrm{PS}_{\boldsymbol{\kappa}}^{\boldsymbol{d}}) = \bigotimes_{q \in [Q]} \mathrm{Unif}\left(\mathbb{S}^{d_q - 1}(r_q)\right) \equiv \mu_{\boldsymbol{d}}^{\boldsymbol{\kappa}}. \tag{29}$$

We have $\boldsymbol{x} \in \mathbb{R}^D$ and $\|\boldsymbol{x}\|_2 = R$ where $D = d^{\eta_1} + \ldots + d^{\eta_Q}$ and $R = (d^{\eta_1 + \kappa_1} + \ldots + d^{\eta_Q + \kappa_Q})^{1/2}$.

We will make the following assumption that will simplify the proofs. Denote

$$\xi \equiv \max_{q \in [Q]} \{\eta_q + \kappa_q\}, \tag{30}$$

then $\xi$ is attained on only one of the sphere, whose coordinate will be denoted $q_\xi$, i.e. $\xi = \eta_{q_\xi} + \kappa_{q_\xi}$ and $\eta_q + \kappa_q < \xi$ for $q \neq q_\xi$.

Let $\sigma : \mathbb{R} \to \mathbb{R}$ be an activation function and $(\boldsymbol{w}_i)_{i \in [N]} \sim_{iid} \mathrm{Unif}(\mathbb{S}^{D-1})$ the weights. We introduce the random feature function class

$$\mathcal{F}_{\mathrm{RF}}(\boldsymbol{W}) = \left\{ \hat{f}_{\mathrm{RF}}(\boldsymbol{x}; \boldsymbol{a}) = \sum_{i=1}^{N} a_i \sigma(\langle \boldsymbol{w}_i, \boldsymbol{x} \rangle \sqrt{D}/R) : \ a_i \in \mathbb{R}, \forall i \in [N] \right\},$$

and the neural tangent function class

$$\mathcal{F}_{\mathrm{NT}}(\boldsymbol{W}) = \left\{ \hat{f}_{\mathrm{RF}}(\boldsymbol{x}; \boldsymbol{a}) = \sum_{i=1}^{N} \langle \boldsymbol{a}_i, \boldsymbol{x} \rangle \sigma'(\langle \boldsymbol{w}_i, \boldsymbol{x} \rangle \sqrt{D}/R) : \ \boldsymbol{a}_i \in \mathbb{R}^D, \forall i \in [N] \right\}.$$

We will denote $\boldsymbol{\theta}_i = \sqrt{D} \boldsymbol{w}_i$. Notice that the normalization in the definition of the function class insures that the scalar product $\langle \boldsymbol{x}, \boldsymbol{\theta}_i \rangle / R$ is of order 1. This corresponds to normalizing the data.

We consider the approximation of $f$ by functions in function classes $\mathcal{F}_{\mathrm{RF}}(\boldsymbol{\Theta})$ and $\mathcal{F}_{\mathrm{NT}}(\boldsymbol{\Theta})$.

## C.2 Reparametrization

Recall $(\boldsymbol{\theta}_i)_{i \in [N]} \sim \mathrm{Unif}(\mathbb{S}^{D-1}(\sqrt{D}))$ independently. We decompose $\boldsymbol{\theta}_i = (\boldsymbol{\theta}_i^{(1)}, \ldots, \boldsymbol{\theta}_i^{(Q)})$ into $Q$ sections corresponding to the $d_q$ coordinates associated to the $q$-th sphere. Let us consider the following reparametrization of $(\boldsymbol{\theta}_i)_{i \in [N]} \overset{i.i.d.}{\sim} \mathrm{Unif}(\mathbb{S}^{D-1}(\sqrt{D}))$:

$$\left(\overline{\boldsymbol{\theta}}_i^{(1)}, \ldots, \overline{\boldsymbol{\theta}}_i^{(Q)}, \tau_i^{(1)}, \ldots, \tau_i^{(Q)}\right),$$

where

$$\overline{\boldsymbol{\theta}}_i^{(q)} \equiv \sqrt{d_q} \boldsymbol{\theta}_i^{(q)} / \|\boldsymbol{\theta}_i^{(q)}\|_2, \qquad \tau_i^{(q)} \equiv \|\boldsymbol{\theta}_i^{(q)}\|_2 / \sqrt{d_q}, \qquad \text{for } q = 1, \ldots, Q.$$

Hence

$$\boldsymbol{\theta}_i = \left(\tau_i^{(1)} \cdot \overline{\boldsymbol{\theta}}_i^{(1)}, \ldots, \tau_i^{(Q)} \cdot \overline{\boldsymbol{\theta}}_i^{(Q)}\right).$$

It is easy to check that the variables $(\overline{\boldsymbol{\theta}}^{(1)}, \ldots, \overline{\boldsymbol{\theta}}^{(Q)})$ are independent and independent of $(\tau_i^{(1)}, \ldots, \tau_i^{(Q)})$, and verify

$$\overline{\boldsymbol{\theta}}_i^{(q)} \sim \mathrm{Unif}(\mathbb{S}^{d_q - 1}(\sqrt{d_q})), \qquad \tau_i^{(q)} \sim d_q^{-1/2} \sqrt{\mathrm{Beta}\left(\frac{d_q}{2}, \frac{D - d_q}{2}\right)}, \qquad \text{for } q = 1, \ldots, Q.$$

We will denote $\overline{\boldsymbol{\theta}}_i \equiv (\overline{\boldsymbol{\theta}}_i^{(1)}, \ldots, \overline{\boldsymbol{\theta}}_i^{(Q)})$ and $\boldsymbol{\tau}_i \equiv (\tau_i^{(1)}, \ldots, \tau_i^{(Q)})$. With these notations, we have

$$\overline{\boldsymbol{\theta}}_i \in \prod_{q \in [Q]} \mathbb{S}^{d_q - 1}(\sqrt{d_q}) \equiv \mathrm{PS}^{\boldsymbol{d}},$$

where $\mathrm{PS}^{\boldsymbol{d}}$ is the 'normalized space of product of spheres', and

$$\left(\overline{\boldsymbol{\theta}}_i\right)_{i \in [N]} \overset{i.i.d.}{\sim} \bigotimes_{q \in [Q]} \mathrm{Unif}(\mathbb{S}^{d_q - 1}(\sqrt{d_q})) \equiv \mu_{\boldsymbol{d}}.$$

Similarly, we will denote the rescaled data $\overline{\boldsymbol{x}} \in \mathrm{PS}^{\boldsymbol{d}}$,

$$\overline{\boldsymbol{x}} = \left(\overline{\boldsymbol{x}}^{(1)}, \ldots, \overline{\boldsymbol{x}}^{(Q)}\right) \sim \bigotimes_{q \in [Q]} \mathrm{Unif}(\mathbb{S}^{d_q - 1}(\sqrt{d_q})),$$

obtained by taking $\overline{\boldsymbol{x}}^{(q)} = \sqrt{d_q} \boldsymbol{x}^{(q)} / r_q = d^{-\kappa_q / 2} \boldsymbol{x}^{(q)}$ for each $q \in [Q]$.

The proof will proceed as follows: first, noticing that $\tau^{(q)}$ concentrates around 1 for every $q = 1, \ldots, Q$, we will restrict ourselves without loss of generality to the following high probability event

$$\mathcal{P}_{d,N,\varepsilon} \equiv \left\{ \boldsymbol{\Theta} \,\middle|\, \tau_i^{(q)} \in [1 - \varepsilon, 1 + \varepsilon], \forall i \in [N], \forall q \in [Q] \right\} \subset \mathbb{S}^{D-1}(\sqrt{D})^N,$$

where $\varepsilon > 0$ will be chosen sufficiently small. Then, we rewrite the activation function

$$\sigma(\langle \cdot, \cdot \rangle / R) : \mathbb{S}^{D-1}(\sqrt{D}) \times \mathrm{PS}^{\boldsymbol{d}}_{\boldsymbol{\kappa}} \to \mathbb{R},$$

as a function, for a random $\boldsymbol{\tau}$ (but close to $(1, \ldots, 1)$)

$$\sigma_{\boldsymbol{d}, \boldsymbol{\tau}} : \mathrm{PS}^{\boldsymbol{d}} \times \mathrm{PS}^{\boldsymbol{d}} \to \mathbb{R},$$

given for $\boldsymbol{\theta} = (\overline{\boldsymbol{\theta}}, \boldsymbol{\tau})$ by

$$\sigma_{\boldsymbol{d}, \boldsymbol{\tau}} \left( \{\langle \overline{\boldsymbol{\theta}}^{(q)}, \overline{\boldsymbol{x}}^{(q)} \rangle / \sqrt{d_q}\}_{q \in [Q]} \right) = \sigma \left( \sum_{q \in [Q]} \frac{\tau^{(q)} r_q}{R} \cdot \frac{\langle \overline{\boldsymbol{\theta}}^{(q)}, \overline{\boldsymbol{x}}^{(q)} \rangle}{\sqrt{d_q}} \right).$$

We can therefore apply the algebra of tensor product of spherical harmonics and use the machinery developed in [4].

### C.3 Notations

Recall the definitions $\boldsymbol{d} = (d_1, \ldots, d_q)$, $\boldsymbol{\kappa} = (\kappa_1, \ldots, \kappa_Q)$, $d_q = d^{\eta_q}$, $r_q = d^{(\eta_q + \kappa_q)/2}$, $D = d^{\eta_1} + \ldots + d^{\eta_Q}$ and $R = (d^{\eta_1 + \kappa_1} + \ldots + d^{\eta_Q + \kappa_Q})^{1/2}$. Let us denote $\xi = \max_{q \in [Q]} \{\eta_q + \kappa_q\}$ and $q_\xi = \arg\min_{q \in [Q]} \{\eta_q + \kappa_q\}$.

Recall that $(\boldsymbol{\theta}_i)_{i \in [N]} \sim \mathrm{Unif}(\mathbb{S}^{D-1}(\sqrt{D}))$ independently. Let $\boldsymbol{\Theta} = (\boldsymbol{\theta}_1, \ldots, \boldsymbol{\theta}_N)$. We denote $\mathbb{E}_{\boldsymbol{\theta}}$ to be the expectation operator with respect to $\boldsymbol{\theta} \sim \mathrm{Unif}(\mathbb{S}^{D-1}(\sqrt{D}))$ and $\mathbb{E}_{\boldsymbol{\Theta}}$ the expectation operator with respect to $\boldsymbol{\Theta} = (\boldsymbol{\theta}_1, \ldots, \boldsymbol{\theta}_N) \sim \mathrm{Unif}(\mathbb{S}^{D-1}(\sqrt{D}))^{\otimes N}$.

We will denote $\mathbb{E}_{\overline{\boldsymbol{\theta}}}$ the expectation operator with respect to $\overline{\boldsymbol{\theta}} \equiv (\overline{\boldsymbol{\theta}}^{(1)}, \ldots, \overline{\boldsymbol{\theta}}^{(Q)}) \sim \mu_{\boldsymbol{d}}$, $\mathbb{E}_{\overline{\boldsymbol{\Theta}}}$ the expectation operator with respect to $\overline{\boldsymbol{\Theta}} = (\overline{\boldsymbol{\theta}}_1, \ldots, \overline{\boldsymbol{\theta}}_N)$, and $\mathbb{E}_{\boldsymbol{\tau}}$ the expectation operator with respect to $\boldsymbol{\tau}$ (we recall $\boldsymbol{\tau} \equiv (\tau^{(1)}, \ldots, \tau^{(Q)})$) or $(\boldsymbol{\tau}_1, \ldots, \boldsymbol{\tau}_N)$ (where the $\boldsymbol{\tau}_i$ are independent) depending on the context. In particular, notice that $\mathbb{E}_{\boldsymbol{\theta}} = \mathbb{E}_{\boldsymbol{\tau}} \mathbb{E}_{\overline{\boldsymbol{\theta}}}$ and $\mathbb{E}_{\boldsymbol{\Theta}} = \mathbb{E}_{\boldsymbol{\tau}} \mathbb{E}_{\overline{\boldsymbol{\Theta}}}$.

We will denote $\mathbb{E}_{\boldsymbol{\Theta}_\varepsilon}$ the expectation operator with respect to $\boldsymbol{\Theta} = (\boldsymbol{\theta}_1, \ldots, \boldsymbol{\theta}_N)$ restricted to $\mathcal{P}_{d,N,\varepsilon}$ and $\mathbb{E}_{\boldsymbol{\tau}_\varepsilon}$ the expectation operator with respect to $\boldsymbol{\tau}$ restricted to $[1 - \varepsilon, 1 + \varepsilon]^Q$. Notice that $\mathbb{E}_{\boldsymbol{\Theta}_\varepsilon} = \mathbb{E}_{\boldsymbol{\tau}_\varepsilon} \mathbb{E}_{\overline{\boldsymbol{\Theta}}}$.

Let $\mathbb{E}_{\boldsymbol{x}}$ to be the expectation operator with respect to $\boldsymbol{x} \sim \mu_{\boldsymbol{\kappa}}^{\boldsymbol{d}}$, and $\mathbb{E}_{\overline{\boldsymbol{x}}}$ the expectation operator with respect to $\overline{\boldsymbol{x}} \sim \mu_{\boldsymbol{d}}$.

## C.4  Generalization error of kernel ridge regression

We consider the Kernel Ridge Regression solution $\hat{a}_i$, namely

$$\hat{a} = (H + \lambda I_n)^{-1} y,$$

where the kernel matrix $H = (H_{ij})_{ij \in [n]}$ is assumed to be given by

$$H_{ij} = \bar{h}_d(\langle x_i, x_j \rangle / R^2) = \mathbb{E}_{\bar{\theta} \sim \mathrm{Unif}(\mathrm{PS}^d)}[\sigma(\langle \bar{\theta}, x \rangle / R) \sigma(\langle \bar{\theta}, y \rangle / R)],$$

and $y = (y_1, \ldots, y_n)^\mathsf{T} = f + \varepsilon$, with

$$f = (f_d(x_1), \ldots, f_d(x_n))^\mathsf{T},$$
$$\varepsilon = (\varepsilon_1, \ldots, \varepsilon_n)^\mathsf{T}.$$

The prediction function at location $x$ gives

$$\hat{f}_\lambda(x) = y^\mathsf{T}(H + \lambda I_n)^{-1} h(x),$$

with

$$h(x) = [\bar{h}_d(\langle x, x_1 \rangle / R^2), \ldots, \bar{h}_d(\langle x, x_n \rangle / R^2)]^\mathsf{T}.$$

The test error of empirical kernel ridge regression is defined as

$$R_{\mathrm{KRR}}(f_d, X, \lambda) \equiv \mathbb{E}_x\left[\left(f_d(x) - y^\mathsf{T}(H + \lambda I_n)^{-1} h(x)\right)^2\right].$$

We define the set $\overline{\mathcal{Q}}_{\mathrm{KRR}}(\gamma) \subseteq \mathbb{Z}_{\geq 0}^Q$ as follows (recall that $\xi \equiv \max_{q \in [Q]}(\eta_q + \kappa_q)$):

$$\overline{\mathcal{Q}}_{\mathrm{KRR}}(\gamma) = \left\{ k \in \mathbb{Z}_{\geq 0}^Q \Big| \sum_{q=1}^Q (\xi - \kappa_q) k_q \leq \gamma \right\}, \tag{31}$$

and the function $m : \mathbb{R}_{\geq 0} \to \mathbb{R}_{\geq 0}$ which at $\gamma$ associates

$$m(\gamma) = \min_{k \notin \overline{\mathcal{Q}}_{\mathrm{KRR}}(\gamma)} \sum_{q \in [Q]} (\xi - \kappa_q) k_q.$$

Notice that by definition $m(\gamma) > \gamma$.

We consider sequences of problems indexed by the integer $d$, and we view the problem parameters (in particular, the dimensions $d_q$, the radii $r_q$, the kernel $h_d$, and so on) as functions of $d$.

**Assumption 1.** *Let $\{h_d\}_{d \geq 1}$ be a sequence of functions $h_d : [-1, 1] \to \mathbb{R}$ such that $H_d(x_1, x_2) = h_d(\langle x_1, x_2 \rangle / d)$ is a positive semidefinite kernel.*

*(a) For $\gamma > 0$ (which is specified in the theorem), we denote $L = \max_{q \in [Q]} \lceil \gamma / \eta_q \rceil$. We assume that $h_d$ is $L$-weakly differentiable. We assume that for $0 \leq k \leq L$, the $k$-th weak derivative verifies almost surely $h_d^{(k)}(u) \leq C$ for some constants $C > 0$ independent of $d$. Furthermore, we assume there exists $k > L$ such that $h_d^{(k)}(0) \geq c > 0$ with $c$ independent of $d$.*

*(b) For $\gamma > 0$ (which is specified in the theorem), we define*

$$\overline{K} = \max_{k \in \overline{\mathcal{Q}}_{\mathrm{KRR}}(\gamma)} |k|.$$

*We assume that $\sigma$ verifies for $k \leq \overline{K}$, $h_d^{(k)}(0) \geq c$, with $c > 0$ independent of $d$.*

**Theorem 1** (Risk of the KRR model). *Let $\{f_d \in L^2(\mathrm{PS}_\kappa^d, \mu_d^\kappa)\}_{d \geq 1}$ be a sequence of functions. Assume $w_d(d^\gamma \log d) \leq n \leq O_d(d^{m(\gamma) - \delta})$ for some $\gamma > 0$ and $\delta > 0$. Let $\{h_d\}_{d \geq 1}$ be a sequence of functions that satisfies Assumption 1 at level $\gamma$. Let $X = (x_i)_{i \in [n]}$ with $(x_i)_{i \in [n]} \sim \mathrm{Unif}(\mathrm{PS}_\kappa^d)$ independently, and $y_i = f_d(x_i) + \varepsilon_i$ and $\varepsilon_i \sim_{iid} \mathsf{N}(0, \tau^2)$ for some $\tau^2 \geq 0$. Then for any $\varepsilon > 0$, and for any $\lambda = O_d(1)$, with high probability we have*

$$\left| R_{\mathrm{KRR}}(f_d, X, \lambda) - \|\mathsf{P}_{\mathcal{Q}^c} f_d\|_{L^2}^2 \right| \leq \varepsilon(\|f_d\|_{L^2}^2 + \tau^2). \tag{32}$$

See Section D for the proof of this Theorem.

## C.5 Approximation error of the random features model

We consider the minimum population error for the random features model

$$R_{\mathrm{RF}}(f_d, \boldsymbol{W}) = \inf_{f \in \mathcal{F}_{\mathrm{RF}}(\boldsymbol{W})} \mathbb{E}\big[(f_*(\boldsymbol{x}) - f(\boldsymbol{x}))^2\big].$$

Let us define the sets:

$$\mathcal{Q}_{\mathrm{RF}}(\gamma) = \Big\{ \boldsymbol{k} \in \mathbb{Z}_{\geq 0}^Q \,\Big|\, \sum_{q=1}^Q (\xi - \kappa_q) k_q < \gamma \Big\}, \tag{33}$$

$$\overline{\mathcal{Q}}_{\mathrm{RF}}(\gamma) = \Big\{ \boldsymbol{k} \in \mathbb{Z}_{\geq 0}^Q \,\Big|\, \sum_{q=1}^Q (\xi - \kappa_q) k_q \leq \gamma \Big\}. \tag{34}$$

**Assumption 2.** *Let $\sigma$ be an activation function.*

(a) *There exists constants $c_0, c_1$, with $c_0 > 0$ and $c_1 < 1$ such that the activation function $\sigma$ verifies $\sigma(u)^2 \leq c_0 \exp(c_1 u^2/2)$ almost surely for $u \in \mathbb{R}$.*

(b) *For $\gamma > 0$ (which is specified in the theorem), we denote $L = \max_{q \in [Q]} \lceil \gamma/\eta_q \rceil$. We assume that $\sigma$ is $L$-weakly differentiable. Define*

$$K = \min_{\boldsymbol{k} \in \mathcal{Q}_{\mathrm{RF}}(\gamma)^c} |\boldsymbol{k}|.$$

*We assume that for $K \leq k \leq L$, the $k$-th weak derivative verifies almost surely $\sigma^{(k)}(u)^2 \leq c_0 \exp(c_1 u^2/2)$ for some constants $c_0 > 0$ and $c_1 < 1$.*

*Furthermore we will assume that $\sigma$ is not a degree-$\lfloor \gamma/\eta_{q_\xi} \rfloor$ polynomial where we recall that $q_\xi$ corresponds to the unique $\arg\min_{q \in [Q]} \{\eta_q + \kappa_q\}$.*

(c) *For $\gamma > 0$ (which is specified in the theorem), we define*

$$\overline{K} = \max_{\boldsymbol{k} \in \overline{\mathcal{Q}}_{\mathrm{RF}}(\gamma)} |\boldsymbol{k}|.$$

*We assume that $\sigma$ verifies for $k \leq \overline{K}$, $\mu_k(\sigma) \neq 0$. Furthermore we assume that for $k \leq \overline{K}$, the $k$-th weak derivative verifies almost surely $\sigma^{(k)}(u)^2 \leq c_0 \exp(c_1 u^2/2)$ for some constants $c_0 > 0$ and $c_1 < 1$.*

Assumption 2.(a) implies that $\sigma \in L^2(\mathbb{R}, \gamma)$ where $\gamma(\mathrm{d}x) = e^{-x^2/2} \mathrm{d}x/\sqrt{2\pi}$ is the standard Gaussian measure. We recall the Hermite decomposition of $\sigma$,

$$\sigma(x) = \sum_{k=0}^\infty \frac{\mu_k(\sigma)}{k!} \mathrm{He}_k(x), \qquad \mu_k(\sigma) \equiv \mathbb{E}_{G \sim \mathsf{N}(0,1)}[\sigma(G) \mathrm{He}_k(G)]. \tag{35}$$

**Theorem 2** (Risk of the RF model). *Let $\{f_d \in L^2(\mathrm{PS}_{\boldsymbol{\kappa}}^{\boldsymbol{d}}, \mu_{\boldsymbol{d}}^{\boldsymbol{\kappa}})\}_{d \geq 1}$ be a sequence of functions. Let $\boldsymbol{W} = (\boldsymbol{w}_i)_{i \in [N]}$ with $(\boldsymbol{w}_i)_{i \in [N]} \sim \mathrm{Unif}(\mathbb{S}^{D-1})$ independently. We have the following results.*

(a) *Assume $N \leq o_d(d^\gamma)$ for a fixed $\gamma > 0$. Let $\sigma$ satisfy Assumptions 2.(a) and 2.(b) at level $\gamma$. Then, for any $\varepsilon > 0$, the following holds with high probability:*

$$\Big| R_{\mathrm{RF}}(f_d, \boldsymbol{W}) - R_{\mathrm{RF}}(\mathsf{P}_{\mathcal{Q}} f_d, \boldsymbol{W}) - \|\mathsf{P}_{\mathcal{Q}^c} f_d\|_{L^2}^2 \Big| \leq \varepsilon \|f_d\|_{L^2} \|\mathsf{P}_{\mathcal{Q}^c} f_d\|_{L^2}, \tag{36}$$

*where $\mathcal{Q} \equiv \mathcal{Q}_{\mathrm{RF}}(\gamma)$ is defined in Equation (33).*

(b) *Assume $N \geq w_d(d^\gamma)$ for some positive constant $\gamma > 0$, and $\sigma$ satisfy Assumptions 2.(a) and 2.(c) at level $\gamma$. Then for any $\varepsilon > 0$, the following holds with high probability:*

$$0 \leq R_{\mathrm{RF}}(\mathsf{P}_{\mathcal{Q}} f_d, \boldsymbol{W}) \leq \varepsilon \|\mathsf{P}_{\mathcal{Q}} f_d\|_{L^2}^2, \tag{37}$$

*where $\mathcal{Q} \equiv \overline{\mathcal{Q}}_{\mathrm{RF}}(\gamma)$ is defined in Equation (34).*

See Section E for the proof of the lower bound (36), and Section F for the proof of the upper bound (37).

**Remark 1.** *This theorems shows that for each $\gamma \notin (\xi - \kappa_1)\mathbb{Z}_{\geq 0} + \ldots + (\xi - \kappa_Q)\mathbb{Z}_{\geq 0}$, we can decompose our functional space as*

$$L^2(\text{PS}_{\boldsymbol{\kappa}}^{\boldsymbol{d}}, \mu_{\boldsymbol{d}}^{\boldsymbol{\kappa}}) = \mathcal{F}(\boldsymbol{\beta}, \boldsymbol{\kappa}, \gamma) \oplus \mathcal{F}^c(\boldsymbol{\beta}, \boldsymbol{\kappa}, \gamma),$$

*where*

$$\mathcal{F}(\boldsymbol{\beta}, \boldsymbol{\kappa}, \gamma) = \bigoplus_{\boldsymbol{k} \in \mathcal{Q}_{\text{RF}}(\gamma)} V_{\boldsymbol{k}}^{\boldsymbol{d}},$$

$$\mathcal{F}^c(\boldsymbol{\beta}, \boldsymbol{\kappa}, \gamma) = \bigoplus_{\boldsymbol{k} \notin \mathcal{Q}_{\text{RF}}(\gamma)} V_{\boldsymbol{k}}^{\boldsymbol{d}},$$

*such that for $N = d^\gamma$, RF model fits the subspace of low degree polynomials $\mathcal{F}(\boldsymbol{\beta}, \boldsymbol{\kappa}, \gamma)$ and cannot fit $\mathcal{F}^c(\boldsymbol{\beta}, \boldsymbol{\kappa}, \gamma)$, i.e.*

$$R_{\text{RF}}(f_d, \boldsymbol{W}) \approx \|\mathsf{P}_{Q_{\text{RF}}(\gamma)^c} f_d\|_{L^2}^2.$$

**Remark 2.** *In other words, we can fit a polynomial of degree $\boldsymbol{k} \in \mathbb{Z}_{\geq 0}^Q$, if and only if*

$$d^{(\xi - \kappa_1)k_1} \cdot \ldots \cdot d^{(\xi - \kappa_Q)k_Q} = d_{1,\text{eff}}^{k_1} \ldots d_{Q,\text{eff}}^{k_Q} = o_d(N).$$

*Each subspace has therefore an effective dimension $d_{q,\text{eff}} \equiv d^{\xi - \kappa_q} = d_q^{(\xi - \kappa_q)/\eta_q} \asymp D^{(\xi - \kappa_q)/\max_{q \in [Q]} \eta_q}$. This can be understood intuitively as follows,*

$$\sigma\left(\langle \boldsymbol{\theta}, \boldsymbol{x} \rangle / R\right) = \sigma\left(\sum_{q \in [Q]} \langle \boldsymbol{\theta}^{(q)}, \boldsymbol{x}^{(q)} \rangle / R\right).$$

*The term $q_\xi$ (recall that $q_\xi = \arg\max_q(\eta_q + \kappa_q)$ and $\xi = \eta_{q_\xi} + \kappa_{q_\xi}$) verifies $\langle \boldsymbol{\theta}^{(q_\xi)}, \boldsymbol{x}^{(q_\xi)} \rangle / R = \Theta_d(1)$ and has the same effective dimension $d_{q_\xi,\text{eff}} = d^{\eta_{q_\xi}}$ has in the uniform case restricted to the sphere $\mathbb{S}^{d^{\eta_q} - 1}(\sqrt{d^{\eta_q}})$ (the scaling of the sphere do not matter because of the global normalization factor $R^{-1}$). However, for $\eta_q + \kappa_q < \xi$, we have $\langle \boldsymbol{\theta}^{(q)}, \boldsymbol{x}^{(q)} \rangle / R = \Theta_d(d^{(\eta_q + \kappa_q - \xi)/2})$ and we will need $d^{\xi - \kappa_q - \eta_q}$ more neurons to capture the dependency on the q-th sphere coordinates. The effective dimension is therefore given by $d_{q,\text{eff}} = d_q \cdot d^{\xi - \kappa_q - \eta_q} = d^{\xi - \kappa_q}$.*

## C.6  Approximation error of the neural tangent model

We consider the minimum population error for the random features model

$$R_{\text{NT}}(f_d, \boldsymbol{W}) = \inf_{f \in \mathcal{F}_{\text{NT}}(\boldsymbol{W})} \mathbb{E}\left[(f_*(\boldsymbol{x}) - f(\boldsymbol{x}))^2\right].$$

For $\boldsymbol{k} \in \mathbb{Z}_{\geq 0}^Q$, we denote by $S(\boldsymbol{k}) \subseteq [Q]$ the subset of indices $q \in [Q]$ such that $k_q > 0$.

We define the sets

$$\mathcal{Q}_{\text{NT}}(\gamma) = \left\{ \boldsymbol{k} \in \mathbb{Z}_{\geq 0}^Q \,\Big|\, \sum_{q=1}^Q (\xi - \kappa_q)k_q < \gamma + \left(\xi - \min_{q \in S(\boldsymbol{k})} \kappa_q\right) \right\}, \tag{38}$$

$$\overline{\mathcal{Q}}_{\text{NT}}(\gamma) = \left\{ \boldsymbol{k} \in \mathbb{Z}_{\geq 0}^Q \,\Big|\, \sum_{q=1}^Q (\xi - \kappa_q)k_q \leq \gamma + \left(\xi - \min_{q \in S(\boldsymbol{k})} \kappa_q\right) \right\}. \tag{39}$$

**Assumption 3.** *Let $\sigma : \mathbb{R} \to \mathbb{R}$ be an activation function.*

(a) *The activation function $\sigma$ is weakly differentiable with weak derivative $\sigma'$. There exists constants $c_0, c_1$, with $c_0 > 0$ and $c_1 < 1$ such that the activation function $\sigma$ verifies $\sigma'(u)^2 \leq c_0 \exp(c_1 u^2/2)$ almost surely for $u \in \mathbb{R}$.*

(b) *For $\gamma > 0$ (which is specified in the theorem), we denote $L = \max_{q \in [Q]} \lceil \gamma/\eta_q \rceil$. We assume that $\sigma'$ is L-weakly differentiable. Define*

$$K = \min_{\boldsymbol{k} \in \mathcal{Q}_{\text{NT}}(\gamma)^c} |\boldsymbol{k}|.$$

*We assume that for $K - 1 \leq k \leq L$, the $k$-th weak derivative verifies almost surely $\sigma^{(k+1)}(u)^2 \leq c_0 \exp(c_1 u^2/2)$ for some constants $c_0 > 0$ and $c_1 < 1$.*

*Furthermore, we assume that $\sigma'$ verifies a non-degeneracy condition. Recall that $\mu_k(h) \equiv \mathbb{E}_{G \sim \mathsf{N}(0,1)}[h(G)\mathrm{He}_k(G)]$ denote the $k$-th coefficient of the Hermite expansion of $h \in L_2(\mathbb{R}, \gamma)$ (with $\gamma$ the standard Gaussian measure). Then there exists $k_1, k_2 \geq 2L + 7[\max_{q \in [Q]} \xi/\eta_q]$ such that $\mu_{k_1}(\sigma'), \mu_{k_2}(\sigma') \neq 0$ and*

$$\frac{\mu_{k_1}(x^2\sigma')}{\mu_{k_1}(\sigma')} \neq \frac{\mu_{k_2}(x^2\sigma')}{\mu_{k_2}(\sigma')} . \tag{40}$$

*(c) For $\gamma > 0$ (which is specified in the theorem), we define*

$$\overline{K} = \max_{\boldsymbol{k} \in \overline{\mathcal{Q}}_{\mathrm{NT}}(\gamma)} |\boldsymbol{k}|.$$

*We assume that $\sigma$ verifies for $k \leq \overline{K} + 1$, $\mu_k(\sigma') = \mu_{k+1}(\sigma) \neq 0$. Furthermore we assume that for $k \leq \overline{K} + 1$, the $k$-th weak derivative verifies almost surely $\sigma^{(k+1)}(u)^2 \leq c_0 \exp(c_1 u^2/2)$ for some constants $c_0 > 0$ and $c_1 < 1$.*

Assumption 3.(a) implies that $\sigma' \in L^2(\mathbb{R}, \gamma)$ where $\gamma(\mathrm{d}x) = e^{-x^2/2}\mathrm{d}x/\sqrt{2\pi}$ is the standard Gaussian measure. We recall the Hermite decomposition of $\sigma'$:

$$\sigma'(x) = \sum_{k=0}^{\infty} \frac{\mu_k(\sigma')}{k!}\mathrm{He}_k(x), \qquad \mu_k(\sigma') \equiv \mathbb{E}_{G \sim \mathsf{N}(0,1)}[\sigma'(G)\mathrm{He}_k(G)]. \tag{41}$$

In the Assumption 3.(b), it is useful to notice that the Hermite coefficients of $x^2\sigma'(x)$ can be computed from the ones of $\sigma'(x)$ using the relation $\mu_k(x^2\sigma') = \mu_{k+2}(\sigma') + [1 + 2k]\mu_k(\sigma') + k(k-1)\mu_{k-2}(\sigma')$.

**Theorem 3** (Risk of the NT model). *Let $\{f_d \in L^2(\mathrm{PS}_{\boldsymbol{\kappa}}^{\boldsymbol{d}}, \mu_{\boldsymbol{d}}^{\boldsymbol{\kappa}})\}_{d \geq 1}$ be a sequence of functions. Let $\boldsymbol{W} = (\boldsymbol{w}_i)_{i \in [N]}$ with $(\boldsymbol{w}_i)_{i \in [N]} \sim \mathrm{Unif}(\mathbb{S}^{D-1})$ independently. We have the following results.*

*(a) Assume $N \leq o_d(d^\gamma)$ for a fixed $\gamma > 0$. Let $\sigma$ satisfy Assumptions 3.(a) and 3.(b) at level $\gamma$. Then, for any $\varepsilon > 0$, the following holds with high probability:*

$$\left| R_{\mathrm{NT}}(f_d, \boldsymbol{W}) - R_{\mathrm{NT}}(\mathsf{P}_{\mathcal{Q}}f_d, \boldsymbol{W}) - \|\mathsf{P}_{\mathcal{Q}^c}f_d\|_{L^2}^2 \right| \leq \varepsilon \|f_d\|_{L^2} \|\mathsf{P}_{\mathcal{Q}^c}f_d\|_{L^2}, \tag{42}$$

*where $\mathcal{Q} \equiv \mathcal{Q}_{\mathrm{NT}}(\gamma)$ is defined in Equation (38).*

*(b) Assume $N \geq w_d(d^\gamma)$ for some positive constant $\gamma > 0$, and $\sigma$ satisfy Assumptions 3.(a) and 3.(c) at level $\gamma$. Then for any $\varepsilon > 0$, the following holds with high probability:*

$$0 \leq R_{\mathrm{NT}}(\mathsf{P}_{\mathcal{Q}}f_d, \boldsymbol{W}) \leq \varepsilon \|\mathsf{P}_{\mathcal{Q}}f_d\|_{L^2}^2, \tag{43}$$

*where $\mathcal{Q} \equiv \overline{\mathcal{Q}}_{\mathrm{NT}}(\gamma)$ is defined in Equation (39).*

See Section G for the proof of lower bound, and Section H for the proof of upper bound.

**Remark 3.** *This theorems shows that each for each $\gamma > 0$ such that $\mathcal{Q}_{\mathrm{NT}}(\gamma)^c \cap \overline{\mathcal{Q}}_{\mathrm{NT}}(\gamma) = \emptyset$, we can decompose our functional space as*

$$L^2(\mathrm{PS}_{\boldsymbol{\kappa}}^{\boldsymbol{d}}, \mu_{\boldsymbol{d}}^{\boldsymbol{\kappa}}) = \mathcal{F}(\boldsymbol{\beta}, \boldsymbol{\kappa}, \gamma) \oplus \mathcal{F}^c(\boldsymbol{\beta}, \boldsymbol{\kappa}, \gamma),$$

*where*

$$\mathcal{F}(\boldsymbol{\beta}, \boldsymbol{\kappa}, \gamma) = \bigoplus_{\boldsymbol{k} \in \mathcal{Q}_{\mathrm{NT}}(\gamma)} \boldsymbol{V}_{\boldsymbol{k}}^{\boldsymbol{d}},$$

$$\mathcal{F}^c(\boldsymbol{\beta}, \boldsymbol{\kappa}, \gamma) = \bigoplus_{\boldsymbol{k} \notin \mathcal{Q}_{\mathrm{NT}}(\gamma)} \boldsymbol{V}_{\boldsymbol{k}}^{\boldsymbol{d}},$$

*such that for $N = d^\gamma$, NT model fits the subspace of low degree polynomials $\mathcal{F}(\boldsymbol{\beta}, \boldsymbol{\kappa}, \gamma)$ and cannot fit $\mathcal{F}^c(\boldsymbol{\beta}, \boldsymbol{\kappa}, \gamma)$ at all, i.e.*

$$R_{\mathrm{NT}}(f_d, \boldsymbol{W}) \approx \|\mathsf{P}_{\mathcal{Q}_{\mathrm{NT}}(\gamma)^c}f_d\|_{L^2}^2.$$

**Remark 4.** *In other words, we can fit a polynomial of degree $\boldsymbol{k} \in \mathbb{Z}_{\geq 0}^Q$, if and only if*

$$d^{(\xi-\kappa_1)k_1} \cdot \ldots \cdot d^{(\xi-\kappa_Q)k_Q} = d_{1,\mathrm{eff}}^{k_1} \ldots d_{Q,\mathrm{eff}}^{k_Q} = o_d(d^\beta N),$$

*where $\beta = \xi - \min_{q \in S(\boldsymbol{k})} \kappa_q$.*

## C.7 Connecting to the theorems in the main text

Let us connect the above general results to the two-spheres setting described in the main text. We consider two spheres with $\eta_1 = \eta$, $\kappa_1 = \kappa$ for the first sphere, and $\eta_2 = 1$, $\kappa_2 = 0$ for the second sphere. We have $\xi = \max(\eta + \kappa, 1)$.

Let $w_d(d^\gamma \log d) \leq n \leq O_d(d^{\gamma+\delta})$ with $\delta > 0$ constant sufficiently small, then by Theorem 1 the function subspace learned by KRR is given by the polynomials of degree $k_1$ in the first sphere coordinates and $k_2$ in the second sphere with

$$\max(\eta, 1 - \kappa)k_1 + \max(\eta + \kappa, 1)k_2 < \gamma.$$

We consider functions that only depend on the first sphere, i.e., $k_2 = 0$ and denote $d_{\text{eff}} = d^{\max(\eta, 1-\kappa)}$. Then the subspace of approximation is given by the $k$ polynomials in the first sphere such that $d_{\text{eff}}^k \leq d^\gamma$. Furthermore, one can check that the Assumptions listed in Theorem 1 in the main text verifies Assumption 1.

Similarly, for $w_d(d^\gamma) \leq N \leq O_d(d^{\gamma+\delta})$ with $\delta > 0$ constant sufficiently small, Theorem 2 implies that the RF models can only approximate $k$ polynomials in the first sphere such that $d_{\text{eff}}^k \leq d^\gamma$. Furthermore, Assumptions listed in Theorem 2 in the main text verifies Assumption 2.

In the case of NT, we only consider $\boldsymbol{k} = (k_1, 0)$ and $S(\boldsymbol{k}) = \{1\}$. We get $\min_{q \in S(\boldsymbol{k})} \kappa_q = \kappa$. The subspace approximated is given by the $k$ polynomials in the first sphere such that $d_{\text{eff}}^k \leq d^\gamma d_{\text{eff}}$. Furthermore, Assumptions listed in Theorem 3 in the main text verifies Assumption 3.

# D Proof of Theorem 1

The proof follows closely the proof of [4, Theorem 4].

## D.1 Preliminaries

Let us rewrite the kernel functions $\{h_d\}_{d\geq 1}$ as functions on the product of normalized spheres: for $\boldsymbol{x} = \{\boldsymbol{x}^{(q)}\}_{q\in[Q]}$ and $\boldsymbol{y} = \{\boldsymbol{y}^{(q)}\}_{q\in[Q]} \in \mathrm{PS}_{\boldsymbol{\kappa}}^{\boldsymbol{d}}$:

$$
\begin{aligned}
h_d(\langle \boldsymbol{y}, \boldsymbol{x} \rangle / R^2) =& h_d\left( \sum_{q\in[Q]} (r_q^2 / R^2 \sqrt{d_q}) \cdot \langle \overline{\boldsymbol{y}}^{(q)}, \overline{\boldsymbol{x}}^{(q)} \rangle / \sqrt{d_q} \right) \\
\equiv& h_{\boldsymbol{d}}\left( \{\langle \overline{\boldsymbol{y}}^{(q)}, \overline{\boldsymbol{x}}^{(q)} \rangle / \sqrt{d_q} \}_{q\in[Q]} \right).
\end{aligned}
\tag{44}
$$

Consider the expansion of $h_{\boldsymbol{d}}$ in terms of tensor product of Gegenbauer polynomials. We have

$$h_d(\langle \boldsymbol{y}, \boldsymbol{x} \rangle / R^2) = \sum_{\boldsymbol{k} \in \mathbb{Z}_{\geq 0}^Q} \lambda_{\boldsymbol{k}}^{\boldsymbol{d}}(h_{\boldsymbol{d}}) B(\boldsymbol{d}, \boldsymbol{k}) Q_{\boldsymbol{k}}^{\boldsymbol{d}}\left( \{\langle \overline{\boldsymbol{y}}^{(q)}, \overline{\boldsymbol{x}}^{(q)} \rangle \}_{q\in[Q]} \right),$$

where

$$\lambda_{\boldsymbol{k}}^{\boldsymbol{d}}(h_{\boldsymbol{d}}) = \mathbb{E}_{\overline{\boldsymbol{x}}}\left[ h_{\boldsymbol{d}}\left( \overline{x}_1^{(1)}, \dots, \overline{x}_1^{(Q)} \right) Q_{\boldsymbol{k}}^{\boldsymbol{d}}\left( \sqrt{d_1} \overline{x}_1^{(1)}, \dots, \sqrt{d_Q} \overline{x}_1^{(Q)} \right) \right],$$

where the expectation is taken over $\overline{\boldsymbol{x}} = (\overline{\boldsymbol{x}}^{(1)}, \dots, \overline{\boldsymbol{x}}^{(Q)}) \sim \mu_{\boldsymbol{d}}$.

**Lemma 2.** *Let $\{h_d\}_{d\geq 1}$ be a sequence of kernel functions that satisfies Assumption 1. Assume $w_d(d^\gamma) \leq n \leq o_d(d^{m(\gamma)})$ for some $\gamma > 0$. Consider $\mathcal{Q} = \overline{\mathcal{Q}}_{\mathrm{KRR}}(\gamma)$ as defined in Eq. (31). Then there exists constants $c, C > 0$ such that for $d$ large enough,*

$$\max_{\boldsymbol{k} \notin \mathcal{Q}} \lambda_{\boldsymbol{k}}^{\boldsymbol{d}}(h_{\boldsymbol{d}}) \leq Cd^{-m(\gamma)},$$

$$\min_{\boldsymbol{k} \in \mathcal{Q}} \lambda_{\boldsymbol{k}}^{\boldsymbol{d}}(h_{\boldsymbol{d}}) \geq cd^{-\gamma}.$$

*Proof of Lemma 2.* Notice that by Lemma 18,

$$\lambda_{\boldsymbol{k}}^{\boldsymbol{d}}(h_{\boldsymbol{d}}) = \left( \prod_{q\in[Q]} \alpha_q^{k_q} \right) \cdot R(\boldsymbol{d}, \boldsymbol{k}) \cdot \mathbb{E}_{\overline{\boldsymbol{x}}}\left[ \left( \prod_{q\in[Q]} \left( 1 - \frac{(\overline{x}_1^{(q)})^2}{d_q} \right)^{k_q} \right) \cdot h_d^{(|\boldsymbol{k}|)}\left( \sum_{q\in[Q]} \alpha_q \overline{x}_1^{(q)} \right) \right],$$

where $\alpha_q = d_q^{-1/2} r_q^2 / R^2 = (1 + o_d(1)) d^{\eta_q/2 + \kappa_q - \xi}$. By Assumption 1.$(a)$, we have

$$\lambda_{\boldsymbol{k}}^{\boldsymbol{d}}(h_{\boldsymbol{d}}) B(\boldsymbol{d}, \boldsymbol{k}) \leq C \prod_{q \in [Q]} d^{(\kappa_q - \xi) k_q}.$$

Furthermore, by Assumption 1.$(b)$ and dominated convergence,

$$\mathbb{E}_{\overline{\boldsymbol{x}}} \left[ \left( \prod_{q \in [Q]} \left( 1 - \frac{(\overline{x}_1^{(q)})^2}{d_q} \right)^{k_q} \right) \cdot h_d^{(|\boldsymbol{k}|)} \left( \sum_{q \in [Q]} \alpha_q \overline{x}_1^{(q)} \right) \right] \to h_d^{(|\boldsymbol{k}|)}(0) \geq c > 0,$$

for $k \geq \overline{K}$. The lemma then follows from the same proof as in Lemma 9 and Lemma 10, where we adapt the proofs of Lemma 19 and 20 to $h_{\boldsymbol{d}}$. $\qquad\square$

### D.2 Proof of Theorem 1

**Step 1. Rewrite the $y$, $E$, $H$, $M$ matrices.**

The test error of empirical kernel ridge regression gives

$$
\begin{aligned}
R_{\mathrm{KRR}}(f_d, \boldsymbol{X}, \lambda) \equiv{} & \mathbb{E}_{\boldsymbol{x}} \left[ \left( f_d(\boldsymbol{x}) - \boldsymbol{y}^\mathsf{T} (\boldsymbol{H} + \lambda \mathbf{I}_n)^{-1} \boldsymbol{h}(\boldsymbol{x}) \right)^2 \right] \\
={} & \mathbb{E}_{\boldsymbol{x}}[f_d(\boldsymbol{x})^2] - 2 \boldsymbol{y}^\mathsf{T} (\boldsymbol{H} + \lambda \mathbf{I}_n)^{-1} \boldsymbol{E} + \boldsymbol{y}^\mathsf{T} (\boldsymbol{H} + \lambda \mathbf{I}_n)^{-1} \boldsymbol{M} (\boldsymbol{H} + \lambda \mathbf{I}_n)^{-1} \boldsymbol{y},
\end{aligned}
$$

where $\boldsymbol{E} = (E_1, \ldots, E_n)^\mathsf{T}$ and $\boldsymbol{M} = (M_{ij})_{ij \in [n]}$, with

$$
\begin{aligned}
E_i ={} & \mathbb{E}_{\boldsymbol{x}}[f_d(\boldsymbol{x}) h_d(\langle \boldsymbol{x}, \boldsymbol{x}_i \rangle / d)], \\
M_{ij} ={} & \mathbb{E}_{\boldsymbol{x}}[h_d(\langle \boldsymbol{x}_i, \boldsymbol{x} \rangle / d) h_d(\langle \boldsymbol{x}_j, \boldsymbol{x} \rangle / d)].
\end{aligned}
$$

Let $B = \sum_{\boldsymbol{k} \in \mathcal{Q}} B(\boldsymbol{d}, \boldsymbol{k})$. Define for any $\boldsymbol{k} \in \mathbb{Z}_{\geq 0}^Q$,

$$
\begin{aligned}
\boldsymbol{D}_{\boldsymbol{k}} ={} & \lambda_{\boldsymbol{k}}^{\boldsymbol{d}}(h_{\boldsymbol{d}}) \mathbf{I}_{B(\boldsymbol{d}, \boldsymbol{k})}, \\
\boldsymbol{Y}_{\boldsymbol{k}} ={} & (Y_{\boldsymbol{k}, \boldsymbol{s}}^{\boldsymbol{d}}(\overline{\boldsymbol{x}}_i))_{i \in [n], \boldsymbol{s} \in [B(\boldsymbol{d}, \boldsymbol{k})]} \in \mathbb{R}^{n \times B(\boldsymbol{d}, \boldsymbol{k})}, \\
\boldsymbol{\lambda}_{\boldsymbol{k}} ={} & (\lambda_{\boldsymbol{k}, \boldsymbol{s}}^{\boldsymbol{d}}(f_d))_{\boldsymbol{s} \in [B(\boldsymbol{d}, \boldsymbol{k})]}^\mathsf{T} \in \mathbb{R}^{B(\boldsymbol{d}, \boldsymbol{k})}, \\
\boldsymbol{D}_{\mathcal{Q}} ={} & \mathrm{diag}\left( \left( \lambda_{\boldsymbol{k}}^{\boldsymbol{d}}(h_{\boldsymbol{d}}) \mathbf{I}_{B(\boldsymbol{d}, \boldsymbol{k})} \right)_{\boldsymbol{k} \in \mathcal{Q}} \right) \in \mathbb{R}^{B \times B} \\
\boldsymbol{Y}_{\mathcal{Q}} ={} & (\boldsymbol{Y}_{\boldsymbol{k}})_{\boldsymbol{k} \in \mathcal{Q}} \in \mathbb{R}^{n \times B}, \\
\boldsymbol{\lambda}_{\mathcal{Q}} ={} & \left( \left( \boldsymbol{\lambda}_{\boldsymbol{k}}^\mathsf{T} \right)_{\boldsymbol{k} \in \mathcal{Q}} \right)^\mathsf{T} \in \mathbb{R}^B.
\end{aligned}
$$

Let the spherical harmonics decomposition of $f_d$ be

$$f_d(\boldsymbol{x}) = \sum_{\boldsymbol{k} \in \mathbb{Z}_{\geq 0}^Q} \sum_{\boldsymbol{s} \in [B(\boldsymbol{d}, \boldsymbol{k})]} \lambda_{\boldsymbol{k}, \boldsymbol{s}}^{\boldsymbol{d}}(f_d) Y_{\boldsymbol{k}, \boldsymbol{s}}^{\boldsymbol{d}}(\overline{\boldsymbol{x}}),$$

and the Gegenbauer decomposition of $h_{\boldsymbol{d}}$ be

$$h_{\boldsymbol{d}}\left(\overline{x}_1^{(1)}, \ldots, \overline{x}_1^{(Q)}\right) = \sum_{\boldsymbol{k} \in \mathbb{Z}_{\geq 0}^Q} \lambda_{\boldsymbol{k}}^{\boldsymbol{d}}(h_{\boldsymbol{d}}) B(\boldsymbol{d}, \boldsymbol{k}) Q_{\boldsymbol{k}}^{\boldsymbol{d}}\left(\sqrt{d_1}\overline{x}_1^{(1)}, \ldots, \sqrt{d_Q}\overline{x}_1^{(Q)}\right).$$

We write the decompositions of vectors $\boldsymbol{f}$, $\boldsymbol{E}$, $\boldsymbol{H}$, and $\boldsymbol{M}$. We have

$$
\begin{aligned}
\boldsymbol{f} ={} & \boldsymbol{Y}_{\mathcal{Q}} \boldsymbol{\lambda}_{\mathcal{Q}} + \sum_{\boldsymbol{k} \in \mathcal{Q}^c} \boldsymbol{Y}_{\boldsymbol{k}} \boldsymbol{\lambda}_{\boldsymbol{k}}, \\
\boldsymbol{E} ={} & \boldsymbol{Y}_{\mathcal{Q}} \boldsymbol{D}_{\mathcal{Q}} \boldsymbol{\lambda}_{\mathcal{Q}} + \sum_{\boldsymbol{k} \in \mathcal{Q}^c} \boldsymbol{Y}_{\boldsymbol{k}} \boldsymbol{D}_{\boldsymbol{k}} \boldsymbol{\lambda}_{\boldsymbol{k}}, \\
\boldsymbol{H} ={} & \boldsymbol{Y}_{\mathcal{Q}} \boldsymbol{D}_{\mathcal{Q}} \boldsymbol{Y}_{\mathcal{Q}}^\mathsf{T} + \sum_{\boldsymbol{k} \in \mathcal{Q}^c} \boldsymbol{Y}_{\boldsymbol{k}} \boldsymbol{D}_{\boldsymbol{k}} \boldsymbol{Y}_{\boldsymbol{k}}^\mathsf{T}, \\
\boldsymbol{M} ={} & \boldsymbol{Y}_{\mathcal{Q}} \boldsymbol{D}_{\mathcal{Q}}^2 \boldsymbol{Y}_{\mathcal{Q}}^\mathsf{T} + \sum_{\boldsymbol{k} \in \mathcal{Q}^c} \boldsymbol{Y}_{\boldsymbol{k}} \boldsymbol{D}_{\boldsymbol{k}}^2 \boldsymbol{Y}_{\boldsymbol{k}}^\mathsf{T}.
\end{aligned}
$$

From Lemma 4, we can rewrite

$$\boldsymbol{H} = \boldsymbol{Y}_{\mathcal{Q}} \boldsymbol{D}_{\mathcal{Q}} \boldsymbol{Y}_{\mathcal{Q}}^{\mathsf{T}} + \kappa_h (\mathbf{I}_n + \boldsymbol{\Delta}_h),$$
$$\boldsymbol{M} = \boldsymbol{Y}_{\mathcal{Q}} \boldsymbol{D}_{\mathcal{Q}}^2 \boldsymbol{Y}_{\mathcal{Q}}^{\mathsf{T}} + \kappa_u \boldsymbol{\Delta}_u,$$

where $\kappa_h = \Theta_d(1)$, $\kappa_u = O_d(d^{-m(\gamma)})$, $\|\boldsymbol{\Delta}_h\|_{\mathrm{op}} = o_{d,\mathbb{P}}(1)$ and $\|\boldsymbol{\Delta}_u\|_{\mathrm{op}} = O_{d,\mathbb{P}}(1)$.

**Step 2. Decompose the risk**

The rest of the proof follows closely from [4, Theorem 4]. We decompose the risk as follows

$$R_{\mathrm{KRR}}(f_d, \boldsymbol{X}, \lambda) = \|f_d\|_{L^2}^2 - 2T_1 + T_2 + T_3 - 2T_4 + 2T_5.$$

where

$$T_1 = \boldsymbol{f}^{\mathsf{T}} (\boldsymbol{H} + \lambda \mathbf{I}_n)^{-1} \boldsymbol{E},$$
$$T_2 = \boldsymbol{f}^{\mathsf{T}} (\boldsymbol{H} + \lambda \mathbf{I}_n)^{-1} \boldsymbol{M} (\boldsymbol{H} + \lambda \mathbf{I}_n)^{-1} \boldsymbol{f},$$
$$T_3 = \boldsymbol{\varepsilon}^{\mathsf{T}} (\boldsymbol{H} + \lambda \mathbf{I}_n)^{-1} \boldsymbol{M} (\boldsymbol{H} + \lambda \mathbf{I}_n)^{-1} \boldsymbol{\varepsilon},$$
$$T_4 = \boldsymbol{\varepsilon}^{\mathsf{T}} (\boldsymbol{H} + \lambda \mathbf{I}_n)^{-1} \boldsymbol{E},$$
$$T_5 = \boldsymbol{\varepsilon}^{\mathsf{T}} (\boldsymbol{H} + \lambda \mathbf{I}_n)^{-1} \boldsymbol{M} (\boldsymbol{H} + \lambda \mathbf{I}_n)^{-1} \boldsymbol{f}.$$

Further, we denote $\boldsymbol{f}_{\mathcal{Q}}$, $\boldsymbol{f}_{\mathcal{Q}^c}$, $\boldsymbol{E}_{\mathcal{Q}}$, and $\boldsymbol{E}_{\mathcal{Q}^c}$,

$$\boldsymbol{f}_{\mathcal{Q}} = \boldsymbol{Y}_{\mathcal{Q}} \boldsymbol{\lambda}_{\mathcal{Q}}, \qquad \boldsymbol{E}_{\mathcal{Q}} = \boldsymbol{Y}_{\mathcal{Q}} \boldsymbol{D}_{\mathcal{Q}} \boldsymbol{\lambda}_{\mathcal{Q}},$$
$$\boldsymbol{f}_{\mathcal{Q}^c} = \sum_{\boldsymbol{k} \in \mathcal{Q}^c} \boldsymbol{Y}_{\boldsymbol{k}} \boldsymbol{\lambda}_{\boldsymbol{k}}, \quad \boldsymbol{E}_{\mathcal{Q}^c} = \sum_{\boldsymbol{k} \in \mathcal{Q}^c} \boldsymbol{Y}_{\boldsymbol{k}} \boldsymbol{D}_{\boldsymbol{k}} \boldsymbol{\lambda}_{\boldsymbol{k}}.$$

**Step 3. Term $T_2$**

Note we have

$$T_2 = T_{21} + T_{22} + T_{23},$$

where

$$T_{21} = \boldsymbol{f}_{\mathcal{Q}}^{\mathsf{T}} (\boldsymbol{H} + \lambda \mathbf{I}_n)^{-1} \boldsymbol{M} (\boldsymbol{H} + \lambda \mathbf{I}_n)^{-1} \boldsymbol{f}_{\mathcal{Q}},$$
$$T_{22} = 2 \boldsymbol{f}_{\mathcal{Q}}^{\mathsf{T}} (\boldsymbol{H} + \lambda \mathbf{I}_n)^{-1} \boldsymbol{M} (\boldsymbol{H} + \lambda \mathbf{I}_n)^{-1} \boldsymbol{f}_{\mathcal{Q}^c},$$
$$T_{23} = \boldsymbol{f}_{\mathcal{Q}^c}^{\mathsf{T}} (\boldsymbol{H} + \lambda \mathbf{I}_n)^{-1} \boldsymbol{M} (\boldsymbol{H} + \lambda \mathbf{I}_n)^{-1} \boldsymbol{f}_{\mathcal{Q}^c}.$$

By Lemma 6, we have

$$\|n (\boldsymbol{H} + \lambda \mathbf{I}_n)^{-1} \boldsymbol{M} (\boldsymbol{H} + \lambda \mathbf{I}_n)^{-1} - \boldsymbol{Y}_{\mathcal{Q}} \boldsymbol{Y}_{\mathcal{Q}}^{\mathsf{T}} / n\|_{\mathrm{op}} = o_{d,\mathbb{P}}(1), \tag{45}$$

hence

$$T_{21} = \boldsymbol{\lambda}_{\mathcal{Q}} \boldsymbol{Y}_{\mathcal{Q}}^{\mathsf{T}} (\boldsymbol{H} + \lambda \mathbf{I}_n)^{-1} \boldsymbol{M} (\boldsymbol{H} + \lambda \mathbf{I}_n)^{-1} \boldsymbol{Y}_{\mathcal{Q}} \boldsymbol{\lambda}_{\mathcal{Q}}$$
$$= \boldsymbol{\lambda}_{\mathcal{Q}}^{\mathsf{T}} \boldsymbol{Y}_{\mathcal{Q}}^{\mathsf{T}} \boldsymbol{Y}_{\mathcal{Q}} \boldsymbol{Y}_{\mathcal{Q}}^{\mathsf{T}} \boldsymbol{Y}_{\mathcal{Q}} \boldsymbol{\lambda}_{\mathcal{Q}} / n^2 + [\|\boldsymbol{Y}_{\mathcal{Q}} \boldsymbol{\lambda}_{\mathcal{Q}}\|_2^2 / n] \cdot o_{d,\mathbb{P}}(1).$$

By Lemma 3, we have (with $\|\boldsymbol{\Delta}\|_2 = o_{d,\mathbb{P}}(1)$)

$$\boldsymbol{\lambda}_{\mathcal{Q}}^{\mathsf{T}} \boldsymbol{Y}_{\mathcal{Q}}^{\mathsf{T}} \boldsymbol{Y}_{\mathcal{Q}} \boldsymbol{Y}_{\mathcal{Q}}^{\mathsf{T}} \boldsymbol{Y}_{\mathcal{Q}} \boldsymbol{\lambda}_{\mathcal{Q}} / n^2 = \boldsymbol{\lambda}_{\mathcal{Q}}^{\mathsf{T}} (\mathbf{I}_B + \boldsymbol{\Delta})^2 \boldsymbol{\lambda}_{\mathcal{Q}} = \|\boldsymbol{\lambda}_{\mathcal{Q}}\|_2^2 (1 + o_{d,\mathbb{P}}(1)).$$

Moreover, we have

$$\|\boldsymbol{Y}_{\mathcal{Q}} \boldsymbol{\lambda}_{\mathcal{Q}}\|_2^2 / n = \boldsymbol{\lambda}_{\mathcal{Q}}^{\mathsf{T}} (\mathbf{I}_B + \boldsymbol{\Delta}) \boldsymbol{\lambda}_{\mathcal{Q}} = \|\boldsymbol{\lambda}_{\mathcal{Q}}\|_2^2 (1 + o_{d,\mathbb{P}}(1)).$$

As a result, we have

$$T_{21} = \|\boldsymbol{\lambda}_{\mathcal{Q}}\|_2^2 (1 + o_{d,\mathbb{P}}(1)) = \|\mathsf{P}_{\mathcal{Q}} f_d\|_{L^2}^2 (1 + o_{d,\mathbb{P}}(1)). \tag{46}$$

By Eq. (45) again, we have

$$T_{23} = \Big( \sum_{\boldsymbol{k} \in \mathcal{Q}^c} \boldsymbol{\lambda}_{\boldsymbol{k}}^{\mathsf{T}} \boldsymbol{Y}_{\boldsymbol{k}}^{\mathsf{T}} \Big) (\boldsymbol{H} + \lambda \mathbf{I}_n)^{-1} \boldsymbol{M} (\boldsymbol{H} + \lambda \mathbf{I}_n)^{-1} \Big( \sum_{\boldsymbol{k} \in \mathcal{Q}^c} \boldsymbol{Y}_{\boldsymbol{k}} \boldsymbol{\lambda}_{\boldsymbol{k}} \Big)$$
$$= \Big( \sum_{\boldsymbol{k} \in \mathcal{Q}^c} \boldsymbol{\lambda}_{\boldsymbol{k}}^{\mathsf{T}} \boldsymbol{Y}_{\boldsymbol{k}}^{\mathsf{T}} \Big) \boldsymbol{Y}_{\mathcal{Q}} \boldsymbol{Y}_{\mathcal{Q}}^{\mathsf{T}} \Big( \sum_{\boldsymbol{k} \in \mathcal{Q}^c} \boldsymbol{Y}_{\boldsymbol{k}} \boldsymbol{\lambda}_{\boldsymbol{k}} \Big) / n^2 + \Big[ \Big\| \sum_{\boldsymbol{k} \in \mathcal{Q}^c} \boldsymbol{Y}_{\boldsymbol{k}} \boldsymbol{\lambda}_{\boldsymbol{k}} \Big\|_2^2 / n \Big] \cdot o_{d,\mathbb{P}}(1).$$

By Lemma 5, we have

$$\mathbb{E}\Big[\Big(\sum_{\boldsymbol{k}\in\mathcal{Q}^c}\boldsymbol{\lambda}_{\boldsymbol{k}}^{\mathsf{T}}\boldsymbol{Y}_{\boldsymbol{k}}^{\mathsf{T}}\Big)\boldsymbol{Y}_{\mathcal{Q}}\boldsymbol{Y}_{\mathcal{Q}}^{\mathsf{T}}\Big(\sum_{\boldsymbol{k}\in\mathcal{Q}^c}\boldsymbol{Y}_{\boldsymbol{k}}\boldsymbol{\lambda}_{\boldsymbol{k}}\Big)\Big]/n^2 = \sum_{\boldsymbol{u},\boldsymbol{v}\in\mathcal{Q}^c}\boldsymbol{\lambda}_{\boldsymbol{u}}^{\mathsf{T}}\{\mathbb{E}[(\boldsymbol{Y}_{\boldsymbol{u}}^{\mathsf{T}}\boldsymbol{Y}_{\mathcal{Q}}\boldsymbol{Y}_{\mathcal{Q}}^{\mathsf{T}}\boldsymbol{Y}_{\boldsymbol{v}})]/n^2\}\boldsymbol{\lambda}_{\boldsymbol{v}}$$

$$=\frac{B}{n}\sum_{\boldsymbol{k}\in\mathcal{Q}^c}\|\boldsymbol{\lambda}_{\boldsymbol{k}}\|_2^2.$$

Moreover

$$\mathbb{E}\Big[\Big\|\sum_{\boldsymbol{k}\in\mathcal{Q}^c}\boldsymbol{Y}_{\boldsymbol{k}}\boldsymbol{\lambda}_{\boldsymbol{k}}\Big\|_2^2/n\Big] = \sum_{\boldsymbol{k}\in\mathcal{Q}^c}\|\boldsymbol{\lambda}_{\boldsymbol{k}}\|_2^2 = \|\mathsf{P}_{\mathcal{Q}^c}f_d\|_{L^2}^2.$$

This gives

$$T_{23} = o_{d,\mathbb{P}}(1)\cdot\|\mathsf{P}_{\mathcal{Q}^c}f_d\|_{L^2}^2. \tag{47}$$

Using Cauchy Schwarz inequality for $T_{22}$, we get

$$T_{22} \le 2(T_{21}T_{23})^{1/2} = o_{d,\mathbb{P}}(1)\cdot\|\mathsf{P}_{\mathcal{Q}}f_d\|_{L^2}\|\mathsf{P}_{\mathcal{Q}^c}f_d\|_{L^2}. \tag{48}$$

As a result, combining Eqs. (46), (48) and (47), we have

$$T_2 = \|\mathsf{P}_{\mathcal{Q}}f_d\|_{L^2}^2 + o_{d,\mathbb{P}}(1)\cdot\|f_d\|_{L^2}^2. \tag{49}$$

**Step 4. Term $T_1$.** Note we have

$$T_1 = T_{11} + T_{12} + T_{13},$$

where

$$T_{11} =\boldsymbol{f}_{\mathcal{Q}}^{\mathsf{T}}(\boldsymbol{H} + \lambda\mathbf{I}_n)^{-1}\boldsymbol{E}_{\mathcal{Q}},$$
$$T_{12} =\boldsymbol{f}_{\mathcal{Q}^c}^{\mathsf{T}}(\boldsymbol{H} + \lambda\mathbf{I}_n)^{-1}\boldsymbol{E}_{\mathcal{Q}},$$
$$T_{13} =\boldsymbol{f}^{\mathsf{T}}(\boldsymbol{H} + \lambda\mathbf{I}_n)^{-1}\boldsymbol{E}_{\mathcal{Q}^c}.$$

By Lemma 7, we have

$$\|\boldsymbol{Y}_{\mathcal{Q}}^{\mathsf{T}}(\boldsymbol{H} + \lambda\mathbf{I}_n)^{-1}\boldsymbol{Y}_{\mathcal{Q}}\boldsymbol{D}_{\mathcal{Q}} - \mathbf{I}_B\|_{\mathrm{op}} = o_{d,\mathbb{P}}(1).$$

so that

$$T_{11} = \boldsymbol{\lambda}_{\mathcal{Q}}^{\mathsf{T}}\boldsymbol{Y}_{\mathcal{Q}}^{\mathsf{T}}(\boldsymbol{H} + \lambda\mathbf{I}_n)^{-1}\boldsymbol{Y}_{\mathcal{Q}}\boldsymbol{D}_{\mathcal{Q}}\boldsymbol{\lambda}_{\mathcal{Q}} = \|\boldsymbol{\lambda}_{\mathcal{Q}}\|_2^2(1 + o_{d,\mathbb{P}}(1)) = \|\mathsf{P}_{\mathcal{Q}}f_d\|_2^2(1 + o_{d,\mathbb{P}}(1)). \tag{50}$$

Using Cauchy Schwarz inequality for $T_{12}$, and by the expression of $\boldsymbol{M} = \boldsymbol{Y}_{\mathcal{Q}}\boldsymbol{D}_{\mathcal{Q}}^2\boldsymbol{Y}_{\mathcal{Q}}^{\mathsf{T}} + \kappa_u\boldsymbol{\Delta}_u$ with $\|\boldsymbol{\Delta}_u\|_{\mathrm{op}} = O_{d,\mathbb{P}}(1)$ and $\kappa_u = O_d(d^{-m(\lambda)})$, we get with high probability

$$|T_{12}| =\Big|\sum_{\boldsymbol{k}\in\mathcal{Q}^c}\boldsymbol{\lambda}_{\boldsymbol{k}}^{\mathsf{T}}\boldsymbol{Y}_{\boldsymbol{k}}^{\mathsf{T}}(\boldsymbol{H} + \lambda\mathbf{I}_n)^{-1}\boldsymbol{Y}_{\mathcal{Q}}\boldsymbol{D}_{\mathcal{Q}}\boldsymbol{\lambda}_{\mathcal{Q}}\Big|$$

$$\le\Big\|\sum_{\boldsymbol{k}\in\mathcal{Q}^c}\boldsymbol{\lambda}_{\boldsymbol{k}}^{\mathsf{T}}\boldsymbol{Y}_{\boldsymbol{k}}^{\mathsf{T}}(\boldsymbol{H} + \lambda\mathbf{I}_n)^{-1}\boldsymbol{Y}_{\mathcal{Q}}\boldsymbol{D}_{\mathcal{Q}}\Big\|_2\|\boldsymbol{\lambda}_{\mathcal{Q}}\|_2$$

$$=\Big[\Big(\sum_{\boldsymbol{k}\in\mathcal{Q}^c}\boldsymbol{\lambda}_{\boldsymbol{k}}^{\mathsf{T}}\boldsymbol{Y}_{\boldsymbol{k}}^{\mathsf{T}}\Big)(\boldsymbol{H} + \lambda\mathbf{I}_n)^{-1}\boldsymbol{Y}_{\mathcal{Q}}\boldsymbol{D}_{\mathcal{Q}}^2\boldsymbol{Y}_{\mathcal{Q}}^{\mathsf{T}}(\boldsymbol{H} + \lambda\mathbf{I}_n)^{-1}\Big(\sum_{\boldsymbol{k}\in\mathcal{Q}^c}\boldsymbol{\lambda}_{\boldsymbol{k}}^{\mathsf{T}}\boldsymbol{Y}_{\boldsymbol{k}}^{\mathsf{T}}\Big)\Big]^{1/2}\|\boldsymbol{\lambda}_{\mathcal{Q}}\|_2 \tag{51}$$

$$\le\Big[\Big(\sum_{\boldsymbol{k}\in\mathcal{Q}^c}\boldsymbol{\lambda}_{\boldsymbol{k}}^{\mathsf{T}}\boldsymbol{Y}_{\boldsymbol{k}}^{\mathsf{T}}\Big)(\boldsymbol{H} + \lambda\mathbf{I}_n)^{-1}\boldsymbol{M}(\boldsymbol{H} + \lambda\mathbf{I}_n)^{-1}\Big(\sum_{\boldsymbol{k}\in\mathcal{Q}^c}\boldsymbol{\lambda}_{\boldsymbol{k}}^{\mathsf{T}}\boldsymbol{Y}_{\boldsymbol{k}}^{\mathsf{T}}\Big)\Big]^{1/2}\|\boldsymbol{\lambda}_{\mathcal{Q}}\|_2$$

$$=T_{23}^{1/2}\|\boldsymbol{\lambda}_{\mathcal{Q}}\|_2 = o_{d,\mathbb{P}}(1)\cdot\|\mathsf{P}_{\mathcal{Q}}f_d\|_{L^2}\|\mathsf{P}_{\mathcal{Q}^c}f_d\|_{L^2}.$$

For term $T_{13}$, we have

$$|T_{13}| =|\boldsymbol{f}^{\mathsf{T}}(\boldsymbol{H} + \lambda\mathbf{I}_n)^{-1}\boldsymbol{E}_{\mathcal{Q}^c}| \le \|\boldsymbol{f}\|_2\|(\boldsymbol{H} + \lambda\mathbf{I}_n)^{-1}\|_{\mathrm{op}}\|\boldsymbol{E}_{\mathcal{Q}^c}\|_2.$$

Note we have $\mathbb{E}[\|\boldsymbol{f}\|_2^2] = n\|f_d\|_{L^2}^2$, and $\|(\boldsymbol{H} + \lambda\mathbf{I}_n)^{-1}\|_{\mathrm{op}} \le 2/(\kappa_h + \lambda)$ with high probability, and

$$\mathbb{E}[\|\boldsymbol{E}_{\mathcal{Q}^c}\|_2^2] = n\sum_{\boldsymbol{k}\in\mathcal{Q}^c}\lambda_{\boldsymbol{k}}^{\boldsymbol{d}}(h_{\boldsymbol{d}})^2\|\mathsf{P}_{\boldsymbol{k}}f_d\|_{L^2}^2 \le n\Big[\max_{\boldsymbol{k}\in\mathcal{Q}^c}\lambda_{\boldsymbol{k}}^{\boldsymbol{d}}(h_{\boldsymbol{d}})^2\Big]\|\mathsf{P}_{\mathcal{Q}^c}f_d\|_{L^2}^2.$$

As a result, we have

$$
\begin{aligned}
|T_{13}| \leq & O_d(1) \cdot \|\mathsf{P}_{\mathcal{Q}^c} f_d\|_{L^2} \|f_d\|_{L^2} \Big[ n^2 \max_{\boldsymbol{k} \in \mathcal{Q}^c} \lambda_{\boldsymbol{k}}^{\boldsymbol{d}}(h_{\boldsymbol{d}})^2 \Big]^{1/2} / (\kappa_h + \lambda) \\
= & o_{d,\mathbb{P}}(1) \cdot \|\mathsf{P}_{\mathcal{Q}^c} f_d\|_{L^2} \|f_d\|_{L^2},
\end{aligned}
\tag{52}
$$

where the last equality used the fact that $n \leq O_d(d^{m(\gamma)-\delta})$ and Lemma 2. Combining Eqs. (50), (51) and (52), we get

$$
T_1 = \|\mathsf{P}_{\mathcal{Q}} f_d\|_{L^2}^2 + o_{d,\mathbb{P}}(1) \cdot \|f_d\|_{L^2}^2.
\tag{53}
$$

**Step 5. Terms $T_3, T_4$ and $T_5$.** By Lemma 6 again, we have

$$
\mathbb{E}_{\boldsymbol{\varepsilon}}[T_3]/\tau^2 = \mathrm{tr}((\boldsymbol{H} + \lambda \mathbf{I}_n)^{-1} \boldsymbol{M} (\boldsymbol{H} + \lambda \mathbf{I}_n)^{-1}) = \mathrm{tr}(\boldsymbol{Y}_{\mathcal{Q}} \boldsymbol{Y}_{\mathcal{Q}}^{\mathsf{T}}/n^2) + o_{d,\mathbb{P}}(1),
$$

By Lemma 3, we have

$$
\mathrm{tr}(\boldsymbol{Y}_{\mathcal{Q}} \boldsymbol{Y}_{\mathcal{Q}}^{\mathsf{T}}/n^2) = \mathrm{tr}(\boldsymbol{Y}_{\mathcal{Q}}^{\mathsf{T}} \boldsymbol{Y}_{\mathcal{Q}})/n^2 = nB/n^2 + o_{d,\mathbb{P}}(1) = o_{d,\mathbb{P}}(1).
$$

This gives

$$
T_3 = o_{d,\mathbb{P}}(1) \cdot \tau^2.
\tag{54}
$$

Let us consider $T_4$ term:

$$
\begin{aligned}
\mathbb{E}_{\boldsymbol{\varepsilon}}[T_4^2]/\tau^2 = & \mathbb{E}_{\boldsymbol{\varepsilon}}[\boldsymbol{\varepsilon}^{\mathsf{T}} (\boldsymbol{H} + \lambda \mathbf{I}_n)^{-1} \boldsymbol{E} \boldsymbol{E}^{\mathsf{T}} (\boldsymbol{H} + \lambda \mathbf{I}_n)^{-1} \boldsymbol{\varepsilon}]/\tau^2 \\
= & \boldsymbol{E}^{\mathsf{T}} (\boldsymbol{H} + \lambda \mathbf{I}_n)^{-2} \boldsymbol{E}.
\end{aligned}
$$

For any integer $L$, denote $\mathcal{L} \equiv [0, L]^Q \cap \mathbb{Z}_{\geq 0}^Q$, and $\boldsymbol{Y}_{\mathcal{L}} = (\boldsymbol{Y}_{\boldsymbol{k}})_{\boldsymbol{k} \in \mathcal{L}}$ and $\boldsymbol{D}_{\mathcal{L}} = (\boldsymbol{D}_{\boldsymbol{k}})_{\boldsymbol{k} \in \mathcal{L}}$. Then notice that by Lemma 3, Lemma 6 and the definition of $\boldsymbol{M}$, we get

$$
\begin{aligned}
\|\boldsymbol{D}_{\mathcal{L}} \boldsymbol{Y}_{\mathcal{L}}^{\mathsf{T}} (\boldsymbol{H} + \lambda \mathbf{I}_n)^{-2} \boldsymbol{Y}_{\mathcal{L}} \boldsymbol{D}_{\mathcal{L}}\|_{\mathrm{op}} = & \|(\boldsymbol{H} + \lambda \mathbf{I}_n)^{-1} \boldsymbol{Y}_{\mathcal{L}} \boldsymbol{D}_{\mathcal{L}}^2 \boldsymbol{Y}_{\mathcal{L}}^{\mathsf{T}} (\boldsymbol{H} + \lambda \mathbf{I}_n)^{-1}\|_{\mathrm{op}} \\
\leq & \|(\boldsymbol{H} + \lambda \mathbf{I}_n)^{-1} \boldsymbol{M} (\boldsymbol{H} + \lambda \mathbf{I}_n)^{-1}\|_{\mathrm{op}}. \\
\leq & \|\boldsymbol{Y}_{\mathcal{Q}} \boldsymbol{Y}_{\mathcal{Q}}^{\mathsf{T}}/n\|_{\mathrm{op}}/n + o_{\mathbb{P},d}(1) \cdot /n \\
= & o_{d,\mathbb{P}}(1)
\end{aligned}
$$

Therefore,

$$
\begin{aligned}
\boldsymbol{E}^{\mathsf{T}} (\boldsymbol{H} + \lambda \mathbf{I}_n)^{-2} \boldsymbol{E} = & \lim_{L \to \infty} \boldsymbol{E}_{\mathcal{L}}^{\mathsf{T}} (\boldsymbol{H} + \lambda \mathbf{I}_n)^{-2} \boldsymbol{E}_{\mathcal{L}} \\
= & \lim_{L \to \infty} \boldsymbol{\lambda}_{\mathcal{L}}^{\mathsf{T}} [\boldsymbol{D}_{\mathcal{L}} \boldsymbol{Y}_{\mathcal{L}}^{\mathsf{T}} (\boldsymbol{H} + \lambda \mathbf{I}_n)^{-2} \boldsymbol{Y}_{\mathcal{L}} \boldsymbol{D}_{\mathcal{L}}] \boldsymbol{\lambda}_{\mathcal{L}} \\
\leq & \|(\boldsymbol{H} + \lambda \mathbf{I}_n)^{-1} \boldsymbol{M} (\boldsymbol{H} + \lambda \mathbf{I}_n)^{-1}\|_{\mathrm{op}} \cdot \lim_{L \to \infty} \|\boldsymbol{\lambda}_{\mathcal{L}}\|_2^2 \\
\leq & o_{d,\mathbb{P}}(1) \cdot \|f_d\|_{L^2}^2,
\end{aligned}
$$

which gives

$$
T_4 = o_{d,\mathbb{P}}(1) \cdot \tau \|f_d\|_{L^2} = o_{d,\mathbb{P}}(1) \cdot (\tau^2 + \|f_d\|_{L^2}^2).
\tag{55}
$$

We decompose $T_5$ using $\boldsymbol{f} = \boldsymbol{f}_{\mathcal{Q}} + \boldsymbol{f}_{\mathcal{Q}^c}$,

$$
T_5 = T_{51} + T_{52},
$$

where

$$
\begin{aligned}
T_{51} = & \boldsymbol{\varepsilon}^{\mathsf{T}} (\boldsymbol{H} + \lambda \mathbf{I}_n)^{-1} \boldsymbol{M} (\boldsymbol{H} + \lambda \mathbf{I}_n)^{-1} \boldsymbol{f}_{\mathcal{Q}}, \\
T_{52} = & \boldsymbol{\varepsilon}^{\mathsf{T}} (\boldsymbol{H} + \lambda \mathbf{I}_n)^{-1} \boldsymbol{M} (\boldsymbol{H} + \lambda \mathbf{I}_n)^{-1} \boldsymbol{f}_{\mathcal{Q}^c}.
\end{aligned}
$$

First notice that

$$
\|\boldsymbol{M}^{1/2} (\boldsymbol{H} + \lambda \mathbf{I}_n)^{-2} \boldsymbol{M}^{1/2}\|_{\mathrm{op}} = \|(\boldsymbol{H} + \lambda \mathbf{I}_n)^{-1} \boldsymbol{M} (\boldsymbol{H} + \lambda \mathbf{I}_n)^{-1}\|_{\mathrm{op}} = o_{d,\mathbb{P}}(1).
$$

Then by Lemma 6, we get

$$
\begin{aligned}
\mathbb{E}_{\boldsymbol{\varepsilon}}[T_{51}^2]/\tau^2 = & \mathbb{E}_{\boldsymbol{\varepsilon}}[\boldsymbol{\varepsilon}^{\mathsf{T}} (\boldsymbol{H} + \lambda \mathbf{I}_n)^{-1} \boldsymbol{M} (\boldsymbol{H} + \lambda \mathbf{I}_n)^{-1} \boldsymbol{f}_{\mathcal{Q}} \boldsymbol{f}_{\mathcal{Q}}^{\mathsf{T}} (\boldsymbol{H} + \lambda \mathbf{I}_n)^{-1} \boldsymbol{M} (\boldsymbol{H} + \lambda \mathbf{I}_n)^{-1} \boldsymbol{\varepsilon}]/\tau^2 \\
= & \boldsymbol{f}_{\mathcal{Q}}^{\mathsf{T}} [(\boldsymbol{H} + \lambda \mathbf{I}_n)^{-1} \boldsymbol{M} (\boldsymbol{H} + \lambda \mathbf{I}_n)^{-1}]^2 \boldsymbol{f}_{\mathcal{Q}} \\
\leq & \|\boldsymbol{M}^{1/2} (\boldsymbol{H} + \lambda \mathbf{I}_n)^{-2} \boldsymbol{M}^{1/2}\|_{\mathrm{op}} \|\boldsymbol{M}^{1/2} (\boldsymbol{H} + \lambda \mathbf{I}_n)^{-1} \boldsymbol{f}_{\mathcal{Q}}\|_2^2 \\
= & o_{d,\mathbb{P}}(1) \cdot T_{21} \\
= & o_{d,\mathbb{P}}(1) \cdot \|\mathsf{P}_{\mathcal{Q}} f_d\|_{L^2}^2.
\end{aligned}
$$

Similarly, we get

$$\mathbb{E}_{\boldsymbol{\varepsilon}}[T_{52}^2]/\tau^2 = o_{d,\mathbb{P}}(1) \cdot T_{23} = \quad o_{d,\mathbb{P}}(1) \cdot \|\mathsf{P}_{\mathcal{Q}^c} f_d\|_{L^2}^2.$$

By Markov's inequality, we deduce that

$$T_5 = o_{d,\mathbb{P}}(1) \cdot \tau(\|\mathsf{P}_{\mathcal{Q}} f_d\|_{L^2} + \|\mathsf{P}_{\mathcal{Q}^c} f_d\|_{L^2}) = o_{d,\mathbb{P}}(1) \cdot (\tau^2 + \|f_d\|_{L^2}^2). \tag{56}$$

**Step 6. Finish the proof.**

Combining Eqs. (53), (49), (54), (55) and (56), we have

$$\begin{aligned}
R_{\mathrm{KRR}}(f_d, \boldsymbol{X}, \lambda) =& \|f_d\|_{L^2}^2 - 2T_1 + T_2 + T_3 - 2T_4 + 2T_5 \\
=& \|\mathsf{P}_{\mathcal{Q}^c} f_d\|_{L^2}^2 + o_{d,\mathbb{P}}(1) \cdot (\|f_d\|_{L^2}^2 + \tau^2),
\end{aligned}$$

which concludes the proof.

### D.3 Auxiliary results

**Lemma 3.** *Let $\{Y_{\boldsymbol{k},\boldsymbol{s}}^{\boldsymbol{d}}\}_{\boldsymbol{k}\in\mathbb{Z}_{\geq 0}^Q, \boldsymbol{s}\in[B(\boldsymbol{d},\boldsymbol{k})]}$ be the collection of tensor product of spherical harmonics on* $\mathrm{PS}^{\boldsymbol{d}}$. *Let $(\overline{\boldsymbol{x}}_i)_{i\in[n]} \sim_{iid} \mathrm{Unif}(\mathrm{PS}^{\boldsymbol{d}})$. Denote*

$$\boldsymbol{Y}_{\boldsymbol{k}} = (Y_{\boldsymbol{k},\boldsymbol{s}}^{\boldsymbol{d}}(\overline{\boldsymbol{x}}_i))_{i\in[n],\boldsymbol{s}\in[B(\boldsymbol{d},\boldsymbol{k})]} \in \mathbb{R}^{n\times B(\boldsymbol{d},\boldsymbol{k})}.$$

*Assume that $n \geq w_d(d^\gamma \log d)$ and consider*

$$\mathcal{R} = \Big\{ \boldsymbol{k} \in \mathbb{Z}_{\geq 0}^Q \Big| \sum_{q\in[Q]} \eta_q k_q < m(\gamma) \Big\}.$$

*Denote $A = \sum_{\boldsymbol{k}\in\mathcal{R}} B(\boldsymbol{d},\boldsymbol{k})$ and*

$$\boldsymbol{Y}_{\mathcal{R}} = (\boldsymbol{Y}_{\boldsymbol{k}})_{\boldsymbol{k}\in\mathcal{R}} \in \mathbb{R}^{n\times A}.$$

*Then we have*

$$\boldsymbol{Y}_{\mathcal{R}}^{\mathsf{T}}\boldsymbol{Y}_{\mathcal{R}}/n = \mathbf{I}_A + \boldsymbol{\Delta},$$

*with $\boldsymbol{\Delta} \in \mathbb{R}^{A\times A}$ and $\mathbb{E}[\|\boldsymbol{\Delta}\|_{\mathrm{op}}] = o_d(1)$.*

*Proof of Lemma 3.* Let $\boldsymbol{\Psi} = \boldsymbol{Y}_{\mathcal{R}}^{\mathsf{T}}\boldsymbol{Y}_{\mathcal{R}}/n \in \mathbb{R}^{A\times A}$. We can rewrite $\boldsymbol{\Psi}$ as

$$\boldsymbol{\Psi} = \frac{1}{n}\sum_{i=1}^n \boldsymbol{h}_i \boldsymbol{h}_i^{\mathsf{T}},$$

where $\boldsymbol{h}_i = (Y_{\boldsymbol{k},\boldsymbol{s}}^{\boldsymbol{d}}(\overline{\boldsymbol{x}}_i))_{\boldsymbol{k}\in\mathcal{R},\boldsymbol{s}\in[B(\boldsymbol{d},\boldsymbol{k})]} \in \mathbb{R}^A$. We use matrix Bernstein inequality. Denote $\boldsymbol{X}_i = \boldsymbol{h}_i\boldsymbol{h}_i - \mathbf{I}_A \in \mathbb{R}^{A\times A}$. Then we have $\mathbb{E}[\boldsymbol{X}_i] = \boldsymbol{0}$, and

$$\begin{aligned}
\|\boldsymbol{X}_i\|_{\mathrm{op}} \leq \|\boldsymbol{h}_i\|_2^2 + 1 &= \sum_{\boldsymbol{k}\in\mathcal{R}}\sum_{\boldsymbol{s}\in[B(\boldsymbol{d},\boldsymbol{k})]} Y_{\boldsymbol{k},\boldsymbol{s}}^{\boldsymbol{d}}(\overline{\boldsymbol{x}}_i)^2 + 1 \\
&= \sum_{\boldsymbol{k}\in\mathcal{R}} B(\boldsymbol{d},\boldsymbol{k}) Q_{\boldsymbol{k}}^{\boldsymbol{d}}\left(\{\langle \overline{\boldsymbol{x}}_i^{(q)}, \overline{\boldsymbol{x}}_i^{(q)}\rangle\}_{q\in[Q]}\right) + 1 = A + 1,
\end{aligned}$$

where we use formula (12) and the normalization $Q_{\boldsymbol{k}}^{\boldsymbol{d}}(d_1,\dots,d_Q) = 1$. Denote $V = \|\sum_{i=1}^n \mathbb{E}[\boldsymbol{X}_i^2]\|_{\mathrm{op}}$. Then we have

$$V = n\|\mathbb{E}[(\boldsymbol{h}_i\boldsymbol{h}_i^{\mathsf{T}} - \mathbf{I}_A)^2]\|_{\mathrm{op}} = n\|\mathbb{E}[\boldsymbol{h}_i\boldsymbol{h}_i^{\mathsf{T}}\boldsymbol{h}_i\boldsymbol{h}_i^{\mathsf{T}} - 2\boldsymbol{h}_i\boldsymbol{h}_i^{\mathsf{T}} + \mathbf{I}_A]\|_{\mathrm{op}} = n\|(A-1)\mathbf{I}_A\|_{\mathrm{op}} = n(A-1),$$

where we used $\boldsymbol{h}_i^{\mathsf{T}}\boldsymbol{h}_i = \|\boldsymbol{h}_i\|_2^2 = A$ and $\mathbb{E}[\boldsymbol{h}_i(\overline{\boldsymbol{x}}_i)\boldsymbol{h}_i^{\mathsf{T}}(\overline{\boldsymbol{x}}_i)] = (\mathbb{E}[Y_{\boldsymbol{k},\boldsymbol{s}}^{\boldsymbol{d}}(\overline{\boldsymbol{x}}_i)Y_{\boldsymbol{k}',\boldsymbol{s}'}^{\boldsymbol{d}}(\overline{\boldsymbol{x}}_i)])_{\boldsymbol{k}\boldsymbol{s},\boldsymbol{k}'\boldsymbol{s}'} = \mathbf{I}_A$. As a result, we have for any $t > 0$,

$$\begin{aligned}
\mathbb{P}(\|\boldsymbol{\Psi} - \mathbf{I}_A\|_{\mathrm{op}} \geq t) \leq& A\exp\{-n^2t^2/[2n(A-1) + 2(A+1)nt/3]\} \\
\leq& \exp\{-(n/A)t^2/[10(1+t)] + \log A\}.
\end{aligned} \tag{57}$$

Notice that there exists $C > 0$ such that $A \leq C\max_{\boldsymbol{k}\in\mathcal{R}}\prod_{q\in[Q]} d^{\eta_q k_q} \leq Cd^\gamma$ (by definition of $m(\gamma)$ and $\mathcal{R}$) and therefore $n \geq w_d(A\log A)$. Integrating the tail bound (57) proves the lemma. $\square$

**Lemma 4.** *Let $\sigma$ be an activation function satisfying Assumption 1. Let $w_d(d^\gamma \log d) \leq n \leq O_d(d^{m(\gamma)-\delta})$ for some $\gamma > 0$ and $\delta > 0$. Then there exists sequences $\kappa_h$ and $\kappa_u$ such that*

$$H = \sum_{\boldsymbol{k} \in \mathbb{Z}_{\geq 0}^Q} \boldsymbol{Y_k D_k Y_k^\mathsf{T}} = \boldsymbol{Y_Q D_Q Y_Q^\mathsf{T}} + \kappa_h(\mathbf{I}_n + \boldsymbol{\Delta}_h), \tag{58}$$

$$M = \sum_{\boldsymbol{k} \in \mathbb{Z}_{\geq 0}^Q} \boldsymbol{Y_k D_k^2 Y_k^\mathsf{T}} = \boldsymbol{Y_Q D_Q^2 Y_Q^\mathsf{T}} + \kappa_m \boldsymbol{\Delta}_m, \tag{59}$$

*where $\kappa_h = \Theta_d(1)$, $\kappa_m = O_d(d^{-m(\gamma)})$, $\|\boldsymbol{\Delta}_h\|_{\mathrm{op}} = o_{d,\mathbb{P}}(1)$ and $\|\boldsymbol{\Delta}_m\|_{\mathrm{op}} = O_{d,\mathbb{P}}(1)$.*

*Proof of Lemma 4.* Define

$$\mathcal{R} = \Big\{ \boldsymbol{k} \in \mathbb{Z}_{\geq 0}^Q \Big| \sum_{q \in [Q]} \eta_q k_q < m(\gamma) \Big\},$$

$$\mathcal{S} = \Big\{ \boldsymbol{k} \in \mathbb{Z}_{\geq 0}^Q \Big| \sum_{q \in [Q]} \eta_q k_q \geq m(\gamma) \Big\},$$

such that $\mathcal{R} \cup \mathcal{S} = \mathbb{Z}_{\geq 0}^Q$. The proof comes from bounding the eigenvalues of the matrix $\boldsymbol{Y_k Y_k^\mathsf{T}}$ for $\boldsymbol{k} \in \mathcal{R}$ and $\boldsymbol{k} \in \mathcal{S}$ separately. From Corollary 1, we have

$$\sup_{\boldsymbol{k} \in \mathcal{S}} \|\boldsymbol{Y_k Y_k^\mathsf{T}}/B(\boldsymbol{d}, \boldsymbol{k}) - \mathbf{I}_n\|_{\mathrm{op}} = o_{d,\mathbb{P}}(1).$$

Hence, we can write

$$\sum_{\boldsymbol{k} \in \mathcal{S}} \boldsymbol{Y_k D_k Y_k^\mathsf{T}} = \kappa_h(\mathbf{I}_n + \boldsymbol{\Delta}_{h,1}), \tag{60}$$

with $\kappa_h = \sum_{\boldsymbol{k} \in \mathcal{S}} \lambda_{\boldsymbol{k}}^{\boldsymbol{d}}(h_{\boldsymbol{d}}) B(\boldsymbol{d}, \boldsymbol{k}) = O_d(1)$. From Assumption 1.(b) and a proof similar to Lemma 20, there exists $\boldsymbol{k} = (0, \ldots, k, \ldots, 0)$ (for $k > L$ at position $q_\xi$) such that $\liminf_{d \to \infty} \lambda_{\boldsymbol{k}}^{\boldsymbol{d}}(h_{\boldsymbol{d}}) B(\boldsymbol{d}, \boldsymbol{k}) > 0$. Hence, $\kappa_h = \Theta_d(1)$.

From Lemma 3 we have for $\boldsymbol{k} \in \mathcal{R} \cap \mathcal{Q}^c$,

$$\boldsymbol{Y_k^\mathsf{T} Y_k}/n = \mathbf{I}_{B(\boldsymbol{d}, \boldsymbol{k})} + \boldsymbol{\Delta},$$

with $\|\boldsymbol{\Delta}\|_{\mathrm{op}} = o_{d,\mathbb{P}}(1)$. We deduce that $\|\boldsymbol{Y_k Y_k^\mathsf{T}}\|_{\mathrm{op}} = O_{d,\mathbb{P}}(n)$. Hence,

$$\|\boldsymbol{Y_k D_k Y_k^\mathsf{T}}\|_{\mathrm{op}} = O_{d,\mathbb{P}}(n \lambda_{\boldsymbol{k}}^{\boldsymbol{d}}(h_{\boldsymbol{d}})) = o_{d,\mathbb{P}}(1),$$

where we used Lemma 2. We deduce that

$$\sum_{\boldsymbol{k} \in \mathcal{R} \cap \mathcal{Q}^c} \boldsymbol{Y_k D_k Y_k^\mathsf{T}} = \kappa_h \boldsymbol{\Delta}_{h,2}, \tag{61}$$

with $\|\boldsymbol{\Delta}_{h,2}\|_{\mathrm{op}} = o_{d,\mathbb{P}}(1)$ where we used $\kappa_h^{-1} = O_d(1)$. Combining Eqs. (60) and (61) yields Eq. (58).

Similarly, we get

$$\sum_{\boldsymbol{k} \in \mathcal{Q}^c} \boldsymbol{Y_k D_k^2 Y_k^\mathsf{T}} = \sum_{\boldsymbol{k} \in \mathcal{R} \cap \mathcal{Q}^c} [\lambda_{\boldsymbol{k}}^{\boldsymbol{d}}(h_{\boldsymbol{d}})^2 n] \boldsymbol{Y_k Y_k^\mathsf{T}}/n + \sum_{\boldsymbol{k} \in \mathcal{S}} [\lambda_{\boldsymbol{k}}^{\boldsymbol{d}}(h_{\boldsymbol{d}})^2 B(\boldsymbol{d}, \boldsymbol{k})] \boldsymbol{Y_k Y_k^\mathsf{T}}/B(\boldsymbol{d}, \boldsymbol{k}).$$

Using Lemma 2, we have $\lambda_{\boldsymbol{k}}^{\boldsymbol{d}}(h_{\boldsymbol{d}})^2 n \leq C d^{-2m(\gamma)} n = O_{d,\mathbb{P}}(d^{-m(\gamma)})$ and $\lambda_{\boldsymbol{k}}^{\boldsymbol{d}}(h_{\boldsymbol{d}})^2 B(\boldsymbol{d}, \boldsymbol{k}) \leq C \lambda_{\boldsymbol{k}}^{\boldsymbol{d}}(h_{\boldsymbol{d}}) \leq C' d^{-m(\gamma)}$. Hence Eq. (59) is verified with

$$\kappa_m = \sum_{\boldsymbol{k} \in \mathcal{R} \cap \mathcal{Q}^c} \lambda_{\boldsymbol{k}}^{\boldsymbol{d}}(h_{\boldsymbol{d}})^2 n + \sum_{\boldsymbol{k} \in \mathcal{S}} \lambda_{\boldsymbol{k}}^{\boldsymbol{d}}(h_{\boldsymbol{d}})^2 B(\boldsymbol{d}, \boldsymbol{k}).$$

$\square$

**Lemma 5.** *Let* $\{Y_{\boldsymbol{k},\boldsymbol{s}}^{\boldsymbol{d}}\}_{\boldsymbol{k}\in\mathbb{Z}_{\geq 0}^{Q},\boldsymbol{s}\in[B(\boldsymbol{d},\boldsymbol{k})]}$ *be the collection of product of spherical harmonics on* $L^2(\mathrm{PS}^{\boldsymbol{d}},\mu_{\boldsymbol{d}})$. *Let* $(\overline{\boldsymbol{x}}_i)_{i\in[n]}\sim_{iid}\mathrm{Unif}(\mathrm{PS}^{\boldsymbol{d}})$. *Denote*

$$\boldsymbol{Y}_{\boldsymbol{k}}=(Y_{\boldsymbol{k},\boldsymbol{s}}^{\boldsymbol{d}}(\overline{\boldsymbol{x}}_i))_{i\in[n],\boldsymbol{s}\in[B(\boldsymbol{d},\boldsymbol{k})]}\in\mathbb{R}^{n\times B(\boldsymbol{d},\boldsymbol{k})}.$$

*Then for* $\boldsymbol{u},\boldsymbol{v},\boldsymbol{t}\in\mathbb{Z}_{\geq 0}^{Q}$ *and* $\boldsymbol{u}\neq\boldsymbol{v}$, *we have*

$$\mathbb{E}[\boldsymbol{Y}_{\boldsymbol{u}}^{\mathsf{T}}\boldsymbol{Y}_{\boldsymbol{t}}\boldsymbol{Y}_{\boldsymbol{t}}^{\mathsf{T}}\boldsymbol{Y}_{\boldsymbol{v}}]=\boldsymbol{0}.$$

*For* $\boldsymbol{u},\boldsymbol{t}\in\mathbb{Z}_{\geq 0}^{Q}$, *we have*

$$\mathbb{E}[\boldsymbol{Y}_{\boldsymbol{u}}^{\mathsf{T}}\boldsymbol{Y}_{\boldsymbol{t}}\boldsymbol{Y}_{\boldsymbol{t}}^{\mathsf{T}}\boldsymbol{Y}_{\boldsymbol{u}}]=[B(\boldsymbol{d},\boldsymbol{t})n+n(n-1)\delta_{\boldsymbol{u},\boldsymbol{t}}]\mathbf{I}_{B(\boldsymbol{d},\boldsymbol{u})}.$$

*Proof.* We have

$$
\begin{aligned}
&\mathbb{E}[\boldsymbol{Y}_{\boldsymbol{u}}^{\mathsf{T}}\boldsymbol{Y}_{\boldsymbol{t}}\boldsymbol{Y}_{\boldsymbol{t}}^{\mathsf{T}}\boldsymbol{Y}_{\boldsymbol{v}}]\\
&=\sum_{i,j\in[n]}\sum_{\boldsymbol{m}\in[B(\boldsymbol{d},\boldsymbol{t})]}(\mathbb{E}[Y_{\boldsymbol{u},\boldsymbol{p}}^{\boldsymbol{d}}(\overline{\boldsymbol{x}}_i)\big(Y_{\boldsymbol{t},\boldsymbol{m}}^{\boldsymbol{d}}(\overline{\boldsymbol{x}}_i)Y_{\boldsymbol{t},\boldsymbol{m}}^{\boldsymbol{d}}(\overline{\boldsymbol{x}}_j)\big)Y_{\boldsymbol{v},\boldsymbol{q}}^{\boldsymbol{d}}(\overline{\boldsymbol{x}}_j)])_{\boldsymbol{p}\in[B(\boldsymbol{d},\boldsymbol{u})],\boldsymbol{q}\in[B(\boldsymbol{d},\boldsymbol{v})]}\\
&=\sum_{i\in[n]}\Big(\mathbb{E}\Big[Y_{\boldsymbol{u},\boldsymbol{p}}^{\boldsymbol{d}}(\overline{\boldsymbol{x}}_i)\Big(\sum_{\boldsymbol{m}\in[B(\boldsymbol{d},\boldsymbol{t})]}Y_{\boldsymbol{t},\boldsymbol{m}}^{\boldsymbol{d}}(\overline{\boldsymbol{x}}_i)Y_{\boldsymbol{t},\boldsymbol{m}}^{\boldsymbol{d}}(\overline{\boldsymbol{x}}_i)\Big)Y_{\boldsymbol{v},\boldsymbol{q}}^{\boldsymbol{d}}(\overline{\boldsymbol{x}}_i)\Big]\Big)_{\boldsymbol{p}\in[B(\boldsymbol{d},\boldsymbol{u})],\boldsymbol{q}\in[B(\boldsymbol{d},\boldsymbol{v})]}\\
&\quad+\sum_{i\neq j\in[n]}\sum_{\boldsymbol{m}\in[B(\boldsymbol{d},\boldsymbol{t})]}(\mathbb{E}[Y_{\boldsymbol{u},\boldsymbol{p}}^{\boldsymbol{d}}(\overline{\boldsymbol{x}}_i)Y_{\boldsymbol{t},\boldsymbol{m}}^{\boldsymbol{d}}(\overline{\boldsymbol{x}}_i)Y_{\boldsymbol{t},\boldsymbol{m}}^{\boldsymbol{d}}(\overline{\boldsymbol{x}}_j)Y_{\boldsymbol{v},\boldsymbol{q}}^{\boldsymbol{d}}(\overline{\boldsymbol{x}}_j)])_{\boldsymbol{p}\in[B(\boldsymbol{d},\boldsymbol{u})],\boldsymbol{q}\in[B(\boldsymbol{d},\boldsymbol{v})]}\qquad(62)\\
&=B(\boldsymbol{d},\boldsymbol{t})\sum_{i\in[n]}(\mathbb{E}[Y_{\boldsymbol{u},\boldsymbol{p}}^{\boldsymbol{d}}(\overline{\boldsymbol{x}}_i)Y_{\boldsymbol{v},\boldsymbol{q}}^{\boldsymbol{d}}(\overline{\boldsymbol{x}}_i)])_{\boldsymbol{p}\in[B(\boldsymbol{d},\boldsymbol{u})],\boldsymbol{q}\in[B(\boldsymbol{d},\boldsymbol{v})]}\\
&\quad+\sum_{i\neq j\in[n]}\sum_{\boldsymbol{m}\in[B(\boldsymbol{d},\boldsymbol{t})]}(\delta_{\boldsymbol{u},\boldsymbol{t}}\delta_{\boldsymbol{p},\boldsymbol{m}}\delta_{\boldsymbol{t},\boldsymbol{v}}\delta_{\boldsymbol{q},\boldsymbol{m}})_{\boldsymbol{p}\in[B(\boldsymbol{d},\boldsymbol{u})],\boldsymbol{q}\in[B(\boldsymbol{d},\boldsymbol{v})]}\\
&=(B(\boldsymbol{d},\boldsymbol{t})n\delta_{\boldsymbol{u},\boldsymbol{v}}\delta_{\boldsymbol{p},\boldsymbol{q}}+n(n-1)\delta_{\boldsymbol{u},\boldsymbol{t}}\delta_{\boldsymbol{t},\boldsymbol{v}}\delta_{\boldsymbol{p},\boldsymbol{q}})_{\boldsymbol{p}\in[B(\boldsymbol{d},\boldsymbol{u})],\boldsymbol{q}\in[B(\boldsymbol{d},\boldsymbol{v})]}.
\end{aligned}
$$

This proves the lemma. $\qquad\square$

**Lemma 6.** *Let* $\sigma$ *be an activation function satisfying Assumption 1. Assume* $\omega_d(d^{\gamma}\log d)\leq n\leq O_d(d^{m(\gamma)-\delta})$ *for some* $\gamma>0$ *and* $\delta>0$. *We have*

$$\|n(\boldsymbol{H}+\lambda\mathbf{I}_n)^{-1}\boldsymbol{M}(\boldsymbol{H}+\lambda\mathbf{I}_n)^{-1}-\boldsymbol{Y}_{\mathcal{Q}}\boldsymbol{Y}_{\mathcal{Q}}^{\mathsf{T}}/n\|_{\mathrm{op}}=o_{d,\mathbb{P}}(1).$$

*Proof of Lemma 6.* Denote

$$\boldsymbol{Y}_{\boldsymbol{k}}=(Y_{\boldsymbol{k},\boldsymbol{s}}^{\boldsymbol{d}}(\overline{\boldsymbol{x}}_i))_{i\in[n],\boldsymbol{s}\in[B(\boldsymbol{d},\boldsymbol{k})]}\in\mathbb{R}^{n\times B(\boldsymbol{d},\boldsymbol{k})}.\qquad(63)$$

Denote $B=\sum_{\boldsymbol{k}\in\mathcal{Q}}B(\boldsymbol{d},\boldsymbol{k})$, and

$$\boldsymbol{Y}_{\mathcal{Q}}=(\boldsymbol{Y}_{\boldsymbol{k}})_{\boldsymbol{k}\in\mathcal{Q}}\in\mathbb{R}^{n\times B},$$

and

$$\boldsymbol{D}_{\mathcal{Q}}=\mathrm{diag}((\lambda_{\boldsymbol{k}}^{\boldsymbol{d}}(h_{\boldsymbol{d}})\mathbf{I}_{B(\boldsymbol{d},\boldsymbol{k})})_{\boldsymbol{k}\in\mathcal{Q}})\in\mathbb{R}^{B\times B}.$$

From Lemma 4, we have

$$
\begin{aligned}
&n(\boldsymbol{H}+\lambda\mathbf{I}_n)^{-1}\boldsymbol{M}(\boldsymbol{H}+\lambda\mathbf{I}_n)^{-1}\\
&=n(\boldsymbol{Y}_{\mathcal{Q}}\boldsymbol{D}_{\mathcal{Q}}\boldsymbol{Y}_{\mathcal{Q}}^{\mathsf{T}}+(\kappa_h+\lambda)\mathbf{I}_n+\kappa_h\boldsymbol{\Delta}_h)^{-1}(\boldsymbol{Y}_{\mathcal{Q}}\boldsymbol{D}_{\mathcal{Q}}^2\boldsymbol{Y}_{\mathcal{Q}}^{\mathsf{T}}+\kappa_m\boldsymbol{\Delta}_m)(\boldsymbol{Y}_{\mathcal{Q}}\boldsymbol{D}_{\mathcal{Q}}\boldsymbol{Y}_{\mathcal{Q}}^{\mathsf{T}}+(\kappa_h+\lambda)\mathbf{I}_n+\kappa_h\boldsymbol{\Delta}_h)^{-1}\\
&=T_1+T_2,
\end{aligned}
$$

where $\|\boldsymbol{\Delta}_h\|_{\mathrm{op}}=o_{d,\mathbb{P}}(1)$, $\|\boldsymbol{\Delta}_u\|_{\mathrm{op}}=O_{d,\mathbb{P}}(1)$ and $\kappa_m=O_d(d^{-m(\gamma)})$, and

$$T_1=n\kappa_m(\boldsymbol{Y}_{\mathcal{Q}}\boldsymbol{D}_{\mathcal{Q}}\boldsymbol{Y}_{\mathcal{Q}}^{\mathsf{T}}+(\kappa_h+\lambda)\mathbf{I}_n+\kappa_h\boldsymbol{\Delta}_h)^{-1}\boldsymbol{\Delta}_m(\boldsymbol{Y}_{\mathcal{Q}}\boldsymbol{D}_{\mathcal{Q}}\boldsymbol{Y}_{\mathcal{Q}}^{\mathsf{T}}+(\kappa_h+\lambda)\mathbf{I}_n+\kappa_h\boldsymbol{\Delta}_h)^{-1},$$
$$T_2=n(\boldsymbol{Y}_{\mathcal{Q}}\boldsymbol{D}_{\mathcal{Q}}\boldsymbol{Y}_{\mathcal{Q}}^{\mathsf{T}}+(\kappa_h+\lambda)\mathbf{I}_n+\kappa_h\boldsymbol{\Delta}_h)^{-1}\boldsymbol{Y}_{\mathcal{Q}}\boldsymbol{D}_{\mathcal{Q}}^2\boldsymbol{Y}_{\mathcal{Q}}^{\mathsf{T}}(\boldsymbol{Y}_{\mathcal{Q}}\boldsymbol{D}_{\mathcal{Q}}\boldsymbol{Y}_{\mathcal{Q}}^{\mathsf{T}}+(\kappa_h+\lambda)\mathbf{I}_n+\kappa_h\boldsymbol{\Delta}_h)^{-1}.$$

Then, we can use the same proof as in [4, Lemma 13] to bound $\|T_1\|_{\mathrm{op}}$ (recall $n = O_d(d^{m(\gamma)-\delta})$)

$$\|T_1\|_{\mathrm{op}} \leq 2n\kappa_m/(\kappa_h + \lambda)^2 \|\boldsymbol{\Delta}_m\|_{\mathrm{op}} = o_{d,\mathbb{P}}(1),$$

and $\|T_2 - \boldsymbol{Y}_{\mathcal{Q}}\boldsymbol{Y}_{\mathcal{Q}}^{\mathsf{T}}/n\|_{\mathrm{op}} = o_{d,\mathbb{P}}(1)$, where we only need to check that

$$\lambda_{\min}(\boldsymbol{D}_{\mathcal{Q}}/[(\kappa_h + \lambda)/n]) = \min_{\boldsymbol{k} \in \mathcal{Q}}[n\lambda_{\boldsymbol{k}}^{\boldsymbol{d}}(h_{\boldsymbol{d}})]/(\kappa_h + \lambda) = w_d(1),$$

which directly follows from Lemma 2. $\hfill\square$

**Lemma 7.** *Let $\sigma$ be an activation function satisfying Assumption 1. Assume $\omega_d(d^\gamma \log d) \leq n \leq O_d(d^{m(\gamma)-\delta})$ for some $\gamma > 0$ and $\delta > 0$. We have*

$$\|\boldsymbol{Y}_{\mathcal{Q}}^{\mathsf{T}}(\boldsymbol{H} + \lambda\mathbf{I}_n)^{-1}\boldsymbol{Y}_{\mathcal{Q}}\boldsymbol{D}_{\mathcal{Q}} - \mathbf{I}_B\|_{\mathrm{op}} = o_{d,\mathbb{P}}(1).$$

*Proof of Lemma 7.*

This lemma can be deduced directly from [4, Lemma 14], by noticing that

$$\lambda_{\min}(\boldsymbol{D}_{\mathcal{Q}}/[(\kappa_h + \lambda)/n] = \min_{\boldsymbol{k} \in \mathcal{Q}}[n\lambda_{\boldsymbol{k}}^{\boldsymbol{d}}(h_{\boldsymbol{d}})]/(\kappa_h + \lambda) = \omega_d(1),$$

from Lemma 2. $\hfill\square$

# E   Proof of Theorem 2.(a): lower bound for the RF model

## E.1   Preliminaries

In the theorems, we show our results in high probability with respect to $\boldsymbol{\Theta}$. Hence, in the proof we will restrict the sample space to the high probability event $\mathcal{P}_\varepsilon \equiv \mathcal{P}_{d,N,\varepsilon}$ for $\varepsilon > 0$ small enough, where

$$\mathcal{P}_{d,N,\varepsilon} \equiv \left\{ \boldsymbol{\Theta} \,\middle|\, \tau_i^{(q)} \in [1-\varepsilon, 1+\varepsilon], \forall i \in [N], \forall q \in [Q] \right\} \subset \left(\mathbb{S}^{D-1}(\sqrt{D})\right)^{\otimes N}. \tag{64}$$

We will denote $\mathbb{E}_{\boldsymbol{\tau}_\varepsilon}$ the expectation over $\boldsymbol{\tau}$ restricted to $\tau^{(q)} \in [1-\varepsilon, 1+\varepsilon]$ for all $q \in [Q]$, and $\mathbb{E}_{\boldsymbol{\Theta}_\varepsilon}$ the expectation over $\boldsymbol{\Theta}$ restricted to the event $\mathcal{P}_\varepsilon$.

**Lemma 8.** *Assume $N = o(d^\gamma)$ for some $\gamma > 0$. We have for any fixed $\varepsilon > 0$,*

$$\mathbb{P}(\mathcal{P}_\varepsilon^c) = o_d(1).$$

*Proof of Lemma 8.* The tail inequality in Lemma 16 and the assumption $N = o(d^\gamma)$ imply that there exists some constants $C, c > 0$ such that

$$\mathbb{P}(\mathcal{P}_{d,N,\varepsilon}^c) \leq \sum_{q \in [Q]} N\mathbb{P}(|\tau^{(q)} - 1| > \varepsilon) \leq \sum_{q \in [Q]} C\exp(\gamma \log(d) - cd^{\eta_q}\varepsilon) = o_d(1).$$

$\hfill\square$

We consider the activation function $\sigma : \mathbb{R} \to \mathbb{R}$. Let $\boldsymbol{\theta} \sim \mathbb{S}^{D-1}(\sqrt{D})$ and $\boldsymbol{x} = \{\boldsymbol{x}^{(q)}\}_{q \in [Q]} \in \mathrm{PS}_{\boldsymbol{\kappa}}^{\boldsymbol{d}}$. We introduce the function $\sigma_{\boldsymbol{d},\boldsymbol{\tau}} : \mathrm{ps}^{\boldsymbol{d}} \to \mathbb{R}$ such that

$$\sigma(\langle\boldsymbol{\theta},\boldsymbol{x}\rangle/R) = \sigma\left(\sum_{q \in [Q]} \tau^{(q)} \cdot (r_q/R) \cdot \langle\overline{\boldsymbol{\theta}}^{(q)}, \overline{\boldsymbol{x}}^{(q)}\rangle/\sqrt{d_q}\right) \tag{65}$$

$$\equiv \sigma_{\boldsymbol{d},\boldsymbol{\tau}}\left(\{\langle\overline{\boldsymbol{\theta}}^{(q)}, \overline{\boldsymbol{x}}^{(q)}\rangle/\sqrt{d_q}\}_{q \in [Q]}\right).$$

Consider the expansion of $\sigma_{\boldsymbol{d},\boldsymbol{\tau}}$ in terms of tensor product of Gegenbauer polynomials. We have

$$\sigma(\langle\boldsymbol{\theta},\boldsymbol{x}\rangle/R) = \sum_{\boldsymbol{k} \in \mathbb{Z}_{\geq 0}^Q} \lambda_{\boldsymbol{k}}^{\boldsymbol{d}}(\sigma_{\boldsymbol{d},\boldsymbol{\tau}}) B(\boldsymbol{d},\boldsymbol{k}) Q_{\boldsymbol{k}}^{\boldsymbol{d}}\left(\{\langle\overline{\boldsymbol{\theta}}^{(q)}, \overline{\boldsymbol{x}}^{(q)}\rangle\}_{q \in [Q]}\right), \tag{66}$$

where

$$\lambda_{\boldsymbol{k}}^{\boldsymbol{d}}(\sigma_{\boldsymbol{d},\boldsymbol{\tau}}) = \mathbb{E}_{\overline{\boldsymbol{x}}}\left[\sigma_{\boldsymbol{d},\boldsymbol{\tau}}\left(\overline{x}_1^{(1)}, \ldots, \overline{x}_1^{(Q)}\right) Q_{\boldsymbol{k}}^{\boldsymbol{d}}\left(\sqrt{d_1}\overline{x}_1^{(1)}, \ldots, \sqrt{d_Q}\overline{x}_1^{(Q)}\right)\right],$$

where the expectation is taken over $\overline{\boldsymbol{x}} = (\overline{\boldsymbol{x}}^{(1)}, \ldots, \overline{\boldsymbol{x}}^{(Q)}) \sim \mu_{\boldsymbol{d}}$.

**Lemma 9.** *Let $\sigma$ be an activation function that satisfies Assumptions 2.(a) and 2.(b). Consider $N \leq o_d(d^\gamma)$ and $\mathcal{Q} = \mathcal{Q}_{\mathrm{RF}}(\gamma)$ as defined in Theorem 2.(a). Then there exists $\varepsilon_0 > 0$ and $d_0$ and a constant $C > 0$ such that for $d \geq d_0$ and $\boldsymbol{\tau} \in [1 - \varepsilon_0, 1 + \varepsilon_0]^Q$,*

$$\max_{\boldsymbol{k} \notin \mathcal{Q}} \lambda_{\boldsymbol{k}}^{\boldsymbol{d}}(\sigma_{\boldsymbol{d},\boldsymbol{\tau}})^2 \leq C d^{-\gamma}.$$

*Proof of Lemma 9.* Notice that by Assumption 2.(b) we can apply Lemma 19 to any $\boldsymbol{k} \in \mathcal{Q}^c$ such that $|\boldsymbol{k}| = k_1 + \ldots + k_Q \leq L$. In particular, there exists $C > 0$, $\varepsilon_0' > 0$ and $d_0'$ such that for any $\boldsymbol{k} \in \mathcal{Q}^c$ with $|\boldsymbol{k}| \leq L$, $d \geq d_0'$ and $\boldsymbol{\tau} \in [1 - \varepsilon_0', 1 + \varepsilon_0']^Q$,

$$\left( \prod_{q \in [Q]} d^{(\xi - \eta_q - \kappa_q)k_q} \right) B(\boldsymbol{d}, \boldsymbol{k}) \lambda_{\boldsymbol{k}}^{\boldsymbol{d}}(\sigma_{\boldsymbol{d},\boldsymbol{\tau}})^2 \leq C < \infty,$$

Furthermore, using that $B(\boldsymbol{d}, \boldsymbol{k}) = \Theta(d_1^{k_1} d_2^{k_2} \ldots d_Q^{k_Q})$, there exists $C' > 0$ such that for $\boldsymbol{k} \in \mathcal{Q}^c$ with $|\boldsymbol{k}| \leq L$,

$$\lambda_{\boldsymbol{k}}^{\boldsymbol{d}}(\sigma_{\boldsymbol{d},\boldsymbol{\tau}})^2 \leq C' \prod_{q \in [Q]} d^{(\eta_q + \kappa_q - \xi)k_q} d_q^{-k_q} = C' \prod_{q \in [Q]} d^{(\kappa_q - \xi)k_q} \leq C' d^{-\gamma}, \tag{67}$$

where we used in the last inequality $\boldsymbol{k} \notin \mathcal{Q}_{\mathrm{RF}}(\gamma)$ implies $(\xi - \kappa_1)k_1 + \ldots + (\xi - \kappa_Q)k_Q \geq \gamma$ by definition.

Furthermore, from Assumption 2 and Lemma 17.(b), there exists $\varepsilon_0'' > 0$, $d_0''$ and $C < \infty$, such that

$$\sup_{d \geq d_0''} \sup_{\boldsymbol{\tau} \in [1-\varepsilon_0'', 1+\varepsilon_0'']^Q} \mathbb{E}_{\overline{\boldsymbol{x}}}\left[ \sigma_{\boldsymbol{d},\boldsymbol{\tau}}\left( \{\langle \boldsymbol{w}^{(q)}, \overline{\boldsymbol{x}}^{(q)}\rangle\}_{q \in [Q]} \right)^2 \right] < C.$$

From the Gegenbauer decomposition (66), this implies that for any $\boldsymbol{k} \in \mathbb{Z}_{\geq 0}^Q$, $d \leq d_0''$ and $\boldsymbol{\tau} \in [1 - \varepsilon_0'', 1 + \varepsilon_0'']^Q$,

$$B(\boldsymbol{d}, \boldsymbol{k}) \lambda_{\boldsymbol{k}}^{\boldsymbol{d}}(\sigma_{\boldsymbol{d},\boldsymbol{\tau}})^2 \leq C.$$

In particular, for $|\boldsymbol{k}| = k_1 + \ldots + k_Q > L = \max_{q \in [Q]} \lceil \gamma / \eta_q \rceil$, we have

$$\lambda_{\boldsymbol{k}}^{\boldsymbol{d}}(\sigma_{\boldsymbol{d},\boldsymbol{\tau}})^2 \leq \frac{C}{B(\boldsymbol{d}, \boldsymbol{k})} \leq C' \prod_{q \in [Q]} d^{-\eta_q k_q} \leq C' \prod_{q \in [Q]} d^{-\gamma k_q / L} \leq C' d^{-\gamma}. \tag{68}$$

Combining Eqs (67) and (68) yields the result. □

### E.2 Proof of Theorem 2.(a): Outline

Let $\mathcal{Q} \equiv \mathcal{Q}_{\mathrm{RF}}(\gamma)$ as defined in Theorem 2.(a) and $\boldsymbol{\Theta} = \sqrt{D}\boldsymbol{W}$ such that $\boldsymbol{\theta}_i = \sqrt{D}\boldsymbol{w}_i \sim_{iid} \mathrm{Unif}(\mathbb{S}^{D-1}(\sqrt{D}))$.

Define the random vectors $\boldsymbol{V} = (V_1, \ldots, V_N)^\mathsf{T}$, $\boldsymbol{V}_{\mathcal{Q}} = (V_{1,\mathcal{Q}}, \ldots, V_{N,\mathcal{Q}})^\mathsf{T}$, $\boldsymbol{V}_{\mathcal{Q}^c} = (V_{1,\mathcal{Q}^c}, \ldots, V_{N,\mathcal{Q}^c})^\mathsf{T}$, with

$$V_{i,\mathcal{Q}} \equiv \mathbb{E}_{\boldsymbol{x}}[[\mathsf{P}_{\mathcal{Q}} f_d](\boldsymbol{x}) \sigma(\langle \boldsymbol{\theta}_i, \boldsymbol{x}\rangle / R)], \tag{69}$$

$$V_{i,\mathcal{Q}^c} \equiv \mathbb{E}_{\boldsymbol{x}}[[\mathsf{P}_{\mathcal{Q}^c} f_d](\boldsymbol{x}) \sigma(\langle \boldsymbol{\theta}_i, \boldsymbol{x}\rangle / R)], \tag{70}$$

$$V_i \equiv \mathbb{E}_{\boldsymbol{x}}[f_d(\boldsymbol{x}) \sigma(\langle \boldsymbol{\theta}_i, \boldsymbol{x}\rangle / R)] = V_{i,\mathcal{Q}} + V_{i,\mathcal{Q}^c}. \tag{71}$$

Define the random matrix $\boldsymbol{U} = (U_{ij})_{i,j \in [N]}$, with

$$U_{ij} = \mathbb{E}_{\boldsymbol{x}}[\sigma(\langle \boldsymbol{x}, \boldsymbol{\theta}_i\rangle / R) \sigma(\langle \boldsymbol{x}, \boldsymbol{\theta}_j\rangle / R)]. \tag{72}$$

In what follows, we write $R_{\mathrm{RF}}(f_d) = R_{\mathrm{RF}}(f_d, \boldsymbol{W}) = R_{\mathrm{RF}}(f_d, \boldsymbol{\Theta}/\sqrt{D})$ for the random features risk, omitting the dependence on the weights $\boldsymbol{W} = \boldsymbol{\Theta}/\sqrt{D}$. By the definition and a simple calculation, we have

$$R_{\mathrm{RF}}(f_d) = \min_{\boldsymbol{a} \in \mathbb{R}^N} \left\{ \mathbb{E}_{\boldsymbol{x}}[f_d(\boldsymbol{x})^2] - 2\langle \boldsymbol{a}, \boldsymbol{V}\rangle + \langle \boldsymbol{a}, \boldsymbol{U}\boldsymbol{a}\rangle \right\} = \mathbb{E}_{\boldsymbol{x}}[f_d(\boldsymbol{x})^2] - \boldsymbol{V}^\mathsf{T} \boldsymbol{U}^{-1} \boldsymbol{V},$$

$$R_{\mathrm{RF}}(\mathsf{P}_{\mathcal{Q}} f_d) = \min_{\boldsymbol{a} \in \mathbb{R}^N} \left\{ \mathbb{E}_{\boldsymbol{x}}[\mathsf{P}_{\mathcal{Q}} f_d(\boldsymbol{x})^2] - 2\langle \boldsymbol{a}, \boldsymbol{V}_{\leq \ell}\rangle + \langle \boldsymbol{a}, \boldsymbol{U}\boldsymbol{a}\rangle \right\} = \mathbb{E}_{\boldsymbol{x}}[\mathsf{P}_{\mathcal{Q}} f_d(\boldsymbol{x})^2] - \boldsymbol{V}_{\mathcal{Q}}^\mathsf{T} \boldsymbol{U}^{-1} \boldsymbol{V}_{\mathcal{Q}}.$$

By orthogonality, we have

$$\mathbb{E}_{\boldsymbol{x}}[f_d(\boldsymbol{x})^2] = \mathbb{E}_{\boldsymbol{x}}[[\mathsf{P}_{\mathcal{Q}}f_d](\boldsymbol{x})^2] + \mathbb{E}_{\boldsymbol{x}}[[\mathsf{P}_{\mathcal{Q}^c}f_d](\boldsymbol{x})^2],$$

which gives

$$
\begin{aligned}
&\left| R_{\mathrm{RF}}(f_d) - R_{\mathrm{RF}}(\mathsf{P}_{\mathcal{Q}}f_d) - \mathbb{E}_{\boldsymbol{x}}[[\mathsf{P}_{\mathcal{Q}^c}f_d](\boldsymbol{x})^2] \right| \\
=& \left| \boldsymbol{V}_{\mathcal{Q}}^{\mathsf{T}}\boldsymbol{U}^{-1}\boldsymbol{V}_{\mathcal{Q}} - \boldsymbol{V}^{\mathsf{T}}\boldsymbol{U}^{-1}\boldsymbol{V} \right| = \left| \boldsymbol{V}_{\mathcal{Q}}^{\mathsf{T}}\boldsymbol{U}^{-1}\boldsymbol{V}_{\mathcal{Q}} - (\boldsymbol{V}_{\mathcal{Q}} + \boldsymbol{V}_{\mathcal{Q}^c})^{\mathsf{T}}\boldsymbol{U}^{-1}(\boldsymbol{V}_{\mathcal{Q}} + \boldsymbol{V}_{\mathcal{Q}^c}) \right| \\
=& \left| 2\boldsymbol{V}^{\mathsf{T}}\boldsymbol{U}^{-1}\boldsymbol{V}_{\mathcal{Q}^c} - \boldsymbol{V}_{\mathcal{Q}^c}^{\mathsf{T}}\boldsymbol{U}^{-1}\boldsymbol{V}_{\mathcal{Q}^c} \right| \le 2\|\boldsymbol{U}^{-1/2}\boldsymbol{V}_{\mathcal{Q}^c}\|_2 \|\boldsymbol{U}^{-1/2}\boldsymbol{V}\|_2 + \|\boldsymbol{U}^{-1}\|_{\mathrm{op}}\|\boldsymbol{V}_{\mathcal{Q}^c}\|_2^2 \\
\le& 2\|\boldsymbol{U}^{-1/2}\|_{\mathrm{op}}\|\boldsymbol{V}_{\mathcal{Q}^c}\|_2\|f_d\|_{L^2} + \|\boldsymbol{U}^{-1}\|_{\mathrm{op}}\|\boldsymbol{V}_{\mathcal{Q}^c}\|_2^2,
\end{aligned}
\tag{73}
$$

where the last inequality used the fact that

$$0 \le R_{\mathrm{RF}}(f_d) = \|f_d\|_{L^2}^2 - \boldsymbol{V}^{\mathsf{T}}\boldsymbol{U}^{-1}\boldsymbol{V},$$

so that

$$\|\boldsymbol{U}^{-1/2}\boldsymbol{V}\|_2^2 = \boldsymbol{V}^{\mathsf{T}}\boldsymbol{U}^{-1}\boldsymbol{V} \le \|f_d\|_{L^2}^2.$$

The Theorem follows from the following two claims

$$\|\boldsymbol{V}_{\mathcal{Q}^c}\|_2 / \|\mathsf{P}_{\mathcal{Q}^c}f_d\|_{L^2} = o_{d,\mathbb{P}}(1), \tag{74}$$

$$\|\boldsymbol{U}^{-1}\|_{\mathrm{op}} = O_{d,\mathbb{P}}(1), \tag{75}$$

This is achieved by the Proposition 1 and 2 stated below.

**Proposition 1** (Expected norm of $\boldsymbol{V}$). *Let $\sigma$ be an activation function satisfying Assumptions 2.(a) and 2.(b) for a fixed $\gamma > 0$. Denote $\mathcal{Q} = \mathcal{Q}_{\mathrm{RF}}(\gamma)$. Let $\varepsilon > 0$ and define $\mathcal{E}_{\mathcal{Q}^c,\varepsilon}$ by*

$$\mathcal{E}_{\mathcal{Q}^c,\varepsilon} \equiv \mathbb{E}_{\boldsymbol{\theta}_\varepsilon}[\langle \mathsf{P}_{\mathcal{Q}^c,0}f_d, \sigma(\langle\boldsymbol{\theta},\cdot\rangle/R)\rangle_{L^2}^2],$$

*where we recall that $\mathbb{E}_{\boldsymbol{\theta}_\varepsilon} = \mathbb{E}_{\boldsymbol{\tau}_\varepsilon}\mathbb{E}_{\overline{\boldsymbol{\theta}}}$ the expectation with respect to $\boldsymbol{\tau}$ restricted to $[1-\varepsilon, 1+\varepsilon]^Q$ and $\overline{\boldsymbol{\theta}} \sim \mathrm{Unif}(\mathrm{PS}^d)$.*

*Then there exists a constant $C > 0$ and $\varepsilon_0 > 0$ (depending only on the constants of Assumptions 2.(a) and 2.(b)) such that for $d$ sufficiently large,*

$$\mathcal{E}_{\mathcal{Q}^c,\varepsilon_0} \le Cd^{-\gamma} \cdot \|\mathsf{P}_{\mathcal{Q}^c}f_d\|_{L^2}^2.$$

**Proposition 2** (Lower bound on the kernel matrix). *Assume $N = o_d(d^\gamma)$ for a fixed integer $\gamma > 0$. Let $(\boldsymbol{\theta}_i)_{i\in[N]} \sim \mathrm{Unif}(\mathbb{S}^{D-1}(\sqrt{D}))$ independently, and $\sigma$ be an activation function satisfying Assumption 2.(a). Let $\boldsymbol{U} \in \mathbb{R}^{N\times N}$ be the kernel matrix defined by Eq. (72). Then there exists a constant $\varepsilon > 0$ that depends on the activation function $\sigma$, such that*

$$\lambda_{\min}(\boldsymbol{U}) \ge \varepsilon,$$

*with high probability as $d \to \infty$.*

The proofs of these two propositions are provided in the next sections.

Proposition 1 shows that there exists $\varepsilon_0 > 0$ such that

$$\mathbb{E}_{\boldsymbol{\Theta}_{\varepsilon_0}}[\|\boldsymbol{V}_{\mathcal{Q}^c}\|_2^2] = N\mathcal{E}_{\mathcal{Q}^c,\varepsilon_0} \le CNd^{-\gamma}\|\mathsf{P}_{\mathcal{Q}^c}f_d\|_{L^2}^2.$$

Hence, by Markov's inequality, we get for any $\varepsilon > 0$,

$$
\begin{aligned}
\mathbb{P}(\|\boldsymbol{V}_{\mathcal{Q}^c}\|_2 \ge \varepsilon \cdot \|\mathsf{P}_{\mathcal{Q}^c}f_d\|_{L^2}) \le& \mathbb{P}(\{\|\boldsymbol{V}_{\mathcal{Q}^c}\|_2 \ge \varepsilon \cdot \|\mathsf{P}_{\mathcal{Q}^c}f_d\|_{L^2}\} \cap \mathcal{P}_{\varepsilon_0}) + \mathbb{P}(\mathcal{P}_{\varepsilon_0}^c) \\
\le& \frac{N\mathcal{E}_{\mathcal{Q}^c,\varepsilon_0}}{\varepsilon^2\|\mathsf{P}_{\mathcal{Q}^c}f_d\|_{L^2}^2} + o_d(1) \\
\le& C'Nd^{-\gamma} + o_d(1),
\end{aligned}
$$

where we used Lemma 8. By assumption, we have $N = o_d(d^\gamma)$, hence Eq. (74) is verified. Furthermore Eq. (75) follows simply from Proposition 2. This proves the theorem.

## E.3 Proof of Proposition 1

We will denote:

$$\overline{f}_d(\overline{\boldsymbol{x}}) = f_d(\boldsymbol{x}),$$

such that $\overline{f}$ is a function on the normalized product of spheres $\mathrm{PS}^{\boldsymbol{d}}$ (Note that we defined $\mathsf{P}_{\boldsymbol{k}}f_d(\boldsymbol{x}) \equiv \mathsf{P}_{\boldsymbol{k}}\overline{f}_d(\overline{\boldsymbol{x}})$ the unambiguous polynomial approximation of $f_d$ with polynomial of degree $\boldsymbol{k}$). We have

$$
\begin{aligned}
V_{i,\mathcal{Q}^c} =& \mathbb{E}_{\boldsymbol{x}}\left[ [\mathsf{P}_{\mathcal{Q}^c}f_d](\boldsymbol{x})\sigma\left( \sum_{q\in[Q]} \langle \boldsymbol{x}^{(q)}, \boldsymbol{\theta}_i^{(q)}\rangle/R \right) \right]\\
=& \mathbb{E}_{\overline{\boldsymbol{x}}}\left[ [\mathsf{P}_{\mathcal{Q}^c}\overline{f}_d](\overline{\boldsymbol{x}})\sigma_{\boldsymbol{d},\boldsymbol{\tau}_i}\left( \{\langle \overline{\boldsymbol{x}}^{(q)}, \overline{\boldsymbol{\theta}}_i^{(q)}\rangle/\sqrt{d_q}\}_{q\in[Q]} \right) \right].
\end{aligned}
$$

We recall the expansion of $\sigma_{\boldsymbol{d},\boldsymbol{\tau}}$ in terms of tensor product of Gegenbauer polynomials

$$
\begin{aligned}
\sigma(\langle \boldsymbol{\theta}, \boldsymbol{x}\rangle/R) =& \sum_{\boldsymbol{k}\in\mathbb{Z}_{\geq 0}^Q} \lambda_{\boldsymbol{k}}^{\boldsymbol{d}}(\sigma_{\boldsymbol{d},\boldsymbol{\tau}})B(\boldsymbol{d},\boldsymbol{k})Q_{\boldsymbol{k}}^{\boldsymbol{d}}\left( \{\langle \overline{\boldsymbol{\theta}}^{(q)}, \overline{\boldsymbol{x}}^{(q)}\rangle\}_{q\in[Q]} \right),\\
\lambda_{\boldsymbol{k}}^{\boldsymbol{d}}(\sigma_{\boldsymbol{d},\boldsymbol{\tau}}) =& \mathbb{E}_{\overline{\boldsymbol{x}}}\left[ \sigma_{\boldsymbol{d},\boldsymbol{\tau}}\left( \overline{x}_1^{(1)}, \dots, \overline{x}_1^{(Q)} \right) Q_{\boldsymbol{k}}^{\boldsymbol{d}}\left( \sqrt{d_1}\overline{x}_1^{(1)}, \dots, \sqrt{d_Q}\overline{x}_1^{(Q)} \right) \right].
\end{aligned}
$$

For any $\boldsymbol{k}\in\mathbb{Z}_{\geq 0}^Q$, the spherical harmonics expansion of $P_{\boldsymbol{k}}\overline{f}_d$ gives

$$P_{\boldsymbol{k}}\overline{f}_d(\overline{\boldsymbol{x}}) = \sum_{\boldsymbol{s}\in[B(\boldsymbol{d},\boldsymbol{k})]} \lambda_{\boldsymbol{k},\boldsymbol{s}}^{\boldsymbol{d}}(\overline{f}_d)Y_{\boldsymbol{k},\boldsymbol{s}}^{\boldsymbol{d}}(\overline{\boldsymbol{x}}).$$

Using Eq. (27) to get the following property

$$
\begin{aligned}
\mathbb{E}_{\overline{\boldsymbol{x}}}\left[ Q_{\boldsymbol{k}'}^{\boldsymbol{d}}\left( \{\langle \overline{\boldsymbol{\theta}}^{(q)}, \overline{\boldsymbol{x}}^{(q)}\rangle\}_{q\in[Q]} \right) Y_{\boldsymbol{k},\boldsymbol{s}}^{\boldsymbol{d}}(\overline{\boldsymbol{x}}) \right] =& \frac{1}{B(\boldsymbol{d},\boldsymbol{k}')}\sum_{\boldsymbol{s}'\in[B(\boldsymbol{d},\boldsymbol{k})]} Y_{\boldsymbol{k}',\boldsymbol{s}'}^{\boldsymbol{d}}(\overline{\boldsymbol{\theta}})\mathbb{E}_{\overline{\boldsymbol{x}}}\left[ Y_{\boldsymbol{k}',\boldsymbol{s}'}^{\boldsymbol{d}}(\overline{\boldsymbol{x}})Y_{\boldsymbol{k},\boldsymbol{s}}^{\boldsymbol{d}}(\overline{\boldsymbol{x}}) \right]\\
=& \frac{1}{B(\boldsymbol{d},\boldsymbol{k})}Y_{\boldsymbol{k},\boldsymbol{s}}^{\boldsymbol{d}}(\overline{\boldsymbol{\theta}})\delta_{\boldsymbol{k},\boldsymbol{k}'},
\end{aligned}
\tag{76}
$$

we get

$$
\begin{aligned}
&\mathbb{E}_{\overline{\boldsymbol{x}}}\left[ [\mathsf{P}_{\boldsymbol{k}}\overline{f}_d](\overline{\boldsymbol{x}})\sigma_{\boldsymbol{d},\boldsymbol{\tau}}\left( \{\langle \overline{\boldsymbol{\theta}}^{(q)}, \overline{\boldsymbol{x}}^{(q)}\rangle/\sqrt{d_q}\}_{q\in[Q]} \right) \right]\\
=& \sum_{\boldsymbol{k}'\geq 0} \lambda_{\boldsymbol{k}'}^{\boldsymbol{d}}(\sigma_{\boldsymbol{d},\boldsymbol{\tau}})B(\boldsymbol{d},\boldsymbol{k}') \sum_{\boldsymbol{s}\in[B(\boldsymbol{d},\boldsymbol{k})]} \lambda_{\boldsymbol{k},\boldsymbol{s}}^{\boldsymbol{d}}(\overline{f}_d)\mathbb{E}_{\overline{\boldsymbol{x}}}\left[ Y_{\boldsymbol{k},\boldsymbol{s}}^{\boldsymbol{d}}(\overline{\boldsymbol{x}})Q_{\boldsymbol{k}}^{\boldsymbol{d}}\left( \{\langle \overline{\boldsymbol{\theta}}^{(q)}, \overline{\boldsymbol{x}}^{(q)}\rangle\}_{q\in[Q]} \right) \right]\\
=& \sum_{\boldsymbol{s}\in[B(\boldsymbol{d},\boldsymbol{k})]} \lambda_{\boldsymbol{k},\boldsymbol{s}}^{\boldsymbol{d}}(\overline{f}_d)\lambda_{\boldsymbol{k}}^{\boldsymbol{d}}(\sigma_{\boldsymbol{d},\boldsymbol{\tau}})Y_{\boldsymbol{k},\boldsymbol{s}}^{\boldsymbol{d}}(\overline{\boldsymbol{\theta}}).
\end{aligned}
$$

Let $\varepsilon_0 > 0$ be a constant as specified in Lemma 9. We consider

$$
\begin{aligned}
\mathcal{E}_{\mathcal{Q}^c,\varepsilon_0} =& \mathbb{E}_{\overline{\boldsymbol{\theta}},\boldsymbol{\tau}_{\varepsilon_0}}\left[\mathbb{E}_{\overline{\boldsymbol{x}}}\left[[\mathsf{P}_{\mathcal{Q}^c}\overline{f}_d](\overline{\boldsymbol{x}})\sigma_{\boldsymbol{d},\boldsymbol{\tau}}\left(\{\langle\overline{\boldsymbol{\theta}}^{(q)},\overline{\boldsymbol{x}}^{(q)}\rangle/\sqrt{d_q}\}_{q\in[Q]}\right)\right]^2\right] \\
=& \sum_{\boldsymbol{k},\boldsymbol{k}'\in\mathcal{Q}^c}\mathbb{E}_{\overline{\boldsymbol{\theta}},\boldsymbol{\tau}_{\varepsilon_0}}\left[\mathbb{E}_{\overline{\boldsymbol{x}}}\left[[\mathsf{P}_{\boldsymbol{k}}\overline{f}_d](\overline{\boldsymbol{x}})\sigma_{\boldsymbol{d},\boldsymbol{\tau}}\left(\{\langle\overline{\boldsymbol{\theta}}^{(q)},\overline{\boldsymbol{x}}^{(q)}\rangle/\sqrt{d_q}\}_{q\in[Q]}\right)\right]\right.\\
&\qquad\qquad\qquad\left.\times\mathbb{E}_{\overline{\boldsymbol{y}}}\left[[\mathsf{P}_{\boldsymbol{k}'}\overline{f}_d](\overline{\boldsymbol{y}})\sigma_{\boldsymbol{d},\boldsymbol{\tau}}\left(\{\langle\overline{\boldsymbol{\theta}}^{(q)},\overline{\boldsymbol{y}}^{(q)}\rangle/\sqrt{d_q}\}_{q\in[Q]}\right)\right]\right] \\
=& \sum_{\boldsymbol{k},\boldsymbol{k}'\in\mathcal{Q}^c}\mathbb{E}_{\boldsymbol{\tau}_{\varepsilon_0}}\left[\lambda_{\boldsymbol{k}}^{\boldsymbol{d}}(\sigma_{\boldsymbol{d},\boldsymbol{\tau}})\lambda_{\boldsymbol{k}'}^{\boldsymbol{d}}(\sigma_{\boldsymbol{d},\boldsymbol{\tau}})\right] \\
&\qquad\qquad\times\sum_{\boldsymbol{s}\in[B(\boldsymbol{d},\boldsymbol{k})]}\sum_{\boldsymbol{s}'\in[B(\boldsymbol{d},\boldsymbol{k}')]}\lambda_{\boldsymbol{k},\boldsymbol{s}}^{\boldsymbol{d}}(\overline{f}_d)\lambda_{\boldsymbol{k}',\boldsymbol{s}'}^{\boldsymbol{d}}(\overline{f}_d)\mathbb{E}_{\overline{\boldsymbol{\theta}}}[Y_{\boldsymbol{k},\boldsymbol{s}}^{\boldsymbol{d}}(\overline{\boldsymbol{\theta}})Y_{\boldsymbol{k}',\boldsymbol{s}'}^{\boldsymbol{d}}(\overline{\boldsymbol{\theta}})] \qquad (77)\\
=& \sum_{\boldsymbol{k}\in\mathcal{Q}^c}\mathbb{E}_{\boldsymbol{\tau}_{\varepsilon_0}}[\lambda_{\boldsymbol{k}}^{\boldsymbol{d}}(\sigma_{\boldsymbol{d},\boldsymbol{\tau}})^2]\sum_{\boldsymbol{s}\in[B(\boldsymbol{d},\boldsymbol{k})]}\lambda_{\boldsymbol{k},\boldsymbol{s}}^{\boldsymbol{d}}(\overline{f}_d)^2\\
\leq& \left[\max_{\boldsymbol{k}\in\mathcal{Q}^c}\mathbb{E}_{\boldsymbol{\tau}_{\varepsilon_0}}[\lambda_{\boldsymbol{k}}^{\boldsymbol{d}}(\sigma_{\boldsymbol{d},\boldsymbol{\tau}})^2]\right]\cdot\sum_{\boldsymbol{k}\in\mathcal{Q}^c}\sum_{\boldsymbol{s}\in[B(\boldsymbol{d},\boldsymbol{k})]}\lambda_{\boldsymbol{k},\boldsymbol{s}}^{\boldsymbol{d}}(\overline{f}_d)^2\\
=& \left[\max_{\boldsymbol{k}\in\mathcal{Q}^c}\mathbb{E}_{\boldsymbol{\tau}_{\varepsilon_0}}[\lambda_{\boldsymbol{k}}^{\boldsymbol{d}}(\sigma_{\boldsymbol{d},\boldsymbol{\tau}})^2]\right]\cdot\|\mathsf{P}_{\mathcal{Q}^c}\overline{f}_d\|_{L^2}.
\end{aligned}
$$

From Lemma 9, there exists a constant $C > 0$ such that for $d$ sufficiently large, we have for any $\boldsymbol{k}\in\mathcal{Q}^c$,

$$
\mathbb{E}_{\boldsymbol{\tau}_{\varepsilon_0}}[\lambda_{\boldsymbol{k}}^{\boldsymbol{d}}(\sigma_{\boldsymbol{d},\boldsymbol{\tau}})^2]\leq\sup_{\boldsymbol{\tau}\in[1-\varepsilon_0,1+\varepsilon_0]^Q}\lambda_{\boldsymbol{k}}^{\boldsymbol{d}}(\sigma_{\boldsymbol{d},\boldsymbol{\tau}})^2\leq Cd^{-\gamma}. \qquad (78)
$$

Combining Eq. (77) and Eq. (78) yields

$$
\mathcal{E}_{\mathcal{Q}^c,\varepsilon_0}\leq Cd^{-\gamma}\cdot\|\mathsf{P}_{\mathcal{Q}^c}\overline{f}_d\|_{L^2}.
$$

## E.4 Proof of Proposition 2

### Step 1. Construction of the activation functions $\hat{\sigma},\bar{\sigma}$.

Without loss of generality, we will assume that $q_\xi = 1$. From Assumption 2.$(b)$, $\sigma$ is not a degree $\lfloor\gamma/\eta_1\rfloor$-polynomial. This is equivalent to having $m\geq\lfloor\gamma/\eta_1\rfloor + 1$ such that $\mu_m(\sigma)\neq 0$. Let us denote

$$
m = \inf\{k\geq\lfloor\gamma/\eta_1\rfloor + 1|\mu_m(\sigma)\neq 0\}.
$$

Recall the expansion of $\sigma_{\boldsymbol{d},\boldsymbol{\tau}}$ in terms of product of Gegenbauer polynomials

$$
\sigma_{\boldsymbol{d},\boldsymbol{\tau}}\left(\{\langle\overline{\boldsymbol{\theta}}^{(q)},\overline{\boldsymbol{x}}^{(q)}\rangle/\sqrt{d_q}\}_{q\in[Q]}\right) = \sum_{\boldsymbol{k}\in\mathbb{Z}_{\geq 0}^Q}\lambda_{\boldsymbol{k}}^{\boldsymbol{d}}(\sigma_{\boldsymbol{d},\boldsymbol{\tau}})B(\boldsymbol{d},\boldsymbol{k})Q_{\boldsymbol{k}}^{\boldsymbol{d}}\left(\{\langle\overline{\boldsymbol{\theta}}^{(q)},\overline{\boldsymbol{x}}^{(q)}\rangle\}_{q\in[Q]}\right),
$$

where

$$
\lambda_{\boldsymbol{k}}^{\boldsymbol{d}}(\sigma_{\boldsymbol{d},\boldsymbol{\tau}}) = \mathbb{E}_{\overline{\boldsymbol{x}}}\left[\sigma_{\boldsymbol{d},\boldsymbol{\tau}}\left(\overline{x}_1^{(1)},\ldots,\overline{x}_1^{(Q)}\right)Q_{\boldsymbol{k}}^{\boldsymbol{d}}\left(\sqrt{d_1}\overline{x}_1^{(1)},\ldots,\sqrt{d_Q}\overline{x}_1^{(Q)}\right)\right].
$$

Denoting $\boldsymbol{m} = (m,0,\ldots,0)\in\mathbb{Z}_{\geq 0}^Q$ and using the Gegenbauer coefficients of $\sigma_{\boldsymbol{d},\boldsymbol{\tau}}$, we define an activation function $\bar{\sigma}_{\boldsymbol{d},\boldsymbol{\tau}}$ which is a degree $m$ polynomial in $\overline{\boldsymbol{x}}^{(1)}$ and do not depend on $\overline{\boldsymbol{x}}^{(q)}$ for $q\geq 2$.

$$
\begin{aligned}
\bar{\sigma}_{\boldsymbol{d},\boldsymbol{\tau}}\left(\{\langle\overline{\boldsymbol{\theta}}^{(q)},\overline{\boldsymbol{x}}^{(q)}\rangle/\sqrt{d_q}\}_{q\in[Q]}\right) &= \lambda_{\boldsymbol{m}}^{\boldsymbol{d}}(\sigma_{\boldsymbol{d},\boldsymbol{\tau}})B(\boldsymbol{d},\boldsymbol{m})Q_{\boldsymbol{m}}^{\boldsymbol{d}}\left(\{\langle\overline{\boldsymbol{\theta}}^{(q)},\overline{\boldsymbol{x}}^{(q)}\rangle/\sqrt{d_q}\}_{q\in[Q]}\right)\\
&= \lambda_{\boldsymbol{m}}^{\boldsymbol{d}}(\sigma_{\boldsymbol{d},\boldsymbol{\tau}})B(d_1,m)Q_m^{(d_1)}(\sqrt{d_1}\overline{x}_1^{(1)}),
\end{aligned}
$$

and an activation function

$$
\hat{\sigma}_{\boldsymbol{d},\boldsymbol{\tau}}\left(\{\langle\overline{\boldsymbol{\theta}}^{(q)},\overline{\boldsymbol{x}}^{(q)}\rangle/\sqrt{d_q}\}_{q\in[Q]}\right) = \sum_{\boldsymbol{k}\neq\boldsymbol{m}\in\mathbb{Z}_{\geq 0}^Q}\lambda_{\boldsymbol{k}}^{\boldsymbol{d}}(\sigma_{\boldsymbol{d},\boldsymbol{\tau}})B(\boldsymbol{d},\boldsymbol{k})Q_{\boldsymbol{k}}^{\boldsymbol{d}}\left(\{\langle\overline{\boldsymbol{\theta}}^{(q)},\overline{\boldsymbol{x}}^{(q)}\rangle/\sqrt{d_q}\}_{q\in[Q]}\right).
$$

**Step 2. The kernel functions $u_d$, $\hat{u}_d$ and $\bar{u}_d$.**

Let $u_d$, $\hat{u}_d$ and $\bar{u}_d$ be defined by

$$
\begin{aligned}
&u_{\boldsymbol{d}}^{\boldsymbol{\tau}_1,\boldsymbol{\tau}_2}\left(\{\langle\overline{\boldsymbol{\theta}}_1^{(q)},\overline{\boldsymbol{\theta}}_2^{(q)}\rangle/\sqrt{d_q}\}_{q\in[Q]}\right)\\
&=\mathbb{E}_{\boldsymbol{x}}[\sigma(\langle\boldsymbol{\theta}_1,\boldsymbol{x}\rangle/R)\sigma(\langle\boldsymbol{\theta}_2,\boldsymbol{x}\rangle/R)]\\
&=\sum_{\boldsymbol{k}\in\mathbb{Z}_{\geq 0}^Q}\lambda_{\boldsymbol{k}}^{\boldsymbol{d}}(\sigma_{\boldsymbol{d},\boldsymbol{\tau}_1})\lambda_{\boldsymbol{k}}^{\boldsymbol{d}}(\sigma_{\boldsymbol{d},\boldsymbol{\tau}_2})B(\boldsymbol{d},\boldsymbol{k})Q_{\boldsymbol{k}}^{\boldsymbol{d}}\left(\{\langle\overline{\boldsymbol{\theta}}_1^{(q)},\overline{\boldsymbol{\theta}}_2^{(q)}\rangle/\sqrt{d_q}\}_{q\in[Q]}\right)
\end{aligned}
\tag{79}
$$

and

$$
\begin{aligned}
&\hat{u}_{\boldsymbol{d}}^{\boldsymbol{\tau}_1,\boldsymbol{\tau}_2}\left(\{\langle\overline{\boldsymbol{\theta}}_1^{(q)},\overline{\boldsymbol{\theta}}_2^{(q)}\rangle/\sqrt{d_q}\}_{q\in[Q]}\right)\\
&=\mathbb{E}_{\boldsymbol{x}}[\hat{\sigma}(\langle\boldsymbol{\theta}_1,\boldsymbol{x}\rangle/R)\hat{\sigma}(\langle\boldsymbol{\theta}_2,\boldsymbol{x}\rangle/R)]\\
&=\sum_{\boldsymbol{k}\neq\boldsymbol{m}\in\mathbb{Z}_{\geq 0}^Q}\lambda_{\boldsymbol{k}}^{\boldsymbol{d}}(\sigma_{\boldsymbol{d},\boldsymbol{\tau}_1})\lambda_{\boldsymbol{k}}^{\boldsymbol{d}}(\sigma_{\boldsymbol{d},\boldsymbol{\tau}_2})B(\boldsymbol{d},\boldsymbol{k})Q_{\boldsymbol{k}}^{\boldsymbol{d}}\left(\{\langle\overline{\boldsymbol{\theta}}_1^{(q)},\overline{\boldsymbol{\theta}}_2^{(q)}\rangle/\sqrt{d_q}\}_{q\in[Q]}\right)
\end{aligned}
\tag{80}
$$

and

$$
\begin{aligned}
\bar{u}_{\boldsymbol{d}}^{\boldsymbol{\tau}_1,\boldsymbol{\tau}_2}\left(\{\langle\overline{\boldsymbol{\theta}}_1^{(q)},\overline{\boldsymbol{\theta}}_2^{(q)}\rangle/\sqrt{d_q}\}_{q\in[Q]}\right)&=\mathbb{E}_{\boldsymbol{x}}[\bar{\sigma}(\langle\boldsymbol{\theta}_1,\boldsymbol{x}\rangle/R)\bar{\sigma}(\langle\boldsymbol{\theta}_2,\boldsymbol{x}\rangle/R)]\\
&=\lambda_{\boldsymbol{m}}^{\boldsymbol{d}}(\sigma_{\boldsymbol{d},\boldsymbol{\tau}_1})\lambda_{\boldsymbol{m}}^{\boldsymbol{d}}(\sigma_{\boldsymbol{d},\boldsymbol{\tau}_2})B(d_1,m)Q_m^{(d_1)}(\langle\overline{\boldsymbol{\theta}}_1^{(1)},\overline{\boldsymbol{\theta}}_2^{(1)}\rangle).
\end{aligned}
\tag{81}
$$

We immediately have $u_{\boldsymbol{d}}^{\boldsymbol{\tau}_1,\boldsymbol{\tau}_2} = \hat{u}_{\boldsymbol{d}}^{\boldsymbol{\tau}_1,\boldsymbol{\tau}_2}+\bar{u}_{\boldsymbol{d}}^{\boldsymbol{\tau}_1,\boldsymbol{\tau}_2}$. Note that all three correspond to positive semi-definite kernels.

**Step 3. Analyzing the kernel matrix.**

Let $\boldsymbol{U},\hat{\boldsymbol{U}},\bar{\boldsymbol{U}}\in\mathbb{R}^{N\times N}$ with

$$
\begin{aligned}
\boldsymbol{U}_{ij}&=u_{\boldsymbol{d}}^{\boldsymbol{\tau}_i,\boldsymbol{\tau}_j}\left(\{\langle\overline{\boldsymbol{\theta}}_i^{(q)},\overline{\boldsymbol{\theta}}_j^{(q)}\rangle/\sqrt{d_q}\}_{q\in[Q]}\right),\\
\hat{\boldsymbol{U}}_{ij}&=\hat{u}_{\boldsymbol{d}}^{\boldsymbol{\tau}_i,\boldsymbol{\tau}_j}\left(\{\langle\overline{\boldsymbol{\theta}}_i^{(q)},\overline{\boldsymbol{\theta}}_j^{(q)}\rangle/\sqrt{d_q}\}_{q\in[Q]}\right),\\
\bar{\boldsymbol{U}}_{ij}&=\bar{u}_{\boldsymbol{d}}^{\boldsymbol{\tau}_i,\boldsymbol{\tau}_j}\left(\{\langle\overline{\boldsymbol{\theta}}_i^{(q)},\overline{\boldsymbol{\theta}}_j^{(q)}\rangle/\sqrt{d_q}\}_{q\in[Q]}\right).
\end{aligned}
$$

Since $\hat{\boldsymbol{U}} = \boldsymbol{U} - \bar{\boldsymbol{U}} \succeq 0$, we immediately have $\boldsymbol{U} \succeq \bar{\boldsymbol{U}}$. In the following, we will lower bound $\bar{\boldsymbol{U}}$.

By the decomposition of $\bar{\boldsymbol{U}}$ in terms of Gegenbauer polynomials (81), we have

$$
\bar{\boldsymbol{U}} = B(d_1,m)\operatorname{diag}\left(\lambda_{\boldsymbol{m}}^{\boldsymbol{d}}(\sigma_{\boldsymbol{d},\boldsymbol{\tau}_i})\right)\cdot\boldsymbol{W}_m\cdot\operatorname{diag}\left(\lambda_{\boldsymbol{m}}^{\boldsymbol{d}}(\sigma_{\boldsymbol{d},\boldsymbol{\tau}_i})\right),
$$

where $\boldsymbol{W}_m\in\mathbb{R}^{N\times N}$ with $W_{m,ij} = Q_m^{(d_1)}(\langle\overline{\boldsymbol{\theta}}_i^{(1)},\overline{\boldsymbol{\theta}}_j^{(1)}\rangle)$. From Proposition 6 (recalling that by definition of $m > \gamma/\eta_1$, i.e. $\gamma < m\eta_1$, we have $N < d^{\eta_1 m-\delta} = d_1^{m-\delta'}$ for some $\delta > 0$), we have

$$
\|\boldsymbol{W}_m - \mathbf{I}_N\|_{\mathrm{op}} = o_{d,\mathbb{P}}(1).
$$

Hence we get

$$
\left\|\overline{\boldsymbol{U}} - B(d_1,m)\operatorname{diag}\left(\lambda_{\boldsymbol{m}}^{\boldsymbol{d}}(\sigma_{\boldsymbol{d},\boldsymbol{\tau}_i})^2\right)\right\|_{\mathrm{op}} = \max_{i\in[N]}\left\{B(d_1,m)\lambda_{\boldsymbol{m}}^{\boldsymbol{d}}(\sigma_{\boldsymbol{d},\boldsymbol{\tau}_i})^2\right\}\cdot o_{d,\mathbb{P}}(1).
\tag{82}
$$

From Assumption 2.$(a)$ and Lemma 20 applied to coefficient $\boldsymbol{m}$, as well as the assumption that $\mu_m(\sigma)\neq 0$, there exists $\varepsilon_0 > 0$ and $C,c > 0$ such that for $d$ large enough,

$$
\begin{aligned}
\sup_{\boldsymbol{\tau}\in[1-\varepsilon_0,1+\varepsilon_0]^Q} B(d_1,m)\lambda_{\boldsymbol{m}}^{\boldsymbol{d}}(\sigma_{\boldsymbol{d},\boldsymbol{\tau}})^2 &\leq C < \infty,\\
\inf_{\boldsymbol{\tau}\in[1-\varepsilon_0,1+\varepsilon_0]^Q} B(d_1,m)\lambda_{\boldsymbol{m}}^{\boldsymbol{d}}(\sigma_{\boldsymbol{d},\boldsymbol{\tau}})^2 &\geq c > 0.
\end{aligned}
\tag{83}
$$

We restrict ourselves to the event $\mathcal{P}_{\varepsilon_0}$ defined in Eq. (64), which happens with high probability (Lemma 8). Hence from Eqs. (82) and (83), we deduce that with high probability

$$
\overline{\boldsymbol{U}} = B(d_1,m)\operatorname{diag}\left(\lambda_{\boldsymbol{m}}^{\boldsymbol{d}}(\sigma_{\boldsymbol{d},\boldsymbol{\tau}_i})^2\right) + o_{d,\mathbb{P}}(1) \succeq \frac{c}{2}\mathbf{I}_N.
$$

We conclude that with high probability

$$
\boldsymbol{U} = \bar{\boldsymbol{U}} + \hat{\boldsymbol{U}} \succeq \overline{\boldsymbol{U}} \succeq \frac{c}{2}\mathbf{I}_N.
$$

# F  Proof of Theorem 2.(b): upper bound for RF model

## F.1  Preliminaries

**Lemma 10.** *Let $\sigma$ be an activation function that satisfies Assumptions 2.(a) and 2.(b). Let $\|\boldsymbol{w}^{(q)}\|_2 = 1$ be unit vectors of $\mathbb{R}^{d_q}$, for $q = 1, \ldots, Q$. Fix $\gamma > 0$ and denote $\mathcal{Q} = \overline{\mathcal{Q}}_{\mathrm{RF}}(\gamma)$. Then there exists $\varepsilon_0 > 0$ and $d_0$ and constants $C, c > 0$ such that for $d \geq d_0$ and $\boldsymbol{\tau} \in [1 - \varepsilon_0, 1 + \varepsilon_0]^Q$,*

$$\mathbb{E}_{\overline{\boldsymbol{x}}}\left[\sigma_{\boldsymbol{d},\boldsymbol{\tau}}\left(\{\langle \boldsymbol{w}^{(q)}, \overline{\boldsymbol{x}}^{(q)}\rangle\}_{q \in [Q]}\right)^2\right] \leq C < \infty, \tag{84}$$

$$\min_{\boldsymbol{k} \in \mathcal{Q}} \lambda_{\boldsymbol{k},0}^{\boldsymbol{d}}(\sigma_{\boldsymbol{d},\boldsymbol{\tau}})^2 \geq cd^{-\gamma} > 0. \tag{85}$$

*Proof of Lemma 10.* The first inequality comes simply from Assumption 2.(a) and Lemma 17.(b). For the second inequality, notice that by Assumption 2.(c) we can apply Lemma 19 to any $\boldsymbol{k} \in \mathcal{Q}$. Hence (using that $\mu_k(\sigma)^2 > 0$ and we can choose $\delta$ sufficiently small), we deduce that there exists $c > 0$, $\varepsilon_0 > 0$ and $d_0$ such that for any $d \geq d_0$, $\boldsymbol{\tau} \in [1 - \varepsilon_0, 1 + \varepsilon_0]^Q$ and $\boldsymbol{k} \in \mathcal{Q}$,

$$\left(\prod_{q \in [Q]} d^{(\xi - \eta_q - \kappa_q)k_q}\right) B(\boldsymbol{d}, \boldsymbol{k})\lambda_{\boldsymbol{k}}^{\boldsymbol{d}}(\sigma_{\boldsymbol{d},\boldsymbol{\tau}})^2 \geq c > 0.$$

Furthermore, using that $B(\boldsymbol{d}, \boldsymbol{k}) = \Theta(d_1^{k_1} d_2^{k_2} \ldots d_Q^{k_Q})$, there exists $c' > 0$ such that for any $\boldsymbol{k} \in \mathcal{Q}$,

$$\lambda_{\boldsymbol{k}}^{\boldsymbol{d}}(\sigma_{\boldsymbol{d},\boldsymbol{\tau}})^2 \geq c' \prod_{q \in [Q]} d^{(\eta_q + \kappa_q - \xi)k_q} d_q^{k_q} = c' \prod_{q \in [Q]} d^{(\kappa_q - \xi)k_q} \geq cd^{-\gamma},$$

where we used in the last inequality $\boldsymbol{k} \in \overline{\mathcal{Q}}_{\mathrm{RF}}(\gamma)$ implies $(\xi - \kappa_1)k_1 + \ldots + (\xi - \kappa_Q)k_Q \leq \gamma$ by definition. $\qquad\square$

## F.2  Properties of the limiting kernel

Similarly to the proof of [4, Theorem 1.(b)], we construct a limiting kernel which is used as a proxy to upper bound the RF risk.

We recall the definition of $\mathrm{PS}^{\boldsymbol{d}} = \prod_{q \in [Q]} \mathbb{S}^{d_q - 1}(\sqrt{d_q})$ and $\mu_{\boldsymbol{d}} = \mathrm{Unif}(\mathrm{PS}^{\boldsymbol{d}})$. Let us denote $\mathcal{L} = L^2(\mathrm{PS}^{\boldsymbol{d}}, \mu_{\boldsymbol{d}})$. Fix $\boldsymbol{\tau} \in \mathbb{R}_{>0}^Q$ and recall the definition for a given $\boldsymbol{\theta} = (\overline{\boldsymbol{\theta}}, \boldsymbol{\tau})$ of $\sigma_{\boldsymbol{d},\boldsymbol{\tau}}(\{\langle \overline{\boldsymbol{\theta}}^{(q)}, \cdot\rangle / \sqrt{d_q}\}) \in \mathcal{L}$,

$$\sigma_{\boldsymbol{d},\boldsymbol{\tau}}\left(\{\langle \overline{\boldsymbol{\theta}}_q, \overline{\boldsymbol{x}}^{(q)}\rangle\}_{q \in [Q]}\right) = \sigma\left(\sum_{q \in [Q]} \tau^{(q)}(r_q/R)\langle \overline{\boldsymbol{\theta}}_q, \overline{\boldsymbol{x}}^{(q)}\rangle\right).$$

Define the operator $\mathbb{T}_{\boldsymbol{\tau}} : \mathcal{L} \to \mathcal{L}$, such that for any $g \in \mathcal{L}$,

$$\mathbb{T}_{\boldsymbol{\tau}} g(\overline{\boldsymbol{\theta}}) = \mathbb{E}_{\overline{\boldsymbol{x}}}\left[\sigma_{\boldsymbol{d},\boldsymbol{\tau}}\left(\{\langle \overline{\boldsymbol{\theta}}^{(q)}, \overline{\boldsymbol{x}}^{(q)}\rangle / \sqrt{d_q}\}_{q \in [Q]}\right) g(\overline{\boldsymbol{x}})\right].$$

It is easy to check that the adjoint operator $\mathbb{T}_{\boldsymbol{\tau}}^* : \mathcal{L} \to \mathcal{L}$ verifies $\mathbb{T}^* = \mathbb{T}$ with variables $\overline{\boldsymbol{x}}$ and $\overline{\boldsymbol{\theta}}$ exchanged.

We define the operator $\mathbb{K}_{\boldsymbol{\tau},\boldsymbol{\tau}'} : \mathcal{L} \to \mathcal{L}$ as $\mathbb{K}_{\boldsymbol{\tau},\boldsymbol{\tau}'} \equiv \mathbb{T}_{\boldsymbol{\tau}}\mathbb{T}_{\boldsymbol{\tau}'}^*$. For $g \in \mathcal{L}$, we can write

$$\mathbb{K}_{\boldsymbol{\tau}_1,\boldsymbol{\tau}_2} g(\overline{\boldsymbol{\theta}}_1) = \mathbb{E}_{\overline{\boldsymbol{\theta}}_2}[K_{\boldsymbol{\tau}_1,\boldsymbol{\tau}_2}(\overline{\boldsymbol{\theta}}_1, \overline{\boldsymbol{\theta}}_2)g(\overline{\boldsymbol{\theta}}_2)],$$

where

$$K_{\boldsymbol{\tau}_1,\boldsymbol{\tau}_2}(\overline{\boldsymbol{\theta}}_1, \overline{\boldsymbol{\theta}}_2) = \mathbb{E}_{\overline{\boldsymbol{x}}}\left[\sigma_{\boldsymbol{d},\boldsymbol{\tau}_1}\left(\{\langle \overline{\boldsymbol{\theta}}_1^{(q)}, \overline{\boldsymbol{x}}^{(q)}\rangle / \sqrt{d_q}\}_{q \in [Q]}\right) \sigma_{\boldsymbol{d},\boldsymbol{\tau}_2}\left(\{\langle \overline{\boldsymbol{\theta}}_2^{(q)}, \overline{\boldsymbol{x}}^{(q)}\rangle / \sqrt{d_q}\}_{q \in [Q]}\right)\right].$$

We recall the decomposition of $\sigma_{\boldsymbol{d},\boldsymbol{\tau}}$ in terms of tensor product of Gegenbauer polynomials

$$\sigma_{\boldsymbol{d},\boldsymbol{\tau}}\left(\{\overline{x}_1^{(q)}\}_{q \in [Q]}\right) = \sum_{\boldsymbol{k} \in \mathbb{Z}_{\geq 0}^Q} \lambda_{\boldsymbol{k}}^{\boldsymbol{d}}(\sigma_{\boldsymbol{d},\boldsymbol{\tau}}) B(\boldsymbol{d}, \boldsymbol{k}) Q_{\boldsymbol{k}}^{\boldsymbol{d}}\left(\{\overline{x}_1^{(q)}\}_{q \in [Q]}\right),$$

$$\lambda_{\boldsymbol{k}}^{\boldsymbol{d}}(\sigma_{\boldsymbol{d},\boldsymbol{\tau}}) = \mathbb{E}_{\overline{\boldsymbol{x}}}\left[\sigma_{\boldsymbol{d},\boldsymbol{\tau}}\left(\{\overline{x}_1^{(q)}\}_{q \in [Q]}\right) Q_{\boldsymbol{k}}^{\boldsymbol{d}}\left(\{\sqrt{d_q}\overline{x}_1^{(q)}\}_{q \in [Q]}\right)\right].$$

Recall that $\{Y^d_{k,s}\}_{k\in\mathbb{Z}^Q_{\geq 0}, s\in[B(d,s)]}$ forms an orthonormal basis of $\mathcal{L}$. From Eq. (76), we have for any $k \geq 0$ and $s \in [B(d,k)]$,

$$\mathbb{T}_{\boldsymbol{\tau}} Y^d_{k,s}(\overline{\boldsymbol{\theta}}) = \sum_{k'\in\mathbb{Z}^Q_{\geq 0}} \lambda^d_{k'}(\sigma_{d,\boldsymbol{\tau}}) B(d,k') \mathbb{E}_{\overline{\boldsymbol{x}}}\left[ Q^d_{k'}\left(\{\langle\overline{\boldsymbol{\theta}}^{(q)}, \overline{\boldsymbol{x}}^{(q)}\rangle\}_{q\in[Q]}\right) Y^d_{k,s}(\overline{\boldsymbol{x}})\right]$$

$$= \lambda^d_k(\sigma_{d,\boldsymbol{\tau}}) Y^d_{k,s}(\overline{\boldsymbol{\theta}}),$$

where we used

$$\mathbb{E}_{\overline{\boldsymbol{x}}}\left[ Q^d_{k'}\left(\{\langle\overline{\boldsymbol{\theta}}^{(q)}, \overline{\boldsymbol{x}}^{(q)}\rangle\}_{q\in[Q]}\right) Y^d_{k,s}(\overline{\boldsymbol{x}})\right] = \frac{\delta_{k,k'}}{B(d,k)} Y_{k,s}(\overline{\boldsymbol{\theta}}).$$

The same equation holds for $\mathbb{T}^*_{\boldsymbol{\tau}}$. Therefore, we directly deduce that

$$\mathbb{K}_{\boldsymbol{\tau},\boldsymbol{\tau}'} Y^d_{k,s}(\overline{\boldsymbol{\theta}}) = (\mathbb{T}_{\boldsymbol{\tau}} \mathbb{T}^*_{\boldsymbol{\tau}'}) Y^d_{k,s}(\overline{\boldsymbol{\theta}}) = \lambda^d_k(\sigma_{d,\boldsymbol{\tau}}) \lambda^d_k(\sigma_{d,\boldsymbol{\tau}'}) Y^d_{k,s}(\overline{\boldsymbol{\theta}}).$$

We deduce that $\{Y^d_{k,s}\}_{k\in\mathbb{Z}^Q_{\geq 0}, s\in[B(d,s)]}$ is an orthonormal basis that diagonalizes the operator $\mathbb{K}_{\boldsymbol{\tau},\boldsymbol{\tau}'}$.

Let $\varepsilon_0 > 0$ be defined as in Lemma 10. We will consider $\boldsymbol{\tau}, \boldsymbol{\tau}' \in [1-\varepsilon_0, 1+\varepsilon_0]^Q$ and restrict ourselves to the subspace $V^d_{\mathcal{Q}}$. From the choice of $\varepsilon_0$ and for $d$ large enough, the eigenvalues $\lambda^d_k(\sigma_{d,\boldsymbol{\tau}}) \lambda^d_k(\sigma_{d,\boldsymbol{\tau}'}) \neq 0$ for any $k \in \mathcal{Q}$. Hence, the operator $\mathbb{K}_{\boldsymbol{\tau},\boldsymbol{\tau}'}|_{V^d_{\mathcal{Q}}}$ is invertible.

### F.3 Proof of Theorem 2.(b)

Without loss of generality, let us assume that $\{f_d\}$ are polynomials contained in $V^d_{\mathcal{Q}}$, i.e. $\overline{f}_d = \mathsf{P}_{\mathcal{Q}}\overline{f}_d$. Consider

$$\hat{f}(\boldsymbol{x}; \boldsymbol{\Theta}, \boldsymbol{a}) = \sum_{i=1}^N a_i \sigma(\langle\boldsymbol{\theta}_i, \boldsymbol{x}\rangle/R).$$

Define $\alpha_{\boldsymbol{\tau}}(\overline{\boldsymbol{\theta}}) \equiv \mathbb{K}^{-1}_{\boldsymbol{\tau},\boldsymbol{\tau}} \mathbb{T}_{\boldsymbol{\tau}} \overline{f}_d(\overline{\boldsymbol{\theta}})$ and choose $a^*_i = N^{-1}\alpha_{\boldsymbol{\tau}_i}(\overline{\boldsymbol{\theta}}_i)$, where we denoted $\overline{\boldsymbol{\theta}}_i = (\overline{\boldsymbol{\theta}}^{(q)}_i)_{q\in[Q]}$ with $\overline{\boldsymbol{\theta}}^{(q)}_i = \boldsymbol{\theta}^{(q)}_i/\tau^{(q)}_i \in \mathbb{S}^{d_q-1}(\sqrt{d_q})$ and $\tau^{(q)}_i = \|\boldsymbol{\theta}^{(q)}_i\|_2/\sqrt{d_q}$ independent of $\overline{\boldsymbol{\theta}}^{(q)}_i$.

Let $\varepsilon_0 > 0$ be defined as in Lemma 10 and consider the expectation over $\mathcal{P}_{\varepsilon_0}$ of the RF risk (in particular, $\boldsymbol{a}^* = (a^*_1, \ldots, a^*_N)$ are well defined):

$$\mathbb{E}_{\boldsymbol{\Theta}_{\varepsilon_0}}[R_{\text{RF}}(f_d, \boldsymbol{\Theta})] = \mathbb{E}_{\boldsymbol{\Theta}_{\varepsilon_0}}\left[ \inf_{\boldsymbol{a}\in\mathbb{R}^N} \mathbb{E}_{\boldsymbol{x}}[(f_d(\boldsymbol{x}) - \hat{f}(\boldsymbol{x}; \boldsymbol{\Theta}, \boldsymbol{a}))^2]\right]$$

$$\leq \mathbb{E}_{\boldsymbol{\Theta}_{\varepsilon_0}}\left[ \mathbb{E}_{\boldsymbol{x}}\left[(f_d(\boldsymbol{x}) - \hat{f}(\boldsymbol{x}; \boldsymbol{\Theta}, \boldsymbol{a}^*(\boldsymbol{\Theta})))^2\right]\right].$$

We can expand the squared loss at $\boldsymbol{a}^*$ as

$$\mathbb{E}_{\boldsymbol{x}}[(f_d(\boldsymbol{x}) - \hat{f}(\boldsymbol{x}; \boldsymbol{\Theta}, \boldsymbol{a}^*))^2] = \|f_d\|^2_{L^2} - 2\sum_{i=1}^N \mathbb{E}_{\boldsymbol{x}}[a^*_i \sigma(\langle\boldsymbol{\theta}_i, \boldsymbol{x}\rangle/R) f_d(\boldsymbol{x})]$$

$$+ \sum_{i,j=1}^N \mathbb{E}_{\boldsymbol{x}}[a^*_i a^*_j \sigma(\langle\boldsymbol{\theta}_i, \boldsymbol{x}\rangle/R) \sigma(\langle\boldsymbol{\theta}_j, \boldsymbol{x}\rangle/R)]. \tag{86}$$

The second term of the expansion (86) around $\boldsymbol{a}^*$ verifies

$$\mathbb{E}_{\boldsymbol{\Theta}_{\varepsilon_0}}\left[ \sum_{i=1}^N \mathbb{E}_{\boldsymbol{x}}\left[a^*_i \sigma(\langle\boldsymbol{\theta}_i, \boldsymbol{x}\rangle/R) f_d(\boldsymbol{x})\right]\right]$$

$$= \mathbb{E}_{\boldsymbol{\tau}_{\varepsilon_0}}\left[ \mathbb{E}_{\overline{\boldsymbol{\theta}}}\left[ \alpha_{\boldsymbol{\tau}}(\overline{\boldsymbol{\theta}}) \mathbb{E}_{\overline{\boldsymbol{x}}}\left[ \sigma_{d,\boldsymbol{\tau}}\left(\{\langle\overline{\boldsymbol{\theta}}^{(q)}, \overline{\boldsymbol{x}}^{(q)}\rangle/\sqrt{d_q}\}_{q\in[Q]}\right) \overline{f}_d(\overline{\boldsymbol{x}})\right]\right]\right] \tag{87}$$

$$= \mathbb{E}_{\boldsymbol{\tau}_{\varepsilon_0}}\left[ \langle\mathbb{K}^{-1}_{\boldsymbol{\tau},\boldsymbol{\tau}} \mathbb{T}_{\boldsymbol{\tau}} \overline{f}_d, \mathbb{T}_{\boldsymbol{\tau}} \overline{f}_d\rangle_{L^2}\right]$$

$$= \|f_d\|^2_{L^2},$$

where we used that for each $\boldsymbol{\tau} \in [1-\varepsilon_0, 1+\varepsilon_0]^Q$, we have $\mathbb{T}^*_{\boldsymbol{\tau}} \mathbb{K}^{-1}_{\boldsymbol{\tau},\boldsymbol{\tau}} \mathbb{T}_{\boldsymbol{\tau}}|_{V^d_{\mathcal{Q}}} = \mathbf{I}|_{V^d_{\mathcal{Q}}}$.

Let us consider the third term in the expansion (86) around $\boldsymbol{a}^*$: the non diagonal term verifies

$$\mathbb{E}_{\boldsymbol{\Theta}_{\varepsilon_0}}\left[\sum_{i\neq j}\mathbb{E}_{\boldsymbol{x}}\left[a_i^* a_j^* \sigma(\langle\boldsymbol{\theta}_i,\boldsymbol{x}\rangle/R)\sigma(\langle\boldsymbol{\theta}_j,\boldsymbol{x}\rangle/R)\right]\right]$$

$$=\left(1-N^{-1}\right)\mathbb{E}_{\boldsymbol{\tau}_{\varepsilon_0}^1,\boldsymbol{\tau}_{\varepsilon_0}^2,\overline{\boldsymbol{\theta}}_1,\overline{\boldsymbol{\theta}}_2}\Big[\alpha_{\boldsymbol{\tau}^1}(\overline{\boldsymbol{\theta}}_1)\alpha_{\boldsymbol{\tau}^2}(\overline{\boldsymbol{\theta}}_2)$$

$$\times E_{\overline{\boldsymbol{x}}}\Big[\sigma_{\boldsymbol{d},\boldsymbol{\tau}^1}\left(\{\langle\overline{\boldsymbol{\theta}}_1^{(q)},\overline{\boldsymbol{x}}^{(q)}\rangle/\sqrt{d_q}\}_{q\in[Q]}\right)\sigma_{\boldsymbol{d},\boldsymbol{\tau}^2}\left(\{\langle\overline{\boldsymbol{\theta}}_2^{(q)},\overline{\boldsymbol{x}}^{(q)}\rangle/\sqrt{d_q}\}_{q\in[Q]}\right)\Big]\Big]$$

$$=\left(1-N^{-1}\right)\mathbb{E}_{\boldsymbol{\tau}_{\varepsilon_0}^1,\boldsymbol{\tau}_{\varepsilon_0}^2,\overline{\boldsymbol{\theta}}_1,\overline{\boldsymbol{\theta}}_2}\Big[\mathbb{K}_{\boldsymbol{\tau}^1,\boldsymbol{\tau}^1}^{-1}\mathbb{T}_{\boldsymbol{\tau}^1}\overline{f}_d(\overline{\boldsymbol{\theta}}_1)\mathbb{K}_{\boldsymbol{\tau}^1,\boldsymbol{\tau}^2}(\overline{\boldsymbol{\theta}}_1,\overline{\boldsymbol{\theta}}_2)\mathbb{K}_{\boldsymbol{\tau}^2,\boldsymbol{\tau}^2}^{-1}T_{\boldsymbol{\tau}^2}\overline{f}_d(\overline{\boldsymbol{\theta}}_2)\Big]$$

$$=\left(1-N^{-1}\right)\mathbb{E}_{\boldsymbol{\tau}_{\varepsilon_0}^1,\boldsymbol{\tau}_{\varepsilon_0}^2}\Big[\langle\mathbb{K}_{\boldsymbol{\tau}^1,\boldsymbol{\tau}^1}^{-1}\mathbb{T}_{\boldsymbol{\tau}^1}\overline{f}_d,\mathbb{K}_{\boldsymbol{\tau}^1,\boldsymbol{\tau}^2}\mathbb{K}_{\boldsymbol{\tau}^2,\boldsymbol{\tau}^2}^{-1}\mathbb{T}_{\boldsymbol{\tau}^2}\overline{f}_d\rangle_{L^2}\Big].$$

For $\boldsymbol{k}\in\mathcal{Q}$ and $\boldsymbol{s}\in[B(\boldsymbol{d},\boldsymbol{k})]$ and $\boldsymbol{\tau}^1,\boldsymbol{\tau}^2\in[1-\varepsilon_0,1+\varepsilon_0]^Q$, we have (for $d$ large enough)

$$\mathbb{T}_{\boldsymbol{\tau}^1}^*\mathbb{K}_{\boldsymbol{\tau}^1,\boldsymbol{\tau}^1}^{-1}\mathbb{K}_{\boldsymbol{\tau}^1,\boldsymbol{\tau}^2}\mathbb{K}_{\boldsymbol{\tau}^2,\boldsymbol{\tau}^2}^{-1}\mathbb{T}_{\boldsymbol{\tau}^2}Y_{\boldsymbol{k},\boldsymbol{s}}^{\boldsymbol{d}}=\left(\mathbb{T}_{\boldsymbol{\tau}^1}^*\mathbb{K}_{\boldsymbol{\tau}^1,\boldsymbol{\tau}^1}^{-1}\mathbb{T}_{\boldsymbol{\tau}^1}\right)\cdot\left(\mathbb{T}_{\boldsymbol{\tau}^2}^*\mathbb{K}_{\boldsymbol{\tau}^2,\boldsymbol{\tau}^2}^{-1}\mathbb{T}_{\boldsymbol{\tau}^2}\right)\cdot Y_{\boldsymbol{k},\boldsymbol{s}}^{\boldsymbol{d}}=Y_{\boldsymbol{k},\boldsymbol{s}}^{\boldsymbol{d}}.$$

Hence for any $\boldsymbol{\tau}^1,\boldsymbol{\tau}^2\in[1-\varepsilon_0,1+\varepsilon_0]^Q$, $\mathbb{T}_{\boldsymbol{\tau}^1}^*\mathbb{K}_{\boldsymbol{\tau}^1,\boldsymbol{\tau}^1}^{-1}\mathbb{K}_{\boldsymbol{\tau}^1,\boldsymbol{\tau}^2}\mathbb{K}_{\boldsymbol{\tau}^2,\boldsymbol{\tau}^2}^{-1}\mathbb{T}_{\boldsymbol{\tau}^2}|_{V_{\mathcal{Q}}^{\boldsymbol{d}}}=\mathbf{I}|_{V_{\mathcal{Q}}^{\boldsymbol{d}}}$. Hence

$$\mathbb{E}_{\boldsymbol{\Theta}_{\varepsilon_0}}\left[\sum_{i\neq j}\mathbb{E}_{\boldsymbol{x}}\left[a_i^* a_j^*\sigma(\langle\boldsymbol{\theta}_i,\boldsymbol{x}\rangle/R)\sigma(\langle\boldsymbol{\theta}_j,\boldsymbol{x}\rangle/R)\right]\right]=\left(1-N^{-1}\right)\|f_d\|_{L^2}^2. \qquad (88)$$

The diagonal term verifies

$$\mathbb{E}_{\boldsymbol{\Theta}_{\varepsilon_0}}\left[\sum_{i\in[N]}\mathbb{E}_{\boldsymbol{x}}\left[(a_i^*)^2\sigma(\langle\boldsymbol{\theta}_i,\boldsymbol{x}\rangle/R)^2\right]\right]$$

$$=N^{-1}\mathbb{E}_{\boldsymbol{\tau}_{\varepsilon_0},\overline{\boldsymbol{\theta}}}\left[\alpha_{\boldsymbol{\tau}}(\overline{\boldsymbol{\theta}})^2 K_{\boldsymbol{\tau},\boldsymbol{\tau}}(\overline{\boldsymbol{\theta}},\overline{\boldsymbol{\theta}})\right]$$

$$\leq N^{-1}\left[\max_{\overline{\boldsymbol{\theta}},\boldsymbol{\tau}\in[1-\varepsilon_0,1+\varepsilon_0]^Q}K_{\boldsymbol{\tau},\boldsymbol{\tau}}(\overline{\boldsymbol{\theta}},\overline{\boldsymbol{\theta}})\right]\cdot\mathbb{E}_{\boldsymbol{\tau}_{\varepsilon_0}}[\|\mathbb{K}_{\boldsymbol{\tau},\boldsymbol{\tau}}^{-1}\mathbb{T}_{\boldsymbol{\tau}}\overline{f}_d\|_{L^2}^2].$$

We have by definition of $\mathbb{K}_{\boldsymbol{\tau},\boldsymbol{\tau}}$

$$\sup_{\boldsymbol{\tau}\in[1-\varepsilon_0,1+\varepsilon_0]^Q}K_{\boldsymbol{\tau},\boldsymbol{\tau}}(\overline{\boldsymbol{\theta}},\overline{\boldsymbol{\theta}})=\sup_{\boldsymbol{\tau}\in[1-\varepsilon_0,1+\varepsilon_0]^Q}\|\sigma_{\boldsymbol{d},\boldsymbol{\tau}}\|_{L^2}^2\leq C,$$

for $d$ large enough (using Lemma 10). Furthermore

$$\|\mathbb{K}_{\boldsymbol{\tau},\boldsymbol{\tau}}^{-1}\mathbb{T}_{\boldsymbol{\tau}}\overline{f}_d\|_{L^2}^2=\sum_{\boldsymbol{k}\in\mathcal{Q}}\frac{1}{\lambda_{\boldsymbol{k}}^{\boldsymbol{d}}(\sigma_{\boldsymbol{d},\boldsymbol{\tau}})^2}\sum_{\boldsymbol{s}\in[B(\boldsymbol{d},\boldsymbol{k})]}\lambda_{\boldsymbol{k},\boldsymbol{s}}^{\boldsymbol{d}}(\overline{f}_d)^2$$

$$\leq\left[\max_{\boldsymbol{k}\in\mathcal{Q}}\frac{1}{\lambda_{\boldsymbol{k}}^{\boldsymbol{d}}(\sigma_{\boldsymbol{d},\boldsymbol{\tau}})^2}\right]\cdot\|\mathsf{P}_{\mathcal{Q}}f_d\|_{L^2}^2.$$

From Lemma 10, we get

$$\mathbb{E}_{\boldsymbol{\tau}_{\varepsilon_0}}[\|\mathbb{K}_{\boldsymbol{\tau},\boldsymbol{\tau}}^{-1}\mathbb{T}_{\boldsymbol{\tau}}\overline{f}_d\|_{L^2}^2]\leq Cd^\gamma\cdot\|\mathsf{P}_{\mathcal{Q}}f_d\|_{L^2}^2.$$

Hence,

$$\mathbb{E}_{\boldsymbol{\Theta}_{\varepsilon_0}}\left[\sum_{i\in[N]}\mathbb{E}_{\boldsymbol{x}}\left[(a_i^*)^2\sigma(\langle\boldsymbol{\theta}_i,\boldsymbol{x}\rangle/R)^2\right]\right]\leq C\frac{d^\gamma}{N}\|\mathsf{P}_{\mathcal{Q}}f_d\|_{L^2}^2. \qquad (89)$$

Combining Eq. (87), Eq. (88) and Eq. (89), we get

$$\mathbb{E}_{\boldsymbol{\Theta}_{\varepsilon_0}}[R_{\mathrm{RF}}(f_d,\boldsymbol{\Theta})]$$

$$\leq\mathbb{E}_{\boldsymbol{\Theta}_{\varepsilon_0}}\left[\mathbb{E}_{\boldsymbol{x}}\left[\left(f_d(\boldsymbol{x})-\hat{f}(\boldsymbol{x};\boldsymbol{\Theta},\boldsymbol{a}^*(\boldsymbol{\Theta}))\right)^2\right]\right]$$

$$=\|f_d\|_{L^2}^2-2\|f_d\|_{L^2}^2+(1-N^{-1})\|f_d\|_{L^2}^2+N^{-1}\mathbb{E}_{\boldsymbol{\tau}_{\varepsilon_0},\overline{\boldsymbol{\theta}}}\left[(\alpha_{\boldsymbol{\tau}}(\overline{\boldsymbol{\theta}}))^2 K_{\boldsymbol{\tau},\boldsymbol{\tau}}(\overline{\boldsymbol{\theta}},\overline{\boldsymbol{\theta}})\right]$$

$$\leq C\frac{d^\gamma}{N}\|\mathsf{P}_{\mathcal{Q}}f_d\|_{L^2}^2.$$

By Markov's inequality, we get for any $\varepsilon > 0$ and $d$ large enough,

$$\mathbb{P}(R_{\mathrm{RF}}(f_d, \mathbf{\Theta}) > \varepsilon \cdot \|f_d\|_{L^2}^2) \leq \mathbb{P}(\{R_{\mathrm{RF}}(f_d, \mathbf{\Theta}) > \varepsilon \cdot \|f_d\|_{L^2}^2\} \cap \mathcal{P}_{\varepsilon_0}) + \mathbb{P}(\mathcal{P}_{\varepsilon_0}^c) \leq C' \frac{d^\gamma}{N} + \mathbb{P}(\mathcal{P}_{\varepsilon_0}^c).$$

The assumption $N = \omega_d(d^\gamma)$ and Lemma 8 conclude the proof.

# G Proof of Theorem 3.(a): lower bound for NT model

## G.1 Preliminaries

We consider the activation function $\sigma : \mathbb{R} \to \mathbb{R}$ with weak derivative $\sigma'$. Consider $\sigma'_{\boldsymbol{d},\boldsymbol{\tau}} : \mathrm{ps}^{\boldsymbol{d}} \to \mathbb{R}$ defined as follows

$$
\begin{aligned}
\sigma'(\langle \boldsymbol{\theta}, \boldsymbol{x} \rangle / R) =& \sigma' \left( \sum_{q \in [Q]} \tau^{(q)} \cdot (r_q/R) \cdot \langle \overline{\boldsymbol{\theta}}^{(q)}, \overline{\boldsymbol{x}}^{(q)} \rangle / \sqrt{d_q} \right) \\
\equiv& \sigma'_{\boldsymbol{d},\boldsymbol{\tau}} \left( \{ \langle \overline{\boldsymbol{\theta}}^{(q)}, \overline{\boldsymbol{x}}^{(q)} \rangle / \sqrt{d_q} \}_{q \in [Q]} \right).
\end{aligned}
\tag{90}
$$

Consider the expansion of $\sigma'_{\boldsymbol{d},\boldsymbol{\tau}}$ in terms of product of Gegenbauer polynomials. We have

$$
\sigma'(\langle \boldsymbol{\theta}, \boldsymbol{x} \rangle / R) = \sum_{\boldsymbol{k} \in \mathbb{Z}_{\geq 0}^Q} \lambda_{\boldsymbol{k}}^{\boldsymbol{d}}(\sigma'_{\boldsymbol{d},\boldsymbol{\tau}}) B(\boldsymbol{d},\boldsymbol{k}) Q_{\boldsymbol{k}}^{\boldsymbol{d}} \left( \{ \langle \overline{\boldsymbol{\theta}}^{(q)}, \overline{\boldsymbol{x}}^{(q)} \rangle \}_{q \in [Q]} \right),
\tag{91}
$$

where

$$
\lambda_{\boldsymbol{k}}^{\boldsymbol{d}}(\sigma'_{\boldsymbol{d},\boldsymbol{\tau}}) = \mathbb{E}_{\overline{\boldsymbol{x}}} \left[ \sigma'_{\boldsymbol{d},\boldsymbol{\tau}} \left( \overline{x}_1^{(1)}, \dots, \overline{x}_1^{(Q)} \right) Q_{\boldsymbol{k}}^{\boldsymbol{d}} \left( \sqrt{d_1} \overline{x}_1^{(1)}, \dots, \sqrt{d_Q} \overline{x}_1^{(Q)} \right) \right],
$$

where the expectation is taken over $\overline{\boldsymbol{x}} = (\overline{\boldsymbol{x}}^{(1)}, \dots, \overline{\boldsymbol{x}}^{(Q)}) \sim \mu_{\boldsymbol{d}}$.

**Lemma 11.** *Let $\sigma$ be an activation function that satisfies Assumptions 3.(a) and 3.(b). Define for $\boldsymbol{k} \in \mathbb{Z}_{\geq 0}^Q$ and $\boldsymbol{\tau} \in \mathbb{R}_{\geq 0}^Q$,*

$$
A_{\boldsymbol{\tau},\boldsymbol{k}}^{(q)} = r_q^2 \cdot [t_{d_q, k_q - 1} \lambda_{\boldsymbol{k}_{q-}}^{\boldsymbol{d}}(\sigma'_{\boldsymbol{d},\boldsymbol{\tau}})^2 B(\boldsymbol{d}, \boldsymbol{k}_{q-}) + s_{d_q, k_q + 1} \lambda_{\boldsymbol{k}_{q+}}^{\boldsymbol{d}}(\sigma'_{\boldsymbol{d},\boldsymbol{\tau}})^2 B(\boldsymbol{d}, \boldsymbol{k}_{q+})],
\tag{92}
$$

*with $\boldsymbol{k}_{q+} = (k_1, \dots, k_q + 1, \dots, k_Q)$ and $\boldsymbol{k}_{q-} = (k_1, \dots, k_q - 1, \dots, k_Q)$, and*

$$
s_{d,k} = \frac{k}{2k + d - 2}, \qquad t_{d,k} = \frac{k + d - 2}{2k + d - 2},
$$

*with the convention $t_{d,-1} = 0$. Then there exists constants $\varepsilon_0 > 0$ and $C > 0$ such that for $d$ large enough, we have for any $\boldsymbol{\tau} \in [1 - \varepsilon_0, 1 + \varepsilon_0]^Q$ and $\boldsymbol{k} \in \mathcal{Q}_{\mathrm{NT}}(\gamma)^c$,*

$$
\frac{A_{\boldsymbol{\tau},\boldsymbol{k}}^{(q)}}{B(\boldsymbol{d},\boldsymbol{k})} \leq \begin{cases} C d^\xi d^{-\gamma - (\xi - \min_{q \in S(\boldsymbol{k})} \kappa_q)} & \text{if } k_q > 0, \\ C d^{\eta_q + 2\kappa_q - \xi} d^{-\gamma - (\xi - \min_{q \in S(\boldsymbol{k})} \kappa_q)} & \text{if } k_q = 0, \end{cases}
$$

*where we recall $S(\boldsymbol{k}) \subset [Q]$ is the subset of indices corresponding to the non zero integers $k_q > 0$.*

*Proof of Lemma 11.* Let us fix an integer $M$ such that $\mathcal{Q} \subset [M]^Q$. We will denote $\mathcal{Q} \equiv \mathcal{Q}_{\mathrm{NT}}(\gamma)$ for simplicity. Following the same proof as in Lemma 9, there exists $\varepsilon_0 > 0$, $d_0$ and $C > 0$ such that for any $d \geq d_0$ and $\boldsymbol{\tau} \in [1 - \varepsilon_0, 1 + \varepsilon_0]^Q$, we have for any $\boldsymbol{k} \in \mathcal{Q}^c \cap [M]^Q$,

$$
\begin{aligned}
\lambda_{\boldsymbol{k}}^{\boldsymbol{d}}(\sigma'_{\boldsymbol{d},\boldsymbol{\tau}})^2 \leq& C d^{-\gamma - (\xi - \min_{q \in S(\boldsymbol{k})} \kappa_q)}, \\
\lambda_{\boldsymbol{k}_{q-}}^{\boldsymbol{d}}(\sigma'_{\boldsymbol{d},\boldsymbol{\tau}})^2 \leq& C d^{\xi - \kappa_q - \gamma - (\xi - \min_{q \in S(\boldsymbol{k})} \kappa_q)}, \\
\lambda_{\boldsymbol{k}_{q+}}^{\boldsymbol{d}}(\sigma'_{\boldsymbol{d},\boldsymbol{\tau}})^2 \leq& C d^{\kappa_q - \xi - \gamma - (\xi - \min_{q \in S(\boldsymbol{k})} \kappa_q)},
\end{aligned}
$$

while for $\boldsymbol{k} \notin [M]^Q$, we get

$$
\max\{\lambda_{\boldsymbol{k}}^{\boldsymbol{d}}(\sigma'_{\boldsymbol{d},\boldsymbol{\tau}})^2, \lambda_{\boldsymbol{k}_{q-}}^{\boldsymbol{d}}(\sigma'_{\boldsymbol{d},\boldsymbol{\tau}})^2, \lambda_{\boldsymbol{k}_{q+}}^{\boldsymbol{d}}(\sigma'_{\boldsymbol{d},\boldsymbol{\tau}})^2\} \leq C d^{-(M-1) \min_{q \in [Q]} \eta_q}.
$$

Injecting this bound in the formula (92) of $A_{\boldsymbol{\tau},\boldsymbol{k}}^{(q)}$, we get for $d \geq d_0$, $\boldsymbol{\tau} \in [1 - \varepsilon_0, 1 + \varepsilon_0]^Q$ and any $\boldsymbol{k} \in \mathcal{Q}^c \cap [M]^Q$: if $k_q > 0$,

$$
\frac{A_{\boldsymbol{\tau},\boldsymbol{k}}^{(q)}}{B(\boldsymbol{d},\boldsymbol{k})} \leq C' d^{\eta_q + \kappa_q} d^{\xi - \kappa_q - \gamma - (\xi - \min_{q \in S(\boldsymbol{k})} \kappa_q)} = C' d^\xi d^{-\gamma - (\xi - \min_{q \in S(\boldsymbol{k})} \kappa_q)},
$$

while for $k_q = 0$,

$$\frac{A^{(q)}_{\boldsymbol{\tau},\boldsymbol{k}}}{B(\boldsymbol{d},\boldsymbol{k})} \leq C' d^{\eta_q + \kappa_q} d^{-\eta_q} d^{\kappa_q + \eta_q - \xi - \gamma - (\xi - \min_{q \in S(\boldsymbol{k})} \kappa_q)} = C' d^{\eta_q + 2\kappa_q - \xi} d^{-\gamma - (\xi - \min_{q \in S(\boldsymbol{k})} \kappa_q)},$$

where we used that for $k_q \in [M]$, there exists a constant $c > 0$ such that $s_{d_q, k_q} \leq c d^{-\eta_q}$ and $t_{d_q, k_q} \leq c$. Similarly, we get for $\boldsymbol{k} \notin [M]^Q$

$$\frac{A^{(q)}_{\boldsymbol{\tau},\boldsymbol{k}}}{B(\boldsymbol{d},\boldsymbol{k})} \leq C'' d^{\kappa_q + \eta_q - (M-1)\min_{q \in [Q]} \eta_q},$$

where we used that $s_{d_q, k}, t_{d_q, k} \leq 1$ for any $k \in \mathbb{Z}_{\geq 0}$. Taking $M$ sufficiently large yields the result.

$\square$

## G.2    Proof of Theorem 3.(a): Outline

The structure of the proof for the NT model is the same as for the RF case, however some parts of the proof requires more work.

We define the random vector $\boldsymbol{V} = (\boldsymbol{V}_1, \dots, \boldsymbol{V}_N)^\mathsf{T} \in \mathbb{R}^{Nd}$, where, for each $j \leq N$, $\boldsymbol{V}_j \in \mathbb{R}^D$, and analogously $\boldsymbol{V}_\mathcal{Q} = (\boldsymbol{V}_{1,\mathcal{Q}}, \dots, \boldsymbol{V}_{N,\mathcal{Q}})^\mathsf{T} \in \mathbb{R}^{ND}$, $\boldsymbol{V}_{\mathcal{Q}^c} = (\boldsymbol{V}_{1,\mathcal{Q}^c}, \dots, \boldsymbol{V}_{N,\mathcal{Q}^c})^\mathsf{T} \in \mathbb{R}^{ND}$, as follows

$$\boldsymbol{V}_{i,\mathcal{Q}} = \mathbb{E}_{\boldsymbol{x}}[[\mathsf{P}_\mathcal{Q} f_d](\boldsymbol{x}) \sigma'(\langle \boldsymbol{\theta}_i, \boldsymbol{x} \rangle / R)\boldsymbol{x}],$$
$$\boldsymbol{V}_{i,\mathcal{Q}^c} = \mathbb{E}_{\boldsymbol{x}}[[\mathsf{P}_{\mathcal{Q}^c} f_d](\boldsymbol{x}) \sigma'(\langle \boldsymbol{\theta}_i, \boldsymbol{x} \rangle / R)\boldsymbol{x}],$$
$$\boldsymbol{V}_i = \mathbb{E}_{\boldsymbol{x}}[f_d(\boldsymbol{x}) \sigma'(\langle \boldsymbol{\theta}_i, \boldsymbol{x} \rangle / R)\boldsymbol{x}] = \boldsymbol{V}_{i,\mathcal{Q}} + \boldsymbol{V}_{i,\mathcal{Q}^c}.$$

We define the random matrix $\boldsymbol{U} = (\boldsymbol{U}_{ij})_{i,j \in [N]} \in \mathbb{R}^{ND \times ND}$, where for each $i, j \leq N$, $\boldsymbol{U}_{ij} \in \mathbb{R}^{D \times D}$, is given by

$$\boldsymbol{U}_{ij} = \mathbb{E}_{\boldsymbol{x}}[\sigma'(\langle \boldsymbol{x}, \boldsymbol{\theta}_i \rangle / R)\sigma'(\langle \boldsymbol{x}, \boldsymbol{\theta}_j \rangle / R)\boldsymbol{x}\boldsymbol{x}^\mathsf{T}]. \tag{93}$$

Proceeding as for the RF model, we obtain

$$\left| R_{\mathrm{NT}}(f_d) - R_{\mathrm{NT}}(\mathsf{P}_\mathcal{Q} f_d) - \|\mathsf{P}_{\mathcal{Q}^c} f_d\|^2_{L^2} \right|$$
$$= \left| \boldsymbol{V}_\mathcal{Q}^\mathsf{T} \boldsymbol{U}^{-1} \boldsymbol{V}_\mathcal{Q} - \boldsymbol{V}^\mathsf{T} \boldsymbol{U}^{-1} \boldsymbol{V} \right| = \left| \boldsymbol{V}_\mathcal{Q}^\mathsf{T} \boldsymbol{U}^{-1} \boldsymbol{V}_\mathcal{Q} - (\boldsymbol{V}_\mathcal{Q} + \boldsymbol{V}_{\mathcal{Q}^c})^\mathsf{T} \boldsymbol{U}^{-1} (\boldsymbol{V}_\mathcal{Q} + \boldsymbol{V}_{\mathcal{Q}^c}) \right|$$
$$= \left| 2\boldsymbol{V}^\mathsf{T} \boldsymbol{U}^{-1} \boldsymbol{V}_{\mathcal{Q}^c} - \boldsymbol{V}_{\mathcal{Q}^c}^\mathsf{T} \boldsymbol{U}^{-1} \boldsymbol{V}_{\mathcal{Q}^c} \right|$$
$$\leq 2\|\boldsymbol{U}^{-1/2} \boldsymbol{V}_{\mathcal{Q}^c}\|_2 \|f_d\|_{L^2} + \boldsymbol{V}_{\mathcal{Q}^c}^\mathsf{T} \boldsymbol{U}^{-1} \boldsymbol{V}_{\mathcal{Q}^c}.$$

We claim that we have

$$\|\boldsymbol{U}^{-1/2} \boldsymbol{V}_{\mathcal{Q}^c}\|_2^2 = \boldsymbol{V}_{\mathcal{Q}^c}^\mathsf{T} \boldsymbol{U}^{-1} \boldsymbol{V}_{\mathcal{Q}^c} = o_{d,\mathbb{P}}(\|\mathsf{P}_{\mathcal{Q}^c} f_d\|^2_{L^2}), . \tag{94}$$

To show this result, we will need the following two propositions.

**Proposition 3** (Expected norm of $\boldsymbol{V}$). *Let $\sigma$ be a weakly differentiable activation function with weak derivative $\sigma'$ and $\mathcal{Q} \subset \mathbb{Z}_{\geq 0}^Q$. Let $\varepsilon > 0$ and define $\mathcal{E}^{(q)}_{\mathcal{Q}^c, \varepsilon}$ by*

$$\mathcal{E}^{(q)}_{\mathcal{Q}^c, \varepsilon} \equiv \mathbb{E}_{\boldsymbol{\theta}_\varepsilon} \left[ \langle \mathbb{E}_{\boldsymbol{x}}[[\mathsf{P}_{\mathcal{Q}^c} f_d](\boldsymbol{x}) \sigma'(\langle \boldsymbol{\theta}, \boldsymbol{x} \rangle / R)\boldsymbol{x}^{(q)}], \mathbb{E}_{\boldsymbol{x}}[[\mathsf{P}_{\mathcal{Q}^c} f_d](\boldsymbol{x}) \sigma'(\langle \boldsymbol{\theta}, \boldsymbol{x} \rangle / R)\boldsymbol{x}^{(q)}] \rangle \right],$$

*where the expectation is taken with respect to $\boldsymbol{x} = (\boldsymbol{x}^{(1)}, \dots, \boldsymbol{x}^{(Q)}) \sim \mu_{\boldsymbol{d}}^{\boldsymbol{\kappa}}$. Then,*

$$\mathcal{E}^{(q)}_{\mathcal{Q}^c, \varepsilon_0} \leq \left[ \max_{\boldsymbol{k} \in \mathcal{Q}^c} B(\boldsymbol{d}, \boldsymbol{k})^{-1} \mathbb{E}_{\boldsymbol{\tau}_{\varepsilon_0}}[A^{(q)}_{\boldsymbol{\tau}, \boldsymbol{k}}] \right] \cdot \|\mathsf{P}_{\mathcal{Q}^c} f_d\|^2_{L^2}.$$

**Proposition 4** (Lower bound on the kernel matrix). *Let $N = o_d(d^\gamma)$ for some $\gamma > 0$, and $(\boldsymbol{\theta}_i)_{i \in [N]} \sim \mathrm{Unif}(\mathbb{S}^{D-1}(\sqrt{D}))$ independently. Let $\sigma$ be an activation that satisfies Assumptions 3.(a) and 3.(b). Let $\boldsymbol{U} \in \mathbb{R}^{ND \times ND}$ be the kernel matrix with $i, j$ block $\boldsymbol{U}_{ij} \in \mathbb{R}^{D \times D}$ defined by Eq. (93). Then there exists two matrices $\boldsymbol{D}$ and $\boldsymbol{\Delta}$ such that*

$$\boldsymbol{U} \succeq \boldsymbol{D} + \boldsymbol{\Delta},$$

*with $\boldsymbol{D} = \mathrm{diag}(\boldsymbol{D}_{ii})$ block diagonal. Furthermore, $\boldsymbol{D}$ and $\boldsymbol{\Delta}$ verifies the following properties:*

(a) $\|\boldsymbol{\Delta}\|_{\mathrm{op}} = o_{d,\mathbb{P}}(d^{-\max_{q\in[Q]}\kappa_q})$

(b) *For each $i \in [N]$, we can decompose the matrix $\boldsymbol{D}_{ii}$ into block matrix form $(\boldsymbol{D}_{ii}^{qq'})_{q,q'\in[Q]} \in \mathbb{R}^{DN\times DN}$ with $\boldsymbol{D}_{ii}^{qq'} \in \mathbb{R}^{d_q N \times d_{q'} N}$ such that*

- *For any $q \in [Q]$, there exists constants $c_q, C_q > 0$ such that we have with high probability*

$$0 < c_q \frac{r_q^2}{d_q} = c_q d^{\kappa_q} \leq \min_{i\in[N]} \lambda_{\min}(\boldsymbol{D}_{ii}^{qq}) \leq \max_{i\in[N]} \lambda_{\max}(\boldsymbol{D}_{ii}^{qq}) \leq C_q \frac{r_q^2}{d_q} = C_q d^{\kappa_q} < \infty, \tag{95}$$

  *as $d \to \infty$.*
- *For any $q \neq q' \in [Q]$, we have*

$$\max_{i\in[N]} \sigma_{\max}(\boldsymbol{D}_{ii}^{qq'}) = o_{d,\mathbb{P}}(r_q r_{q'}/\sqrt{d_q d_{q'}}). \tag{96}$$

The proofs of these two propositions are provided in the next sections.

From Proposition 4, we can upper bound Eq. (94) as follows

$$\boldsymbol{V}_{\mathcal{Q}^c}^{\mathsf{T}} \boldsymbol{U}^{-1} \boldsymbol{V}_{\mathcal{Q}^c} \preceq \boldsymbol{V}_{\mathcal{Q}^c}^{\mathsf{T}} (\boldsymbol{D}+\boldsymbol{\Delta})^{-1} \boldsymbol{V}_{\mathcal{Q}^c} = \boldsymbol{V}_{\mathcal{Q}^c}^{\mathsf{T}} \boldsymbol{D}^{-1} \boldsymbol{V}_{\mathcal{Q}^c} - \boldsymbol{V}_{\mathcal{Q}^c}^{\mathsf{T}} \boldsymbol{D}^{-1} \boldsymbol{\Delta} (\boldsymbol{D}+\boldsymbol{\Delta})^{-1} \boldsymbol{V}_{\mathcal{Q}^c}. \tag{97}$$

Let us fix $\varepsilon_0 > 0$ as prescribed in Lemma 11. We decompose the vector $\boldsymbol{V}_{i,\mathcal{Q}^c} = (\boldsymbol{V}_{i,\mathcal{Q}^c}^{(q)})_{q\in[Q]}$ where

$$\boldsymbol{V}_{i,\mathcal{Q}^c}^{(q)} = \mathbb{E}_{\boldsymbol{x}}[[\mathsf{P}_{\mathcal{Q}^c} f_d](\boldsymbol{x})\sigma'(\langle\boldsymbol{\theta}_i,\boldsymbol{x}\rangle/R)\boldsymbol{x}^{(q)}].$$

We denote $\boldsymbol{V}_{\mathcal{Q}^c}^{(q)} = (\boldsymbol{V}_{1,\mathcal{Q}^c}^{(q)}, \ldots, \boldsymbol{V}_{N,\mathcal{Q}^c}^{(q)}) \in \mathbb{R}^{d_q N}$. From Proposition 3, we have

$$\frac{d_q}{r_q^2} \mathbb{E}_{\boldsymbol{\Theta}_{\varepsilon_0}}[\|\boldsymbol{V}_{\mathcal{Q}^c}^{(q)}\|_2^2] \leq \left[\max_{\boldsymbol{k}\in\mathcal{Q}^c} Nd^{-\kappa_q} B(\boldsymbol{d},\boldsymbol{k})^{-1} \mathbb{E}_{\boldsymbol{\tau}_{\varepsilon_0}}[A_{\boldsymbol{\tau},\boldsymbol{k}}^{(q)}]\right] \cdot \|\mathsf{P}_{\mathcal{Q}^c} f_d\|_{L^2}^2.$$

Hence, using the upper bounds on $A_{\boldsymbol{\tau},\boldsymbol{k}}^{(q)}$ in Lemma 11, we get for $\boldsymbol{k} \in \mathcal{Q}^c$ with $k_q > 0$:

$$Nd^{-\kappa_q} B(\boldsymbol{d},\boldsymbol{k})^{-1} \mathbb{E}_{\boldsymbol{\tau}_{\varepsilon_0}}[A_{\boldsymbol{\tau},\boldsymbol{k}}^{(q)}] \leq CNd^{-\kappa_q} d^{-\gamma+\min_{q\in S(\boldsymbol{k})}\kappa_q} = o_d(1),$$

where we used that $N = o_d(d^\gamma)$ and $\kappa_q \geq \min_{q\in S(\boldsymbol{k})}\kappa_q$ (we have $k_q > 0$ and therefore $q \in S(\boldsymbol{k})$ by definition). Similarly for $\boldsymbol{k} \in \mathcal{Q}^c$ with $k_q = 0$:

$$Nd^{-\kappa_q} B(\boldsymbol{d},\boldsymbol{k})^{-1} \mathbb{E}_{\boldsymbol{\tau}_{\varepsilon_0}}[A_{\boldsymbol{\tau},\boldsymbol{k}}^{(q)}] \leq CNd^{\eta_q+\kappa_q-\xi} d^{-\gamma-(\xi-\min_{q\in S(\boldsymbol{k})}\kappa_q)} = o_d(1),$$

where we used that by definition of $\xi$ we have $\eta_q + \kappa_q \leq \xi$ and $\min_{q\in S(\boldsymbol{k})}\kappa_q \leq \xi$. We deduce that

$$\frac{d_q}{r_q^2} \mathbb{E}_{\boldsymbol{\Theta}_{\varepsilon_0}}[\|\boldsymbol{V}_{\mathcal{Q}^c}^{(q)}\|_2^2] = o_d(1) \cdot \|\mathsf{P}_{\mathcal{Q}^c} f_d\|_{L^2}^2,$$

and therefore by Markov's inequality that

$$\frac{d_q}{r_q^2} \|\boldsymbol{V}_{\mathcal{Q}^c}^{(q)}\|_2^2 = o_{d,\mathbb{P}}(1) \cdot \|\mathsf{P}_{\mathcal{Q}^c} f_d\|_{L^2}^2. \tag{98}$$

Notice that the properties (95) and (96) imply that there exists $c > 0$ such that with high probability $\lambda_{\min}(\boldsymbol{D}) \geq \min_{i\in[N]} \lambda_{\min}(\boldsymbol{D}_{ii}) \geq c$. In particular, we deduce that $\|(\boldsymbol{D}+\boldsymbol{\Delta})^{-1}\|_{\mathrm{op}} \leq c^{-1}/2$ with high probability. Combining these bounds and Eq. (98) and recalling that $\|\boldsymbol{\Delta}\|_{\mathrm{op}} = o_{d,\mathbb{P}}(d^{-\max_{q\in[Q]}\kappa_q})$ show that

$$|\boldsymbol{V}_{\mathcal{Q}^c}^{\mathsf{T}} \boldsymbol{D}^{-1} \boldsymbol{\Delta} (\boldsymbol{D}+\boldsymbol{\Delta})^{-1} \boldsymbol{V}_{\mathcal{Q}^c}| \leq \|\boldsymbol{D}^{-1}\|_{\mathrm{op}} \|(\boldsymbol{D}+\boldsymbol{\Delta})^{-1}\|_{\mathrm{op}} \sum_{q\in[Q]} \|\boldsymbol{\Delta}\|_{\mathrm{op}} \|\boldsymbol{V}_{\mathcal{Q}^c}^{(q)}\|_2^2 \tag{99}$$

$$= o_{d,\mathbb{P}}(1) \cdot \|\mathsf{P}_{\mathcal{Q}^c} f_d\|_{L^2}^2.$$

We are now left to show $\boldsymbol{V}_{\mathcal{Q}^c}^\mathsf{T}\boldsymbol{D}^{-1}\boldsymbol{V}_{\mathcal{Q}^c} = o_{d,\mathbb{P}}(1)$. For each $i \in [N]$, denote $\boldsymbol{B}_{ii} = \boldsymbol{D}_{ii}^{-1}$ and notice that we can apply Lemma 24 to $\boldsymbol{B}_{ii}$ and get

$$\max_{i\in[N]}\|\boldsymbol{B}_{ii}^{qq}\|_{\mathrm{op}} = O_{d,\mathbb{P}}\left(\frac{d_q}{r_q^2}\right), \qquad \max_{i\in[N]}\|\boldsymbol{B}_{ii}^{qq'}\|_{\mathrm{op}} = o_{d,\mathbb{P}}\left(\frac{\sqrt{d_qd_{q'}}}{r_qr_{q'}}\right).$$

Therefore,

$$\boldsymbol{V}_{\mathcal{Q}^c}^\mathsf{T}\boldsymbol{D}^{-1}\boldsymbol{V}_{\mathcal{Q}^c} = \sum_{i\in[N]}\sum_{q,q'\in[Q]}(\boldsymbol{V}_{i,\mathcal{Q}^c}^{(q)})^\mathsf{T}\boldsymbol{B}_{ii}^{qq'}\boldsymbol{V}_{i,\mathcal{Q}^c}^{(q')}$$

$$\leq \sum_{q,q'\in[Q]}O_{d,\mathbb{P}}(1)\cdot\left(\frac{d_q}{r_q^2}\|\boldsymbol{V}_{\mathcal{Q}^c}^{(q)}\|_2^2\right)^{1/2}\left(\frac{d_{q'}}{r_{q'}^2}\|\boldsymbol{V}_{\mathcal{Q}^c}^{(q')}\|_2^2\right)^{1/2}. \tag{100}$$

Using Eq. (98) in Eq. (100), we get

$$\boldsymbol{V}_{\mathcal{Q}^c}^\mathsf{T}\boldsymbol{D}^{-1}\boldsymbol{V}_{\mathcal{Q}^c} = o_{d,\mathbb{P}}(1)\cdot\|\mathsf{P}_{\mathcal{Q}^c}f_d\|_{L^2}^2. \tag{101}$$

Combining Eq. (99) and Eq. (101) yields Eq. (94). This proves the theorem.

### G.3 Proof of Proposition 3

*Proof of Proposition 3.* Let us consider $\varepsilon_0 > 0$ as prescribed in Lemma 11. We have for $q \in [Q]$

$$\mathcal{E}_{\mathcal{Q}^c,\varepsilon_0}^{(q)} = \mathbb{E}_{\boldsymbol{\theta}_\varepsilon}\left[\langle\mathbb{E}_{\boldsymbol{x}}[[\mathsf{P}_{\mathcal{Q}^c}f_d](\boldsymbol{x})\sigma'(\langle\boldsymbol{\theta},\boldsymbol{x}\rangle/R)\boldsymbol{x}^{(q)}], \mathbb{E}_{\boldsymbol{y}}[[\mathsf{P}_{\mathcal{Q}^c}f_d](\boldsymbol{y})\sigma'(\langle\boldsymbol{\theta},\boldsymbol{y}\rangle/R)\boldsymbol{y}^{(q)}]\rangle\right]$$

$$= \mathbb{E}_{\boldsymbol{\tau}_{\varepsilon_0}}\left[\mathbb{E}_{\overline{\boldsymbol{x}},\overline{\boldsymbol{y}}}\left[[\mathsf{P}_{\mathcal{Q}^c}\overline{f}_d](\overline{\boldsymbol{x}})[\mathsf{P}_{\mathcal{Q}^c}\overline{f}](\overline{\boldsymbol{y}})H_{\boldsymbol{\tau}}^{(q)}(\overline{\boldsymbol{x}},\overline{\boldsymbol{y}})\right]\right],$$

where we denoted $H_{\boldsymbol{\tau}}^{(q)}$ the kernel given by

$$H_{\boldsymbol{\tau}}^{(q)}(\overline{\boldsymbol{x}},\overline{\boldsymbol{y}})$$
$$= \mathbb{E}_{\overline{\boldsymbol{\theta}}}\left[\sigma'_{\boldsymbol{d},\boldsymbol{\tau}}\left(\{\langle\overline{\boldsymbol{\theta}}^{(q)},\overline{\boldsymbol{x}}^{(q)}\rangle/\sqrt{d_q}\}_{q\in[Q]}\right)\sigma'_{\boldsymbol{d},\boldsymbol{\tau}}\left(\{\langle\overline{\boldsymbol{\theta}}^{(q)},\overline{\boldsymbol{y}}^{(q)}\rangle/\sqrt{d_q}\}_{q\in[Q]}\right)\right]\langle\boldsymbol{x}^{(q)},\boldsymbol{y}^{(q)}\rangle,$$

Then we have

$$H_{\boldsymbol{\tau}}^{(q)}(\overline{\boldsymbol{x}},\overline{\boldsymbol{y}}) = \sum_{\boldsymbol{k}\in\mathbb{Z}_{\geq0}^Q}A_{\boldsymbol{\tau},\boldsymbol{k}}^{(q)}Q_{\boldsymbol{k}}^{\boldsymbol{d}}\left(\{\langle\overline{\boldsymbol{x}}^{(q)},\overline{\boldsymbol{y}}^{(q)}\rangle\}_{q\in[Q]}\right), \tag{102}$$

where $A_{\boldsymbol{\tau},\boldsymbol{k}}^{(q)}$ is given in Lemma 12. Hence we get

$$\mathcal{E}_{\mathcal{Q}^c,\varepsilon_0}^{(q)} = \mathbb{E}_{\boldsymbol{\tau}_{\varepsilon_0}}[\mathbb{E}_{\overline{\boldsymbol{x}},\overline{\boldsymbol{y}}}[P_{\mathcal{Q}^c}\overline{f}_d(\overline{\boldsymbol{x}})P_{\mathcal{Q}^c}\overline{f}_d(\overline{\boldsymbol{y}})H_{\boldsymbol{\tau}}^{(q)}(\overline{\boldsymbol{x}},\overline{\boldsymbol{y}})]]$$

$$= \sum_{\boldsymbol{k}\in\mathbb{Z}_{\geq0}^Q}\mathbb{E}_{\boldsymbol{\tau}_{\varepsilon_0}}[A_{\boldsymbol{\tau},\boldsymbol{k}}^{(q)}]\mathbb{E}_{\overline{\boldsymbol{x}},\overline{\boldsymbol{y}}}\left[Q_{\boldsymbol{k}}^{\boldsymbol{d}}\left(\{\langle\overline{\boldsymbol{x}}^{(q)},\overline{\boldsymbol{y}}^{(q)}\rangle\}_{q\in[Q]}\right)[P_{\mathcal{Q}^c}\overline{f}_d](\overline{\boldsymbol{x}})[P_{\mathcal{Q}^c}\overline{f}_d](\overline{\boldsymbol{y}})\right].$$

We have

$$\mathbb{E}_{\overline{\boldsymbol{x}},\overline{\boldsymbol{y}}}\left[Q_{\boldsymbol{k}}^{\boldsymbol{d}}\left(\{\langle\overline{\boldsymbol{x}}^{(q)},\overline{\boldsymbol{y}}^{(q)}\rangle\}_{q\in[Q]}\right)[P_{\mathcal{Q}^c}\overline{f}_d](\overline{\boldsymbol{x}})[P_{\mathcal{Q}^c}\overline{f}_d](\overline{\boldsymbol{y}})\right]$$

$$= \sum_{\boldsymbol{l},\boldsymbol{l}'\in\mathcal{Q}^c}\sum_{\boldsymbol{s}\in[B(\boldsymbol{d},\boldsymbol{l})]}\sum_{\boldsymbol{s}'\in[B(\boldsymbol{d},\boldsymbol{l}')]}\lambda_{\boldsymbol{l},\boldsymbol{s}}^{\boldsymbol{d}}(\overline{f}_d)\lambda_{\boldsymbol{l}',\boldsymbol{s}'}^{\boldsymbol{d}}(\overline{f}_d)\mathbb{E}_{\overline{\boldsymbol{x}},\overline{\boldsymbol{y}}}\left[Q_{\boldsymbol{k}}^{\boldsymbol{d}}\left(\{\langle\overline{\boldsymbol{x}}^{(q)},\overline{\boldsymbol{y}}^{(q)}\rangle\}_{q\in[Q]}\right)Y_{\boldsymbol{l},\boldsymbol{s}}^{\boldsymbol{d}}(\overline{\boldsymbol{x}})Y_{\boldsymbol{l}',\boldsymbol{s}'}^{\boldsymbol{d}}(\overline{\boldsymbol{y}})\right]$$

$$= \delta_{\boldsymbol{k}\in\mathcal{Q}^c}\sum_{\boldsymbol{s}\in[B(\boldsymbol{d},\boldsymbol{k})]}\frac{\lambda_{\boldsymbol{k},\boldsymbol{s}}^{\boldsymbol{d}}(\overline{f}_d)^2}{B(\boldsymbol{d},\boldsymbol{k})},$$

where we used in the third line

$$\mathbb{E}_{\overline{\boldsymbol{x}},\overline{\boldsymbol{y}}}\left[Q_{\boldsymbol{k}}^{\boldsymbol{d}}\left(\{\langle\overline{\boldsymbol{x}}^{(q)},\overline{\boldsymbol{y}}^{(q)}\rangle\}_{q\in[Q]}\right)Y_{\boldsymbol{l},\boldsymbol{s}}^{\boldsymbol{d}}(\overline{\boldsymbol{x}})Y_{\boldsymbol{l}',\boldsymbol{s}'}^{\boldsymbol{d}}(\overline{\boldsymbol{y}})\right]$$

$$= \mathbb{E}_{\overline{\boldsymbol{y}}}\left[\mathbb{E}_{\overline{\boldsymbol{x}}}\left[Q_{\boldsymbol{k}}^{\boldsymbol{d}}\left(\{\langle\overline{\boldsymbol{x}}^{(q)},\overline{\boldsymbol{y}}^{(q)}\rangle\}_{q\in[Q]}\right)Y_{\boldsymbol{l},\boldsymbol{s}}^{\boldsymbol{d}}(\overline{\boldsymbol{x}})\right]Y_{\boldsymbol{l}',\boldsymbol{s}'}^{\boldsymbol{d}}(\overline{\boldsymbol{y}})\right]$$

$$= \frac{\delta_{\boldsymbol{k},\boldsymbol{l}}}{B(\boldsymbol{d},\boldsymbol{k})}\mathbb{E}_{\overline{\boldsymbol{y}}}\left[Y_{\boldsymbol{k},\boldsymbol{s}}^{\boldsymbol{d}}(\overline{\boldsymbol{y}})Y_{\boldsymbol{l}',\boldsymbol{s}'}^{\boldsymbol{d}}(\overline{\boldsymbol{y}})\right]$$

$$= \frac{\delta_{\boldsymbol{k},\boldsymbol{l}}\delta_{\boldsymbol{k},\boldsymbol{l}'}\delta_{\boldsymbol{s},\boldsymbol{s}'}}{B(\boldsymbol{d},\boldsymbol{k})}.$$

We conclude that

$$\mathcal{E}_{\mathcal{Q}^c,\varepsilon_0}^{(q)} = \sum_{\boldsymbol{k}\in\mathbb{Z}_{\geq 0}^Q} \mathbb{E}_{\boldsymbol{\tau}_{\varepsilon_0}}[A_{\boldsymbol{\tau},\boldsymbol{k}}^{(q)}] \sum_{\boldsymbol{s}\in[B(\boldsymbol{d},\boldsymbol{k})]} \frac{\lambda_{\boldsymbol{k},\boldsymbol{s}}^{\boldsymbol{d}}(\overline{f}_d)^2}{B(\boldsymbol{d},\boldsymbol{k})} \delta_{\boldsymbol{k}\in\mathcal{Q}^c}$$

$$= \sum_{\boldsymbol{k}\in\mathcal{Q}^c} \frac{\mathbb{E}_{\boldsymbol{\tau}_{\varepsilon_0}}[A_{\boldsymbol{\tau},\boldsymbol{k}}^{(q)}]}{B(\boldsymbol{d},\boldsymbol{k})} \|P_{\boldsymbol{k}} f_d\|_{L^2}^2$$

$$\leq \left[\max_{\boldsymbol{k}\in\mathcal{Q}^c} B(\boldsymbol{d},\boldsymbol{k})^{-1}\mathbb{E}_{\boldsymbol{\tau}_{\varepsilon_0}}[A_{\boldsymbol{\tau},\boldsymbol{k}}^{(q)}]\right] \cdot \|\mathsf{P}_{\mathcal{Q}^c} f_d\|_{L^2}^2.$$

$\square$

**Lemma 12.** *Let $\sigma$ be a weakly differentiable activation function with weak derivative $\sigma'$. For a fixed $\boldsymbol{\tau}\in\mathbb{R}_{\geq 0}^Q$, define the kernels for $q\in[Q]$,*

$$H_{\boldsymbol{\tau}}^{(q)}(\overline{\boldsymbol{x}},\overline{\boldsymbol{y}})$$
$$=\frac{r_q^2}{d_q}\mathbb{E}_{\overline{\boldsymbol{\theta}}}\left[\sigma'_{\boldsymbol{d},\boldsymbol{\tau}}\left(\{\langle\overline{\boldsymbol{\theta}}^{(q)},\overline{\boldsymbol{x}}^{(q)}\rangle/\sqrt{d_q}\}_{q\in[Q]}\right)\sigma'_{\boldsymbol{d},\boldsymbol{\tau}}\left(\{\langle\overline{\boldsymbol{\theta}}^{(q)},\overline{\boldsymbol{y}}^{(q)}\rangle/\sqrt{d_q}\}_{q\in[Q]}\right)\right]\langle\overline{\boldsymbol{x}}^{(q)},\overline{\boldsymbol{y}}^{(q)}\rangle.$$

*Then, we have the following decomposition in terms of product of Gegenbauer polynomials,*

$$H_{\boldsymbol{\tau}}^{(q)}(\overline{\boldsymbol{x}},\overline{\boldsymbol{y}}) = \sum_{\boldsymbol{k}\in\mathbb{Z}_{\geq 0}^2} A_{\boldsymbol{\tau},\boldsymbol{k}}^{(q)} Q_{\boldsymbol{k}}^{\boldsymbol{d}}\left(\{\langle\overline{\boldsymbol{x}}^{(q)},\overline{\boldsymbol{y}}^{(q)}\rangle\}_{q\in[Q]}\right),$$

*where*

$$A_{\boldsymbol{\tau},\boldsymbol{k}}^{(q)} = r_q^2 \cdot [t_{d_q,k_q-1}\lambda_{\boldsymbol{k}_{q-}}^{\boldsymbol{d}}(\sigma'_{\boldsymbol{d},\boldsymbol{\tau}})^2 B(\boldsymbol{d},\boldsymbol{k}_{q-}) + s_{d_q,k_q+1}\lambda_{\boldsymbol{k}_{q+}}^{\boldsymbol{d}}(\sigma'_{\boldsymbol{d},\boldsymbol{\tau}})^2 B(\boldsymbol{d},\boldsymbol{k}_{q+})],$$

*with $\boldsymbol{k}_{q+} = (k_1,\ldots,k_q+1,\ldots,k_Q)$ and $\boldsymbol{k}_{q-} = (k_1,\ldots,k_q-1,\ldots,k_Q)$, and*

$$s_{d,k} = \frac{k}{2k+d-2}, \qquad t_{d,k} = \frac{k+d-2}{2k+d-2},$$

*with the convention $t_{d,-1} = 0$.*

*Proof of Lemma 12.* Recall the decomposition of $\sigma'$ in terms of tensor product of Gegenbauer polynomials,

$$\sigma'(\langle\boldsymbol{\theta},\boldsymbol{x}\rangle/R) = \sum_{\boldsymbol{k}\in\mathbb{Z}_{\geq 0}^Q} \lambda_{\boldsymbol{k}}^{\boldsymbol{d}}(\sigma'_{\boldsymbol{d},\boldsymbol{\tau}}) B(\boldsymbol{d},\boldsymbol{k}) Q_{\boldsymbol{k}}^{\boldsymbol{d}}\left(\{\langle\overline{\boldsymbol{\theta}}^{(q)},\overline{\boldsymbol{x}}^{(q)}\rangle\}_{q\in[Q]}\right),$$

$$\lambda_{\boldsymbol{k}}^{\boldsymbol{d}}(\sigma'_{\boldsymbol{d},\boldsymbol{\tau}}) = \mathbb{E}_{\overline{\boldsymbol{x}}}\left[\sigma'_{\boldsymbol{d},\boldsymbol{\tau}}\left(\overline{x}_1^{(1)},\ldots,\overline{x}_1^{(Q)}\right) Q_{\boldsymbol{k}}^{\boldsymbol{d}}\left(\sqrt{d_1}\overline{x}_1^{(1)},\ldots,\sqrt{d_q}\overline{x}_1^{(Q)}\right)\right],$$

Injecting this decomposition into the definition of $H_{\boldsymbol{\tau}}^{(q)}$ yields

$$H_{\boldsymbol{\tau}}^{(q)}(\overline{\boldsymbol{x}},\overline{\boldsymbol{y}})$$
$$=\frac{r_q^2}{d_q}\mathbb{E}_{\overline{\boldsymbol{\theta}}}\left[\sigma'_{\boldsymbol{d},\boldsymbol{\tau}}\left(\{\langle\overline{\boldsymbol{\theta}}^{(q)},\overline{\boldsymbol{x}}^{(q)}\rangle/\sqrt{d_q}\}_{q\in[Q]}\right)\sigma'_{\boldsymbol{d},\boldsymbol{\tau}}\left(\{\langle\overline{\boldsymbol{\theta}}^{(q)},\overline{\boldsymbol{y}}^{(q)}\rangle/\sqrt{d_q}\}_{q\in[Q]}\right)\right]\langle\overline{\boldsymbol{x}}^{(q)},\overline{\boldsymbol{y}}^{(q)}\rangle$$
$$=\frac{r_q^2}{d_q}\sum_{\boldsymbol{k},\boldsymbol{k}'\in\mathbb{Z}_{\geq 0}^Q} \lambda_{\boldsymbol{k}}^{\boldsymbol{d}}(\sigma'_{\boldsymbol{d},\boldsymbol{\tau}})\lambda_{\boldsymbol{k}'}^{\boldsymbol{d}}(\sigma'_{\boldsymbol{d},\boldsymbol{\tau}})B(\boldsymbol{d},\boldsymbol{k})B(\boldsymbol{d},\boldsymbol{k}')\times$$
$$\mathbb{E}_{\overline{\boldsymbol{\theta}}}\left[Q_{\boldsymbol{k}}^{\boldsymbol{d}}\left(\{\langle\overline{\boldsymbol{\theta}}^{(q)},\overline{\boldsymbol{x}}^{(q)}\rangle\}_{q\in[Q]}\right) Q_{\boldsymbol{k}'}^{\boldsymbol{d}}\left(\{\langle\overline{\boldsymbol{\theta}}^{(q)},\overline{\boldsymbol{y}}^{(q)}\rangle\}_{q\in[Q]}\right)\right]\langle\overline{\boldsymbol{x}}^{(q)},\overline{\boldsymbol{y}}^{(q)}\rangle.$$

Recalling Eq. (26), we have

$$\mathbb{E}_{\overline{\boldsymbol{\theta}}}\left[Q_{\boldsymbol{k}}^{\boldsymbol{d}}\left(\{\langle\overline{\boldsymbol{\theta}}^{(q)},\overline{\boldsymbol{x}}^{(q)}\rangle\}_{q\in[Q]}\right) Q_{\boldsymbol{k}'}^{\boldsymbol{d}}\left(\{\langle\overline{\boldsymbol{\theta}}^{(q)},\overline{\boldsymbol{y}}^{(q)}\rangle\}_{q\in[Q]}\right)\right] = \delta_{\boldsymbol{k},\boldsymbol{k}'}\frac{Q_{\boldsymbol{k}}^{\boldsymbol{d}}\left(\{\langle\overline{\boldsymbol{x}}^{(q)},\overline{\boldsymbol{y}}^{(q)}\rangle\}_{q\in[Q]}\right)}{B(\boldsymbol{d},\boldsymbol{k})}.$$

Hence,

$$
\begin{aligned}
&H_{\boldsymbol{\tau}}^{(q)}(\overline{\boldsymbol{x}}, \overline{\boldsymbol{y}}) \\
&= \frac{r_q^2}{d_q} \sum_{\boldsymbol{k} \in \mathbb{Z}_{\geq 0}^Q} \lambda_{\boldsymbol{k}}^{\boldsymbol{d}} (\sigma_{\boldsymbol{d}, \boldsymbol{\tau}}')^2 B(\boldsymbol{d}, \boldsymbol{k}) Q_{\boldsymbol{k}}^{\boldsymbol{d}} \left( \{ \langle \overline{\boldsymbol{x}}^{(q)}, \overline{\boldsymbol{y}}^{(q)} \rangle \}_{q \in [Q]} \right) \langle \overline{\boldsymbol{x}}^{(q)}, \overline{\boldsymbol{y}}^{(q)} \rangle \\
&= r_q^2 \sum_{\boldsymbol{k} \in \mathbb{Z}_{\geq 0}^Q} \lambda_{\boldsymbol{k}}^{\boldsymbol{d}} (\sigma_{\boldsymbol{d}, \boldsymbol{\tau}}')^2 B(\boldsymbol{d}, \boldsymbol{k}) \left[ Q_{k_q}^{(d_q)} (\langle \overline{\boldsymbol{x}}^{(q)}, \overline{\boldsymbol{y}}^{(q)} \rangle) \langle \overline{\boldsymbol{x}}^{(q)}, \overline{\boldsymbol{y}}^{(q)} \rangle / d_q \right] \prod_{q' \neq q} Q_{k_{q'}}^{\boldsymbol{d}} (\langle \overline{\boldsymbol{x}}^{(q')}, \overline{\boldsymbol{y}}^{(q')} \rangle).
\end{aligned}
$$

By the recurrence relationship for Gegenbauer polynomials (13), we have

$$
\frac{t}{d_q} Q_{k_q}^{(d_q)}(t) = s_{d_q, k_q} Q_{k_q - 1}^{(d_q)}(t) + t_{d_q, k_q} Q_{k_q + 1}^{(d_q)}(t),
$$

where (we use the convention $t_{d_q, -1} = 0$)

$$
s_{d_q, k_q} = \frac{k_q}{2k_q + d_q - 2}, \qquad t_{d_q, k_q} = \frac{k_q + d_q - 2}{2k_q + d_q - 2}.
$$

Hence we get,

$$
\begin{aligned}
&H_{\boldsymbol{\tau}}^{(q)}(\overline{\boldsymbol{x}}, \overline{\boldsymbol{y}}) \\
&= r_q^2 \sum_{\boldsymbol{k} \in \mathbb{Z}_{\geq 0}^Q} \lambda_{\boldsymbol{k}}^{\boldsymbol{d}} (\sigma_{\boldsymbol{d}, \boldsymbol{\tau}}')^2 B(\boldsymbol{d}, \boldsymbol{k}) \left[ Q_{k_q}^{(d_q)} (\langle \overline{\boldsymbol{x}}^{(q)}, \overline{\boldsymbol{y}}^{(q)} \rangle) \langle \overline{\boldsymbol{x}}^{(q)}, \overline{\boldsymbol{y}}^{(q)} \rangle / d_q \right] \prod_{q' \neq q} Q_{k_{q'}}^{\boldsymbol{d}} (\langle \overline{\boldsymbol{x}}^{(q')}, \overline{\boldsymbol{y}}^{(q')} \rangle) \\
&= r_q^2 \sum_{\boldsymbol{k} \in \mathbb{Z}_{\geq 0}^Q} \lambda_{\boldsymbol{k}}^{\boldsymbol{d}} (\sigma_{\boldsymbol{d}, \boldsymbol{\tau}}')^2 B(\boldsymbol{d}, \boldsymbol{k}) \left[ s_{d_q, k_q} Q_{k_q - 1}^{(d_q)} (\langle \overline{\boldsymbol{x}}^{(q)}, \overline{\boldsymbol{y}}^{(q)} \rangle) + t_{d_q, k_q} Q_{k_q + 1}^{(d_q)} (\langle \overline{\boldsymbol{x}}^{(q)}, \overline{\boldsymbol{y}}^{(q)} \rangle) \right] \\
&\qquad\qquad\qquad\qquad\qquad\qquad\qquad\qquad \times \prod_{q' \neq q} Q_{k_{q'}}^{\boldsymbol{d}} (\langle \overline{\boldsymbol{x}}^{(q')}, \overline{\boldsymbol{y}}^{(q')} \rangle) \\
&= \sum_{\boldsymbol{k} \in \mathbb{Z}_{\geq 0}^2} A_{\boldsymbol{\tau}, \boldsymbol{k}}^{(q)} Q_{\boldsymbol{k}}^{\boldsymbol{d}} \left( \{ \langle \overline{\boldsymbol{x}}^{(q)}, \overline{\boldsymbol{y}}^{(q)} \rangle \}_{q \in [Q]} \right),
\end{aligned}
$$

where we get by matching the coefficients,

$$
A_{\boldsymbol{\tau}, \boldsymbol{k}}^{(q)} = r_q^2 \cdot [t_{d_q, k_q - 1} \lambda_{\boldsymbol{k}_{q-}}^{\boldsymbol{d}} (\sigma_{\boldsymbol{d}, \boldsymbol{\tau}}')^2 B(\boldsymbol{d}, \boldsymbol{k}_{q-}) + s_{d_q, k_q + 1} \lambda_{\boldsymbol{k}_{q+}}^{\boldsymbol{d}} (\sigma_{\boldsymbol{d}, \boldsymbol{\tau}}')^2 B(\boldsymbol{d}, \boldsymbol{k}_{q+})],
$$

with $\boldsymbol{k}_{q+} = (k_1, \ldots, k_q + 1, \ldots, k_Q)$ and $\boldsymbol{k}_{q-} = (k_1, \ldots, k_q - 1, \ldots, k_Q)$. $\qquad \square$

## G.4 Proof of Proposition 4

### G.4.1 Preliminaries

**Lemma 13.** *Let $\psi : \mathbb{R}^Q \to \mathbb{R}$ be a function such that $\psi(\{\langle \boldsymbol{e}_q, \cdot \rangle\}_{q \in [Q]}) \in L^2(\mathrm{PS}^{\boldsymbol{d}}, \mu_{\boldsymbol{d}})$. We will consider for integers $\boldsymbol{i} = (i_1, \ldots, i_Q) \in \mathbb{Z}_{\geq 0}^Q$, the associated function $\psi^{(\boldsymbol{i})}$ given by:*

$$
\psi^{(\boldsymbol{i})} \left( \overline{x}_1^{(1)}, \ldots, \overline{x}_1^{(Q)} \right) = \left( \overline{x}_1^{(1)} \right)^{i_1} \cdots \left( \overline{x}_1^{(Q)} \right)^{i_Q} \psi \left( \overline{x}_1^{(1)}, \ldots, \overline{x}_1^{(Q)} \right).
$$

*Assume that $\psi^{(\boldsymbol{i})}(\{\langle \boldsymbol{e}_q, \cdot \rangle\}_{q \in [Q]}) \in L^2(\mathrm{PS}^{\boldsymbol{d}}, \mu_{\boldsymbol{d}})$. Let $\{\lambda_{\boldsymbol{k}}^{\boldsymbol{d}}(\psi)\}_{\boldsymbol{k} \in \mathbb{Z}_{\geq 0}^Q}$ be the coefficients of the expansion of $\psi$ in terms of the product of Gegenbauer polynomials*

$$
\psi \left( \overline{x}_1^{(1)}, \ldots, \overline{x}_1^{(Q)} \right) = \sum_{\boldsymbol{k} \in \mathbb{Z}_{\geq 0}^Q} \lambda_{\boldsymbol{k}}^{\boldsymbol{d}}(\psi) B(\boldsymbol{d}, \boldsymbol{k}) Q_{\boldsymbol{k}}^{\boldsymbol{d}} \left( \sqrt{d_1} \overline{x}_1^{(1)}, \ldots, \sqrt{d_Q} \overline{x}_1^{(Q)} \right),
$$

$$
\lambda_{\boldsymbol{k}}^{\boldsymbol{d}}(\psi) = \mathbb{E}_{\overline{\boldsymbol{x}}} \left[ \psi \left( \overline{x}_1^{(1)}, \ldots, \overline{x}_1^{(Q)} \right) Q_{\boldsymbol{k}}^{\boldsymbol{d}} \left( \sqrt{d_1} \overline{x}_1^{(1)}, \ldots, \sqrt{d_Q} \overline{x}_1^{(Q)} \right) \right].
$$

*Then we can write*

$$
\psi^{(\boldsymbol{i})} \left( \overline{x}_1^{(1)}, \ldots, \overline{x}_1^{(Q)} \right) = \sum_{\boldsymbol{k} \in \mathbb{Z}_{\geq 0}^Q} \lambda_{\boldsymbol{k}}^{\boldsymbol{d}, \boldsymbol{i}}(\psi) B(\boldsymbol{d}, \boldsymbol{k}) Q_{\boldsymbol{k}}^{\boldsymbol{d}} \left( \sqrt{d_1} \overline{x}_1^{(1)}, \ldots, \sqrt{d_Q} \overline{x}_1^{(Q)} \right),
$$

where the coefficients $\lambda_{\boldsymbol{k}}^{\boldsymbol{d},\boldsymbol{i}}(\psi)$ are given recursively: denoting $\boldsymbol{i}_{q+} = (i_1,\ldots,i_q+1,\ldots,i_Q)$, if $k_q = 0$,

$$\lambda_{\boldsymbol{k}}^{\boldsymbol{d},\boldsymbol{i}_{q+}}(\psi) = \sqrt{d_q}\lambda_{\boldsymbol{k}_{q+}}^{\boldsymbol{d},\boldsymbol{i}}(\psi),$$

and for $k_q > 0$,

$$\lambda_{\boldsymbol{k}}^{\boldsymbol{d},\boldsymbol{i}_{q+}}(\psi) = \sqrt{d_q}\frac{k_q + d_q - 2}{2k_q + d_q - 2}\lambda_{\boldsymbol{k}_{q+}}^{\boldsymbol{d},\boldsymbol{i}}(\psi) + \sqrt{d_q}\frac{k_q}{2k_q + d_q - 2}\lambda_{\boldsymbol{k}_{q-}}^{\boldsymbol{d},\boldsymbol{i}}(\psi),$$

where we recall the notations $\boldsymbol{k}_{q+} = (k_1,\ldots,k_q+1,\ldots,k_Q)$ and $\boldsymbol{k}_{q-} = (k_1,\ldots,k_q-1,\ldots,k_Q)$.

*Proof of Lemma 13.* We recall the following two formulas for $k \geq 1$ (see Section B.2):

$$\frac{x}{d}Q_k^{(d)}(x) = \frac{k}{2k + d - 2}Q_{k-1}^{(d)}(x) + \frac{k + d - 2}{2k + d - 2}Q_{k+1}^{(d)}(x),$$

$$B(d, k) = \frac{2k + d - 2}{k}\binom{k + d - 3}{k - 1}.$$

Furthermore, we have $Q_0^{(d)}(x) = 1$, $Q_1^{(d)}(x) = x/d$ and therefore therefore $xQ_0^{(d)}(x) = dQ_1^{(d)}(x)$. Similarly to the proof of [4, Lemma 6], we insert these expressions in the expansion of the function $\psi$. Matching the coefficients of the expansion yields the result. $\qquad\square$

Let $\boldsymbol{u} : \mathbb{S}^{D-1}(\sqrt{D}) \times \mathbb{S}^{D-1}(\sqrt{D}) \to \mathbb{R}^{D \times D}$ be a matrix-valued function defined by

$$\boldsymbol{u}(\boldsymbol{\theta}_1, \boldsymbol{\theta}_2) = \mathbb{E}_{\boldsymbol{x}}[\sigma'(\langle\boldsymbol{\theta}_1, \boldsymbol{x}\rangle/R)\sigma'(\langle\boldsymbol{\theta}_2, \boldsymbol{x}\rangle/R)\boldsymbol{x}\boldsymbol{x}^\mathsf{T}].$$

We can write this function as a $Q$ by $Q$ block matrix function $\boldsymbol{u} = (\boldsymbol{u}^{(qq')})_{q,q'\in[Q]}$, where $\boldsymbol{u}^{(qq')} : \mathbb{S}^{D-1}(\sqrt{D}) \times \mathbb{S}^{D-1}(\sqrt{D}) \to \mathbb{R}^{d_q \times d_{q'}}$ are given by

$$\boldsymbol{u}^{(qq')}(\boldsymbol{\theta}_1, \boldsymbol{\theta}_2) = \mathbb{E}_{\boldsymbol{x}}[\sigma'(\langle\boldsymbol{\theta}_1, \boldsymbol{x}\rangle/R)\sigma'(\langle\boldsymbol{\theta}_2, \boldsymbol{x}\rangle/R)\boldsymbol{x}^{(q)}(\boldsymbol{x}^{(q')})^\mathsf{T}].$$

We have the following lemma which is a generalization of [4, Lemma 7], that shows essentially the same decomposition of the matrix $\boldsymbol{u}(\boldsymbol{\theta}_1, \boldsymbol{\theta}_2)$ as by integration by part if we had $\boldsymbol{x} \sim \mathsf{N}(0, \mathbf{I})$.

**Lemma 14.** *For* $q \in [Q]$, *there exists functions* $u_1^{(qq)}, u_2^{(qq)}, u_{3,1}^{(qq)}, u_{3,2}^{(qq)} : \mathbb{S}^{D-1}(\sqrt{D}) \times \mathbb{S}^{D-1}(\sqrt{D}) \to \mathbb{R}$ *such that*

$$\begin{aligned}\boldsymbol{u}^{(qq)}(\boldsymbol{\theta}_1, \boldsymbol{\theta}_2) =&u_1^{(qq)}(\boldsymbol{\theta}_1, \boldsymbol{\theta}_2)\mathbf{I}_{d_q} + u_2^{(qq)}(\boldsymbol{\theta}_1, \boldsymbol{\theta}_2)[\boldsymbol{\theta}_1^{(q)}(\boldsymbol{\theta}_2^{(q)})^\mathsf{T} + \boldsymbol{\theta}_2^{(q)}(\boldsymbol{\theta}_1^{(q)})^\mathsf{T}] \\ &+ u_{3,1}^{(qq)}(\boldsymbol{\theta}_1, \boldsymbol{\theta}_2)\boldsymbol{\theta}_1^{(q)}(\boldsymbol{\theta}_1^{(q)})^\mathsf{T} + u_{3,2}^{(qq)}(\boldsymbol{\theta}_1, \boldsymbol{\theta}_2)\boldsymbol{\theta}_2^{(q)}(\boldsymbol{\theta}_2^{(q)})^\mathsf{T}.\end{aligned}$$

*For* $q, q' \in [Q]$, *there exists functions* $u_{2,1}^{(qq')}, u_{2,2}^{(qq')}, u_{3,1}^{(qq')}, u_{3,2}^{(qq')} : \mathbb{S}^{D-1}(\sqrt{D}) \times \mathbb{S}^{D-1}(\sqrt{D}) \to \mathbb{R}$ *such that*

$$\begin{aligned}\boldsymbol{u}^{(qq')}(\boldsymbol{\theta}_1, \boldsymbol{\theta}_2) =&u_{2,1}^{(qq')}(\boldsymbol{\theta}_1, \boldsymbol{\theta}_2)\boldsymbol{\theta}_1^{(q)}(\boldsymbol{\theta}_2^{(q')})^\mathsf{T} + u_{2,2}^{(qq')}(\boldsymbol{\theta}_1, \boldsymbol{\theta}_2)\boldsymbol{\theta}_2^{(q)}(\boldsymbol{\theta}_1^{(q')})^\mathsf{T} \\ &+ u_{3,1}^{(qq')}(\boldsymbol{\theta}_1, \boldsymbol{\theta}_2)\boldsymbol{\theta}_1^{(q)}(\boldsymbol{\theta}_1^{(q')})^\mathsf{T} + u_{3,2}^{(qq')}(\boldsymbol{\theta}_1, \boldsymbol{\theta}_2)\boldsymbol{\theta}_2^{(q)}(\boldsymbol{\theta}_2^{(q')})^\mathsf{T}.\end{aligned}$$

*Proof of Lemma 14.* Denote $\gamma^{(q)} = \langle\overline{\boldsymbol{\theta}}_1^{(q)}, \overline{\boldsymbol{\theta}}_2^{(q)}\rangle/d_q$. Let us rotate each sphere $q \in [Q]$ such that

$$\begin{aligned}\boldsymbol{\theta}_1^{(q)} &= \left(\tau_1^{(q)}\sqrt{d_q}, 0, \ldots, 0\right), \\ \boldsymbol{\theta}_2^{(q)} &= \left(\tau_2^{(q)}\sqrt{d_q}\gamma^{(q)}, \tau_2^{(q)}\sqrt{d_q}\sqrt{1 - (\gamma^{(q)})^2}, 0, \ldots, 0\right).\end{aligned} \tag{103}$$

**Step 1: $\boldsymbol{u}^{(qq)}$.**

Let us start with $\boldsymbol{u}^{(qq)}$. For clarity, we will denote (in the rotated basis (103))

$$\begin{aligned}\alpha_1 &= \langle\boldsymbol{\theta}_1, \boldsymbol{x}\rangle/R = \sum_{q\in[Q]} \tau_1^{(q)}\sqrt{d_q}/R \cdot \overline{x}_1^{(q)}, \\ \alpha_2 &= \langle\boldsymbol{\theta}_2, \boldsymbol{x}\rangle/R = \sum_{q\in[Q]} \left[\tau_2^{(q)}\sqrt{d_q}\gamma^{(q)}/R \cdot \overline{x}_1^{(q)} + \tau_2^{(q)}\sqrt{d_q}\sqrt{1 - (\gamma^{(q)})^2}/R \cdot \overline{x}_2^{(q)}\right].\end{aligned}$$

Then it is easy to show that we can rewrite

$$\boldsymbol{u}^{(qq)}(\boldsymbol{\theta}_1, \boldsymbol{\theta}_2) = \mathbb{E}_{\boldsymbol{x}}[\sigma'(\alpha_1)\sigma'(\alpha_2)\boldsymbol{x}^{(q)}(\boldsymbol{x}^{(q)})^\mathsf{T} = \begin{bmatrix} \boldsymbol{u}_{1:2,1:2}^{(qq)} & \mathbf{0} \\ \mathbf{0} & \mathbb{E}_{\boldsymbol{x}}[\sigma'(\alpha_1)\sigma'(\alpha_2)(x_3^{(q)})^2]\mathbf{I}_{d_q-2} \end{bmatrix},$$

with

$$\boldsymbol{u}_{1:2,1:2}^{(qq)} = \begin{bmatrix} \mathbb{E}_{\boldsymbol{x}}[\sigma'(\alpha_1)\sigma'(\alpha_2)(x_1^{(q)})^2] & \mathbb{E}_{\boldsymbol{x}}[\sigma'(\alpha_1)\sigma'(\alpha_2)x_1^{(q)}x_2^{(q)}] \\ \mathbb{E}_{\boldsymbol{x}}[\sigma'(\alpha_1)\sigma'(\alpha_2)x_2^{(q)}x_1^{(q)}] & \mathbb{E}_{\boldsymbol{x}}[\sigma'(\alpha_1)\sigma'(\alpha_2)(x_2^{(q)})^2] \end{bmatrix}.$$

**Case (a):** $\boldsymbol{\theta}_1^{(q)} \neq \boldsymbol{\theta}_2^{(q)}$.

Given any functions $u_1^{(qq)}, u_2^{(qq)}, u_{3,1}^{(qq)}, u_{3,2}^{(qq)} : \mathbb{S}^{D-1}(\sqrt{D}) \times \mathbb{S}^{D-1}(\sqrt{D}) \to \mathbb{R}$, we define

$$\tilde{\boldsymbol{u}}^{(qq)}(\boldsymbol{\theta}_1, \boldsymbol{\theta}_2) = u_1^{(qq)}(\boldsymbol{\theta}_1, \boldsymbol{\theta}_2)\mathbf{I}_{d_1} + u_2^{(qq)}(\boldsymbol{\theta}_1, \boldsymbol{\theta}_2)[\boldsymbol{\theta}_1^{(q)}(\boldsymbol{\theta}_2^{(q)})^\mathsf{T} + \boldsymbol{\theta}_2^{(q)}(\boldsymbol{\theta}_1^{(q)})^\mathsf{T}]$$
$$+ u_{3,1}^{(qq)}(\boldsymbol{\theta}_1, \boldsymbol{\theta}_2)\boldsymbol{\theta}_1^{(q)}(\boldsymbol{\theta}_1^{(q)})^\mathsf{T} + u_{3,2}^{(qq)}(\boldsymbol{\theta}_1, \boldsymbol{\theta}_2)\boldsymbol{\theta}_2^{(q)}(\boldsymbol{\theta}_2^{(q)})^\mathsf{T}.$$

In the rotated basis (103), we have

$$\tilde{\boldsymbol{u}}^{(qq)}(\boldsymbol{\theta}_1, \boldsymbol{\theta}_2) = \begin{bmatrix} \tilde{\boldsymbol{u}}_{1:2,1:2}^{(qq)} & \mathbf{0} \\ \mathbf{0} & u_1^{(qq)}(\boldsymbol{\theta}_1, \boldsymbol{\theta}_2)\mathbf{I}_{d_q-2} \end{bmatrix},$$

where (we dropped the dependency on $(\boldsymbol{\theta}_1, \boldsymbol{\theta}_2)$ for clarity)

$$\boldsymbol{u}_{11}^{(qq)} = u_1^{(qq)} + 2\tau_1^{(q)}\tau_2^{(q)}d_q\gamma^{(q)}u_2^{(qq)} + (\tau_1^{(q)})^2 d_q u_{3,1}^{(qq)} + (\tau_2^{(q)})^2 d_q (\gamma^{(q)})^2 u_{3,2}^{(qq)},$$

$$\boldsymbol{u}_{12}^{(qq)} = \tau_1^{(q)}\tau_2^{(q)}d_q\sqrt{1-(\gamma^{(q)})^2}\,u_2^{(qq)} + (\tau_2^{(q)})^2 d_q\gamma^{(q)}\sqrt{1-(\gamma^{(q)})^2}\,u_{3,2}^{(qq)},$$

$$\boldsymbol{u}_{22}^{(qq)} = u_1^{(qq)} + (\tau_2^{(q)})^2 d_q(1-(\gamma^{(q)})^2)u_{3,2}^{(qq)}.$$

We see that $\boldsymbol{u}^{(qq)}$ and $\tilde{\boldsymbol{u}}^{(qq)}$ will be equal if and only if we have the following equalities:

$$\mathrm{Tr}(\boldsymbol{u}^{(qq)}(\boldsymbol{\theta}_1, \boldsymbol{\theta}_2)) = \mathrm{Tr}(\tilde{\boldsymbol{u}}^{(qq)}(\boldsymbol{\theta}_1, \boldsymbol{\theta}_2))$$
$$= d_q u_1^{(qq)} + 2\tau_1^{(q)}\tau_2^{(q)}d_q\gamma^{(q)}u_2^{(qq)} + (\tau_1^{(q)})^2 d_q u_{3,1}^{(qq)} + (\tau_2^{(q)})^2 d_q u_{3,2}^{(qq)},$$

$$\langle\boldsymbol{\theta}_1^{(q)}, \boldsymbol{u}^{(qq)}(\boldsymbol{\theta}_1, \boldsymbol{\theta}_2)\boldsymbol{\theta}_2^{(q)}\rangle = \langle\boldsymbol{\theta}_1^{(q)}, \tilde{\boldsymbol{u}}^{(qq)}(\boldsymbol{\theta}_1, \boldsymbol{\theta}_2)\boldsymbol{\theta}_2^{(q)}\rangle$$
$$= \tau_1^{(q)}\tau_2^{(q)}d_q\gamma^{(q)}u_1^{(qq)} + (\tau_1^{(q)})^2(\tau_2^{(q)})^2 d_q^2(1+(\gamma^{(q)})^2)u_2^{(qq)}$$
$$+ (\tau_1^{(q)})^3\tau_2^{(q)}d_q^2\gamma^{(q)}u_{3,1}^{(qq)} + \tau_1^{(q)}(\tau_2^{(q)})^3 d_q^2\gamma^{(q)}u_{3,1}^{(qq)},$$

$$\langle\boldsymbol{\theta}_1^{(q)}, \boldsymbol{u}^{(qq)}(\boldsymbol{\theta}_1, \boldsymbol{\theta}_2)\boldsymbol{\theta}_1^{(q)}\rangle = \langle\boldsymbol{\theta}_1^{(q)}, \tilde{\boldsymbol{u}}^{(qq)}(\boldsymbol{\theta}_1, \boldsymbol{\theta}_2)\boldsymbol{\theta}_1^{(q)}\rangle$$
$$= (\tau_1^{(q)})^2 d_q u_1^{(qq)} + 2(\tau_1^{(q)})^3\tau_2^{(q)}d_q^2\gamma^{(q)}u_2^{(qq)}$$
$$+ (\tau_1^{(q)})^4 d_q^2 u_{3,1}^{(qq)} + (\tau_1^{(q)})^2(\tau_2^{(q)})^2 d_q^2(\gamma^{(q)})^2 u_{3,2}^{(qq)},$$

$$\langle\boldsymbol{\theta}_2^{(q)}, \boldsymbol{u}^{(qq)}(\boldsymbol{\theta}_1, \boldsymbol{\theta}_2)\boldsymbol{\theta}_2^{(q)}\rangle = \langle\boldsymbol{\theta}_2^{(q)}, \tilde{\boldsymbol{u}}^{(qq)}(\boldsymbol{\theta}_1, \boldsymbol{\theta}_2)\boldsymbol{\theta}_2^{(q)}\rangle$$
$$= (\tau_2^{(q)})^2 d_q u_1^{(qq)} + 2\tau_1^{(q)}(\tau_2^{(q)})^3 d_q^2\gamma^{(q)}u_2^{(qq)}$$
$$+ (\tau_1^{(q)})^2(\tau_2^{(q)})^2 d_q^2(\gamma^{(q)})^2 u_{3,1}^{(qq)} + (\tau_2^{(q)})^4 d_q^2 u_{3,2}^{(qq)}.$$

Hence $\tilde{\boldsymbol{u}}^{(qq)} = \boldsymbol{u}^{(qq)}$ if and only if

$$\begin{bmatrix} u_1^{(qq)} \\ u_2^{(qq)} \\ u_{3,1}^{(qq)} \\ u_{3,2}^{(qq)} \end{bmatrix} = d_q^{-1}(\boldsymbol{M}^{(qq)})^{-1} \times \begin{bmatrix} \mathrm{Tr}(\boldsymbol{u}^{(qq)}(\boldsymbol{\theta}_1, \boldsymbol{\theta}_2)) \\ \langle\boldsymbol{\theta}_1^{(q)}, \boldsymbol{u}^{(qq)}(\boldsymbol{\theta}_1, \boldsymbol{\theta}_2)\boldsymbol{\theta}_2^{(q)}\rangle \\ \langle\boldsymbol{\theta}_1^{(q)}, \boldsymbol{u}^{(qq)}(\boldsymbol{\theta}_1, \boldsymbol{\theta}_2)\boldsymbol{\theta}_1^{(q)}\rangle \\ \langle\boldsymbol{\theta}_2^{(q)}, \boldsymbol{u}^{(qq)}(\boldsymbol{\theta}_1, \boldsymbol{\theta}_2)\boldsymbol{\theta}_2^{(q)}\rangle \end{bmatrix}, \tag{104}$$

where

$$\boldsymbol{M}^{(qq)} = \begin{bmatrix} 1 & 2\tau_1^{(q)}\tau_2^{(q)}\gamma^{(q)} & (\tau_1^{(q)})^2 & (\tau_2^{(q)})^2 \\ \tau_1^{(q)}\tau_2^{(q)}\gamma^{(q)} & (\tau_1^{(q)})^2(\tau_2^{(q)})^2 d_q(1+(\gamma^{(q)})^2) & (\tau_1^{(q)})^3\tau_2^{(q)}d_q\gamma^{(q)} & \tau_1^{(q)}(\tau_2^{(q)})^3 d_q\gamma^{(q)} \\ (\tau_1^{(q)})^2 & 2(\tau_1^{(q)})^3\tau_2^{(q)}d_q\gamma^{(q)} & (\tau_1^{(q)})^4 d_q & (\tau_1^{(q)})^2(\tau_2^{(q)})^2 d_q(\gamma^{(q)})^2 \\ (\tau_2^{(q)})^2 & 2\tau_1^{(q)}(\tau_2^{(q)})^3 d_q\gamma^{(q)} & (\tau_1^{(q)})^2(\tau_2^{(q)})^2 d_q(\gamma^{(q)})^2 & (\tau_2^{(q)})^4 d_q \end{bmatrix}$$

is invertible almost surely (for $\tau_1^{(q)}, \tau_2^{(q)} \neq 0$ and $\gamma^{(q)} \neq 1$).

**Case (b):** $\boldsymbol{\theta}_1^{(q)} = \boldsymbol{\theta}_2^{(q)}$.

Similarly, for some fixed $\alpha$ and $\beta$, we define

$$\tilde{\boldsymbol{u}}^{(qq)}(\boldsymbol{\theta}_1, \boldsymbol{\theta}_1) = \alpha \mathbf{I}_{d_q} + \beta \boldsymbol{\theta}_1^{(q)}(\boldsymbol{\theta}_1^{(q)})^\mathsf{T}.$$

Then $\boldsymbol{u}^{(qq)}(\boldsymbol{\theta}_1, \boldsymbol{\theta}_1)$ and $\tilde{\boldsymbol{u}}^{(qq)}(\boldsymbol{\theta}_1, \boldsymbol{\theta}_1)$ are equal if and only if

$$\begin{bmatrix} \alpha \\ \beta \end{bmatrix} = d_q^{-1}(\boldsymbol{M}_\parallel^{(qq)})^{-1} \times \begin{bmatrix} \mathrm{Tr}(\boldsymbol{u}^{(qq)}(\boldsymbol{\theta}_1, \boldsymbol{\theta}_1)) \\ \langle \boldsymbol{\theta}_1^{(q)}, \boldsymbol{u}^{(qq)}(\boldsymbol{\theta}_1, \boldsymbol{\theta}_1)\boldsymbol{\theta}_1^{(q)} \rangle \end{bmatrix},$$

where

$$\boldsymbol{M}_\parallel^{(qq)} = \begin{bmatrix} 1 & (\tau_1^{(q)})^2 \\ (\tau_1^{(q)})^2 & (\tau_1^{(q)})^4 d_q \end{bmatrix}.$$

**Step 2:** $\boldsymbol{u}^{(qq')}$ for $q \neq q'$.

Similarly to the two previous steps, we define for any functions $u_{2,1}^{(qq')}, u_{2,2}^{(qq')}, u_{3,1}^{(qq')}, u_{3,2}^{(qq')}$ : $\mathbb{S}^{D-1}(\sqrt{D}) \times \mathbb{S}^{D-1}(\sqrt{D}) \to \mathbb{R}$,

$$\tilde{\boldsymbol{u}}^{(qq')}(\boldsymbol{\theta}_1, \boldsymbol{\theta}_2) = u_{2,1}^{(qq')}(\boldsymbol{\theta}_1, \boldsymbol{\theta}_2)\boldsymbol{\theta}_1^{(q)}(\boldsymbol{\theta}_2^{(q')})^\mathsf{T} + u_{2,2}^{(qq')}(\boldsymbol{\theta}_1, \boldsymbol{\theta}_2)\boldsymbol{\theta}_2^{(q)}(\boldsymbol{\theta}_1^{(q')})^\mathsf{T}$$

$$+ u_{3,1}^{(qq')}(\boldsymbol{\theta}_1, \boldsymbol{\theta}_2)\boldsymbol{\theta}_1^{(q)}(\boldsymbol{\theta}_1^{(q')})^\mathsf{T} + u_{3,2}^{(qq')}(\boldsymbol{\theta}_1, \boldsymbol{\theta}_2)\boldsymbol{\theta}_2^{(q)}(\boldsymbol{\theta}_2^{(q')})^\mathsf{T}.$$

We can rewrite $\tilde{\boldsymbol{u}}^{(qq')}$ as

$$\tilde{\boldsymbol{u}}^{(qq')}(\boldsymbol{\theta}_1, \boldsymbol{\theta}_2) = \begin{bmatrix} \tilde{\boldsymbol{u}}_{1:2,1:2}^{(qq')} & \mathbf{0} \\ \mathbf{0} & \mathbf{0} \end{bmatrix},$$

where

$$\tilde{\boldsymbol{u}}_{11}^{(qq')} = u_{2,1}^{(qq')}\tau_1^{(q)}\tau_2^{(q')}\gamma^{(q')} + u_{2,2}^{(qq')}\tau_2^{(q)}\tau_1^{(q')}\gamma^{(q)} + u_{3,1}^{(qq')}\tau_1^{(q)}\tau_1^{(q')} + u_{3,2}^{(qq')}\tau_2^{(q)}\tau_2^{(q')}\gamma^{(q)}\gamma^{(q')},$$

$$\tilde{\boldsymbol{u}}_{12}^{(qq')} = u_{2,1}^{(qq')}\tau_1^{(q)}\tau_2^{(q')}\sqrt{1 - (\gamma^{(q')})^2} + u_{3,2}^{(qq')}\tau_2^{(q)}\tau_2^{(q')}\gamma^{(q)}\sqrt{1 - (\gamma^{(q')})^2},$$

$$\tilde{\boldsymbol{u}}_{21}^{(qq')} = u_{2,2}^{(qq')}\tau_2^{(q)}\tau_1^{(q')}\sqrt{1 - (\gamma^{(q)})^2} + u_{3,2}^{(qq')}\tau_2^{(q)}\tau_2^{(q')}\sqrt{1 - (\gamma^{(q)})^2}\gamma^{(q')},$$

$$\tilde{\boldsymbol{u}}_{22}^{(qq')} = u_{3,2}^{(qq')}\tau_2^{(q)}\tau_2^{(q')}\sqrt{1 - (\gamma^{(q)})^2}\sqrt{1 - (\gamma^{(q')})^2}.$$

**Case (a):** $\boldsymbol{\theta}_1^{(q)} \neq \boldsymbol{\theta}_2^{(q)}$.

We have equality $\tilde{\boldsymbol{u}}^{(qq')} = \boldsymbol{u}^{(qq')}$ if and only if

$$\begin{bmatrix} u_{2,1}^{(qq')} \\ u_{2,2}^{(qq')} \\ u_{3,1}^{(qq')} \\ u_{3,2}^{(qq')} \end{bmatrix} = (d_q d_{q'})^{-1}(\boldsymbol{M}^{(qq')})^{-1} \times \begin{bmatrix} \langle \boldsymbol{\theta}_1^{(q)}, \boldsymbol{u}^{(qq')}(\boldsymbol{\theta}_1, \boldsymbol{\theta}_2)\boldsymbol{\theta}_1^{(q')} \rangle \\ \langle \boldsymbol{\theta}_1^{(q)}, \boldsymbol{u}^{(qq')}(\boldsymbol{\theta}_1, \boldsymbol{\theta}_2)\boldsymbol{\theta}_2^{(q')} \rangle \\ \langle \boldsymbol{\theta}_2^{(q)}, \boldsymbol{u}^{(qq')}(\boldsymbol{\theta}_1, \boldsymbol{\theta}_2)\boldsymbol{\theta}_1^{(q')} \rangle \\ \langle \boldsymbol{\theta}_2^{(q)}, \boldsymbol{u}^{(qq')}(\boldsymbol{\theta}_1, \boldsymbol{\theta}_2)\boldsymbol{\theta}_2^{(q')} \rangle \end{bmatrix},$$

where $\boldsymbol{M}^{(qq')}$ is given by

$$\begin{bmatrix} (\tau_1^{(q)})^2\tau_1^{(q')}\tau_2^{(q')}\gamma^{(q')} & \tau_1^{(q)}\tau_2^{(q)}(\tau_1^{(q')})^2\gamma^{(q)} & (\tau_1^{(q)})^2(\tau_1^{(q')})^2 & \tau_1^{(q)}\tau_2^{(q)}\tau_1^{(q')}\tau_2^{(q')}\gamma^{(q)}\gamma^{(q')} \\ (\tau_1^{(q)})^2(\tau_2^{(q')})^2 & \tau_1^{(q)}\tau_2^{(q)}\tau_1^{(q')}\tau_2^{(q')}\gamma^{(q)}\gamma^{(q')} & (\tau_1^{(q)})^2\tau_1^{(q')}\tau_2^{(q')}\gamma^{(q')} & \tau_1^{(q)}\tau_2^{(q)}(\tau_2^{(q')})^2\gamma^{(q)} \\ \tau_1^{(q)}\tau_2^{(q)}\tau_1^{(q')}\tau_2^{(q')}\gamma^{(q)}\gamma^{(q')} & (\tau_2^{(q)})^2(\tau_1^{(q')})^2 & \tau_1^{(q)}\tau_2^{(q)}(\tau_1^{(q')})^2\gamma^{(q)} & (\tau_2^{(q)})^2\tau_1^{(q')}\tau_2^{(q')}\gamma^{(q')} \\ \tau_1^{(q)}\tau_2^{(q)}(\tau_2^{(q')})^2\gamma^{(q)} & (\tau_2^{(q)})^2\tau_1^{(q')}\tau_2^{(q')}\gamma^{(q')} & \tau_1^{(q)}\tau_2^{(q)}\tau_1^{(q')}\tau_2^{(q')}\gamma^{(q)}\gamma^{(q')} & (\tau_2^{(q)})^2(\tau_2^{(q')})^2 \end{bmatrix},$$

which is invertible almost surely (for $\tau_1^{(q)}, \tau_2^{(q)} \neq 0$ and $\gamma^{(q)} \neq 1$).

**Case (b):** $\boldsymbol{\theta}_1^{(q)} = \boldsymbol{\theta}_2^{(q)}$.

It is straightforward to check that

$$\boldsymbol{u}^{(qq')}(\boldsymbol{\theta}_1, \boldsymbol{\theta}_1) = \beta \boldsymbol{\theta}_1^{(q)} (\boldsymbol{\theta}_1^{(q')})^\mathsf{T},$$

where

$$\beta = (d_q d_{q'})^{-1} (\tau_1^{(q)} \tau_1^{(q')})^{-2} \left\langle \boldsymbol{\theta}_1^{(q)}, \boldsymbol{u}^{(qq')}(\boldsymbol{\theta}_1, \boldsymbol{\theta}_1) \boldsymbol{\theta}_1^{(q')} \right\rangle.$$

$\square$

### G.4.2   Proof of Proposition 4

**Step 1. Construction of the activation function $\hat{\sigma}$.**

Recall the definition of $\sigma_{\boldsymbol{d},\boldsymbol{\tau}}$ in Eq. (90) and its expansion in terms of tensor product of Gegenbauer polynomials:

$$\sigma'(\langle \boldsymbol{\theta}, \boldsymbol{x} \rangle / R) = \sum_{\boldsymbol{k} \in \mathbb{Z}_{\geq 0}^Q} \lambda_{\boldsymbol{k}}^{\boldsymbol{d}}(\sigma'_{\boldsymbol{d},\boldsymbol{\tau}}) B(\boldsymbol{d}, \boldsymbol{k}) Q_{\boldsymbol{k}}^{\boldsymbol{d}} \left( \{ \langle \overline{\boldsymbol{\theta}}^{(q)}, \overline{\boldsymbol{x}}^{(q)} \rangle \}_{q \in [Q]} \right),$$

$$\lambda_{\boldsymbol{k}}^{\boldsymbol{d}}(\sigma'_{\boldsymbol{d},\boldsymbol{\tau}}) = \mathbb{E}_{\overline{\boldsymbol{x}}} \left[ \sigma'_{\boldsymbol{d},\boldsymbol{\tau}} \left( \overline{x}_1^{(1)}, \ldots, \overline{x}_1^{(Q)} \right) Q_{\boldsymbol{k}}^{\boldsymbol{d}} \left( \sqrt{d_1} \overline{x}_1^{(1)}, \ldots, \sqrt{d_Q} \overline{x}_1^{(Q)} \right) \right].$$

We recall the definition of $q_\xi = \arg \max_{q \in [Q]} \{ \eta_q + \kappa_q \}$. Let $l_2 > l_1 \geq 2L + 5$ be two indices that satisfy the conditions of Assumption 3.$(b)$ and we define $\boldsymbol{l}_1 = (0, \ldots, 0, l_1, 0, \ldots, 0)$ ($l_1$ at position $q_\xi$) and $\boldsymbol{l}_2 = (0, \ldots, 0, l_2, 0, \ldots, 0)$ ($l_2$ at position $q_\xi$). Using the Gegenbauer coefficients of $\sigma'$, we define a new activation function $\hat{\sigma}'$ by

$$\hat{\sigma}'(\langle \boldsymbol{\theta}, \boldsymbol{x} \rangle / R) = \sum_{\boldsymbol{k} \in \mathbb{Z}_{\geq 0}^Q \setminus \{\boldsymbol{l}_1, \boldsymbol{l}_2\}} \lambda_{\boldsymbol{k}}^{\boldsymbol{d}}(\sigma'_{\boldsymbol{d},\boldsymbol{\tau}}) B(\boldsymbol{d}, \boldsymbol{k}) Q_{\boldsymbol{k}}^{\boldsymbol{d}} \left( \{ \langle \overline{\boldsymbol{\theta}}^{(q)}, \overline{\boldsymbol{x}}^{(q)} \rangle \}_{q \in [Q]} \right) \tag{105}$$

$$+ \sum_{t=1,2} (1 - \delta_t) \lambda_{\boldsymbol{l}_t}^{\boldsymbol{d}}(\sigma'_{\boldsymbol{d},\boldsymbol{\tau}}) B(d_{q_\xi}, l_t) Q_{l_t}^{(d_{q_\xi})} (\langle \overline{\boldsymbol{\theta}}^{(q_\xi)}, \overline{\boldsymbol{x}}^{(q_\xi)} \rangle), \tag{106}$$

for some $\delta_1, \delta_2$ that we will fix later (with $|\delta_t| \leq 1$).

**Step 2. The functions $\boldsymbol{u}, \hat{\boldsymbol{u}}$ and $\bar{\boldsymbol{u}}$.**

Let $\boldsymbol{u}$ and $\hat{\boldsymbol{u}}$ be the matrix-valued functions associated respectively to $\sigma'$ and $\hat{\sigma}'$

$$\boldsymbol{u}(\boldsymbol{\theta}_1, \boldsymbol{\theta}_2) = \mathbb{E}_{\boldsymbol{x}}[\sigma'(\langle \boldsymbol{\theta}_1, \boldsymbol{x} \rangle / R) \sigma'(\langle \boldsymbol{\theta}_2, \boldsymbol{x} \rangle / R) \boldsymbol{x} \boldsymbol{x}^\mathsf{T}], \tag{107}$$

$$\hat{\boldsymbol{u}}(\boldsymbol{\theta}_1, \boldsymbol{\theta}_2) = \mathbb{E}_{\boldsymbol{x}}[\hat{\sigma}'(\langle \boldsymbol{\theta}_1, \boldsymbol{x} \rangle / R) \hat{\sigma}'(\langle \boldsymbol{\theta}_2, \boldsymbol{x} \rangle / R) \boldsymbol{x} \boldsymbol{x}^\mathsf{T}]. \tag{108}$$

From Lemma 14, there exists functions $u_1^{ab}, u_{2,1}^{ab}, u_{2,2}^{ab}, u_{3,1}^{ab}, u_{3,2}^{ab}$ and $\hat{u}_1^{ab}, \hat{u}_{2,1}^{ab}, \hat{u}_{2,2}^{ab}, \hat{u}_{3,1}^{ab}, \hat{u}_{3,2}^{ab}$ (for $a, b \in [Q]$), which decompose $\boldsymbol{u}$ and $\hat{\boldsymbol{u}}$ along $\boldsymbol{\theta}_1$ and $\boldsymbol{\theta}_2$ vectors. We define $\bar{\boldsymbol{u}} = \boldsymbol{u} - \hat{\boldsymbol{u}}$. Then we have the same decomposition for $\bar{u}_{k,j}^{ab} = u_{k,j}^{ab} - \hat{u}_{k,j}^{ab}$ for $a, b \in [Q], k = 1, 2, 3, j = 1, 2$.

**Step 3. Construction of the kernel matrices.**

Let $\boldsymbol{U}, \hat{\boldsymbol{U}}, \bar{\boldsymbol{U}} \in \mathbb{R}^{ND \times ND}$ with $i, j$-th block (for $i, j \in [N]$) given by

$$\boldsymbol{U}_{ij} = \boldsymbol{u}(\boldsymbol{\theta}_i, \boldsymbol{\theta}_j), \tag{109}$$

$$\hat{\boldsymbol{U}}_{ij} = \hat{\boldsymbol{u}}(\boldsymbol{\theta}_i, \boldsymbol{\theta}_j), \tag{110}$$

$$\bar{\boldsymbol{U}}_{ij} = \bar{\boldsymbol{u}}(\boldsymbol{\theta}_i, \boldsymbol{\theta}_j) = \boldsymbol{u}(\boldsymbol{\theta}_i, \boldsymbol{\theta}_j) - \hat{\boldsymbol{u}}(\boldsymbol{\theta}_i, \boldsymbol{\theta}_j). \tag{111}$$

Note that we have $\boldsymbol{U} = \hat{\boldsymbol{U}} + \bar{\boldsymbol{U}}$. By Eq. (110) and (108), it is easy to see that $\hat{\boldsymbol{U}} \succeq 0$. Then we have $\boldsymbol{U} \succeq \bar{\boldsymbol{U}}$. In the following, we would like to lower bound matrix $\bar{\boldsymbol{U}}$.

We decompose $\bar{\boldsymbol{U}}$ as

$$\bar{\boldsymbol{U}} = \boldsymbol{D} + \boldsymbol{\Delta},$$

where $\boldsymbol{D} \in \mathbb{R}^{DN \times DN}$ is a block-diagonal matrix, with

$$\boldsymbol{D} = \mathrm{diag}(\bar{\boldsymbol{U}}_{11}, \ldots, \bar{\boldsymbol{U}}_{NN}), \tag{112}$$

and $\boldsymbol{\Delta} \in \mathbb{R}^{DN \times DN}$ is formed by blocks $\boldsymbol{\Delta}_{ij} \in \mathbb{R}^{D \times D}$ for $i, j \in [n]$, defined by

$$\boldsymbol{\Delta}_{ij} = \begin{cases} 0, & i = j, \\ \bar{\boldsymbol{U}}_{ij}, & i \neq j. \end{cases} \tag{113}$$

In the rest of the proof, we will prove that $\|\boldsymbol{\Delta}\|_{\mathrm{op}} = o_{d,\mathbb{P}}(d^{-\max_{q \in [Q]} \kappa_q})$ and the block matrix $\boldsymbol{D}$ verifies the properties (95) and (96).

**Step 4. Prove that $\|\boldsymbol{\Delta}\|_{\mathrm{op}} = o_{d,\mathbb{P}}(d^{-\max_{q \in [Q]} \kappa_q})$.**

We will prove in fact that $\|\boldsymbol{\Delta}\|_F^2 = o_{d,\mathbb{P}}(d^{-2\max_{q \in [Q]} \kappa_q})$. For the rest of the proof, we fix $\varepsilon_0 \in (0, 1)$ and we restrict ourselves without loss of generality to the set $\mathcal{P}_{\varepsilon_0}$.

Let us start with $\bar{\boldsymbol{u}}^{(qq)}$ for $q \in [Q]$. Denoting $\gamma_{ij}^{(q)} = \langle \overline{\boldsymbol{\theta}}_i^{(q)}, \overline{\boldsymbol{\theta}}_j^{(q)} \rangle / d_q < 1$, we get, from Eq. (104),

$$\begin{bmatrix} \bar{u}_1^{(qq)}(\boldsymbol{\theta}_i, \boldsymbol{\theta}_j) \\ \bar{u}_2^{(qq)}(\boldsymbol{\theta}_i, \boldsymbol{\theta}_j) \\ \bar{u}_{3,1}^{(qq)}(\boldsymbol{\theta}_i, \boldsymbol{\theta}_j) \\ \bar{u}_{3,2}^{(qq)}(\boldsymbol{\theta}_i, \boldsymbol{\theta}_j) \end{bmatrix} = \begin{bmatrix} u_1(\boldsymbol{\theta}_i, \boldsymbol{\theta}_j) - \hat{u}_1(\boldsymbol{\theta}_i, \boldsymbol{\theta}_j) \\ u_2(\boldsymbol{\theta}_i, \boldsymbol{\theta}_j) - \hat{u}_2(\boldsymbol{\theta}_i, \boldsymbol{\theta}_j) \\ u_{3,1}(\boldsymbol{\theta}_i, \boldsymbol{\theta}_j) - \hat{u}_{3,1}(\boldsymbol{\theta}_i, \boldsymbol{\theta}_j) \\ u_{3,2}(\boldsymbol{\theta}_i, \boldsymbol{\theta}_j) - \hat{u}_{3,2}(\boldsymbol{\theta}_i, \boldsymbol{\theta}_j) \end{bmatrix} = d_q^{-1}(\boldsymbol{M}_{ij}^{(qq)})^{-1} \times \begin{bmatrix} \mathrm{Tr}(\bar{\boldsymbol{u}}^{(qq)}(\boldsymbol{\theta}_1, \boldsymbol{\theta}_2)) \\ \langle \boldsymbol{\theta}_1^{(q)}, \bar{\boldsymbol{u}}^{(qq)}(\boldsymbol{\theta}_1, \boldsymbol{\theta}_2)\boldsymbol{\theta}_2^{(q)} \rangle \\ \langle \boldsymbol{\theta}_1^{(q)}, \bar{\boldsymbol{u}}^{(qq)}(\boldsymbol{\theta}_1, \boldsymbol{\theta}_2)\boldsymbol{\theta}_1^{(q)} \rangle \\ \langle \boldsymbol{\theta}_2^{(q)}, \bar{\boldsymbol{u}}^{(qq)}(\boldsymbol{\theta}_1, \boldsymbol{\theta}_2)\boldsymbol{\theta}_2^{(q)} \rangle \end{bmatrix}, \tag{114}$$

where $\boldsymbol{M}_{ij}^{(qq)}$ is given by

$$\begin{bmatrix} 1 & 2\tau_1^{(q)}\tau_2^{(q)}\gamma_{ij}^{(q)} & (\tau_1^{(q)})^2 & (\tau_2^{(q)})^2 \\ \tau_1^{(q)}\tau_2^{(q)}\gamma_{ij}^{(q)} & (\tau_1^{(q)})^2(\tau_2^{(q)})^2 d_q(1 + (\gamma_{ij}^{(q)})^2) & (\tau_1^{(q)})^3\tau_2^{(q)}d_q\gamma_{ij}^{(q)} & \tau_1^{(q)}(\tau_2^{(q)})^3 d_q\gamma_{ij}^{(q)} \\ (\tau_1^{(q)})^2 & 2(\tau_1^{(q)})^3\tau_2^{(q)}d_q\gamma^{(q)} & (\tau_1^{(q)})^4 d_q & (\tau_1^{(q)})^2(\tau_2^{(q)})^2 d_q(\gamma_{ij}^{(q)})^2 \\ (\tau_2^{(q)})^2 & \tau_1^{(q)}(\tau_2^{(q)})^3 d_q\gamma_{ij}^{(q)} & (\tau_1^{(q)})^2(\tau_2^{(q)})^2 d_q(\gamma^{(q)})^2 & (\tau_2^{(q)})^4 d_q \end{bmatrix}. \tag{115}$$

Using the notations of Lemma 13, we get

$$\begin{aligned} \mathrm{Tr}(\boldsymbol{U}_{ij}^{(qq)}) &= \mathbb{E}_{\boldsymbol{x}}[\sigma'(\langle \boldsymbol{\theta}_i, \boldsymbol{x} \rangle / R)\sigma'(\langle \boldsymbol{\theta}_j, \boldsymbol{x} \rangle / R)\|\boldsymbol{x}^{(q)}\|_2^2] \\ &= r_q^2 \sum_{\boldsymbol{k} \in \mathbb{Z}_{\geq 0}^Q} \lambda_{\boldsymbol{k}}^{\boldsymbol{d}}(\sigma'_{\boldsymbol{d},\boldsymbol{\tau}_i})\lambda_{\boldsymbol{k}}^{\boldsymbol{d}}(\sigma'_{\boldsymbol{d},\boldsymbol{\tau}_j})B(\boldsymbol{d}, \boldsymbol{k})Q_{\boldsymbol{k}}^{\boldsymbol{d}}\left(\{\langle \overline{\boldsymbol{\theta}}_i^{(q)}, \overline{\boldsymbol{\theta}}_j^{(q)} \rangle\}_{q \in [Q]}\right), \end{aligned}$$

$$\begin{aligned} \langle \boldsymbol{\theta}_i^{(q)}, \boldsymbol{U}_{ij}^{(qq)}\boldsymbol{\theta}_j^{(q)} \rangle &= \mathbb{E}_{\boldsymbol{x}}[\sigma'(\langle \boldsymbol{\theta}_i, \boldsymbol{x} \rangle / R)\langle \boldsymbol{\theta}_i^{(q)}, \boldsymbol{z}^{(q)} \rangle \sigma'(\langle \boldsymbol{\theta}_j, \boldsymbol{x} \rangle / R)\langle \boldsymbol{\theta}_j^{(q)}, \boldsymbol{x}^{(q)} \rangle] \\ &= r_q^2 \tau_i^{(q)}\tau_j^{(q)} \sum_{\boldsymbol{k} \in \mathbb{Z}_{\geq 0}^Q} \lambda_{\boldsymbol{k}}^{\boldsymbol{d},\boldsymbol{1}_q}(\sigma'_{\boldsymbol{d},\boldsymbol{\tau}_i})\lambda_{\boldsymbol{k}}^{\boldsymbol{d},\boldsymbol{1}_q}(\sigma'_{\boldsymbol{d},\boldsymbol{\tau}_j})B(\boldsymbol{d}, \boldsymbol{k})Q_{\boldsymbol{k}}^{\boldsymbol{d}}\left(\{\langle \overline{\boldsymbol{\theta}}_i^{(q)}, \overline{\boldsymbol{\theta}}_j^{(q)} \rangle\}_{q \in [Q]}\right), \end{aligned}$$

$$\begin{aligned} \langle \boldsymbol{\theta}_i^{(q)}, \boldsymbol{U}_{ij}^{(qq)}\boldsymbol{\theta}_i^{(q)} \rangle &= \mathbb{E}_{\boldsymbol{x}}[\sigma'(\langle \boldsymbol{\theta}_i, \boldsymbol{x} \rangle / R)\langle \boldsymbol{\theta}_i^{(q)}, \boldsymbol{x}^{(q)} \rangle^2 \sigma'(\langle \boldsymbol{\theta}_j, \boldsymbol{x} \rangle / R)] \\ &= r_q^2 (\tau_i^{(q)})^2 \sum_{\boldsymbol{k} \in \mathbb{Z}_{\geq 0}^Q} \lambda_{\boldsymbol{k}}^{\boldsymbol{d},\boldsymbol{2}_q}(\sigma'_{\boldsymbol{d},\boldsymbol{\tau}_i})\lambda_{\boldsymbol{k}}^{\boldsymbol{d}}(\sigma'_{\boldsymbol{d},\boldsymbol{\tau}_j})B(\boldsymbol{d}, \boldsymbol{k})Q_{\boldsymbol{k}}^{\boldsymbol{d}}\left(\{\langle \overline{\boldsymbol{\theta}}_i^{(q)}, \overline{\boldsymbol{\theta}}_j^{(q)} \rangle\}_{q \in [Q]}\right), \end{aligned}$$

$$\begin{aligned} \langle \boldsymbol{\theta}_j^{(q)}, \boldsymbol{U}_{ij}^{(qq)}\boldsymbol{\theta}_j^{(q)} \rangle &= \mathbb{E}_{\boldsymbol{x}}[\sigma'(\langle \boldsymbol{\theta}_i, \boldsymbol{x} \rangle / R)\sigma'(\langle \boldsymbol{\theta}_j, \boldsymbol{x} \rangle / R)\langle \boldsymbol{\theta}_j^{(q)}, \boldsymbol{x}^{(q)} \rangle^2] \\ &= r_q^2 (\tau_j^{(q)})^2 \sum_{\boldsymbol{k} \in \mathbb{Z}_{\geq 0}^Q} \lambda_{\boldsymbol{k}}^{\boldsymbol{d}}(\sigma'_{\boldsymbol{d},\boldsymbol{\tau}_i})\lambda_{\boldsymbol{k}}^{\boldsymbol{d},\boldsymbol{2}_q}(\sigma'_{\boldsymbol{d},\boldsymbol{\tau}_j})B(\boldsymbol{d}, \boldsymbol{k})Q_{\boldsymbol{k}}^{\boldsymbol{d}}\left(\{\langle \overline{\boldsymbol{\theta}}_i^{(q)}, \overline{\boldsymbol{\theta}}_j^{(q)} \rangle\}_{q \in [Q]}\right), \end{aligned}$$

where we denoted $\boldsymbol{1}_q = (0, \ldots, 0, 1, 0, \ldots, 0)$ (namely the $q$'th coordinate vector in $\mathbb{R}^Q$) and $\boldsymbol{2}_q = (0, \ldots, 0, 2, 0, \ldots, 0) = 2\boldsymbol{1}_q$.

We get similar expressions for $\hat{\boldsymbol{U}}_{ij}$ with $\lambda_{\boldsymbol{k}}^{\boldsymbol{d}}(\sigma'_{\boldsymbol{d},\boldsymbol{\tau}})$ replaced by $\lambda_{\boldsymbol{k}}^{\boldsymbol{d}}(\hat{\sigma}'_{\boldsymbol{d},\boldsymbol{\tau}})$. Because we defined $\sigma'$ and $\hat{\sigma}'$ by only modifying the $l_1$-th and $l_2$-th coefficients, we get

$$\begin{aligned} \mathrm{Tr}(\bar{\boldsymbol{U}}_{ij}^{(qq)}) &= \mathrm{Tr}(\boldsymbol{U}_{ij}^{(qq)} - \hat{\boldsymbol{U}}_{ij}^{(qq)}) \\ &= r_q^2 \sum_{t=1,2} \delta_t(2 - \delta_t)\lambda_{\boldsymbol{l}_t}^{\boldsymbol{d}}(\sigma'_{\boldsymbol{d},\boldsymbol{\tau}_i})\lambda_{\boldsymbol{l}_t}^{\boldsymbol{d}}(\sigma'_{\boldsymbol{d},\boldsymbol{\tau}_j})B(\boldsymbol{d}, \boldsymbol{l}_t)Q_{\boldsymbol{l}_t}^{\boldsymbol{d}}\left(\{d_q\gamma_{ij}^{(q)}\}_{q \in [Q]}\right). \end{aligned} \tag{116}$$

Recalling that $\lambda_{\boldsymbol{k}}^{\boldsymbol{d},\boldsymbol{1}_q}$ only depend on $\lambda_{\boldsymbol{k-1}_q}^{\boldsymbol{d}}$ and $\lambda_{\boldsymbol{k+1}_q}^{\boldsymbol{d}}$, and $\lambda_{\boldsymbol{k}}^{\boldsymbol{d},\boldsymbol{2}_q}$ on $\lambda_{\boldsymbol{k-2}_q}^{\boldsymbol{d}}$, $\lambda_{\boldsymbol{k}}^{\boldsymbol{d}}$ and $\lambda_{\boldsymbol{k+2}_q}^{\boldsymbol{d}}$, (Lemma 13), we get

$$
\begin{aligned}
&\langle \boldsymbol{\theta}_i^{(q)}, \bar{\boldsymbol{U}}_{ij}^{(qq)}\boldsymbol{\theta}_j^{(q)}\rangle \\
&= r_q^2 \tau_i^{(q)}\tau_j^{(q)} \sum_{t=\{1,2\},\boldsymbol{k}\in\{\boldsymbol{l}_t\pm\boldsymbol{1}_q\}} \delta_t(2-\delta_t)\lambda_{\boldsymbol{k}}^{\boldsymbol{d},\boldsymbol{1}_q}(\sigma'_{\boldsymbol{d},\tau_i})\lambda_{\boldsymbol{k}}^{\boldsymbol{d},\boldsymbol{1}_q}(\sigma'_{\boldsymbol{d},\tau_j})B(\boldsymbol{d},\boldsymbol{k})Q_{\boldsymbol{k}}^{\boldsymbol{d}}\left(\{d_q\gamma_{ij}^{(q)}\}_{q\in[Q]}\right), \\
&\langle \boldsymbol{\theta}_i^{(q)}, \bar{\boldsymbol{U}}_{ij}^{(qq)}\boldsymbol{\theta}_i^{(q)}\rangle \\
&= r_q^2 (\tau_i^{(q)})^2 \sum_{t\in\{1,2\},\boldsymbol{k}\in\{\boldsymbol{l}_t,\boldsymbol{l}_t\pm\boldsymbol{2}_q\}} \delta_t(2-\delta_t)\lambda_{\boldsymbol{k}}^{\boldsymbol{d},\boldsymbol{2}_q}(\sigma'_{\boldsymbol{d},\tau_i})\lambda_{\boldsymbol{k}}^{\boldsymbol{d}}(\sigma'_{\boldsymbol{d},\tau_j})B(\boldsymbol{d},\boldsymbol{k})Q_{\boldsymbol{k}}^{\boldsymbol{d}}\left(\{d_q\gamma_{ij}^{(q)}\}_{q\in[Q]}\right), \\
&\langle \boldsymbol{\theta}_j^{(q)}, \bar{\boldsymbol{U}}_{ij}^{(qq)}\boldsymbol{\theta}_j^{(q)}\rangle \\
&= r_q^2 (\tau_j^{(q)})^2 \sum_{t\in\{1,2\},\boldsymbol{k}\in\{\boldsymbol{l}_t,\boldsymbol{l}_t\pm\boldsymbol{2}_q\}} \delta_t(2-\delta_t)\lambda_{\boldsymbol{k}}^{\boldsymbol{d}}(\sigma'_{\boldsymbol{d},\tau_i})\lambda_{\boldsymbol{k}}^{\boldsymbol{d},\boldsymbol{2}_q}(\sigma'_{\boldsymbol{d},\tau_j})B(\boldsymbol{d},\boldsymbol{k})Q_{\boldsymbol{k}}^{\boldsymbol{d}}\left(\{d_q\gamma_{ij}^{(q)}\}_{q\in[Q]}\right),
\end{aligned}
\tag{117}
$$

where we used the convention $\lambda_{\boldsymbol{k}}^{\boldsymbol{d}}(\sigma'_{\boldsymbol{d},\tau})=0$ if one of the coordinates verifies $k_q<0$.

From Lemma 13, Lemma 19 and Lemma 20, we get for $t=1,2$ and $q\neq q_\xi$:

$$
\begin{aligned}
\lim_{(d,\tau_i,\tau_j)\to(+\infty,\boldsymbol{1},\boldsymbol{1})} \lambda_{\boldsymbol{l}_t}^{\boldsymbol{d}}(\sigma'_{\boldsymbol{d},\tau_i})\lambda_{\boldsymbol{l}_t}^{\boldsymbol{d}}(\sigma'_{\boldsymbol{d},\tau_j})B(\boldsymbol{d},\boldsymbol{l}_t) &= \frac{\mu_{l_t}(\sigma')^2}{l_t!}, \\
\lim_{(d,\tau_i,\tau_j)\to(+\infty,\boldsymbol{1},\boldsymbol{1})} \lambda_{\boldsymbol{l}_t+\boldsymbol{1}_q}^{\boldsymbol{d},\boldsymbol{1}_q}(\sigma'_{\boldsymbol{d},\tau_i})\lambda_{\boldsymbol{l}_t+\boldsymbol{1}_q}^{\boldsymbol{d},\boldsymbol{1}_q}(\sigma'_{\boldsymbol{d},\tau_j})B(\boldsymbol{d},\boldsymbol{l}_t+\boldsymbol{1}_q) &= \frac{\mu_{l_t}(\sigma')^2}{l_t!}, \\
\lim_{(d,\tau_i,\tau_j)\to(+\infty,\boldsymbol{1},\boldsymbol{1})} \lambda_{\boldsymbol{l}_t}^{\boldsymbol{d},\boldsymbol{2}_q}(\sigma'_{\boldsymbol{d},\tau_i})\lambda_{\boldsymbol{l}_t}^{\boldsymbol{d}}(\sigma'_{\boldsymbol{d},\tau_j})B(\boldsymbol{d},\boldsymbol{l}_t) &= \frac{\mu_{l_t}(\sigma')^2}{l_t!}, \\
\lim_{(d,\tau_i,\tau_j)\to(+\infty,\boldsymbol{1},\boldsymbol{1})} \lambda_{\boldsymbol{l}_t+\boldsymbol{2}_q}^{\boldsymbol{d},\boldsymbol{2}_q}(\sigma'_{\boldsymbol{d},\tau_i})\lambda_{\boldsymbol{l}_t+\boldsymbol{2}_q}^{\boldsymbol{d}}(\sigma'_{\boldsymbol{d},\tau_j})B(\boldsymbol{d},\boldsymbol{l}_t+\boldsymbol{2}_q) &= 0,
\end{aligned}
\tag{118}
$$

while for $q=q_\xi$ and $u\in\{-1,1\}$,

$$
\begin{aligned}
&\lim_{(d,\tau_i,\tau_j)\to(+\infty,\boldsymbol{1},\boldsymbol{1})} \lambda_{\boldsymbol{l}_t+u\boldsymbol{1}_{q_\xi}}^{\boldsymbol{d},\boldsymbol{1}_{q_\xi}}(\sigma'_{\boldsymbol{d},\tau_i})[B(\boldsymbol{d},\boldsymbol{l}_t+u\boldsymbol{1}_{q_\xi})(l_t+u)!]^{1/2} \\
&= \mu_{l_t+u+1}(\sigma') + (l_t+u)\mu_{l_t+u-1}(\sigma'),
\end{aligned}
\tag{119}
$$

and for $v\in\{-2,0,2\}$,

$$
\begin{aligned}
&\lim_{(d,\tau_i,\tau_j)\to(+\infty,\boldsymbol{1},\boldsymbol{1})} \lambda_{\boldsymbol{l}_t+v\boldsymbol{1}_{q_\xi}}^{\boldsymbol{d},\boldsymbol{2}_{q_\xi}}(\sigma'_{\boldsymbol{d},\tau_i})[B(\boldsymbol{d},\boldsymbol{l}_t+v\boldsymbol{1}_{q_\xi})(l_t+v)!]^{1/2} \\
&= \mu_{l_t+v+2}(\sigma') + (2l_t+2v+1)\mu_{l_t+v}(\sigma') + (l_t+v)(l_t+v-1)\mu_{l_t+v-2}(\sigma').
\end{aligned}
\tag{120}
$$

From Lemma (26), we recall that the coefficients of the $k$-th Gegenbauer polynomial $Q_k^{(d)}(x)=\sum_{s=0}^k p_{k,s}^{(d)}x^s$ satisfy

$$
p_{k,s}^{(d)} = O_d(d^{-k/2-s/2}).
\tag{121}
$$

Furthermore, Lemma 27 shows that $\max_{i\neq j}|\langle \overline{\boldsymbol{\theta}}_i^{(q)}, \overline{\boldsymbol{\theta}}_j^{(q)}\rangle| = O_{d,\mathbb{P}}(\sqrt{d_q\log d_q})$. We deduce that

$$
\max_{i\neq j}|Q_{k_q}^{(d_q)}(\langle \overline{\boldsymbol{\theta}}_i^{(q)}, \overline{\boldsymbol{\theta}}_j^{(q)}\rangle)| = \tilde{O}_{d,\mathbb{P}}(d_q^{-k_q/2})
\tag{122}
$$

Plugging the estimates (118) and (122) into Eqs. (116) and (117), we obtain that

$$
\begin{aligned}
&\max_{i\neq j}\left\{|\mathrm{Tr}(\bar{\boldsymbol{U}}_{ij}^{(qq)})|,\ |\langle \boldsymbol{\theta}_i^{(q)}, \bar{\boldsymbol{U}}_{ij}^{(qq)}\boldsymbol{\theta}_j^{(q)}\rangle|,\ |\langle \boldsymbol{\theta}_i^{(q)}, \bar{\boldsymbol{U}}_{ij}^{(qq)}\boldsymbol{\theta}_i^{(q)}\rangle,\ |\langle \boldsymbol{\theta}_j^{(q)}, \bar{\boldsymbol{U}}_{ij}^{(qq)}\boldsymbol{\theta}_j^{(q)}\rangle|\right\} \\
&= \tilde{O}_{d,\mathbb{P}}(d^{2\xi}d^{-\eta_q l_1/2}).
\end{aligned}
\tag{123}
$$

From Eq. (115), using the fact that $\max_{i\neq j}|\gamma_{ij}^{(q)}| = O_{d,\mathbb{P}}(\sqrt{(\log d_q)/d_q})$ and Cramer's rule for matrix inversion, it is easy to see that

$$
\max_{i\neq j}\max_{l,k\in[4]}\left|((\boldsymbol{M}_{ij}^{(qq)})^{-1})_{lk}\right| = O_{d,\mathbb{P}}(1).
\tag{124}
$$

We deduce from (123), (114) and (124) that for $a \in [3], b \in [2]$,

$$\max_{i \neq j}\{|\bar{u}_{a,b}^{(qq)}(\boldsymbol{\theta}_i^{(q)}, \boldsymbol{\theta}_j^{(q)})|\} = \tilde{O}_{d,\mathbb{P}}(d^{2\xi}d^{-\eta_q l_1/2}). \tag{125}$$

As a result, combining Eq. (125) with Eq. (111) in the expression of $\overline{u}^{(qq)}$ given in Lemma 14, we get

$$\max_{i \neq j}\|\bar{\boldsymbol{U}}_{ij}^{(qq)}\|_F^2$$
$$= \max_{i \neq j}\|\bar{u}_1^{(qq)}\mathbf{I}_{d_q} + \bar{u}_2^{(qq)}[\boldsymbol{\theta}_i^{(q)}(\boldsymbol{\theta}_j^{(q)})^\mathsf{T} + \boldsymbol{\theta}_j^{(q)}(\boldsymbol{\theta}_i^{(q)})^\mathsf{T}] + \bar{u}_{3,1}^{(qq)}\boldsymbol{\theta}_i^{(q)}(\boldsymbol{\theta}_i^{(q)})^\mathsf{T} + \bar{u}_{3,2}^{(qq)}\boldsymbol{\theta}_j^{(q)}(\boldsymbol{\theta}_j^{(q)})^\mathsf{T}\|_F^2$$
$$\leq \tilde{O}_{d,\mathbb{P}}(d^{6\xi}d^{-\eta_q l_1}).$$

A similar computation shows that

$$\max_{i \neq j}\|\bar{\boldsymbol{U}}_{ij}^{(qq')}\|_F^2 \leq \tilde{O}_{d,\mathbb{P}}(d^{6\xi}d^{-\eta_q l_1}).$$

By the expression of $\boldsymbol{\Delta}$ given by (113), we conclude that

$$\|\boldsymbol{\Delta}\|_{\mathrm{op}}^2 \leq \|\boldsymbol{\Delta}\|_F^2 = \sum_{q,q' \in [Q]} \sum_{i,j=1, i \neq j}^{N} \|\bar{U}_{ij}^{(qq')}\|_F^2 = \tilde{O}_{d,\mathbb{P}}(N^2 d^{6\xi - \eta_q l_1}).$$

By assumption, $N = o_d(d^\gamma)$. Hence, since by assumption $\eta_q l_1 \geq 2\gamma + 7\xi$, we deduce that $\|\boldsymbol{\Delta}\|_{\mathrm{op}} = o_{d,\mathbb{P}}(d^{-\xi}) = o_{d,\mathbb{P}}(d^{-\max_{q \in [Q]} \kappa_q})$.

**Step 5. Checking the properties of matrix $D$.**

By Lemma 14, we can express $\bar{\boldsymbol{U}}_{ii}$ as a block matrix with

$$\bar{\boldsymbol{U}}_{ii}^{(qq)} = \alpha^{(q)}\mathbf{I}_{d_q} + \beta^{(q)}\boldsymbol{\theta}_i^{(q)}(\boldsymbol{\theta}_i^{(q)})^\mathsf{T}, \qquad \bar{\boldsymbol{U}}_{ii}^{(qq')} = \beta^{(qq')}\boldsymbol{\theta}_i^{(q)}(\boldsymbol{\theta}_i^{(q')})^\mathsf{T},$$

with coefficients given by

$$\begin{bmatrix} \alpha^{(q)} \\ \beta^{(q)} \end{bmatrix} = [d_q(d_q - 1)(\tau_i^{(q)})^4]^{-1} \begin{bmatrix} d_q(\tau_i^{(q)})^4 & -(\tau_i^{(q)})^2 \\ -(\tau_i^{(q)})^2 & 1 \end{bmatrix} \times \begin{bmatrix} \mathrm{Tr}(\bar{\boldsymbol{U}}_{ii}^{(qq)}) \\ \langle \boldsymbol{\theta}_i^{(q)}, \bar{\boldsymbol{U}}_{ii}^{(qq)}\boldsymbol{\theta}_i^{(q)} \rangle \end{bmatrix}, \tag{126}$$
$$\beta^{(qq')} = (d_q d_{q'})^{-1}(\tau_i^{(q)}\tau_i^{(q')})^{-2}\langle \boldsymbol{\theta}_i^{(q)}, \bar{\boldsymbol{U}}_{ii}^{(qq')}(\boldsymbol{\theta}_i, \boldsymbol{\theta}_i)\boldsymbol{\theta}_i^{(q')} \rangle.$$

Let us first focus on the $q = q_\xi$ sphere. Using Eqs. (116) and (117) with the expressions (119) and (120), we get the following convergence in probability (using that $\{\tau_i^{(q)}\}_{i \in [N]}$ concentrates on 1),

$$\sup_{i \in [N]} \left| r_{q_\xi}^{-2}\mathrm{Tr}(\bar{\boldsymbol{U}}_{ii}^{(q_\xi q_\xi)}) - F_1(\boldsymbol{\delta}) \right| \xrightarrow{\mathbb{P}} 0,$$
$$\sup_{i \in [N]} \left| r_{q_\xi}^{-2}\langle \boldsymbol{\theta}_i^{(q_\xi)}, \bar{\boldsymbol{U}}_{ii}^{(q_\xi q_\xi)}\boldsymbol{\theta}_i^{(q_\xi)} \rangle - F_2(\boldsymbol{\delta}) \right| \xrightarrow{\mathbb{P}} 0, \tag{127}$$

where we denoted $\boldsymbol{\delta} = (\delta_1, \delta_2)$ (where $\delta_1, \delta_2$ first appears in the definition of $\hat{\sigma}$ in Eq. (105), and till now $\delta_1, \delta_2$ are still not determined) and, similarly to the proof of [4, Proposition 5] and letting $\mu_k \equiv \mu_k(\sigma')$, we have

$$F_1(\boldsymbol{\delta}) = \sum_{t \in \{1,2\}} \delta_t(2 - \delta_t)\frac{\mu_{l_t}^2}{l_t!}, \tag{128}$$

while, for $l_2 \neq l_1 + 2$

$$F_2(\boldsymbol{\delta}) = \sum_{t \in \{1,2\}} \left\{ \frac{1}{(l_t - 1)!}\left[(\mu_{l_t} + (l_t - 1)\mu_{l_t - 2})^2 - ((1 - \delta_t)\mu_{l_t} + (l_t - 1)\mu_{l_t - 2})^2\right]\right.$$
$$\left. + \frac{1}{(l_t + 1)!}\left[(\mu_{l_t+2} + (l_t + 1)\mu_{l_t})^2 - (\mu_{l_t+2} + (1 - \delta_t)(l_t + 1)\mu_{l_t})^2\right]\right\},$$

while, for $l_2 = l_1 + 2$

$$F_2(\boldsymbol{\delta}) = \frac{1}{(l_1 - 1)!}\Big[(\mu_{l_1} + (l_1 - 1)\mu_{l_1 - 2})^2 - ((1 - \delta_1)\mu_{l_1} + (l_1 - 1)\mu_{l_1 - 2})^2\Big]$$
$$+ \frac{1}{(l_1 + 1)!}\Big[(\mu_{l_1 + 2} + (l_1 + 1)\mu_{l_1})^2 - ((1 - \delta_2)\mu_{l_1 + 2} + (1 - \delta_1)(l_1 + 1)\mu_{l_1})^2\Big]$$
$$+ \frac{1}{(l_2 + 1)!}\Big[(\mu_{l_2 + 2} + (l_2 + 1)\mu_{l_2})^2 - (\mu_{l_2 + 2} + (1 - \delta_2)(l_2 + 1)\mu_{l_2})^2\Big].$$

We have from Eq. (126),

$$\lambda_{\min}(\bar{U}_{ii}^{(qq)})$$
$$= \min\left\{\alpha^{(q)}, \alpha^{(q)} + \beta^{(q)} d_q(\tau_i^{(q)})^2\right\}$$
$$= \min\left\{\frac{1}{d_q - 1}\mathrm{Tr}(\bar{U}_{ii}^{(qq)}) - \frac{1}{d_q(d_q - 1)(\tau_i^{(q)})^2}\langle\boldsymbol{\theta}_i^{(q)}, \bar{U}_{ii}^{(qq)}\boldsymbol{\theta}_i^{(q)}\rangle, \frac{1}{d_q(\tau_i^{(q)})^2}\langle\boldsymbol{\theta}_i^{(q)}, \bar{U}_{ii}^{(qq)}\boldsymbol{\theta}_i^{(q)}\rangle\right\}.$$

Hence, using Eq. (127), we get

$$\sup_{i\in[N]}\left|\frac{d_{q_\xi}}{r_{q_\xi}^2}\lambda_{\min}(\bar{U}_{ii}^{(q_\xi q_\xi)}) - \min\{F_1(\boldsymbol{\delta}), F_2(\boldsymbol{\delta})\}\right| \xrightarrow{\mathbb{P}} 0. \qquad (129)$$

Following the same reasoning as in [4, Proposition 5], we can verify that under Assumption 3.$(b)$, we have $\nabla F_1(\mathbf{0}), \nabla F_2(\mathbf{0}) \neq \mathbf{0}$ and $\det(\nabla F_1(\mathbf{0}), \nabla F_2(\mathbf{0})) \neq 0$. We can therefore find $\boldsymbol{\delta} = (\delta_1, \delta_2)$ such that $F_1(\boldsymbol{\delta}) > 0$, $F_2(\boldsymbol{\delta}) > 0$. Furthermore,

$$\sup_{i\in[N]}\left|\frac{d_{q_\xi}}{r_{q_\xi}^2}\lambda_{\max}(\bar{U}_{ii}^{(q_\xi q_\xi)}) - \max\{F_1(\boldsymbol{\delta}), F_2(\boldsymbol{\delta})\}\right| \xrightarrow{\mathbb{P}} 0. \qquad (130)$$

Similarly, we get for $q \neq q_\xi$ from Eqs. (116) and (117) with the expressions (118) (recalling that $\{\tau_i^{(q)}\}_{i\in[N]}$ concentrates on 1),

$$\sup_{i\in[N]}\left|r_q^{-2}\mathrm{Tr}(\bar{U}_{ii}^{(qq)}) - F_1(\boldsymbol{\delta})\right| \xrightarrow{\mathbb{P}} 0,$$
$$\sup_{i\in[N]}\left|r_q^{-2}\langle\boldsymbol{\theta}_i^{(q)}, \bar{U}_{ii}^{(qq)}\boldsymbol{\theta}_i^{(q)}\rangle - F_1(\boldsymbol{\delta})\right| \xrightarrow{\mathbb{P}} 0, \qquad (131)$$
$$\sup_{i\in[N]}\left|(r_q r_{q'})^{-1}\langle\boldsymbol{\theta}_i^{(q)}, \bar{U}_{ii}^{(qq')}\boldsymbol{\theta}_i^{(q')}\rangle\right| \xrightarrow{\mathbb{P}} 0.$$

We deduce that for $q \neq q_\xi$ and $q \neq q'$,

$$\sup_{i\in[N]}\left|\frac{d_q}{r_q^2}\lambda_{\min}(\bar{U}_{ii}^{(qq)}) - F_1(\boldsymbol{\delta})\right| \xrightarrow{\mathbb{P}} 0,$$
$$\sup_{i\in[N]}\left|\frac{d_q}{r_q^2}\lambda_{\max}(\bar{U}_{ii}^{(qq)}) - F_1(\boldsymbol{\delta})\right| \xrightarrow{\mathbb{P}} 0,$$
$$\sup_{i\in[N]}\left|\frac{(d_q d_{q'})^{1/2}}{r_q r_{q'}}\sigma_{\max}(\bar{U}_{ii}^{(qq')})\right| \xrightarrow{\mathbb{P}} 0,$$

which finishes to prove properties (95) and (96).

# H Proof of Theorem 3.(b): upper bound for NT model

## H.1 Preliminaries

**Lemma 15.** *Let $\sigma$ be an activation function that satisfies Assumptions 3.(a) and 3.(c) for some level $\gamma > 0$. Let $\mathcal{Q} = \overline{\mathcal{Q}}_{\mathrm{NT}}(\gamma)$ as defined in Eq. (39). Define for integer $\boldsymbol{k} \in \mathbb{Z}_{\geq 0}^Q$ and $\boldsymbol{\tau}, \boldsymbol{\tau}' \in \mathbb{R}_{\geq 0}^Q$,*

$$
\begin{aligned}
A_{(\boldsymbol{\tau},\boldsymbol{\tau}'),\boldsymbol{k}}^{(q)} = r_q^2 \cdot \Big[ & t_{d_q,k_q-1} \lambda_{\boldsymbol{k}_{q-}}^{\boldsymbol{d}}(\sigma_{\boldsymbol{d},\boldsymbol{\tau}}') \lambda_{\boldsymbol{k}_{q-}}^{\boldsymbol{d}}(\sigma_{\boldsymbol{d},\boldsymbol{\tau}'}') B(\boldsymbol{d}, \boldsymbol{k}_{q-}) \\
& + s_{d_q,k_q+1} \lambda_{\boldsymbol{k}_{q+}}^{\boldsymbol{d}}(\sigma_{\boldsymbol{d},\boldsymbol{\tau}}') \lambda_{\boldsymbol{k}_{q+}}^{\boldsymbol{d}}(\sigma_{\boldsymbol{d},\boldsymbol{\tau}'}') B(\boldsymbol{d}, \boldsymbol{k}_{q+}) \Big].
\end{aligned}
\tag{132}
$$

*with $\boldsymbol{k}_{q+} = (k_1, \ldots, k_q + 1, \ldots, k_Q)$ and $\boldsymbol{k}_{q-} = (k_1, \ldots, k_q - 1, \ldots, k_Q)$, and*

$$
s_{d,k} = \frac{k}{2k+d-2}, \qquad t_{d,k} = \frac{k+d-2}{2k+d-2},
$$

*with the convention $t_{d,-1} = 0$.*

*Then there exists constants $\varepsilon_0 > 0$ and $C > 0$ such that for $d$ large enough, we have for any $\boldsymbol{\tau}, \boldsymbol{\tau}' \in [1 - \varepsilon_0, 1 + \varepsilon_0]^Q$,*

$$
\max_{\boldsymbol{k} \in \mathcal{Q}} \frac{B(\boldsymbol{d}, \boldsymbol{k})}{A_{(\boldsymbol{\tau},\boldsymbol{\tau}'),\boldsymbol{k}}^{(q)}} \leq C d^{\gamma - \kappa_q}.
$$

*Proof of Lemma 15.* From Assumptions 3.(a) and 3.(c) and Lemma 19, there exists $c > 0$ and $\varepsilon_0 > 0$ such that for any $\boldsymbol{\tau}, \boldsymbol{\tau}' \in [1 - \varepsilon_0, 1 + \varepsilon_0]^Q$ and $\boldsymbol{k} \in \mathcal{Q}$,

$$
\lambda_{\boldsymbol{k}}^{\boldsymbol{d}}(\sigma_{\boldsymbol{d},\boldsymbol{\tau}}') \lambda_{\boldsymbol{k}}^{\boldsymbol{d}}(\sigma_{\boldsymbol{d},\boldsymbol{\tau}'}') \geq c \prod_{q \in [Q]} d^{k_q(\kappa_q - \xi)}.
$$

Hence for $k_q > 0$, we get $\lambda_{\boldsymbol{k}}^{\boldsymbol{d}}(\sigma_{\boldsymbol{d},\boldsymbol{\tau}}') \lambda_{\boldsymbol{k}}^{\boldsymbol{d}}(\sigma_{\boldsymbol{d},\boldsymbol{\tau}'}') \geq c d^{-\gamma - \xi + \kappa_q}$, and for $k_q = 0$, we get $\lambda_{\boldsymbol{k}}^{\boldsymbol{d}}(\sigma_{\boldsymbol{d},\boldsymbol{\tau}}') \lambda_{\boldsymbol{k}}^{\boldsymbol{d}}(\sigma_{\boldsymbol{d},\boldsymbol{\tau}'}') \geq c d^{-\gamma + \xi - \eta_q - \kappa_q}$. Carefully injecting these bounds in Eq. (132) yields the lemma. $\qquad\square$

## H.2 Proof of Theorem 3.(b): outline

In this proof, we will consider $Q$ sub-classes of functions corresponding to the NT model restricted to the $q$-th sphere:

$$
\mathcal{F}_{\mathrm{NT}^{(q)}}(\boldsymbol{W}) \equiv \Big\{ f(\boldsymbol{x}) = \sum_{i=1}^N \langle \boldsymbol{a}_i, \boldsymbol{x}^{(q)} \rangle \sigma'(\langle \boldsymbol{w}_i, \boldsymbol{x} \rangle / R) \; : \; \boldsymbol{a}_i \in \mathbb{R}^{d_q}, i \in [N] \Big\}.
$$

We define similarly the risk associated to this sub-model

$$
R_{\mathrm{NT}^{(q)}}(f_d, \boldsymbol{W}) = \inf_{f \in \mathcal{F}_{\mathrm{NT}^{(q)}}(\boldsymbol{W})} \mathbb{E}[(f_d(\boldsymbol{x}) - f(\boldsymbol{x}))^2].
$$

and approximation subspace

$$
\begin{aligned}
\overline{\mathcal{Q}}_{\mathrm{NT}^{(q)}}(\gamma) = & \Big\{ \boldsymbol{k} \in \mathbb{Z}_{\geq 0}^Q \Big| k_q > 0 \text{ and } \sum_{q \in [Q]} k_q(\xi - \kappa_q) \leq \gamma + (\xi - \kappa_q) \Big\} \\
& \cup \Big\{ \boldsymbol{k} \in \mathbb{Z}_{\geq 0}^Q \Big| k_q = 0 \text{ and } \sum_{q \in [Q]} k_q(\xi - \kappa_q) \leq \gamma - (\xi - \kappa_q - \eta_q) \Big\}.
\end{aligned}
\tag{133}
$$

**Theorem 4.** *Let $\{f_d \in L^2(\mathrm{PS}_{\boldsymbol{\kappa}}^{\boldsymbol{d}}, \mu_{\boldsymbol{d}}^{\boldsymbol{\kappa}})\}_{d \geq 1}$ be a sequence of functions. Let $\boldsymbol{W} = (\boldsymbol{w}_i)_{i \in [N]}$ with $(\boldsymbol{w}_i)_{i \in [N]} \sim \mathrm{Unif}(\mathbb{S}^{D-1})$ independently. Assume $N \geq \omega_d(d^\gamma)$ for some positive constant $\gamma > 0$, and $\sigma$ satisfy Assumptions 3.(a) and 3.(c) at level $\gamma$. Then for any $\varepsilon > 0$, the following holds with high probability:*

$$
0 \leq R_{\mathrm{NT}^{(q)}}(\mathsf{P}_{\mathcal{Q}} f_d, \boldsymbol{W}) \leq \varepsilon \| \mathsf{P}_{\mathcal{Q}} f_d \|_{L^2}^2,
\tag{134}
$$

*where $\mathcal{Q} \equiv \overline{\mathcal{Q}}_{\mathrm{NT}^{(q)}}(\gamma)$ is defined in Equation (133).*

**Remark 5.** *From the proof of Theorem 3.(a), we have a matching lower bound for $\mathcal{F}_{\mathrm{NT}^{(q)}}$.*

We recall

$$\overline{\mathcal{Q}}_{\mathrm{NT}}(\gamma) = \Big\{ \boldsymbol{k} \in \mathbb{Z}_{\geq 0}^{Q} \, \Big| \, \sum_{q=1}^{Q} (\xi - \kappa_q) k_q \leq \gamma + \Big( \xi - \min_{q \in S(\boldsymbol{k})} \kappa_q \Big) \Big\}.$$

Notice that

$$\overline{\mathcal{Q}}_{\mathrm{NT}}(\gamma) = \bigcup_{q \in Q} \overline{\mathcal{Q}}_{\mathrm{NT}^{(q)}}(\gamma)$$

Denote $q_{\boldsymbol{k}} = \arg\min_{q \in S(\boldsymbol{k})} \kappa_q$, such that $\boldsymbol{k} \in \overline{\mathcal{Q}}_{\mathrm{NT}^{(q_{\boldsymbol{k}})}}$ for any $\boldsymbol{k} \in \overline{\mathcal{Q}}_{\mathrm{NT}}(\gamma)$. Furthermore, notice that by definition for any $f \in L^2(\mathrm{PS}_{\boldsymbol{\kappa}}^{\boldsymbol{d}}, \mu_{\boldsymbol{d}}^{\boldsymbol{\kappa}})$ and $q \in [Q]$,

$$R_{\mathrm{NT}}(f, \boldsymbol{W}) \leq R_{\mathrm{NT}^{(q)}}(f, \boldsymbol{W}).$$

Let us deduce Theorem 3.(b) from Theorem 4. Denote $\mathcal{Q} = \overline{\mathcal{Q}}_{\mathrm{NT}}(\gamma)$. We divide the $N$ neurons in $|\mathcal{Q}|$ sections of size $N' = N/|\mathcal{Q}|$, i.e. $\boldsymbol{W} = (\boldsymbol{W}_{\boldsymbol{k}})_{\boldsymbol{k} \in \mathcal{Q}}$ where $\boldsymbol{W}_{\boldsymbol{k}} \in \mathbb{R}^{N' \times d}$. For any $\varepsilon > 0$, we get from Theorem 4 that with high probability

$$R_{\mathrm{NT}}(\mathsf{P}_{\mathcal{Q}} f_d, \boldsymbol{W}) \leq \sum_{\boldsymbol{k} \in \mathcal{Q}} R_{\mathrm{NT}^{(q_{\boldsymbol{k}})}}(\mathsf{P}_{\boldsymbol{k}} f, \boldsymbol{W}_{\boldsymbol{k}}) \leq \sum_{\boldsymbol{k} \in \mathcal{Q}} \varepsilon \|\mathsf{P}_{\boldsymbol{k}} f_d\|_{L^2}^2 = \varepsilon \|\mathsf{P}_{\mathcal{Q}} f_d\|_{L^2}^2.$$

## H.3 Proof of Theorem 4

### H.3.1 Properties of the limiting kernel

Similarly to the proof of Theorem 2.(b), we construct a limiting kernel which is used as a proxy to upper bound the $\mathrm{NT}^{(q)}$ risk.

We recall the definition of $\mathrm{PS}^{\boldsymbol{d}} = \prod_{q \in [Q]} \mathbb{S}^{d_q - 1}(\sqrt{d_q})$. We introduce $\mathcal{L} = L^2(\mathrm{PS}^{\boldsymbol{d}} \to \mathbb{R}, \mu_{\boldsymbol{d}})$ and $\mathcal{L}_{d_q} = L^2(\mathrm{PS}^{\boldsymbol{d}} \to \mathbb{R}^{d_q}, \mu_{\boldsymbol{d}})$. For a given $\boldsymbol{\theta} \in \mathbb{S}^{D-1}(\sqrt{D})$ and associated vector $\boldsymbol{\tau} \in \mathbb{R}_{\geq 0}^{Q}$, recall the definition of $\sigma'_{\boldsymbol{d},\boldsymbol{\tau}} \in \mathcal{L}$:

$$\sigma'_{\boldsymbol{d},\boldsymbol{\tau}}\left(\{\langle \overline{\boldsymbol{\theta}}^{(q)}, \overline{\boldsymbol{x}}^{(q)}\rangle / \sqrt{d_q}\}_{q \in [Q]}\right) = \sigma'\left(\sum_{q \in [Q]} \tau^{(q)} \cdot (r_q / R) \cdot \langle \overline{\boldsymbol{\theta}}^{(q)}, \overline{\boldsymbol{x}}^{(q)}\rangle / \sqrt{d_q}\right)$$

For any $\boldsymbol{\tau} \in \mathbb{R}_{\geq 0}^{Q}$, define the operator $\mathbb{T}_{\boldsymbol{\tau}} : \mathcal{L} \to \mathcal{L}_{d_q}$, such that for any $g \in \mathcal{L}$,

$$\mathbb{T}_{\boldsymbol{\tau}} g(\overline{\boldsymbol{\theta}}) = \frac{r_q}{\sqrt{d_q}} \mathbb{E}_{\overline{\boldsymbol{x}}}\Big[ \overline{\boldsymbol{x}}^{(q)} \sigma'_{\boldsymbol{d},\boldsymbol{\tau}}\left(\{\langle \overline{\boldsymbol{\theta}}^{(q)}, \overline{\boldsymbol{x}}^{(q)}\rangle / \sqrt{d_q}\}_{q \in [Q]}\right) g(\overline{\boldsymbol{x}}) \Big].$$

The adjoint operator $\mathbb{T}_{\boldsymbol{\tau}}^* : \mathcal{L}_{d_q} \to \mathcal{L}$ verifies for any $h \in \mathcal{L}_{d_q}$,

$$\mathbb{T}_{\boldsymbol{\tau}}^* h(\overline{\boldsymbol{x}}) = \frac{r_q}{\sqrt{d_q}} (\overline{\boldsymbol{x}}^{(q)})^{\mathsf{T}} \mathbb{E}_{\overline{\boldsymbol{\theta}}}\Big[ \sigma'_{\boldsymbol{d},\boldsymbol{\tau}}\left(\{\langle \overline{\boldsymbol{\theta}}^{(q)}, \overline{\boldsymbol{x}}^{(q)}\rangle / \sqrt{d_q}\}_{q \in [Q]}\right) h(\overline{\boldsymbol{\theta}}) \Big].$$

We define the operator $\mathbb{K}_{\boldsymbol{\tau},\boldsymbol{\tau}'} : \mathcal{L}_{d_q} \to \mathcal{L}_{d_q}$ as $\mathbb{K}_{\boldsymbol{\tau},\boldsymbol{\tau}'} \equiv \mathbb{T}_{\boldsymbol{\tau}} \mathbb{T}_{\boldsymbol{\tau}'}^*$. For $h \in \mathcal{L}_{d_q}$, we can write

$$\mathbb{K}_{\boldsymbol{\tau},\boldsymbol{\tau}'} h(\overline{\boldsymbol{\theta}}_1) = \mathbb{E}_{\overline{\boldsymbol{\theta}}_2}[\mathbb{K}_{\boldsymbol{\tau},\boldsymbol{\tau}'}(\overline{\boldsymbol{\theta}}_1, \overline{\boldsymbol{\theta}}_2) h(\overline{\boldsymbol{\theta}}_2)],$$

where

$$\mathbb{K}_{\boldsymbol{\tau}_1, \boldsymbol{\tau}_2}(\overline{\boldsymbol{\theta}}_1, \overline{\boldsymbol{\theta}}_2)$$
$$= \frac{r_q^2}{d_q} \mathbb{E}_{\overline{\boldsymbol{x}}}\Big[ \overline{\boldsymbol{x}}^{(q)} (\overline{\boldsymbol{x}}^{(q)})^{\mathsf{T}} \sigma'_{\boldsymbol{d},\boldsymbol{\tau}_1}\left(\{\langle \overline{\boldsymbol{\theta}}_1^{(q)}, \overline{\boldsymbol{x}}^{(q)}\rangle / \sqrt{d_q}\}_{q \in [Q]}\right) \sigma'_{\boldsymbol{d},\boldsymbol{\tau}_2}\left(\{\langle \overline{\boldsymbol{\theta}}_2^{(q)}, \overline{\boldsymbol{x}}^{(q)}\rangle / \sqrt{d_q}\}_{q \in [Q]}\right) \Big].$$

Define $\mathbb{H}_{\boldsymbol{\tau},\boldsymbol{\tau}'} : \mathcal{L} \to \mathcal{L}$ as $\mathbb{H}_{\boldsymbol{\tau},\boldsymbol{\tau}'} \equiv \mathbb{T}_{\boldsymbol{\tau}}^* \mathbb{T}_{\boldsymbol{\tau}'}$. For $g \in \mathcal{L}$, we can write

$$\mathbb{H}_{\boldsymbol{\tau},\boldsymbol{\tau}'} g(\overline{\boldsymbol{x}}_1) = \mathbb{E}_{\overline{\boldsymbol{x}}_2}[\mathbb{H}_{\boldsymbol{\tau},\boldsymbol{\tau}'}(\overline{\boldsymbol{x}}_1, \overline{\boldsymbol{x}}_2) g(\overline{\boldsymbol{x}}_2)]$$

where

$$\mathbb{H}_{\boldsymbol{\tau},\boldsymbol{\tau}'}(\overline{\boldsymbol{x}}_1,\overline{\boldsymbol{x}}_2)$$
$$=\frac{r_q^2}{d_q}\mathbb{E}_{\overline{\boldsymbol{\theta}}}\Big[\sigma'_{\boldsymbol{d},\boldsymbol{\tau}}\Big(\{\langle\overline{\boldsymbol{\theta}}^{(q)},\overline{\boldsymbol{x}}_1^{(q)}\rangle/\sqrt{d_q}\}_{q\in[Q]}\Big)\,\sigma'_{\boldsymbol{d},\boldsymbol{\tau}'}\Big(\{\langle\overline{\boldsymbol{\theta}}^{(q)},\overline{\boldsymbol{x}}_2^{(q)}\rangle/\sqrt{d_q}\}_{q\in[Q]}\Big)\Big]\langle\overline{\boldsymbol{x}}_1^{(q)},\overline{\boldsymbol{x}}_2^{(q)}\rangle.$$

We recall the decomposition of $\sigma'_{\boldsymbol{d},\boldsymbol{\tau}}$ in terms of tensor product of Gegenbauer polynomials:

$$\sigma'_{\boldsymbol{d},\boldsymbol{\tau}}(\overline{x}_1^{(1)},\ldots,\overline{x}_1^{(Q)})=\sum_{\boldsymbol{k}\in\mathbb{Z}_{\geq 0}^Q}\lambda_{\boldsymbol{k}}^{\boldsymbol{d}}(\sigma'_{\boldsymbol{d},\boldsymbol{\tau}})B(\boldsymbol{d},\boldsymbol{k})Q_{\boldsymbol{k}}^{\boldsymbol{d}}\Big(\sqrt{d_1}\overline{x}_1^{(1)},\ldots,\sqrt{d_Q}\overline{x}_1^{(Q)}\Big),$$

$$\lambda_{\boldsymbol{k}}^{\boldsymbol{d}}(\sigma'_{\boldsymbol{d},\boldsymbol{\tau}})=\mathbb{E}_{\overline{\boldsymbol{x}}}\Big[\sigma'_{\boldsymbol{d},\boldsymbol{\tau}}(\overline{x}_1^{(1)},\ldots,\overline{x}_1^{(Q)})Q_{\boldsymbol{k}}^{\boldsymbol{d}}\Big(\sqrt{d_1}\overline{x}_1^{(1)},\ldots,\sqrt{d_Q}\overline{x}_1^{(Q)}\Big)\Big].$$

Following the same computations as in Lemma 12, we get

$$\mathbb{H}_{\boldsymbol{\tau},\boldsymbol{\tau}'}(\overline{\boldsymbol{x}}_1,\overline{\boldsymbol{x}}_2)=\sum_{\boldsymbol{k}\in\mathbb{Z}_{\geq 0}^Q}A_{(\boldsymbol{\tau},\boldsymbol{\tau}'),\boldsymbol{k}}^{(q)}Q_{\boldsymbol{k}}^{\boldsymbol{d}}\Big(\{\langle\overline{\boldsymbol{x}}_1^{(q)},\overline{\boldsymbol{x}}_2^{(q)}\rangle\}_{q\in[Q]}\Big),$$

where

$$A_{(\boldsymbol{\tau},\boldsymbol{\tau}'),\boldsymbol{k}}^{(q)}=r_q^2\cdot\Big[t_{d_q,k_q-1}\lambda_{\boldsymbol{k}_{q-}}^{\boldsymbol{d}}(\sigma'_{\boldsymbol{d},\boldsymbol{\tau}})\lambda_{\boldsymbol{k}_{q-}}^{\boldsymbol{d}}(\sigma'_{\boldsymbol{d},\boldsymbol{\tau}'})B(\boldsymbol{d},\boldsymbol{k}_{q-})$$
$$+\,s_{d_q,k_q+1}\lambda_{\boldsymbol{k}_{q+}}^{\boldsymbol{d}}(\sigma'_{\boldsymbol{d},\boldsymbol{\tau}})\lambda_{\boldsymbol{k}_{q+}}^{\boldsymbol{d}}(\sigma'_{\boldsymbol{d},\boldsymbol{\tau}'})B(\boldsymbol{d},\boldsymbol{k}_{q+})\Big],$$

(135)

with $\boldsymbol{k}_{q+}=(k_1,\ldots,k_q+1,\ldots,k_Q)$ and $\boldsymbol{k}_{q-}=(k_1,\ldots,k_q-1,\ldots,k_Q)$, and convention $t_{d_q,-1}=0$,

$$s_{d_q,k_q}=\frac{k_q}{2k_q+d_q-2},\qquad t_{d_q,k_q}=\frac{k_q+d_q-2}{2k_q+d_q-2}.$$

Recall that for $\boldsymbol{k}\in\mathbb{Z}_{\geq 0}^Q$ and $\boldsymbol{s}\in[B(\boldsymbol{d},\boldsymbol{k})]$, $Y_{\boldsymbol{k},\boldsymbol{s}}^{\boldsymbol{d}}=\bigotimes_{q\in[Q]}Y_{k_q s_q}^{(d_q)}$ forms an orthogonal basis of $\mathcal{L}$ and that

$$\mathbb{E}_{\overline{\boldsymbol{x}}_2}\Big[Q_{\boldsymbol{k}}^{\boldsymbol{d}}\left(\{\langle\overline{\boldsymbol{x}}_1,\overline{\boldsymbol{x}}_2\rangle\}_{q\in[Q]}\right)Y_{\boldsymbol{k},\boldsymbol{s}}^{\boldsymbol{d}}(\overline{\boldsymbol{x}}_2)\Big]=\frac{1}{B(\boldsymbol{d},\boldsymbol{k})}Y_{\boldsymbol{k},\boldsymbol{s}}^{\boldsymbol{d}}(\overline{\boldsymbol{x}}_1)\delta_{\boldsymbol{k},\boldsymbol{s}}.$$

We deduce that

$$\mathbb{H}_{\boldsymbol{\tau},\boldsymbol{\tau}'}Y_{\boldsymbol{k},\boldsymbol{s}}^{\boldsymbol{d}}(\overline{\boldsymbol{x}}_1)=\sum_{\boldsymbol{k}'\in\mathbb{Z}_{\geq 0}^Q}A_{(\boldsymbol{\tau},\boldsymbol{\tau}'),\boldsymbol{k}'}^{(q)}\mathbb{E}_{\overline{\boldsymbol{x}}_2}\Big[Q_{\boldsymbol{k}'}^{\boldsymbol{d}}\left(\{\langle\overline{\boldsymbol{x}}_1,\overline{\boldsymbol{x}}_2\rangle\}_{q\in[Q]}\right)Y_{\boldsymbol{k},\boldsymbol{s}}^{\boldsymbol{d}}(\overline{\boldsymbol{x}}_2)\Big]=\frac{A_{(\boldsymbol{\tau},\boldsymbol{\tau}'),\boldsymbol{k}}^{(q)}}{B(\boldsymbol{d},\boldsymbol{k})}Y_{\boldsymbol{k},\boldsymbol{s}}^{\boldsymbol{d}}(\overline{\boldsymbol{x}}_1).$$

Consider $\{\mathbb{T}_{\boldsymbol{\tau}}Y_{\boldsymbol{k},\boldsymbol{s}}^{\boldsymbol{d}}\}_{\boldsymbol{k}\in\mathbb{Z}_{\geq 0}^Q,\boldsymbol{s}\in[B(\boldsymbol{d},\boldsymbol{k})]}$. We have:

$$\langle\mathbb{T}_{\boldsymbol{\tau}}Y_{\boldsymbol{k},\boldsymbol{s}}^{\boldsymbol{d}},\mathbb{T}_{\boldsymbol{\tau}'}Y_{\boldsymbol{k}',\boldsymbol{s}'}^{\boldsymbol{d}}\rangle_{L^2}=\langle Y_{\boldsymbol{k},\boldsymbol{s}}^{\boldsymbol{d}},\mathbb{H}_{\boldsymbol{\tau},\boldsymbol{\tau}'}Y_{\boldsymbol{k}',\boldsymbol{s}'}^{\boldsymbol{d}}\rangle_{L^2}=\frac{A_{(\boldsymbol{\tau},\boldsymbol{\tau}'),\boldsymbol{k}}^{(q)}}{B(\boldsymbol{d},\boldsymbol{k})}\delta_{\boldsymbol{k},\boldsymbol{k}'}\delta_{\boldsymbol{s},\boldsymbol{s}'},$$

$$\mathbb{K}_{\boldsymbol{\tau},\boldsymbol{\tau}'}\mathbb{T}_{\boldsymbol{\tau}''}Y_{\boldsymbol{k},\boldsymbol{s}}^{\boldsymbol{d}}=\mathbb{T}_{\boldsymbol{\tau}}\mathbb{H}_{\boldsymbol{\tau}',\boldsymbol{\tau}''}Y_{\boldsymbol{k},\boldsymbol{s}}^{\boldsymbol{d}}=\frac{A_{(\boldsymbol{\tau}',\boldsymbol{\tau}''),\boldsymbol{k}}^{(q)}}{B(\boldsymbol{d},\boldsymbol{k})}\mathbb{T}_{\boldsymbol{\tau}}Y_{\boldsymbol{k},\boldsymbol{s}}^{\boldsymbol{d}}.$$

Hence $\{\mathbb{T}_{\boldsymbol{\tau}''}Y_{\boldsymbol{k},\boldsymbol{s}}^{(\boldsymbol{d})}\}$ forms an orthogonal basis that diagonalizes $\mathbb{K}_{\boldsymbol{\tau}',\boldsymbol{\tau}''}$ (notice that $\mathbb{T}_{\boldsymbol{\tau}}Y_{\boldsymbol{k},\boldsymbol{s}}^{\boldsymbol{d}}$ is parallel to $\mathbb{T}_{\boldsymbol{\tau}'}Y_{\boldsymbol{k},\boldsymbol{s}}^{\boldsymbol{d}}$ for any $\boldsymbol{\tau},\boldsymbol{\tau}'\in\mathbb{R}_{\geq 0}^Q$). Let us consider the subspace $\mathbb{T}_{\boldsymbol{\tau}}(V_{\mathcal{Q}}^{\boldsymbol{d}})$, the image of $V_{\mathcal{Q}}^{\boldsymbol{d}}$ by the operator $\mathbb{T}_{\boldsymbol{\tau}}$. From Assumptions 3.(a) and 3.(b) and Lemma 19, there exists $\varepsilon_0\in(0,1)$ and $d_0$ such that for any $\boldsymbol{\tau},\boldsymbol{\tau}'\in[1-\varepsilon_0,1+\varepsilon_0]^Q$ and $d\geq d_0$, we have $A_{(\boldsymbol{\tau},\boldsymbol{\tau}'),\boldsymbol{k}}^{(q)}>0$ for any $\boldsymbol{k}\in\mathcal{Q}$, and therefore the inverse $\mathbb{K}_{\boldsymbol{\tau},\boldsymbol{\tau}'}^{-1}|_{\mathbb{T}_{\boldsymbol{\tau}}(V_{\mathcal{Q}}^{\boldsymbol{d}})}$ (restricted to $\mathbb{T}_{\boldsymbol{\tau}}(V_{\mathcal{Q}}^{\boldsymbol{d}})$) is well defined.

### H.3.2  Proof of Theorem 4

Let us assume that $\{f_d\}$ is contained in $\bigoplus_{\boldsymbol{k}\in\mathcal{Q}}V_{\boldsymbol{k}}^{\boldsymbol{d}}$, i.e. $\overline{f}_d=\mathsf{P}_{\mathcal{Q}}\overline{f}_d$.

Consider

$$\hat{f}(\boldsymbol{x};\boldsymbol{\Theta},\boldsymbol{a})=\sum_{i=1}^N\langle\boldsymbol{a}_i,\boldsymbol{x}^{(q)}\rangle\sigma'(\langle\boldsymbol{\theta}_i,\boldsymbol{x}\rangle/R).$$

Define $\boldsymbol{\alpha}_{\boldsymbol{\tau}}(\overline{\boldsymbol{\theta}}) \equiv \mathbb{K}_{\boldsymbol{\tau},\boldsymbol{\tau}}^{-1} \mathbb{T}_{\boldsymbol{\tau}} \overline{f}_d(\overline{\boldsymbol{\theta}})$ and choose $\boldsymbol{a}_i^* = N^{-1} \boldsymbol{\alpha}_{\boldsymbol{\tau}_i}(\overline{\boldsymbol{\theta}}_i)$, where we denoted $\overline{\boldsymbol{\theta}}_i = (\overline{\boldsymbol{\theta}}_i^{(q)})_{q \in [Q]}$ with $\overline{\boldsymbol{\theta}}_i^{(q)} = \boldsymbol{\theta}_i^{(q)}/\tau_i^{(q)} \in \mathbb{S}^{d_q-1}(\sqrt{d_q})$ independent of $\boldsymbol{\tau}_i$.

Fix $\varepsilon_0 > 0$ as prescribed in Lemma 15 and consider the expectation over $\mathcal{P}_{\varepsilon_0}$ of the $\mathrm{NT}^{(q)}$ risk (in particular, $\boldsymbol{a}^* = (\boldsymbol{a}_1^*, \ldots, \boldsymbol{a}_N^*) \in \mathbb{R}^{Nd_q}$ are well defined):

$$
\begin{aligned}
\mathbb{E}_{\boldsymbol{\Theta}_{\varepsilon_0}}[R_{\mathrm{NT}^{(q)}}(f_d, \boldsymbol{\Theta})] &= \mathbb{E}_{\boldsymbol{\Theta}_{\varepsilon_0}} \Big[ \inf_{\boldsymbol{a} \in \mathbb{R}^{Nd_q}} \mathbb{E}_{\boldsymbol{x}}[(f_d(\boldsymbol{x}) - \hat{f}(\boldsymbol{x}; \boldsymbol{\Theta}, \boldsymbol{a}))^2] \Big] \\
&\leq \mathbb{E}_{\boldsymbol{\Theta}_{\varepsilon_0}} \Big[ \mathbb{E}_{\boldsymbol{x}} \Big[ (f_d(\boldsymbol{x}) - \hat{f}(\boldsymbol{x}; \boldsymbol{\Theta}, \boldsymbol{a}^*(\boldsymbol{\Theta})))^2 \Big] \Big].
\end{aligned}
$$

We can expand the squared loss at $\boldsymbol{a}$ as

$$
\begin{aligned}
\mathbb{E}_{\boldsymbol{x}}[(f_d(\boldsymbol{x}) - \hat{f}(\boldsymbol{x}))^2] = &\|f_d\|_{L^2}^2 - 2 \sum_{i=1}^N \mathbb{E}_{\boldsymbol{x}}[\langle \boldsymbol{a}_i, \boldsymbol{x}^{(q)} \rangle \sigma'(\langle \boldsymbol{\theta}_i, \boldsymbol{x} \rangle/R) f_d(\boldsymbol{x})] \\
&+ \sum_{i,j=1}^N \mathbb{E}_{\boldsymbol{x}}[\langle \boldsymbol{a}_i, \boldsymbol{x}^{(q)} \rangle \langle \boldsymbol{a}_j, \boldsymbol{x}^{(q)} \rangle \sigma'(\langle \boldsymbol{\theta}_i, \boldsymbol{x} \rangle/R) \sigma'(\langle \boldsymbol{\theta}_j, \boldsymbol{x} \rangle/R)].
\end{aligned}
\tag{136}
$$

The second term of the expansion (136) around $\boldsymbol{a}^*$ verifies

$$
\begin{aligned}
&\mathbb{E}_{\boldsymbol{\Theta}_{\varepsilon_0}} \Big[ \sum_{i=1}^N \mathbb{E}_{\boldsymbol{x}}[\langle \boldsymbol{a}_i^*, \boldsymbol{x}^{(q)} \rangle \sigma'(\langle \boldsymbol{\theta}_i, \boldsymbol{x} \rangle/R) f_d(\boldsymbol{x})] \Big] \\
&= \mathbb{E}_{\boldsymbol{\tau}_{\varepsilon_0}} \Big[ \mathbb{E}_{\overline{\boldsymbol{\theta}}} \Big[ \boldsymbol{\alpha}_{\boldsymbol{\tau}}(\overline{\boldsymbol{\theta}})^{\mathsf{T}} \mathbb{E}_{\overline{\boldsymbol{x}}} \Big[ \boldsymbol{x}^{(q)} \sigma'_{\boldsymbol{d},\boldsymbol{\tau}} \Big( \{\langle \overline{\boldsymbol{\theta}}^{(q)}, \overline{\boldsymbol{x}}^{(q)} \rangle / \sqrt{d_q}\}_{q \in [Q]} \Big) \overline{f}_d(\overline{\boldsymbol{x}}) \Big] \Big] \Big] \\
&= \mathbb{E}_{\boldsymbol{\tau}_{\varepsilon_0}} \Big[ \langle \mathbb{K}_{\boldsymbol{\tau},\boldsymbol{\tau}}^{-1} \mathbb{T}_{\boldsymbol{\tau}} \overline{f}_d, \mathbb{T}_{\boldsymbol{\tau}} \overline{f}_d \rangle_{L^2} \Big] \\
&= \|f_d\|_{L^2}^2,
\end{aligned}
\tag{137}
$$

where we used that for each $\boldsymbol{\tau} \in [1-\varepsilon_0, 1+\varepsilon_0]^Q$, we have $\mathbb{T}_{\boldsymbol{\tau}}^* \mathbb{K}_{\boldsymbol{\tau},\boldsymbol{\tau}}^{-1} \mathbb{T}_{\boldsymbol{\tau}} = \mathbf{I}|_{V_{\mathcal{Q}}^d}$.

Let us consider the third term in the expansion (136) around $\boldsymbol{a}^*$: the non diagonal term verifies

$$
\begin{aligned}
&\mathbb{E}_{\boldsymbol{\Theta}_{\varepsilon_0}} \Big[ \sum_{i \neq j} \mathbb{E}_{\boldsymbol{x}}[\langle \boldsymbol{a}_i^*, \boldsymbol{x}^{(q)} \rangle \langle \boldsymbol{a}_j^*, \boldsymbol{x}^{(q)} \rangle \sigma'(\langle \boldsymbol{\theta}_i, \boldsymbol{x} \rangle/R) \sigma'(\langle \boldsymbol{\theta}_j, \boldsymbol{x} \rangle/R)] \Big] \\
&= (1 - N^{-1}) \mathbb{E}_{\boldsymbol{\tau}_{\varepsilon_0}^1, \boldsymbol{\tau}_{\varepsilon_0}^2, \overline{\boldsymbol{\theta}}_1, \overline{\boldsymbol{\theta}}_2} \Big[ \boldsymbol{\alpha}_{\boldsymbol{\tau}^1}(\overline{\boldsymbol{\theta}}_1)^{\mathsf{T}} E_{\overline{\boldsymbol{x}}} \Big[ \sigma'_{\boldsymbol{d},\boldsymbol{\tau}} \Big( \{\langle \overline{\boldsymbol{\theta}}_1^{(q)}, \overline{\boldsymbol{x}}^{(q)} \rangle / \sqrt{d_q}\}_{q \in [Q]} \Big) \\
&\qquad \times \sigma'_{\boldsymbol{d},\boldsymbol{\tau}'} \Big( \{\langle \overline{\boldsymbol{\theta}}_2^{(q)}, \overline{\boldsymbol{x}}^{(q)} \rangle / \sqrt{d_q}\}_{q \in [Q]} \Big) \boldsymbol{x}^{(q)} (\boldsymbol{x}^{(q)})^{\mathsf{T}} \Big] \boldsymbol{\alpha}_{\boldsymbol{\tau}^2}(\overline{\boldsymbol{\theta}}_2) \Big] \\
&= (1 - N^{-1}) \mathbb{E}_{\boldsymbol{\tau}_{\varepsilon_0}^1, \boldsymbol{\tau}_{\varepsilon_0}^2, \overline{\boldsymbol{\theta}}_1, \overline{\boldsymbol{\theta}}_2} \Big[ \mathbb{K}_{\boldsymbol{\tau}^1,\boldsymbol{\tau}^1}^{-1} \mathbb{T}_{\boldsymbol{\tau}^1} \overline{f}_d(\overline{\boldsymbol{\theta}}_1)^{\mathsf{T}} \mathbb{K}_{\boldsymbol{\tau}^1,\boldsymbol{\tau}^2}(\overline{\boldsymbol{\theta}}_1, \overline{\boldsymbol{\theta}}_2) \mathbb{K}_{\boldsymbol{\tau}^2,\boldsymbol{\tau}^2}^{-1} \mathbb{T}_{\boldsymbol{\tau}^2} \overline{f}_d(\overline{\boldsymbol{\theta}}_2) \Big] \\
&= (1 - N^{-1}) \mathbb{E}_{\boldsymbol{\tau}_{\varepsilon_0}^1, \boldsymbol{\tau}_{\varepsilon_0}^2} \Big[ \langle \mathbb{K}_{\boldsymbol{\tau}^1,\boldsymbol{\tau}^1}^{-1} \mathbb{T}_{\boldsymbol{\tau}^1} \overline{f}_d, \mathbb{K}_{\boldsymbol{\tau}^1,\boldsymbol{\tau}^2} \mathbb{K}_{\boldsymbol{\tau}^2,\boldsymbol{\tau}^2}^{-1} \mathbb{T}_{\boldsymbol{\tau}^2} \overline{f}_d \rangle_{L^2} \Big].
\end{aligned}
$$

For $\boldsymbol{k} \in \mathcal{Q}$ and $\boldsymbol{s} \in [B(\boldsymbol{d}, \boldsymbol{k})]$ and $\boldsymbol{\tau}^1, \boldsymbol{\tau}^2 \in [1-\varepsilon_0, 1+\varepsilon_0]^Q$, we have

$$
\mathbb{T}_{\boldsymbol{\tau}^1}^* \mathbb{K}_{\boldsymbol{\tau}^1,\boldsymbol{\tau}^1}^{-1} \mathbb{K}_{\boldsymbol{\tau}^1,\boldsymbol{\tau}^2} \mathbb{K}_{\boldsymbol{\tau}^2,\boldsymbol{\tau}^2}^{-1} \mathbb{T}_{\boldsymbol{\tau}^2} Y_{\boldsymbol{k},\boldsymbol{s}}^{\boldsymbol{d}} = \Big( \mathbb{T}_{\boldsymbol{\tau}^1}^* \mathbb{K}_{\boldsymbol{\tau}^1,\boldsymbol{\tau}^1}^{-1} \mathbb{T}_{\boldsymbol{\tau}^1} \Big) \cdot \Big( \mathbb{T}_{\boldsymbol{\tau}^2}^* \mathbb{K}_{\boldsymbol{\tau}^2,\boldsymbol{\tau}^2}^{-1} \mathbb{T}_{\boldsymbol{\tau}^2} \Big) \cdot Y_{\boldsymbol{k},\boldsymbol{s}}^{\boldsymbol{d}} = Y_{\boldsymbol{k},\boldsymbol{s}}^{\boldsymbol{d}}.
$$

Hence for any $\boldsymbol{\tau}^1, \boldsymbol{\tau}^2 \in [1-\varepsilon_0, 1+\varepsilon_0]^Q$, $\mathbb{T}_{\boldsymbol{\tau}^1}^* \mathbb{K}_{\boldsymbol{\tau}^1,\boldsymbol{\tau}^1}^{-1} \mathbb{K}_{\boldsymbol{\tau}^1,\boldsymbol{\tau}^2} \mathbb{K}_{\boldsymbol{\tau}^2,\boldsymbol{\tau}^2}^{-1} \mathbb{T}_{\boldsymbol{\tau}^2} = \mathbf{I}|_{V_{\mathcal{Q}}^d}$. Hence

$$
\mathbb{E}_{\boldsymbol{\Theta}_{\varepsilon_0}} \Big[ \sum_{i \neq j} \mathbb{E}_{\boldsymbol{x}}[\langle \boldsymbol{a}_i^*, \boldsymbol{x}^{(q)} \rangle \langle \boldsymbol{a}_j^*, \boldsymbol{x}^{(q)} \rangle \sigma'(\langle \boldsymbol{\theta}_i, \boldsymbol{x} \rangle/R) \sigma'(\langle \boldsymbol{\theta}_j, \boldsymbol{x} \rangle/R)] \Big] = (1 - N^{-1}) \|f_d\|_{L^2}^2. \tag{138}
$$

The diagonal term verifies

$$
\begin{aligned}
&\mathbb{E}_{\boldsymbol{\Theta}_{\varepsilon_0}} \Big[ \sum_{i \in [N]} \mathbb{E}_{\boldsymbol{x}}[\langle \boldsymbol{a}_i^*, \boldsymbol{x}^{(q)} \rangle^2 \sigma'(\langle \boldsymbol{\theta}_i, \boldsymbol{x} \rangle/R) \sigma'(\langle \boldsymbol{\theta}_j, \boldsymbol{x} \rangle/R)] \Big] \\
&= N^{-1} \mathbb{E}_{\boldsymbol{\tau}_{\varepsilon_0}, \overline{\boldsymbol{\theta}}} \Big[ \boldsymbol{\alpha}_{\boldsymbol{\tau}}(\overline{\boldsymbol{\theta}})^{\mathsf{T}} \mathbb{K}_{\boldsymbol{\tau},\boldsymbol{\tau}}(\overline{\boldsymbol{\theta}}, \overline{\boldsymbol{\theta}}) \boldsymbol{\alpha}_{\boldsymbol{\tau}}(\overline{\boldsymbol{\theta}}) \Big] \\
&\leq N^{-1} \Big[ \max_{\overline{\boldsymbol{\theta}}, \boldsymbol{\tau} \in [1-\varepsilon_0, 1+\varepsilon_0]^Q} \|\mathbb{K}_{\boldsymbol{\tau},\boldsymbol{\tau}}(\overline{\boldsymbol{\theta}}, \overline{\boldsymbol{\theta}})\|_{\mathrm{op}} \Big] \cdot \mathbb{E}_{\boldsymbol{\tau}_{\varepsilon_0}}[\|\mathbb{K}_{\boldsymbol{\tau},\boldsymbol{\tau}}^{-1} \mathbb{T}_{\boldsymbol{\tau}} \overline{f}_d\|_{L^2}^2].
\end{aligned}
$$

We have, from Lemma 14,

$$\mathbb{K}_{\boldsymbol{\tau},\boldsymbol{\tau}}(\overline{\boldsymbol{\theta}},\overline{\boldsymbol{\theta}}) = \alpha^{(q)}\mathbf{I}_{d_q} + \beta^{(q)}\overline{\boldsymbol{\theta}}^{(q)}(\overline{\boldsymbol{\theta}}^{(q)})^2$$

where

$$\begin{bmatrix}\alpha^{(q)}\\\beta^{(q)}\end{bmatrix} = [d_q(d_q-1)(\tau^{(q)})^4]^{-1}\begin{bmatrix}d_q(\tau^{(q)})^4 & -(\tau^{(q)})^2\\-(\tau^{(q)})^2 & 1\end{bmatrix}$$
$$\times \begin{bmatrix}\mathbb{E}_{\boldsymbol{x}}[\langle\boldsymbol{x}^{(q)},\boldsymbol{x}^{(q)}\rangle\sigma'_{\boldsymbol{d},\boldsymbol{\tau}}(\overline{x}_1^{(1)},\ldots,\overline{x}_1^{(Q)})^2]\\\mathbb{E}_{\boldsymbol{x}}[(\overline{x}_1^{(q)})^2\sigma'_{\boldsymbol{d},\boldsymbol{\tau}}(\overline{x}_1^{(1)},\ldots,\overline{x}_1^{(Q)})^2]\end{bmatrix}.$$

Hence from Lemma 17 and for $\varepsilon_0$ small enough, there exists $C > 0$ such that for $d$ large enough

$$\sup_{\boldsymbol{\tau}\in[1-\varepsilon_0,1+\varepsilon_0]^Q}\|\mathbb{K}_{\boldsymbol{\tau},\boldsymbol{\tau}}(\overline{\boldsymbol{\theta}},\overline{\boldsymbol{\theta}})\|_{\mathrm{op}} \leq C\frac{r_q^2}{d_q} = Cd^{\kappa_q}.$$

Furthermore

$$\|\mathbb{K}_{\boldsymbol{\tau},\boldsymbol{\tau}}^{-1}\mathbb{T}_{\boldsymbol{\tau}}\overline{f}_d\|_{L^2}^2 = \sum_{\boldsymbol{k}\in\mathcal{Q}}\frac{B(\boldsymbol{d},\boldsymbol{k})}{A_{(\boldsymbol{\tau},\boldsymbol{\tau}),\boldsymbol{k}}^{(q)}}\sum_{\boldsymbol{s}\in[B(\boldsymbol{d},\boldsymbol{k})]}\lambda_{\boldsymbol{k},\boldsymbol{s}}^{\boldsymbol{d}}(\overline{f}_d)^2$$
$$\leq \left[\max_{k\in\mathcal{Q}}\frac{B(\boldsymbol{d},\boldsymbol{k})}{A_{(\boldsymbol{\tau},\boldsymbol{\tau}),\boldsymbol{k}}^{(q)}}\right]\cdot\|\mathsf{P}_{\mathcal{Q}}f_d\|_{L^2}^2.$$

From Lemma 15, we get

$$\mathbb{E}_{\boldsymbol{\tau}_{\varepsilon_0}}[\|\mathbb{K}_{\boldsymbol{\tau},\boldsymbol{\tau}}^{-1}\mathbb{T}_{\boldsymbol{\tau}}\overline{f}_d\|_{L^2}^2] \leq Cd^{\gamma-\kappa_q}\cdot\|\mathsf{P}_{\mathcal{Q}}f_d\|_{L^2}^2.$$

Hence,

$$\mathbb{E}_{\boldsymbol{\Theta}_{\varepsilon_0}}\Big[\sum_{i\in[N]}\mathbb{E}_{\boldsymbol{x}}[\langle\boldsymbol{a}_i^*,\boldsymbol{x}^{(q)}\rangle^2\sigma'(\langle\boldsymbol{\theta}_i,\boldsymbol{x}\rangle/R)\sigma'(\langle\boldsymbol{\theta}_j,\boldsymbol{x}\rangle/R)]\Big] \leq C\frac{d^\gamma}{N}\|\mathsf{P}_{\mathcal{Q}}f_d\|_{L^2}^2. \qquad (139)$$

Combining Eq. (137), Eq. (138) and Eq. (139), we get

$$\mathbb{E}_{\boldsymbol{\Theta}_{\varepsilon_0}}\Big[\mathbb{E}_{\boldsymbol{x}}\Big[(f_d(\boldsymbol{x})-\hat{f}(\boldsymbol{x};\boldsymbol{\Theta},\boldsymbol{a}^*(\boldsymbol{\Theta})))^2\Big]\Big]$$
$$= \|f_d\|_{L^2}^2 - 2\|f_d\|_{L^2}^2 + (1-N^{-1})\|f_d\|_{L^2}^2 + N^{-1}\mathbb{E}_{\boldsymbol{\tau}_{\varepsilon_0},\overline{\boldsymbol{\theta}}}\Big[\boldsymbol{\alpha}_{\boldsymbol{\tau}}(\overline{\boldsymbol{\theta}})^\mathsf{T}\mathbb{K}_{\boldsymbol{\tau},\boldsymbol{\tau}}(\overline{\boldsymbol{\theta}},\overline{\boldsymbol{\theta}})\boldsymbol{\alpha}_{\boldsymbol{\tau}}(\overline{\boldsymbol{\theta}})\Big]$$
$$\leq C\frac{d^\gamma}{N}\|\mathsf{P}_{\mathcal{Q}}f_d\|_{L^2}^2.$$

By Markov's inequality, we get for any $\varepsilon > 0$ and $d$ large enough,

$$\mathbb{P}(R_{\mathrm{NT}^{(q)}}(f_d,\boldsymbol{\Theta}) > \varepsilon\cdot\|f_d\|_{L^2}^2) \leq \mathbb{P}(\{R_{\mathrm{NT}^{(q)}}(f_d,\boldsymbol{\Theta}) > \varepsilon\cdot\|f_d\|_{L^2}^2\}\cap\mathcal{P}_{\varepsilon_0}) + \mathbb{P}(\mathcal{P}_{\varepsilon_0}^c)$$
$$\leq C'\frac{d^\gamma}{N} + \mathbb{P}(\mathcal{P}_{\varepsilon_0}^c).$$

The assumption that $N = \omega_d(d^\gamma)$ and Lemma 8 conclude the proof.

# I  Proof of Theorem 4 in the main text

**Step 1. Show that $R_{\mathrm{NN},2N}(f_*) \leq \inf_{\boldsymbol{W} \in \mathbb{R}^{N \times d}} R_{\mathrm{NT},N}(f_*, \boldsymbol{W})$.**

Define the neural tangent model with $N$ neurons by $\hat{f}_{\mathrm{NT},N}(\boldsymbol{x}; \boldsymbol{s}; \boldsymbol{W}) = \sum_{i=1}^N \langle \boldsymbol{s}_i, \boldsymbol{x} \rangle \sigma'(\langle \boldsymbol{w}_i, \boldsymbol{x} \rangle)$ and the neural networks with $N$ neurons by $\hat{f}_{\mathrm{NN},N}(\boldsymbol{x}; \boldsymbol{W}, \boldsymbol{b}) = \sum_{i=1}^N b_i \sigma(\langle \boldsymbol{w}_i, \boldsymbol{x} \rangle)$. For any $\boldsymbol{W} \in \mathbb{R}^{N \times d}$, $\boldsymbol{s} \in \mathbb{R}^N$, and $\varepsilon > 0$, we define

$$\hat{g}_N(\boldsymbol{x}; \boldsymbol{W}, \boldsymbol{s}, \varepsilon) \equiv \varepsilon^{-1}\Big( \hat{f}_{\mathrm{NN},N}(\boldsymbol{x}; \boldsymbol{W} + \varepsilon\boldsymbol{s}, \boldsymbol{1}) - \hat{f}_{\mathrm{NN},N}(\boldsymbol{x}; \boldsymbol{W}, \boldsymbol{1}) \Big),$$

$$\mathcal{E}(\boldsymbol{x}; \boldsymbol{W}, \boldsymbol{s}, \varepsilon) = \hat{g}_N(\boldsymbol{x}; \boldsymbol{W}, \boldsymbol{s}, \varepsilon) - \hat{f}_{\mathrm{NT},N}(\boldsymbol{x}; \boldsymbol{s}; \boldsymbol{W}).$$

Then by Taylor expansion, there exists $(\tilde{\boldsymbol{w}}_i)_{i \in [N]}$ such that

$$|\mathcal{E}(\boldsymbol{x}; \boldsymbol{W}, \boldsymbol{s}, \varepsilon)| = \frac{\varepsilon}{2}\Big| \sum_{i=1}^N \langle \boldsymbol{s}_i, \boldsymbol{x} \rangle^2 \sigma''(\langle \tilde{\boldsymbol{w}}_i, \boldsymbol{x} \rangle) \Big|.$$

By the boundedness assumption of $\sup_{x \in \mathbb{R}} |\sigma''(x)|$, we have

$$\lim_{\varepsilon \to 0+} \|\mathcal{E}(\cdot; \boldsymbol{W}, \boldsymbol{s}, \varepsilon)\|_{L^2}^2 = 0,$$

and hence

$$\lim_{\varepsilon \to 0+} \|f_* - \hat{g}_N(\cdot; \boldsymbol{W}, \boldsymbol{s}, \varepsilon)\|_{L^2}^2 = \|f_* - \hat{f}_{\mathrm{NT},N}(\cdot; \boldsymbol{s}; \boldsymbol{W})\|_{L^2}^2.$$

Note that $\hat{g}_N$ can be regarded as a function in $\mathcal{F}_{\mathrm{NN}}^{2N}$ and $\hat{f}_{\mathrm{NT},N} \in \mathcal{F}_{\mathrm{NN}}^N(\boldsymbol{W})$, this implies that

$$R_{\mathrm{NN},2N}(f_*) \leq \inf_{\boldsymbol{W} \in \mathbb{R}^{N \times d}} R_{\mathrm{NT},N}(f_*, \boldsymbol{W}). \tag{140}$$

**Step 2. Give upper bound of $\inf_{\boldsymbol{W} \in \mathbb{R}^{N \times d}} R_{\mathrm{NT},N}(f_*, \boldsymbol{W})$.** We take $\overline{\boldsymbol{W}} = (\bar{\boldsymbol{w}}_i)_{i \leq N}$ with $\bar{\boldsymbol{w}}_i = \boldsymbol{U}\bar{\boldsymbol{v}}_i$, where $\bar{\boldsymbol{v}}_i \sim \mathrm{Unif}(\mathbb{S}^{d_0-1}(r^{-1}))$, and denote $\overline{\boldsymbol{V}} = (\bar{\boldsymbol{v}}_i)_{i \leq N}$. Then we have

$$\mathcal{G}_{\mathrm{NT}}^N(\overline{\boldsymbol{V}}) \equiv \Big\{ f(\boldsymbol{x}) = \bar{f}(\boldsymbol{U}^{\mathsf{T}}\boldsymbol{x}) : \bar{f}(\boldsymbol{z}) = \sum_{i=1}^N \langle \bar{\boldsymbol{s}}_i, \boldsymbol{z} \rangle \sigma'(\langle \bar{\boldsymbol{v}}_i, \boldsymbol{z} \rangle), \bar{\boldsymbol{s}}_i \in \mathbb{R}^{d_0}, i \leq N \Big\} \subseteq \mathcal{F}_{\mathrm{NT}}^N(\overline{\boldsymbol{W}}).$$

It is easy to see that, when $f_*(\boldsymbol{x}) = \varphi(\boldsymbol{U}^{\mathsf{T}}\boldsymbol{x})$, we have

$$\inf_{\hat{f} \in \mathcal{G}_{\mathrm{NT}}^N(\overline{\boldsymbol{V}})} \mathbb{E}[(f_*(\boldsymbol{x}) - \hat{f}(\boldsymbol{x}))^2] = \inf_{\hat{f} \in \mathcal{F}_{\mathrm{NT}}^N(\overline{\boldsymbol{V}})} \mathbb{E}[(\varphi(\boldsymbol{z}) - \hat{f}(\boldsymbol{z}))^2],$$

where $\mathcal{F}_{\mathrm{NT}}^N(\overline{\boldsymbol{V}})$ is the class of neural tangent model on $\mathbb{R}^{d_0}$

$$\mathcal{F}_{\mathrm{NT}}^N(\overline{\boldsymbol{V}}) = \Big\{ \bar{f}(\boldsymbol{z}) = \sum_{i=1}^N \langle \bar{\boldsymbol{s}}_i, \boldsymbol{z} \rangle \sigma'(\langle \bar{\boldsymbol{v}}_i, \boldsymbol{z} \rangle) : \bar{\boldsymbol{s}}_i \in \mathbb{R}^{d_0}, i \leq N \Big\}.$$

Moreover, by Theorem 3 in the main text, when $d_0^{\ell+\delta} \leq N \leq d_0^{\ell+1-\delta}$ for some $\delta > 0$ independent of $N, d$, we have

$$\inf_{\hat{f} \in \mathcal{F}_{\mathrm{NT}}^N(\overline{\boldsymbol{V}})} \mathbb{E}[(\varphi(\boldsymbol{z}) - \hat{f}(\boldsymbol{z}))^2] = (1 + o_{d,\mathbb{P}}(1)) \cdot \|\mathsf{P}_{>\ell+1}\varphi\|_{L^2}^2 = (1 + o_{d,\mathbb{P}}(1)) \cdot \|\mathsf{P}_{>\ell+1}f_*\|_{L^2}^2.$$

As a consequence, we have

$$\inf_{\boldsymbol{W} \in \mathbb{R}^{N \times d}} R_{\mathrm{NT},N}(f_*, \boldsymbol{W}) \leq \inf_{\hat{f} \in \mathcal{F}_{\mathrm{NT}}^N(\overline{\boldsymbol{W}})} \mathbb{E}[(f_*(\boldsymbol{x}) - \hat{f}(\boldsymbol{x}))^2] \leq \inf_{\hat{f} \in \mathcal{G}_{\mathrm{NT}}^N(\overline{\boldsymbol{V}})} \mathbb{E}[(f_*(\boldsymbol{x}) - \hat{f}(\boldsymbol{x}))^2]$$

$$= \inf_{\hat{f} \in \mathcal{F}_{\mathrm{NT}}^N(\overline{\boldsymbol{V}})} \mathbb{E}[(\varphi(\boldsymbol{z}) - \hat{f}(\boldsymbol{z}))^2] = (1 + o_{d,\mathbb{P}}(1)) \cdot \|\mathsf{P}_{>\ell+1}f_*\|_{L^2}^2.$$

Combining with Eq. (140) gives that, when $d_0^{\ell+\delta} \leq N \leq d_0^{\ell+1-\delta}$, we have

$$R_{\mathrm{NN},N}(f_*) \leq (1 + o_d(1)) \cdot \|\mathsf{P}_{>\ell+1}f_*\|_{L^2}^2.$$

**Step 3. Show that $R_{\mathrm{NN},N}(f_*)$ is independent of $\kappa$.**

We let $\tilde{r} = d^{\tilde{\kappa}/2}$ and $\mathring{r} = d^{\mathring{\kappa}/2}$ for some $\tilde{\kappa} \neq \mathring{\kappa}$. Suppose we have $\tilde{\boldsymbol{x}} = \boldsymbol{U}\tilde{\boldsymbol{z}}_1 + \boldsymbol{U}^\perp \boldsymbol{z}_2$ and $\mathring{\boldsymbol{x}} = \boldsymbol{U}\mathring{\boldsymbol{z}}_1 + \boldsymbol{U}^\perp \boldsymbol{z}_2$, where $\tilde{\boldsymbol{z}}_1 \sim \mathrm{Unif}(\mathbb{S}^{d_0-1}(\tilde{r}\sqrt{d_0}))$, $\mathring{\boldsymbol{z}}_1 \sim \mathrm{Unif}(\mathbb{S}^{d_0-1}(\mathring{r}\sqrt{d_0}))$, and $\boldsymbol{z}_2 \sim \mathrm{Unif}(\mathbb{S}^{d-d_0-1}(\sqrt{d-d_0}))$. Moreover, we let $\tilde{f}_*(\tilde{\boldsymbol{x}}) = \varphi(\boldsymbol{U}^\mathsf{T}\tilde{\boldsymbol{x}}/\tilde{r})$ and $\mathring{f}_*(\mathring{\boldsymbol{x}}) = \varphi(\boldsymbol{U}^\mathsf{T}\mathring{\boldsymbol{x}}/\mathring{r})$ for some function $\varphi : \mathbb{R}^{d_0} \to \mathbb{R}$.

Then, for any $\tilde{\boldsymbol{W}} = (\tilde{\boldsymbol{w}}_i)_{i\leq N} \subseteq \mathbb{R}^d$ and $\tilde{\boldsymbol{b}} = (\tilde{b}_i)_{i\leq N} \subseteq \mathbb{R}$, there exists $(\tilde{\boldsymbol{v}}_{1,i})_{i\leq N} \subseteq \mathbb{R}^{d_0}$ and $(\tilde{\boldsymbol{v}}_{2,i})_{i\leq N} \subseteq \mathbb{R}^{d-d_0}$ such that $\tilde{\boldsymbol{w}}_i = \boldsymbol{U}\tilde{\boldsymbol{v}}_{1,i} + \boldsymbol{U}^\perp \tilde{\boldsymbol{v}}_{2,i}$. We define $\mathring{\boldsymbol{v}}_{1,i} = \tilde{r} \cdot \tilde{\boldsymbol{v}}_{1,i}/\mathring{r}$, $\mathring{\boldsymbol{w}}_i = \boldsymbol{U}\mathring{\boldsymbol{v}}_{1,i} + \boldsymbol{U}^\perp \tilde{\boldsymbol{v}}_{2,i}$, $\mathring{\boldsymbol{W}} = (\mathring{\boldsymbol{w}}_i)_{i\leq N}$, and $\mathring{\boldsymbol{b}} = \tilde{\boldsymbol{b}}$. Then we have

$$\mathbb{E}_{\mathring{\boldsymbol{x}}}[(\mathring{f}_*(\mathring{\boldsymbol{x}}) - f_{\mathrm{NN},N}(\mathring{\boldsymbol{x}}; \mathring{\boldsymbol{W}}, \mathring{\boldsymbol{b}}))^2] = \mathbb{E}_{\tilde{\boldsymbol{x}}}[(\tilde{f}_*(\tilde{\boldsymbol{x}}) - f_{\mathrm{NN},N}(\tilde{\boldsymbol{x}}; \tilde{\boldsymbol{W}}, \tilde{\boldsymbol{b}}))^2].$$

On the other hand, for any $\mathring{\boldsymbol{W}} = (\mathring{\boldsymbol{w}}_i)_{i\leq N} \subseteq \mathbb{R}^d$ and $\mathring{\boldsymbol{b}} = (\mathring{b}_i)_{i\leq N} \subseteq \mathbb{R}$, we can find $\tilde{\boldsymbol{W}} = (\tilde{\boldsymbol{w}}_i)_{i\leq N} \subseteq \mathbb{R}^d$ and $\tilde{\boldsymbol{b}} = (\tilde{b}_i)_{i\leq N} \subseteq \mathbb{R}$ such that the above equation holds. This proves that $R_{\mathrm{NN},N}(f_*)$ is independent of $\kappa$.

# J Convergence of the Gegenbauer coefficients

In this section, we prove a string of lemmas that are used to show convergence of the Gegenbauer coefficients.

## J.1 Technical lemmas

First recall that for $q \in [Q]$ we denote $\tau^{(q)} \equiv \|\boldsymbol{\theta}^{(q)}\|_2/\sqrt{d_q}$ where $\boldsymbol{\theta}^{(q)}$ are the $d_q$ coordinates of $\boldsymbol{\theta} \sim \text{Unif}(\mathbb{S}^{D-1}(\sqrt{D}))$ associated to the $q$-th sphere of $\text{PS}^{\boldsymbol{d}}$. We show that $\tau^{(q)}$ is $(1/d_q)$-sub-Gaussian.

**Lemma 16.** *There exists constants $c, C > 0$ such that for any $\varepsilon > 0$,*

$$\mathbb{P}(|\tau^{(q)} - 1| > \varepsilon) \leq C \exp(-cd_q\varepsilon^2).$$

*Proof of Lemma 16.* Let $\boldsymbol{G} \sim \mathsf{N}(0, \mathbf{I}_D)$. We consider the random vector $\boldsymbol{U} \equiv \boldsymbol{G}/\|\boldsymbol{G}\|_2 \in \mathbb{R}^D$. We have $\boldsymbol{U} \sim \text{Unif}(\mathbb{S}^{D-1}(1))$. We denote $N_{d_q} = G_1^2 + \ldots + G_{d_q}^2$ and $N_D = G_1^2 + \ldots + G_D^2$. The random variable $\tau^{(q)}$ has the same distribution as

$$\tau^{(q)} \equiv \|\boldsymbol{\theta}^{(q)}\|_2/\sqrt{d_q} \overset{\text{d}}{=} \frac{\sqrt{N_{d_q}/d_q}}{\sqrt{N_D/D}}.$$

Hence,

$$
\begin{aligned}
\mathbb{P}(|\tau^{(q)} - 1| > \varepsilon) =& \mathbb{P}\left(\left|\frac{\sqrt{N_{d_q}/d_q}}{\sqrt{N_D/D}} - 1\right| > \varepsilon\right) \\
\leq& \mathbb{P}\left(\left|\sqrt{N_{d_q}/d_q} - 1\right| > \varepsilon/2\right) + \mathbb{P}\left(\left|\sqrt{N_D/D} - 1\right| > \varepsilon/(2 + 2\varepsilon)\right),
\end{aligned}
\tag{141}
$$

where we used the fact that

$$|a - 1| \leq \frac{\varepsilon}{2} \text{ and } |b - 1| \leq \frac{\varepsilon}{2 + 2\varepsilon} \Rightarrow \left|\frac{a}{b} - 1\right| \leq \varepsilon.$$

Let us first consider $N_{d_q}$ with $\varepsilon \in (0, 2]$. The $G_i^2$ are sub-exponential random variables with

$$\mathbb{E}\left[e^{\lambda(G_i^2 - 1)}\right] \leq e^{2\lambda^2}, \qquad \forall |\lambda| < 1/4.$$

From standard sub-exponential concentration inequality, we get

$$\mathbb{P}\left(\left|N_{d_q}/d_q - 1\right| > \varepsilon\right) \leq 2 \exp\left(-d_q\varepsilon \min(1, \varepsilon)/8\right). \tag{142}$$

Hence, for $\varepsilon \in (0, 2]$, we have

$$\mathbb{P}\left(\left|\sqrt{N_{d_q}/d_q} - 1\right| > \varepsilon/2\right) \leq \mathbb{P}\left(\left|N_{d_q}/d_q - 1\right| > \varepsilon/2\right) \leq 2 \exp\left(-d_q\varepsilon^2/32\right),$$

while for $\varepsilon > 2$,

$$
\begin{aligned}
\mathbb{P}\left(\left|\sqrt{N_{d_q}/d_q} - 1\right| > \varepsilon/2\right) \leq \mathbb{P}\left(N_{d_q}/d_q > (\varepsilon/2 + 1)^2\right) \leq& \mathbb{P}\left(N_{d_q}/d_q - 1 > \varepsilon^2/4\right) \\
\leq& \exp\left(-d_q\varepsilon^2/32\right).
\end{aligned}
$$

In the case of $N_D$, applying (142) with $\varepsilon/(2 + 2\varepsilon) \leq 1$ shows that

$$
\begin{aligned}
\mathbb{P}\left(\left|\sqrt{N_D/D} - 1\right| > \varepsilon/(2 + 2\varepsilon)\right) \leq& \mathbb{P}\left(\left|N_D/D - 1\right| > \varepsilon/(2 + 2\varepsilon)\right) \\
\leq& 2 \exp\left(-D\varepsilon^2/(32(1 + \varepsilon)^2)\right).
\end{aligned}
$$

Combining the above bounds into (141) yields for $\varepsilon \geq 0$,

$$
\begin{aligned}
\mathbb{P}(|\tau^{(q)} - 1| > \varepsilon) \leq& 2 \exp\left(-d_q\varepsilon^2/32\right) + 2 \exp\left(-D\varepsilon^2/(32(1 + \varepsilon)^2)\right) \\
\leq& 4 \exp\left(-\varepsilon^2 \min\left(d_q, D/(1 + \varepsilon)^2\right)/32\right).
\end{aligned}
$$

Notice that $|\tau^{(q)} - 1| \leq \sqrt{D/d_q} - 1$ and we only need to consider $\varepsilon \in [0, \sqrt{D/d_q} - 1]$. We conclude that for any $\varepsilon \geq 0$, we have

$$\mathbb{P}(|\tau^{(q)} - 1| > \varepsilon) \leq 4\exp\left(-d_q\varepsilon^2/32\right).$$

$\square$

We consider an activation function $\sigma : \mathbb{R} \to \mathbb{R}$. Fix $\boldsymbol{\theta} \in \mathbb{S}^{D-1}(\sqrt{D})$ and recall that $\boldsymbol{x} = (\boldsymbol{x}^{(1)}, \dots \boldsymbol{x}^{(Q)}) \in \mathrm{PS}_{\boldsymbol{\kappa}}^{\boldsymbol{d}}$. We recall that $\boldsymbol{x} \sim \mathrm{Unif}(\mathrm{PS}_{\boldsymbol{\kappa}}^{\boldsymbol{d}}) = \mu_{\boldsymbol{d}}^{\boldsymbol{\kappa}}$ while $\overline{\boldsymbol{x}} \sim \mathrm{Unif}(\mathrm{PS}^{\boldsymbol{d}}) = \mu_{\boldsymbol{d}}$. Therefore, for a given $\overline{\boldsymbol{\theta}}$, $\{\langle\overline{\boldsymbol{\theta}}^{(q)}, \overline{\boldsymbol{x}}^{(q)}\rangle/\sqrt{d_q}\}_{q\in[Q]} \sim \tilde{\mu}_{\boldsymbol{d}}^1$ as defined in Eq. (23). Therefore we reformulate $\sigma(\langle\boldsymbol{\theta}, \cdot\rangle/R)$ as a function $\sigma_{\boldsymbol{d},\boldsymbol{\tau}}$ from $\mathrm{ps}^{\boldsymbol{d}}$ to $\mathbb{R}$:

$$\begin{aligned}
\sigma(\langle\boldsymbol{\theta}, \boldsymbol{x}\rangle/R) &= \sigma\left(\sum_{q\in[Q]} \tau^{(q)} \cdot (r_q/R) \cdot \langle\overline{\boldsymbol{\theta}}^{(q)}, \overline{\boldsymbol{x}}^{(q)}\rangle/\sqrt{d_q}\right) \\
&\equiv \sigma_{\boldsymbol{d},\boldsymbol{\tau}}\left(\{\langle\overline{\boldsymbol{\theta}}^{(q)}, \overline{\boldsymbol{x}}^{(q)}\rangle/\sqrt{d_q}\}_{q\in[Q]}\right).
\end{aligned}$$

(143)

We will denote in the rest of this section $\alpha_q = \tau^{(q)} r_q/R$ for $q = 1, \dots, Q$. Notice in particular that $\alpha_q \propto d^{\eta_q + \kappa_q - \xi}$ where we recall that $\xi = \max_{q\in[Q]}\{\eta_q + \kappa_q\}$. Without loss of generality, we will assume that the (unique) maximum is attained on the first sphere, i.e. $\xi = \eta_1 + \kappa_1$ and $\xi > \eta_q + \kappa_q$ for $q \geq 2$.

**Lemma 17.** *Assume $\sigma$ is an activation function with $\sigma(u)^2 \leq c_0\exp(c_1 u^2/2)$ almost surely, for some constants $c_0 > 1$ and $c_1 < 1$. We consider the function $\sigma_{\boldsymbol{d},\boldsymbol{\tau}} : \mathrm{ps}^{\boldsymbol{d}} \to \mathbb{R}$ associated to $\sigma$, as defined in Eq. (65).*

*Then*

*(a) $\mathbb{E}_{G\sim\mathsf{N}(0,1)}[\sigma(G)^2] < \infty$.*

*(b) Let $\boldsymbol{w}^{(q)}$ be unit vectors in $\mathbb{R}^{d_q}$ for $q = 1, \dots, Q$. There exists $\varepsilon_0 = \varepsilon_0(c_1)$ and $d_0 = d_0(c_1)$ such that, for $\overline{\boldsymbol{x}} = (\boldsymbol{x}^{(1)}, \dots, \boldsymbol{x}^{(Q)}) \sim \mu_{\boldsymbol{d}}^{\boldsymbol{\kappa}}$,*

$$\sup_{d\geq d_0}\sup_{\boldsymbol{\tau}\in[1-\varepsilon_0, 1+\varepsilon_0]^Q}\mathbb{E}_{\overline{\boldsymbol{x}}}\left[\sigma_{\boldsymbol{d},\boldsymbol{\tau}}\left(\{\langle\boldsymbol{w}^{(q)}, \overline{\boldsymbol{x}}^{(q)}\rangle\}_{q\in[Q]}\right)^2\right] < \infty.$$

(144)

*(c) Let $\boldsymbol{w}^{(q)}$ be unit vectors in $\mathbb{R}^{d_q}$ for $q = 1, \dots, Q$. Fix integers $\boldsymbol{k} = (k_1, \dots, k_Q) \in \mathbb{Z}_{\geq 0}^Q$. Then for any $\delta > 0$, there exists constants $\varepsilon_0 = \varepsilon_0(c_1, \delta)$ and $d_0 = d_0(c_1, \delta)$, and a coupling of $G \sim \mathsf{N}(0, 1)$ and $\overline{\boldsymbol{x}} = (\boldsymbol{x}^{(1)}, \dots, \boldsymbol{x}^{(Q)}) \sim \mu_{\boldsymbol{d}}^{\boldsymbol{\kappa}}$ such that for any $d \geq d_0$ and $\boldsymbol{\tau} \in [1-\varepsilon_0, 1+\varepsilon_0]^Q$*

$$\mathbb{E}_{\overline{\boldsymbol{x}}, G}\left[\left(\left[\prod_{q\in[Q]}\left(1 - \langle\boldsymbol{w}^{(q)}, \overline{\boldsymbol{x}}^{(q)}\rangle^2/d_q\right)^{k_q}\right]\sigma_{\boldsymbol{d},\boldsymbol{\tau}}\left(\{\langle\boldsymbol{w}^{(q)}, \overline{\boldsymbol{x}}^{(q)}\rangle\}_{q\in[Q]}\right) - \sigma(G)\right)^2\right] < \delta.$$

(145)

*Proof of Lemma 17.* Part $(a)$ is straightforward.

For part $(b)$, recall that the probability distribution of $\langle\boldsymbol{w}^{(q)}, \overline{\boldsymbol{x}}^{(q)}\rangle$ when $\overline{\boldsymbol{x}}^{(q)} \sim \mathrm{Unif}(\mathbb{S}^{d_q-1}(\sqrt{d_q}))$ is given by

$$\tilde{\tau}_{d_q-1}^1(\mathrm{d}x) = C_{d_q}\left(1 - \frac{x^2}{d_q}\right)^{\frac{d_q-3}{2}}\mathbf{1}_{x\in[-\sqrt{d_q}, \sqrt{d_q}]}\mathrm{d}x,$$

(146)

$$C_{d_q} = \frac{\Gamma(d_q - 1)}{2^{d_q-2}\sqrt{d_q}\,\Gamma((d_q-1)/2)^2}.$$

(147)

A simple calculation shows that $C_n \to (2\pi)^{-1/2}$ as $n \to \infty$, and hence $\sup_n C_n \le \overline{C} < \infty$. Therefore for $\boldsymbol{\tau} \in [1 - \varepsilon, 1 + \varepsilon]^Q$, we have

$$\mathbb{E}_{\overline{\boldsymbol{x}}}\left[\sigma_{\boldsymbol{d},\boldsymbol{\tau}}\left(\{\langle \boldsymbol{w}^{(q)}, \overline{\boldsymbol{x}}^{(q)}\rangle\}_{q\in[Q]}\right)^2\right]$$

$$= \int_{\prod_{q\in[Q]}[-\sqrt{d_q},\sqrt{d_q}]} \sigma_{\boldsymbol{d},\boldsymbol{\tau}}\left(\overline{x}_1^{(1)},\ldots,\overline{x}_1^{(Q)}\right)^2 \prod_{q\in[Q]}\left(C_{d_q}\left(1 - \frac{(\overline{x}_1^{(q)})^2}{d_q}\right)^{\frac{d_q-3}{2}} \mathrm{d}\overline{x}_1^{(q)}\right)$$

$$\le \overline{C}^Q \int_{\mathbb{R}^Q} c_0 \exp\left(c_1\left(\sum_{q\in[Q]}\alpha_q\overline{x}_1^{(q)}\right)^2/2\right) \prod_{q\in[Q]}\left(\exp\left(-\frac{d_q-3}{2d_q}(\overline{x}_1^{(q)})^2\right)\mathrm{d}\overline{x}_1^{(q)}\right)$$

$$= c_0\overline{C}^Q \int_{\mathbb{R}^Q} \exp\left(-\overline{\boldsymbol{x}}_1^{\mathsf{T}}\boldsymbol{M}\overline{\boldsymbol{x}}_1/2\right)\left(\prod_{q\in[Q]}\mathrm{d}\overline{x}_1^{(q)}\right)$$

where we denoted $\overline{\boldsymbol{x}}_1 = (\overline{x}_1^{(1)},\ldots,\overline{x}_1^{(Q)})$ and $\boldsymbol{M} \in \mathbb{R}^{Q\times Q}$ with

$$M_{qq} = \frac{d_q-3}{d_q} - c_1^2\alpha_q^2, \qquad M_{qq'} = -c_1\alpha_q\alpha_{q'}, \qquad \text{for } q \ne q' \in [Q].$$

Recalling the definition of $\alpha_q = \tau^{(q)}r_q/R$, with $r_q = d^{(\eta_q+\kappa_q)/2}$ and $R = d^{\xi/2}(1+o_d(1))$. Hence for any $\varepsilon > 0$, uniformly on $\boldsymbol{\tau} \in [1-\varepsilon, 1+\varepsilon]^Q$, we have $\alpha_q \to 0$ for $q \ge 2$ and $\limsup_{d\to\infty}|\alpha_1-1| \le \varepsilon$. Hence if we choose $\varepsilon_0 < c_1^{-1} - 1$, there exists $c > 0$ such that for $d$ sufficiently large $\boldsymbol{M} \succeq c\mathbf{I}_Q$ and for any $\boldsymbol{\tau} \in [1-\varepsilon_0, 1+\varepsilon_0]^Q$

$$\mathbb{E}_{\overline{\boldsymbol{x}}}\left[\sigma_{\boldsymbol{d},\boldsymbol{\tau}}\left(\{\langle \boldsymbol{w}^{(q)}, \overline{\boldsymbol{x}}^{(q)}\rangle\}_{q\in[Q]}\right)^2\right] \le c_0\overline{C}^Q \int_{\mathbb{R}^Q} \exp\left(-c\|\overline{\boldsymbol{x}}_1\|_2^2/2\right)\left(\prod_{q\in[Q]}\mathrm{d}\overline{x}_1^{(q)}\right) < \infty.$$

Finally, for part $(c)$, without loss of generality we will take $\boldsymbol{w}^{(q)} = \boldsymbol{e}_1^{(q)}$ so that $\langle \boldsymbol{w}^{(q)}, \overline{\boldsymbol{x}}^{(q)}\rangle = \overline{x}_1^{(q)}$. From part $(b)$, there exists $\varepsilon > 0$ and $d_0$ such that

$$\sup_{d\ge d_0}\sup_{\boldsymbol{\tau}\in[1-\varepsilon,1+\varepsilon]}\mathbb{E}_{\overline{\boldsymbol{x}}}\mathbb{E}_{\overline{\boldsymbol{x}},G}\left[\left[\prod_{q\in[Q]}\left(1 - \langle \boldsymbol{w}^{(q)}, \overline{\boldsymbol{x}}^{(q)}\rangle^2/d_q\right)^{2k_q}\right]\sigma_{\boldsymbol{d},\boldsymbol{\tau}}\left(\{\langle \boldsymbol{w}^{(q)}, \overline{\boldsymbol{x}}^{(q)}\rangle\}_{q\in[Q]}\right)^2\right]$$

$$\le \sup_{d\ge d_0}\sup_{\boldsymbol{\tau}\in[1-\varepsilon,1+\varepsilon]}\mathbb{E}_{\overline{\boldsymbol{x}}}\left[\sigma_{\boldsymbol{d},\boldsymbol{\tau}}\left(\{\langle \boldsymbol{w}^{(q)}, \overline{\boldsymbol{x}}^{(q)}\rangle\}_{q\in[Q]}\right)^2\right] < \infty.$$

Consider $\boldsymbol{G} \sim \mathsf{N}(0,\mathbf{I}_Q)$ and an arbitrary coupling between $\overline{\boldsymbol{x}}$ and $\boldsymbol{G}$. For any $M > 0$ we can choose $\sigma_M$ bounded continuous so that for any $d$ and $\boldsymbol{\tau} \in [1-\varepsilon, 1+\varepsilon]^Q$,

$$\mathbb{E}_{\overline{\boldsymbol{x}},\boldsymbol{G}}\left[\left(\prod_{q\in[Q]}\left(1 - (\overline{x}_1^{(q)})^2/d_q\right)^{k_q}\cdot\sigma\left(\sum_{q\in[Q]}\alpha_q\overline{x}_1^{(q)}\right) - \prod_{q\in[Q]}\left(1 - G_q^2/d_q\right)^{k_q}\cdot\sigma\left(\sum_{q\in[Q]}\alpha_qG_q\right)\right)^2\right]$$

$$\le \mathbb{E}_{\overline{\boldsymbol{x}},\boldsymbol{G}}\left[\left(\prod_{q\in[Q]}\left(1 - (\overline{x}_1^{(q)})^2/d_q\right)^{k_q}\cdot\sigma_M\left(\sum_{q\in[Q]}\alpha_q\overline{x}_1^{(q)}\right) - \prod_{q\in[Q]}\left(1 - G_q^2/d_q\right)^{k_q}\cdot\sigma_M\left(\sum_{q\in[Q]}\alpha_qG_q\right)\right)^2\right] + \frac{1}{M}.$$

$$\tag{148}$$

It is therefore sufficient to prove the claim for $\sigma_M$. Letting $\boldsymbol{\xi}_q \sim \mathsf{N}(0, \mathbf{I}_{d_q-1})$ independently for each $q \in [Q]$ and independent of $\boldsymbol{G}$, we construct the coupling via

$$\overline{x}_1^{(q)} = \frac{G_q\sqrt{d_q}}{\sqrt{G_q^2 + \|\boldsymbol{\xi}_q\|_2^2}}, \qquad \overline{\boldsymbol{x}}_{-1}^{(q)} = \frac{\boldsymbol{\xi}_q\sqrt{d_q}}{\sqrt{G_q^2 + \|\boldsymbol{\xi}_q\|_2^2}}, \qquad q \in [Q], \tag{149}$$

where we set $\overline{\boldsymbol{x}}^{(q)} = (\overline{x}_1^{(q)}, \overline{\boldsymbol{x}}_{-1}^{(q)})$ for each $q \in [Q]$. We thus have $(\overline{x}_1^{(q)}, \overline{\boldsymbol{x}}_{-1}^{(q)}) \to \boldsymbol{G}$ almost surely, hence the limit superior of Eq. (148) is by weak convergence bounded by $1/M$ for any arbitrary $M$. Furthermore, noticing that $\alpha_q \to 0$ uniformly on $\boldsymbol{\tau} \in [1 - \varepsilon, 1 + \varepsilon]^Q$ for $q \geq 2$, we have by bounded convergence

$$\lim_{d \to \infty} \sup_{\boldsymbol{\tau} \in [1-\varepsilon, 1+\varepsilon]^Q} \mathbb{E}_{\overline{\boldsymbol{x}}, \boldsymbol{G}} \left[ \left( \prod_{q \in [Q]} \left( 1 - G_q^2/d_q \right)^{k_q} \cdot \sigma \left( \sum_{q \in [Q]} \alpha_q G_q \right) - \sigma(\alpha_1 G_1) \right)^2 \right] = 0. \tag{150}$$

We further have $\lim_{(d, \tau^{(1)}) \to (\infty, 1)} \alpha_1 = 1$. Hence, by bounded convergence,

$$\lim_{(d, \tau^{(1)}) \to (\infty, 1)} \mathbb{E}_{G_1} \left[ (\sigma(\alpha_1 G_1) - \sigma(G_1))^2 \right] = 0. \tag{151}$$

Combining Eq. (148) with the coupling (149) and Eqs (150) and (151) yields the result. $\qquad\square$

Consider the expansion of $\sigma_{\boldsymbol{d}, \boldsymbol{\tau}}$ in terms of tensor product of Gegenbauer polynomials. We have

$$\sigma(\langle \boldsymbol{\theta}, \boldsymbol{x} \rangle / R) = \sum_{\boldsymbol{k} \in \mathbb{Z}_{\geq 0}^Q} \lambda_{\boldsymbol{k}}^{\boldsymbol{d}}(\sigma_{\boldsymbol{d}, \boldsymbol{\tau}}) B(\boldsymbol{d}, \boldsymbol{k}) Q_{\boldsymbol{k}}^{\boldsymbol{d}} \left( \{ \langle \overline{\boldsymbol{\theta}}^{(q)}, \overline{\boldsymbol{x}}^{(q)} \rangle \}_{q \in [Q]} \right),$$

where

$$\lambda_{\boldsymbol{k}}^{\boldsymbol{d}}(\sigma_{\boldsymbol{d}, \boldsymbol{\tau}}) = \mathbb{E}_{\overline{\boldsymbol{x}}} \left[ \sigma_{\boldsymbol{d}, \boldsymbol{\tau}} \left( \overline{x}_1^{(1)}, \ldots, \overline{x}_1^{(Q)} \right) Q_{\boldsymbol{k}}^{\boldsymbol{d}} \left( \sqrt{d_1} \overline{x}_1^{(1)}, \ldots, \sqrt{d_Q} \overline{x}_1^{(Q)} \right) \right].$$

with the expectation taken over $\overline{\boldsymbol{x}} = (\overline{\boldsymbol{x}}^{(1)}, \ldots, \overline{\boldsymbol{x}}^{(Q)}) \sim \mu_{\boldsymbol{d}} \equiv \mathrm{Unif}(\mathrm{PS}^{\boldsymbol{d}})$. We will need the following lemma, which is direct consequence of Rodrigues formula, to get the scaling of the Gegenbauer coeffects of $\sigma_{\boldsymbol{d}, \boldsymbol{\tau}}$.

**Lemma 18.** *Let $\boldsymbol{k} = (k_1, \ldots, k_Q) \in \mathbb{Z}_{\geq 0}^Q$ and denote $|\boldsymbol{k}| = k_1 + \ldots + k_Q$. Assume that the activation function $\sigma$ is $|\boldsymbol{k}|$-times weakly differentiable and denote $\sigma^{(|\boldsymbol{k}|)}$ its $|\boldsymbol{k}|$-weak derivative. Let $\alpha_q = \tau^{(q)} r_q / R$ for $q = 1, \ldots, Q$. Then*

$$\lambda_{\boldsymbol{k}}^{\boldsymbol{d}}(\sigma_{\boldsymbol{d}, \boldsymbol{\tau}}) = \left( \prod_{q \in [Q]} \alpha_q^{k_q} \right) \cdot R(\boldsymbol{d}, \boldsymbol{k}) \cdot \mathbb{E}_{\overline{\boldsymbol{x}}} \left[ \left( \prod_{q \in [Q]} \left( 1 - \frac{(\overline{x}_1^{(q)})^2}{d_q} \right)^{k_q} \right) \cdot \sigma^{(|\boldsymbol{k}|)} \left( \sum_{q \in [Q]} \alpha_q \overline{x}_1^{(q)} \right) \right], \tag{152}$$

*where $\overline{\boldsymbol{x}} \sim \mathrm{Unif}(\mathrm{PS}^{\boldsymbol{d}})$ and*

$$R(\boldsymbol{d}, \boldsymbol{k}) = \prod_{q \in [Q]} \frac{d_q^{k_q/2} \Gamma((d_q - 1)/2)}{2^{k_q} \Gamma(k_q + (d_q - 1)/2)}.$$

*Furthermore,*

$$\lim_{d \to \infty} B(\boldsymbol{d}, \boldsymbol{k}) R(\boldsymbol{d}, \boldsymbol{k})^2 = \frac{1}{\boldsymbol{k}!}, \tag{153}$$

*where $\boldsymbol{k}! = k_1! \ldots k_Q!$.*

*Proof of Lemma 18.* We have

$$\lambda_{\boldsymbol{k}}^{\boldsymbol{d}}(\sigma_{\boldsymbol{d}, \boldsymbol{\tau}})$$
$$= \mathbb{E}_{\overline{\boldsymbol{x}}} \left[ \sigma_{\boldsymbol{d}, \boldsymbol{\tau}} \left( \overline{x}_1^{(1)}, \ldots, \overline{x}_1^{(Q)} \right) Q_{\boldsymbol{k}}^{\boldsymbol{d}} \left( \sqrt{d_1} \overline{x}_1^{(1)}, \ldots, \sqrt{d_Q} \overline{x}_1^{(Q)} \right) \right]$$
$$= \mathbb{E}_{\overline{\boldsymbol{x}}^{(1)}, \ldots, \overline{\boldsymbol{x}}^{(Q-1)}} \left[ \mathbb{E}_{\overline{\boldsymbol{x}}^{(Q)}} \left[ \sigma \left( \sum_{q \in [Q-1]} \alpha_q \overline{x}_1^{(q)} + \alpha_Q \overline{x}_1^{(Q)} \right) Q_{k_Q}^{(d_Q)}(\sqrt{d_Q} \overline{x}_1^{(Q)}) \right] \prod_{q \in [Q-1]} Q_{k_q}^{(d_q)}(\sqrt{d_q} \overline{x}_1^{(q)}) \right], \tag{154}$$

where we used the definition (24) of tensor product of Gegenbauer polynomials.

Consider the integration with respect to $\overline{\boldsymbol{x}}^{(Q)}$. Denote for ease of notations $u = \alpha_1 \overline{x}_1^{(1)} + \ldots + \alpha_{Q-1}\overline{x}_1^{(Q-1)}$. We use the Rodrigues formula for the Gegenbauer polynomials (see Eq. (14)):

$$
\mathbb{E}_{\overline{\boldsymbol{x}}^{(Q)}\sim\mathrm{Unif}(\mathbb{S}^{d_Q-1}(\sqrt{d_Q}))}\left[\sigma\left(u + \alpha_Q\overline{x}_1^{(Q)}\right)Q_{k_Q}^{(d_Q)}\left(\sqrt{d_Q}\overline{x}_1^{(Q)}\right)\right]
$$

$$
=\frac{\omega_{d_Q-2}}{\omega_{d_Q-1}}\int_{[-1,1]}\sigma\left(\alpha_Q\sqrt{d_Q}t+u\right)Q_{k_Q}^{(d_Q)}(d_Qt)(1-t^2)^{(d_Q-3)/2}\mathrm{d}t
$$

$$
=(-1/2)^{k_Q}\frac{\Gamma((d_Q-1)/2)}{\Gamma(k_Q+(d_Q-1)/2)}\cdot\frac{\omega_{d_Q-2}}{\omega_{d_Q-1}}\int_{[-1,1]}\sigma\left(\alpha_Q\sqrt{d_Q}t+u\right)\left(\frac{\mathrm{d}}{\mathrm{d}t}\right)^{k_Q}(1-t^2)^{k_Q+(d_Q-3)/2}\mathrm{d}t
$$

$$
=\alpha_Q^{k_Q}2^{-k_Q}d_Q^{k_Q/2}\frac{\Gamma((d_Q-1)/2)}{\Gamma(k_Q+(d_Q-1)/2)}\cdot\frac{\omega_{d_Q-2}}{\omega_{d_Q-1}}\int_{[-1,1]}(1-t^2)^{k_Q}\sigma^{(k_Q)}\left(\alpha_Q\sqrt{d_Q}t+u\right)(1-t^2)^{(d_Q-3)/2}\mathrm{d}t
$$

$$
=\alpha_Q^{k_Q}\frac{d_Q^{k_Q/2}\Gamma((d_Q-1)/2)}{2^{k_Q}\Gamma(k_Q+(d_Q-1)/2)}\mathbb{E}_{\overline{\boldsymbol{x}}^{(Q)}\sim\mathrm{Unif}(\mathbb{S}^{d_Q-1}(\sqrt{d_Q}))}\left[\left(1-(\overline{x}_1^{(Q)})^2/d_Q\right)^{k_Q}\sigma^{(k_Q)}\left(\alpha_Q\overline{x}_1^{(Q)}+u\right)\right].
$$

$$(155)$$

Iterating Eq. (155) over $q \in [Q]$ and Eq. (154) yield the desired formula (152).

Furthermore, for each $q \in [Q]$,

$$
k_q!B(d_q,k_q)=(2k_q+d_q-2)\prod_{j=0}^{k_q-2}(j+d_q-1),
$$

$$
\frac{\Gamma((d_q-1)/2)}{2^{k_q}\Gamma(k_q+(d_q-1)/2)}=\prod_{j=0}^{k_q-1}\frac{1}{2j+d_q-1}.
$$

Combining these two equations yields

$$
\begin{aligned}
&k_q!B(d_q,k_q)\frac{d_q^{k_q}\Gamma((d_q-1)/2)^2}{2^{2k_q}\Gamma(k_q+(d_q-1)/2)^2}\\
&=\frac{2k_q+d_q-2}{2k_q+d_q-3}\cdot\left(\prod_{j=0}^{k_q-2}\frac{j+d_q-1}{2j+d_q-1}\right)\cdot\left(\prod_{j=0}^{k_q-1}\frac{d_q}{2j+d_q-1}\right),
\end{aligned}
\tag{156}
$$

which converges to 1 when $d_q \to \infty$. We deduce that

$$
\lim_{d\to\infty}B(\boldsymbol{d},\boldsymbol{k})R(\boldsymbol{d},\boldsymbol{k})^2=\frac{1}{\boldsymbol{k}!}.
$$

$\square$

## J.2 Proof of convergence in probability of the Gegenbauer coefficients

**Lemma 19.** *Let* $\boldsymbol{k} = (k_1,\ldots,k_Q) \in \mathbb{Z}_{\geq 0}^Q$ *and denote* $|\boldsymbol{k}| = k_1+\ldots+k_Q$. *Assume that the activation function* $\sigma$ *is* $|\boldsymbol{k}|$*-times weakly differentiable and denote* $\sigma^{(|\boldsymbol{k}|)}$ *its* $|\boldsymbol{k}|$*-weak derivative. Assume furthermore that there exist constants* $c_0 > 0$ *and* $c_1 < 1$ *such that* $\sigma^{(|\boldsymbol{k}|)}(u)^2 \leq c_0\exp(c_1u^2/2)$ *almost surely.*

*Then for any* $\delta > 0$, *there exists* $\varepsilon_0 \in (0,1)$ *and* $d_0$ *such that for any* $d \geq d_0$ *and* $\boldsymbol{\tau} \in [1-\varepsilon_0,1+\varepsilon_0]^Q$,

$$
\left|\left(\prod_{q\in[Q]}d^{(\xi-\eta_q-\kappa_q)k_q}\right)B(\boldsymbol{d},\boldsymbol{k})\lambda_{\boldsymbol{k}}^{\boldsymbol{d}}(\sigma_{\boldsymbol{d},\boldsymbol{\tau}})^2-\frac{\mu_{|\boldsymbol{k}|}(\sigma)^2}{\boldsymbol{k}!}\right|\leq\delta.
$$

*Proof of Lemma 19.* From Lemma 18, we have

$$
\left( \prod_{q \in [Q]} d^{(\xi - \eta_q - \kappa_q)k_q} \right) B(\boldsymbol{d}, \boldsymbol{k}) \lambda_{\boldsymbol{k}}^{\boldsymbol{d}}(\sigma_{\boldsymbol{d},\boldsymbol{\tau}})^2
$$

$$
= \left( \prod_{q \in [Q]} \alpha_q^{2k_q} d^{(\xi - \eta_q - \kappa_q)k_q} \right) \cdot [B(\boldsymbol{d}, \boldsymbol{k}) R(\boldsymbol{d}, \boldsymbol{k})^2] \tag{157}
$$

$$
\times \mathbb{E}_{\overline{\boldsymbol{x}}} \left[ \prod_{q \in [Q]} \left( 1 - \frac{(\overline{x}_1^{(q)})^2}{d_q} \right)^{k_q} \cdot \sigma^{(|\boldsymbol{k}|)} \left( \sum_{q \in [Q]} \alpha_q \overline{x}_1^{(q)} \right) \right]^2 .
$$

Recall $\alpha_q = \tau^{(q)} r_q / R$ with $r_q = d^{(\kappa_q + \eta_q)/2}$ and $R = d^{\xi/2}(1 + o_d(1))$. Hence, we have

$$
\lim_{(d,\boldsymbol{\tau}) \to (\infty, \mathbf{1})} \prod_{q \in [Q]} \alpha_q^{2k_q} d^{(\xi - \eta_q - \kappa_q)k_q} = 1. \tag{158}
$$

Furthermore, from Lemma 18, we have

$$
\lim_{d \to \infty} B(\boldsymbol{d}, \boldsymbol{k}) R(\boldsymbol{d}, \boldsymbol{k})^2 = \frac{1}{\boldsymbol{k}!}. \tag{159}
$$

We can apply Lemma 17 to the activation function $\sigma^{(|\boldsymbol{k}|)}$. In particular part $(c)$ of the lemma implies that there exists $\varepsilon_0 \in (0,1)$ such that for $d$ sufficiently large, we have for any $\boldsymbol{\tau} \in [1 - \varepsilon_0, 1 + \varepsilon_0]^Q$,

$$
\left| E_{\overline{\boldsymbol{x}}} \left[ \left( \prod_{q \in [Q]} \left( 1 - \frac{(\overline{x}_1^{(q)})^2}{d_q} \right)^{k_q} \right) \cdot \sigma^{(|\boldsymbol{k}|)} \left( \sum_{q \in [Q]} \alpha_q \overline{x}_1^{(q)} \right) \right] - \mathbb{E}_G[\sigma^{(|\boldsymbol{k}|)}(G)] \right| \le \delta/2. \tag{160}
$$

From Eq. (18), we have $\mathbb{E}_G[\sigma^{(|\boldsymbol{k}|)}(G)] = \mu_{|\boldsymbol{k}|}(\sigma)$. Combining Eqs. (158) and (160) into Eq. (157) yields the result. $\qquad\square$

**Lemma 20.** *Let $k$ be a non negative integer and denote $\boldsymbol{k} = (k, 0, \dots, 0) \in \mathbb{Z}_{\ge 0}^Q$, where we recall that without loss of generality we choose $q = 1$ as the unique $\arg\max_{q \in [Q]}\{\eta_q + \kappa_q\}$. Assume that the activation function $\sigma$ verifies $\sigma(u)^2 \le c_0 \exp(c_1 u^2/2)$ almost surely for some constants $c_0 > 0$ and $c_1 < 1$.*

*Then for any $\delta > 0$, there exists $\varepsilon_0 = \varepsilon_0(c_1, \delta)$ and $d_0 = d_0(c_1, \delta)$ such that for any $d \ge d_0$ and $\boldsymbol{\tau} \in [1 - \varepsilon_0, 1 + \varepsilon_0]^Q$,*

$$
\left| B(d_1, k) \lambda_{\boldsymbol{k}}^{\boldsymbol{d}}(\sigma_{\boldsymbol{d},\boldsymbol{\tau}})^2 - \frac{\mu_k(\sigma)^2}{k!} \right| \le \delta.
$$

*Proof of Lemma 20.* Recall the correspondence (19) between Gegenbauer and Hermite polynomials. Note for any monomial $m_l(x) = x^k$, we can apply Lemma 17.$(c)$ to $m_l(\overline{x}_1^{(q_\xi)})\sigma$ and find a coupling such that for any $\eta > 0$, there exists $\varepsilon_0 > 0$ and

$$
\lim_{d \to \infty} \sup_{\boldsymbol{\tau} \in [1 - \varepsilon_0, 1 + \varepsilon_0]^Q} \mathbb{E}_{\overline{\boldsymbol{x}}, G} \left[ \left( m_k(\overline{x}_1^{(q_\xi)}) \sigma_{\boldsymbol{d},\boldsymbol{\tau}}(\overline{x}_1^{(1)}, \dots, \overline{x}_1^{(Q)}) - m_k(G)\sigma(G) \right)^2 \right] \le \eta. \tag{161}
$$

We have

$$
[B(d_1, k) k!]^{1/2} \lambda_{\boldsymbol{k}}^{\boldsymbol{d}}(\sigma_{\boldsymbol{d},\boldsymbol{\tau}}) = \mathbb{E}_{\overline{\boldsymbol{x}}} [\sigma_{\boldsymbol{d},\boldsymbol{\tau}}(\overline{x}_1^{(1)}, \dots, \overline{x}_1^{(Q)}) Q_k^{(d_1)}(\sqrt{d_1} \overline{x}_1^{(1)}) [B(d_1, k) k!]^{1/2}].
$$

Using the asymptotic correspondence between Gegenbauer polynomials and Hermite polynomials (19)

$$
\lim_{d \to \infty} \mathrm{Coeff}\{Q_k^{(d)}(\sqrt{d}x) B(d, k)^{1/2}\} = \mathrm{Coeff}\left\{ \frac{1}{(k!)^{1/2}} \mathrm{He}_k(x) \right\},
$$

and Eq. (161), we get for any $\delta > 0$, there exists $\varepsilon_0 > 0$ such that for $d$ sufficiently large, we have for any $\boldsymbol{\tau} \in [1 - \varepsilon_0, 1 + \varepsilon_0]^Q$,

$$
\left| E_{\overline{\boldsymbol{x}}} \left[ \sigma_{\boldsymbol{d},\boldsymbol{\tau}}(\overline{x}_1^{(1)}, \dots, \overline{x}_1^{(Q)}) Q_k^{(d_1)}(\sqrt{d_1} \overline{x}_1^{(1)}) [B(d_1, k) k!]^{1/2} \right] - \mathbb{E}_G[\sigma(G)\mathrm{He}_k(G)] \right| \le \delta,
$$

which concludes the proof. $\qquad\square$

## K  Bound on the operator norm of Gegenbauer polynomials

**Proposition 5** (Bound on the Gram matrix). *Let $\boldsymbol{k} \in \mathbb{Z}_{\geq 0}^Q$ and denote $\gamma = \sum_{q \in [Q]} \eta_q k_q$. Let $n \leq d^\gamma / e^{A_d \sqrt{\log d}}$ for any $A_d \to \infty$. Let $(\overline{\boldsymbol{x}}_i)_{i \in [n]}$ with $\overline{\boldsymbol{x}}_i = (\{\overline{\boldsymbol{x}}_i^{(q)}\}_{q \in [Q]}) \sim \mathrm{Unif}(\mathrm{PS}^{\boldsymbol{d}})$ independently, and $Q_{k_q}^{(d_q)}$ be the $k_q$'th Gegenbauer polynomial with domain $[-d_q, d_q]$. Consider the random matrix $\boldsymbol{W} = (\boldsymbol{W}_{ij})_{i,j \in [n]} \in \mathbb{R}^{n \times n}$, with*

$$\boldsymbol{W}_{ij} = Q_{\boldsymbol{k}}^{\boldsymbol{d}}(\{\langle \overline{\boldsymbol{x}}_i^{(q)}, \overline{\boldsymbol{x}}_j^{(q)} \rangle\}_{q \in [Q]}) = \prod_{q \in [Q]} Q_{k_q}^{(d_q)}(\langle \overline{\boldsymbol{x}}_i^{(q)}, \overline{\boldsymbol{x}}_j^{(q)} \rangle).$$

*Then we have*

$$\lim_{d,n \to \infty} \mathbb{E}[\|\boldsymbol{W} - \mathbf{I}_n\|_{\mathrm{op}}] = 0.$$

**Corollary 1** (Uniform bound on the Gram matrix). *Let $n \leq d^\gamma / e^{A_d \sqrt{\log d}}$ for some $\gamma > 0$ and any $A_d \to \infty$. Let $(\overline{\boldsymbol{x}}_i)_{i \in [N]}$ with $\overline{\boldsymbol{x}}_i = (\{\overline{\boldsymbol{x}}_i^{(q)}\}_{q \in [Q]}) \sim \mathrm{Unif}(\mathrm{PS}^{\boldsymbol{d}})$ independently. Consider for any $\boldsymbol{k} \in \mathbb{Z}_{\geq 0}^Q$, the random matrix $\boldsymbol{W}_{\boldsymbol{k}} = ((\boldsymbol{W}_{\boldsymbol{k}})_{ij})_{i,j \in [n]} \in \mathbb{R}^{n \times n}$ as defined in Proposition 5. Denote:*

$$\mathcal{Q} = \Big\{ \boldsymbol{k} \in \mathbb{Z}_{\geq 0}^Q \Big| \sum_{q \in [Q]} \eta_q k_q < \gamma \Big\}.$$

*Then we have*

$$\sup_{\boldsymbol{k} \in \mathcal{Q}^c} \mathbb{E}[\|\boldsymbol{W}_{\boldsymbol{k}} - \mathbf{I}_n\|_{\mathrm{op}}] = o_{d,\mathbb{P}}(1).$$

*Proof of Corollary 1.* For each $q \in [Q]$, we consider $\boldsymbol{\Delta}^{(q)} = \boldsymbol{W}_k^{(q)} - \mathbf{I}_n$ where $\boldsymbol{W}_k^{(q)} = ((\boldsymbol{W}_k^{(q)})_{ij})_{i,j \in [n]}$ with

$$(\boldsymbol{W}_k^{(q)})_{ij} = Q_k^{(d_q)}(\langle \overline{\boldsymbol{x}}_i^{(q)}, \overline{\boldsymbol{x}}_j^{(q)} \rangle).$$

Then, defining $\gamma_q \equiv \gamma / \eta_q$, we have

$$\mathbb{E}\Big[ \sup_{k \geq 2\gamma_q + 3} \|\boldsymbol{W}_k^{(q)} - \mathbf{I}_n\|_{\mathrm{op}}^2 \Big] \leq \mathbb{E}\Big[ \sum_{k \geq 2\gamma_q + 3} \|\boldsymbol{W}_k^{(q)} - \mathbf{I}_n\|_F^2 \Big]$$

$$= n(n-1) \sum_{k \geq 2\gamma_q + 3} \mathbb{E}[Q_k^{(d_q)}(\langle \overline{\boldsymbol{x}}^{(q)}, \overline{\boldsymbol{y}}^{(q)} \rangle)^2] = n(n-1) \sum_{k \geq 2\gamma_q + 3} B(d_q, k)^{-1}.$$

For $d$ sufficiently large, there exists $C > 0$ such that for any $p \geq m \equiv \lceil 2\gamma_q + 3 \rceil$:

$$\frac{B(d_q, m)}{B(d_q, p)} = \prod_{k=m}^{p-1} \frac{(2k + d_q - 2)}{(2k + d_q)} \cdot \frac{(k+1)}{(k + d_q - 2)} \leq \prod_{k=m}^{p-1} \frac{1}{1 + (d_q - 3)/(k+1)}$$

$$\leq \prod_{k=m}^{p-1} e^{-\frac{m+1}{d_q - 2 + m} \cdot \frac{d_q - 2}{k+1}} \leq \frac{C}{p^2}.$$

Hence, there exists constant $C'$, such that for large $d$, we have

$$\sum_{k \geq 2\gamma_q + 3} B(d_q, k)^{-1} \leq C' \cdot B(d_q, m)^{-1}.$$

Recalling that $B(d_q, m) = \Theta_d(d^{\eta_q m}) = \omega_d(d^{2\gamma})$, and $n = o_d(d^\gamma)$, we deduce

$$\mathbb{E}\Big[ \sup_{k \geq 2\gamma_q + 3} \|\boldsymbol{W}_k^{(q)} - \mathbf{I}_n\|_{\mathrm{op}}^2 \Big] = o_d(1). \tag{162}$$

Let us now consider $\boldsymbol{\Delta} = \boldsymbol{W}_{\boldsymbol{k}} - \mathbf{I}_n$. We will denote $\boldsymbol{\Delta}^{(q)} = \boldsymbol{W}_{k_q}^{(q)} - \mathbf{I}_n$. Then it is easy to check (recall the diagonal elements of $\boldsymbol{W}_{k_q}^{(d_q)}$ are equal to one) that for any $q \in [Q]$

$$\boldsymbol{\Delta} = \Big( \bigodot_{q' \neq q} \boldsymbol{W}_{k_{q'}}^{(q')} \Big) \odot \boldsymbol{\Delta}^{(q)}$$

where $\boldsymbol{A} \odot \boldsymbol{B}$ denotes the Hadamard product, or entrywise product, $(\boldsymbol{A} \odot \boldsymbol{B})_{i,j \in [n]} = (A_{ij} B_{ij})_{i,j \in [n]}$. We recall the following inequality on the operator norm of Hadamard product of two matrices, with $\boldsymbol{A}$ positive definite:

$$\|\boldsymbol{A} \odot \boldsymbol{B}\|_{\mathrm{op}} \leq \Big( \max_{ij} \boldsymbol{A}_{ij} \Big) \|\boldsymbol{B}\|_{\mathrm{op}}.$$

Hence, in particular

$$\|\boldsymbol{\Delta}\|_{\mathrm{op}} \leq \Big( \prod_{q' \neq q} \max_{ij} [(\boldsymbol{W}_{k_{q'}}^{(q')})_{ij}] \Big) \|\boldsymbol{\Delta}^{(q)}\|_{\mathrm{op}}$$

Consider $\mathcal{I} = [0, 2\gamma_1 + 3[ \times \ldots \times [0, 2\gamma_Q + 3[ \cap \mathbb{Z}_{\geq 0}^Q$. Then, from Eq. (162), we get directly

$$\sup_{\boldsymbol{k} \in \mathcal{I}^c} \|\boldsymbol{W}_{\boldsymbol{k}} - \mathbf{I}_n\|_{\mathrm{op}} = o_{d,\mathbb{P}}(1). \tag{163}$$

Furthermore, $\mathcal{I} \cap \mathcal{Q}$ is finite and from Proposition 5, we directly get

$$\sup_{\boldsymbol{k} \in \mathcal{I} \cap \mathcal{Q}} \|\boldsymbol{W}_{\boldsymbol{k}} - \mathbf{I}_n\|_{\mathrm{op}} = o_{d,\mathbb{P}}(1). \tag{164}$$

Combining bounds (163) and (164) yields the result.

$\square$

## K.1  Proof of Proposition 5

The proof follows closely the proof of the uniform case presented in [4]. For completeness, we copy here the relevant lemmas.

### Step 1. Bounding operator norm by moments.

Denote $\boldsymbol{\Delta} = \boldsymbol{W} - \mathbf{I}_n$. We define for each $q \in [Q]$, $\boldsymbol{W}_{k_q}^{(d_q)} = (Q_{k_q}^{(d_q)}(\langle \overline{\boldsymbol{x}}_i^{(q)}, \overline{\boldsymbol{x}}_j^{(q)} \rangle))_{ij \in [n]}$ and $\boldsymbol{\Delta}^{(q)} = \boldsymbol{W}_{k_q}^{(d_q)} - \mathbf{I}_n$. Then it is easy to check (recall the diagonal elements of $\boldsymbol{W}_{k_q}^{(d_q)}$ are equal to one)

$$\boldsymbol{\Delta} = \boldsymbol{\Delta}^{(1)} \odot \ldots \odot \boldsymbol{\Delta}^{(Q)},$$

where $\boldsymbol{A} \odot \boldsymbol{B}$ denotes the Hadamard product, or entrywise product, $(\boldsymbol{A} \odot \boldsymbol{B})_{i,j \in [n]} = (A_{ij} B_{ij})_{i,j \in [n]}$. For any sequence of integers $p = p(d)$, we have

$$\mathbb{E}[\|\boldsymbol{\Delta}\|_{\mathrm{op}}] \leq \mathbb{E}[\mathrm{Tr}(\boldsymbol{\Delta}^{2p})^{1/(2p)}] \leq \mathbb{E}[\mathrm{Tr}(\boldsymbol{\Delta}^{2p})]^{1/(2p)} \tag{165}$$

To prove the proposition, it suffices to show that for any sequence $A_d \to \infty$, we have

$$\lim_{d,n \to \infty, n = O_d(d^\gamma e^{-A_d \sqrt{\log d}})} \mathbb{E}[\mathrm{Tr}(\boldsymbol{\Delta}^{2p})]^{1/(2p)} = 0. \tag{166}$$

In the following, we calculate $\mathbb{E}[\mathrm{Tr}(\boldsymbol{\Delta}^{2p})]$. We have

$$\mathbb{E}[\mathrm{Tr}(\boldsymbol{\Delta}^{2p})] = \sum_{\boldsymbol{i} = (i_1, \ldots, i_{2p}) \in [n]^{2p}} \mathbb{E}[\Delta_{i_1 i_2} \Delta_{i_2 i_3} \ldots \Delta_{i_{2p} i_1}]$$

$$= \sum_{\boldsymbol{i} = (i_1, \ldots, i_{2p}) \in [n]^{2p}} \prod_{q \in [Q]} \mathbb{E}[\Delta_{i_1 i_2}^{(q)} \Delta_{i_2 i_3}^{(q)} \ldots \Delta_{i_{2p} i_1}^{(q)}],$$

where we used that $\overline{\boldsymbol{x}}^{(q)}$ and $\overline{\boldsymbol{x}}^{(q')}$ are independent for $q \neq q'$.

We will denote for any $\boldsymbol{i} = (i_1, \ldots, i_k) \in [n]^k$, define for each $q \in [Q]$

$$M_{\boldsymbol{i}}^{(q)} = \begin{cases} \mathbb{E}[\Delta_{i_1 i_2}^{(q)} \cdots \Delta_{i_k i_1}^{(q)}] & k \geq 2, \\ 1 & k = 1. \end{cases}$$

Similarly, we define $M_{\boldsymbol{i}}$ associated to $\boldsymbol{\Delta}$,

$$M_{\boldsymbol{i}} = \prod_{q \in [Q]} M_{\boldsymbol{i}}^{(q)}.$$

To calculate these quantities, we will apply repeatedly the following identity, which is an immediate consequence of Eq. (11). For any $i_1, i_2, i_3$ distinct, we have

$$\mathbb{E}_{\boldsymbol{\theta}_{i_2}}[\Delta_{i_1 i_2}^{(q)} \Delta_{i_2 i_3}^{(q)}] = \frac{1}{B(d_q, k_q)} \Delta_{i_1 i_3}^{(q)}.$$

Throughout the proof, we will denote by $C, C', C''$ constants that may depend on $k$ but not on $p, d, n$. The value of these constants is allowed to change from line to line.

**Step 2. The induced graph and equivalence of index sequences.**

For any index sequence $\boldsymbol{i} = (i_1, i_2, \ldots, i_{2p}) \in [n]^{2p}$, we defined an undirected multigraph $G_{\boldsymbol{i}} = (V_{\boldsymbol{i}}, E_{\boldsymbol{i}})$ associated to index sequence $\boldsymbol{i}$. The vertex set $V_{\boldsymbol{i}}$ is the set of distinct elements in $i_1, \ldots, i_{2p}$. The edge set $E_{\boldsymbol{i}}$ is formed as follows: for any $j \in [2p]$ we add an edge between $i_j$ and $i_{j+1}$ (with convention $2p + 1 \equiv 1$). Notice that this could be a self-edge, or a repeated edge: $G_{\boldsymbol{i}} = (V_{\boldsymbol{i}}, E_{\boldsymbol{i}})$ will be –in general– a multigraph. We denote $v(\boldsymbol{i}) = |V_{\boldsymbol{i}}|$ to be the number of vertices of $G_{\boldsymbol{i}}$, and $e(\boldsymbol{i}) = |E_{\boldsymbol{i}}|$ to be the number of edges (counting multiplicities). In particular, $e(\boldsymbol{i}) = k$ for $\boldsymbol{i} \in [n]^k$. We define

$$\mathcal{T}_\star(p) = \{\boldsymbol{i} \in [n]^{2p} : G_{\boldsymbol{i}} \text{ does not have self edge}\}.$$

For any two index sequences $\boldsymbol{i}_1, \boldsymbol{i}_2$, we say they are equivalent $\boldsymbol{i}_1 \asymp \boldsymbol{i}_2$, if the two graphs $G_{\boldsymbol{i}_1}$ and $G_{\boldsymbol{i}_2}$ are isomorphic, i.e. there exists an edge-preserving bijection of their vertices (ignoring vertex labels). We denote the equivalent class of $\boldsymbol{i}$ to be

$$\mathcal{C}(\boldsymbol{i}) = \{\boldsymbol{j} : \boldsymbol{j} \asymp \boldsymbol{i}\}.$$

We define the quotient set $\mathcal{Q}(p)$ by

$$\mathcal{Q}(p) = \{\mathcal{C}(\boldsymbol{i}) : \boldsymbol{i} \in [n]^{2p}\}.$$

The following Lemma was proved in [4, Proposition 3]

**Lemma 21.** *The following properties holds for all sufficiently large $n$ and $d$:*

(a) *For any equivalent index sequences $\boldsymbol{i} = (i_1, \ldots, i_{2p}) \asymp \boldsymbol{j} = (j_1, \ldots, j_{2p})$, we have $M_{\boldsymbol{i}}^{(q)} = M_{\boldsymbol{j}}^{(q)}$.*

(b) *For any index sequence $\boldsymbol{i} \in [n]^{2p} \setminus \mathcal{T}_\star(p)$, we have $M_{\boldsymbol{i}} = 0$.*

(c) *For any index sequence $\boldsymbol{i} \in \mathcal{T}_\star(p)$, the degree of any vertex in $G_{\boldsymbol{i}}$ must be even.*

(d) *The number of equivalent classes $|\mathcal{Q}(p)| \leq (2p)^{2p}$.*

(e) *Recall that $v(\boldsymbol{i}) = |V_{\boldsymbol{i}}|$ denotes the number of distinct elements in $\boldsymbol{i}$. Then, for any $\boldsymbol{i} \in [n]^{2p}$, the number of elements in the corresponding equivalence class satisfies $|\mathcal{C}(\boldsymbol{i})| \leq v(\boldsymbol{i})^{v(\boldsymbol{i})} \cdot n^{v(\boldsymbol{i})} \leq p^p n^{v(\boldsymbol{i})}$.*

In view of property $(a)$ in the last lemma, given an equivalence class $\mathcal{C} = \mathcal{C}(\boldsymbol{i})$, we will write $M_\mathcal{C} = M_{\boldsymbol{i}}$ for the corresponding value.

**Step 3. The skeletonization process.**

For multi-graph $G$, we say that one of its vertices is *redundant*, if it has degree 2. For any index sequence $\boldsymbol{i} \in \mathcal{T}_\star(p) \subset [n]^{2p}$ (i.e. such that $G_{\boldsymbol{i}}$ does not have self-edges), we denote by $r(\boldsymbol{i}) \in \mathbb{N}_+$ to be the redundancy of $\boldsymbol{i}$, and by $\mathrm{sk}(\boldsymbol{i})$ to be the skeleton of $\boldsymbol{i}$, both defined by the following skeletonization process. Let $\boldsymbol{i}_0 = \boldsymbol{i} \in [n]^{2p}$. For any integer $s \geq 0$, if $G_{\boldsymbol{i}_s}$ has no redundant vertices then stop and set $\mathrm{sk}(\boldsymbol{i}) = \boldsymbol{i}_s$. Otherwise, select a redundant vertex $\boldsymbol{i}_s(\ell)$ arbitrarily (the $\ell$-th element of $\boldsymbol{i}_s$). If $\boldsymbol{i}_s(\ell - 1) \neq \boldsymbol{i}_s(\ell + 1)$, then remove $\boldsymbol{i}_s(\ell)$ from the graph (and from the sequence), together with its adjacent edges, and connect $\boldsymbol{i}_s(\ell - 1)$ and $\boldsymbol{i}_s(\ell + 1)$ with an edge, and denote $\boldsymbol{i}_{s+1}$ to be the resulting index sequence, i.e., $\boldsymbol{i}_{s+1} = (\boldsymbol{i}_s(1), \ldots, \boldsymbol{i}_s(\ell - 1), \boldsymbol{i}_s(\ell + 2), \ldots, \boldsymbol{i}_s(\text{end}))$. If $\boldsymbol{i}_s(\ell - 1) = \boldsymbol{i}_s(\ell + 1)$, then remove $\boldsymbol{i}_s(\ell)$ from the graph (and from the sequence), together with its adjacent edges, and denote $\boldsymbol{i}_{s+1}$ to be the resulting index sequence, i.e., $\boldsymbol{i}_{s+1} = (\boldsymbol{i}_s(1), \ldots, \boldsymbol{i}_s(\ell - 1), \boldsymbol{i}_s(\ell + 1), \boldsymbol{i}_s(\ell + 2), \ldots, \boldsymbol{i}_s(\text{end}))$. (Here $\ell + 1$, and $\ell - 1$ have to be interpreted modulo $|\boldsymbol{i}_s|$, the length of $\boldsymbol{i}_s$.) The redundancy of $\boldsymbol{i}$, denoted by $r(\boldsymbol{i})$, is the number of vertices removed during the skeletonization process.

It is easy to see that the outcome of this process is independent of the order in which we select vertices.

**Lemma 22.** *For the above skeletonization process, the following properties hold*

(a) *If $\boldsymbol{i} \asymp \boldsymbol{j} \in [n]^p$, then $\mathrm{sk}(\boldsymbol{i}) \asymp \mathrm{sk}(\boldsymbol{j})$. That is, the skeletons of equivalent index sequences are equivalent.*

(b) *For any $\boldsymbol{i} = (i_1, \dots, i_k) \in [n]^k$, and $q \in [Q]$, we have*

$$M_{\boldsymbol{i}}^{(q)} = \frac{M_{\mathrm{sk}(\boldsymbol{i})}^{(q)}}{B(d_q, k_q)^{r(\boldsymbol{i})}}.$$

(c) *For any $\boldsymbol{i} \in \mathcal{T}_\star(p) \subset [n]^{2p}$, its skeleton is either formed by a single element, or an index sequence whose graph has the property that every vertex has degree greater or equal to $4$.*

Given an index sequence $\boldsymbol{i} \in \mathcal{T}_\star(p) \subset [n]^{2p}$, we say $\boldsymbol{i}$ is of type 1, if $\mathrm{sk}(\boldsymbol{i})$ contains only one index. We say $\boldsymbol{i}$ is of type 2 if $\mathrm{sk}(\boldsymbol{i})$ is not empty (so that by Lemma 22, $G_{\mathrm{sk}(\boldsymbol{i})}$ can only contain vertices with degree greater or equal to $4$). Denote the class of type 1 index sequence (respectively type 2 index sequence) by $\mathcal{T}_1(p)$ (respectively $\mathcal{T}_2(p)$). We also denote by $\widetilde{\mathcal{T}}_a(p)$, $a \in \{1, 2\}$ the set of equivalence classes of sequences in $\mathcal{T}_a(p)$. This definition makes sense since the equivalence class of the skeleton of a sequence only depends on the equivalence class of the sequence itself.

### Step 4. Type 1 index sequences.

Recall that $v(\boldsymbol{i})$ is the number of vertices in $G_{\boldsymbol{i}}$, and $e(\boldsymbol{i})$ is the number of edges in $G_{\boldsymbol{i}}$ (which coincides with the length of $\boldsymbol{i}$). We consider $\boldsymbol{i} \in \mathcal{T}_1(p)$. Since for $\boldsymbol{i} \in \mathcal{T}_1(p)$, every edge of $G_{\boldsymbol{i}}$ must be at most a double edge. Indeed, if $(u_1, u_2)$ had multiplicity larger than 2 in $G_{\boldsymbol{i}}$, neither $u_1$ nor $u_2$ could be deleted during the skeletonization process, contradicting the assumption that $\mathrm{sk}(\boldsymbol{i})$ contains a single vertex. Therefore, we must have $\min_{\boldsymbol{i} \in \mathcal{T}_1} v(\boldsymbol{i}) = p + 1$. According the Lemma 22.(b), for every $\boldsymbol{i} \in \mathcal{T}_1(p)$, we have

$$M_{\boldsymbol{i}} = \prod_{q \in [Q]} M_{\boldsymbol{i}}^{(q)} = \prod_{q \in [Q]} 1/B(d_q, k_q)^{v(\boldsymbol{i})-1} = \frac{1}{B(\boldsymbol{d}, \boldsymbol{k})^{v(\boldsymbol{i})-1}}.$$

Note by Lemma 21.(e), the number of elements in the equivalence class of $\boldsymbol{i}$ is $|\mathcal{C}(\boldsymbol{i})| \le p^p \cdot n^{v(\boldsymbol{i})}$. Hence we get

$$\max_{\boldsymbol{i} \in \mathcal{T}_1(p)} \left[ |\mathcal{C}(\boldsymbol{i})| |M_{\boldsymbol{i}}| \right] \le \sup_{\boldsymbol{i} \in \mathcal{T}_1(p)} \left[ p^p n^{v(\boldsymbol{i})} / B(\boldsymbol{d}, \boldsymbol{k})^{v(\boldsymbol{i})-1} \right] = p^p n^{p+1} / B(\boldsymbol{d}, \boldsymbol{k})^p. \tag{167}$$

Therefore, denoting $K = \sum_{q \in [Q]} \eta_q k_q$,

$$\sum_{\boldsymbol{i} \in \mathcal{T}_1(p)} M_{\boldsymbol{i}} = \sum_{\mathcal{C} \in \widetilde{\mathcal{T}}_1(p)} |\mathcal{C}| |M_{\mathcal{C}}| \tag{168}$$

$$\le |\mathcal{Q}(p)| p^p \frac{n^{p+1}}{B(\boldsymbol{d}, \boldsymbol{k})^p} \le (Cp)^{3p} n^{p+1} d^{-Kp}. \tag{169}$$

where in the last step we used Lemma 21 and the fact that for $q \in [Q]$, $B(d_q, k_q) \ge C_0 d_q^{k_q}$ for some $C_0 > 0$.

### Step 5. Type 2 index sequences.

We have the following simple lemma bounding $M_{\boldsymbol{i}}$, copied from [4, Proposition 3]. This bound is useful when $\boldsymbol{i}$ is a skeleton.

**Lemma 23.** *For any $q \in [Q]$, there exists constants $C$ and $d_0$ depending uniquely on $k_q$ such that, for any $d \ge d_0(k_q)$, and any index sequence $\boldsymbol{i} \in [n]^m$ with $2 \le m \le d_q/(4k_q)$, we have*

$$|M_{\boldsymbol{i}}^{(q)}| \le \left( Cm^{k_q} \cdot d_q^{-k_q} \right)^{m/2}.$$

Suppose $\boldsymbol{i} \in \mathcal{T}_2(p)$, and denote $v(\boldsymbol{i})$ to be the number of vertices in $G_{\boldsymbol{i}}$. We have, for a sequence $p = o_d(d)$, and each $q \in [Q]$

$$
\begin{aligned}
|M_{\boldsymbol{i}}^{(q)}| &\overset{(1)}{=} \frac{|M_{\mathrm{sk}(\boldsymbol{i})}^{(q)}|}{B(d_q, k_q)^{r(\boldsymbol{i})}} \\
&\overset{(2)}{\leq} \left( \frac{Ce(\mathrm{sk}(\boldsymbol{i}))}{d_q} \right)^{k_q \cdot e(\mathrm{sk}(\boldsymbol{i}))/2} (C'd_q)^{-r(\boldsymbol{i})k_q} \\
&\overset{(3)}{\leq} \left( \frac{Cp}{d_q} \right)^{k_q \cdot e(\mathrm{sk}(\boldsymbol{i}))/2} (C'd_q)^{-r(\boldsymbol{i})k_q} \\
&\overset{(4)}{\leq} \left( \frac{Cp}{d_q} \right)^{k_q \cdot v(\mathrm{sk}(\boldsymbol{i}))} (C'd_q)^{-r(\boldsymbol{i})k_q} \\
&\overset{(5)}{\leq} C^{v(\boldsymbol{i})} p^{k_q \cdot v(\mathrm{sk}(\boldsymbol{i}))} d_q^{-(v(\mathrm{sk}(\boldsymbol{i}))+r(\boldsymbol{i})) \cdot k_q} \\
&\overset{(6)}{\leq} (Cp)^{k_q \cdot v(\boldsymbol{i})} d_q^{-v(\boldsymbol{i})k_q} .
\end{aligned}
$$

Here $(1)$ holds by Lemma 22.$(b)$; $(2)$ by Lemma 23, and the fact that $\mathrm{sk}(\boldsymbol{i}) \in [n]^{e(\mathrm{sk}(\boldsymbol{i}))}$, together by $B(d_q, k_q) \geq C_0 d_q^{k_q}$; $(3)$ because $e(\mathrm{sk}(\boldsymbol{i})) \leq 2p$; $(4)$ by Lemma 22.$(c)$, implying that for $\boldsymbol{i} \in \mathcal{T}_2(p)$, each vertex of $G_{\mathrm{sk}(\boldsymbol{i})}$ has degree greater or equal to 4, so that $v(\mathrm{sk}(\boldsymbol{i})) \leq e(\mathrm{sk}(\boldsymbol{i}))/2$ (notice that for $d \geq d_0(k_q)$ we can assume $Cp/d_q < 1$). Finally, $(5)$ follows since $r(\boldsymbol{i}), v(\mathrm{sk}(\boldsymbol{i})) \leq v(\boldsymbol{i})$, and $(6)$ the definition of $r(\boldsymbol{i})$ implying $r(\boldsymbol{i}) = v(\boldsymbol{i}) - v(\mathrm{sk}(\boldsymbol{i}))$.

Hence we get

$$
|M_{\boldsymbol{i}}| \leq \prod_{q \in [Q]} (Cp)^{k_q \cdot v(\boldsymbol{i})} d_q^{-v(\boldsymbol{i})k_q}
$$

Note by Lemma 21.$(e)$, the number of elements in equivalent class $|\mathcal{C}(\boldsymbol{i})| \leq p^{v(\boldsymbol{i})} \cdot n^{v(\boldsymbol{i})}$. Since $v(\boldsymbol{i})$ depends only on the equivalence class of $\boldsymbol{i}$, we will write, with a slight abuse of notation $v(\boldsymbol{i}) = v(\mathcal{C}(\boldsymbol{i}))$. Notice that the number of equivalence classes with $v(\mathcal{C}) = v$ is upper bounded by the number multi-graphs with $v$ vertices and $2p$ edges, which is at most $v^{4p}$. Denoting $\alpha = \max_{q \in [Q]} \{ 1/\eta_q \}$, we have

$$
\sum_{\boldsymbol{i} \in \mathcal{T}_2(p)} M_{\boldsymbol{i}} \leq \sum_{\mathcal{C} \in \widetilde{\mathcal{T}}_2(p)} |\mathcal{C}||M_{\mathcal{C}}| \tag{170}
$$

$$
\leq \sum_{\mathcal{C} \in \widetilde{\mathcal{T}}_2(p)} (Cp^\alpha)^{(K+1)v(\mathcal{C})} \left( \frac{n}{d^K} \right)^{v(\mathcal{C})} \tag{171}
$$

$$
\leq \sum_{v=2}^{2p} v^{4p} \left( \frac{Cnp^{\alpha(K+1)}}{d^K} \right)^v . \tag{172}
$$

Define $\varepsilon = Cnp^{\alpha(K+1)}/d^K$. We will assume hereafter that $p$ is selected such that

$$
2p \leq -\log \left( \frac{Cnp^{\alpha(K+1)}}{d^K} \right) . \tag{173}
$$

By calculus and condition (173), the function $F(v) = v^{4p} \varepsilon^v$ is maximized over $v \in [2, 2p]$ at $v = 2$, whence

$$
\sum_{\boldsymbol{i} \in \mathcal{T}_2(p)} M_{\boldsymbol{i}} \leq 2p \, F(2) \leq C^p \left( \frac{n}{d^K} \right)^2 . \tag{174}
$$

**Step 6. Concluding the proof.**

Using Eqs. (169) and (174), we have, for any $p = o_d(d)$ satisfying Eq. (173), we have

$$
\mathbb{E}[\mathrm{Tr}(\boldsymbol{\Delta}^{2p})] = \sum_{\boldsymbol{i}=(i_1,\dots,i_{2p}) \in [N]^{2p}} M_{\boldsymbol{i}} = \sum_{\boldsymbol{i} \in \mathcal{T}_1(p)} M_{\boldsymbol{i}} + \sum_{\boldsymbol{i} \in \mathcal{T}_2(p)} M_{\boldsymbol{i}} \tag{175}
$$

$$
\leq (Cp)^{3p} \frac{n^{p+1}}{d^{Kp}} + C^p \left( \frac{n}{d^K} \right)^2 . \tag{176}
$$

Form Eq. (165), we obtain

$$\mathbb{E}[\|\mathbf{\Delta}\|_{\mathrm{op}}] \leq C \left\{ p^{3/2} n^{1/(2p)} \sqrt{\frac{n}{d^K}} + \left( \frac{n}{d^K} \right)^{1/p} \right\}. \tag{177}$$

Finally setting $n = d^K e^{-2A\sqrt{\log d}}$ and $p = (K/A)\sqrt{\log d}$, this yields

$$\mathbb{E}[\|\mathbf{\Delta}\|_{\mathrm{op}}] \leq C \left\{ e^{-\frac{A}{4}\sqrt{\log d}} + e^{-2A^2/K} \right\}. \tag{178}$$

Therefore, as long as $A \to \infty$, we have $\mathbb{E}[\|\mathbf{\Delta}\|_{\mathrm{op}}] \to 0$. It is immediate to check that the above choice of $p$ satisfies the required conditions $p = o_d(d)$ and Eq. (173) for all $d$ large enough.

## L    Technical lemmas

We put here one technical lemma that is used in the proof of Theorem 3.(a).

**Lemma 24.** *Let $\boldsymbol{D} = (\boldsymbol{D}^{qq'})_{q,q'\in[Q]} \in \mathbb{R}^{DN\times DN}$ be a symmetric $Q$ by $Q$ block matrix with $\boldsymbol{D}^{qq'} \in \mathbb{R}^{d_q N\times d_{q'} N}$. Denote $\boldsymbol{B} = \boldsymbol{D}^{-1}$. Assume that $\boldsymbol{D}$ satisfies the following properties:*

*1. For any $q \in [Q]$, there exists $c_q, C_q > 0$ such that we have with high probability*

$$0 < \frac{r_q^2}{d_q}c_q = d^{\kappa_q}c_q \leq \lambda_{\min}(\boldsymbol{D}^{qq}) \leq \lambda_{\max}(\boldsymbol{D}^{qq}) \leq \frac{r_q^2}{d_q}C_q = d^{\kappa_q}C_q < \infty,$$

*as $d \to \infty$.*

*2. For any $q \neq q' \in [Q]$, we have $\sigma_{\max}(\boldsymbol{D}^{qq'}) = o_{d,\mathbb{P}}(r_q r_{q'}/\sqrt{d_q d_{q'}}) = o_{d,\mathbb{P}}(d^{(\kappa_q+\kappa_{q'})/2})$.*

*Then for any $q \neq q' \in [Q]$, we have*

$$\|\boldsymbol{B}^{qq}\|_{\mathrm{op}} = O_{d,\mathbb{P}}\left(\frac{d_q}{r_q^2}\right) = O_{d,\mathbb{P}}(d^{-\kappa_q}), \qquad \|\boldsymbol{B}^{qq'}\|_{\mathrm{op}} = o_{d,\mathbb{P}}\left(\frac{\sqrt{d_q d_{q'}}}{r_q r_{q'}}\right) = o_{d,\mathbb{P}}(d^{-(\kappa_q+\kappa_{q'})/2}).$$
$$(179)$$

*Proof of Lemma 24.* Let us show the result recursively on the integer $Q$. Note that the case $Q = 1$ is direct.

Consider $\boldsymbol{D} = (\boldsymbol{D}^{qq'})_{q,q'\in[Q]}$. Denote $\tilde{D} = D - d_Q$, $\boldsymbol{A} = (\boldsymbol{D}^{qq'})_{q,q'\in[Q-1]} \in \mathbb{R}^{\tilde{D}N\times\tilde{D}N}$ and $\boldsymbol{C} = [(\boldsymbol{D}^{1Q})^{\mathsf{T}},\ldots,(\boldsymbol{D}^{(Q-1)Q})^{\mathsf{T}}]^{\mathsf{T}} \in \mathbb{R}^{d_Q N\times \tilde{D}N}$ such that

$$\boldsymbol{D} = \begin{bmatrix} \boldsymbol{A} & \boldsymbol{C} \\ \boldsymbol{C}^{\mathsf{T}} & \boldsymbol{D}^{QQ} \end{bmatrix}.$$

Assume that $\boldsymbol{A}^{-1}$ verifies Eq. (179). Denote

$$\boldsymbol{B} = \begin{bmatrix} \boldsymbol{R} & \boldsymbol{T} \\ \boldsymbol{T}^{\mathsf{T}} & \boldsymbol{B}^{QQ} \end{bmatrix}.$$

From the two by two blockmatrix inversion, we have:

$$\boldsymbol{B}^{QQ} = (\boldsymbol{D}^{QQ} - \boldsymbol{C}^{\mathsf{T}}\boldsymbol{A}^{-1}\boldsymbol{C})^{-1},$$
$$\boldsymbol{T} = -\boldsymbol{A}^{-1}\boldsymbol{C}\boldsymbol{B}^{QQ}.$$

We have

$$\left\|\boldsymbol{C}^{\mathsf{T}}\boldsymbol{A}^{-1}\boldsymbol{C}\right\|_{\mathrm{op}} \leq \sum_{q,q'\in[Q-1]} \left\|(\boldsymbol{D}^{qQ})^{\mathsf{T}}(\boldsymbol{A}^{-1})_{qq'}\boldsymbol{D}^{q'Q}\right\|_{\mathrm{op}}$$

$$= \sum_{q,q'\in[Q-1]} o_{d,\mathbb{P}}\left(\frac{r_q r_Q}{\sqrt{d_q d_Q}}\right) \cdot O_{d,\mathbb{P}}\left(\frac{\sqrt{d_q d_{q'}}}{r_q r_{q'}}\right) \cdot o_{d,\mathbb{P}}\left(\frac{r_{q'} r_Q}{\sqrt{d_{q'} d_Q}}\right)$$

$$= o_{d,\mathbb{P}}(r_Q^2/d_Q),$$

where we used in the second line the properties on $\boldsymbol{D}$ and our assumption on $\boldsymbol{A}^{-1}$. Hence $\boldsymbol{D}^{QQ} - \boldsymbol{C}^{\mathsf{T}}\boldsymbol{A}^{-1}\boldsymbol{C} \preceq (r_q^2/d_q)(c_q - o_{d,\mathbb{P}}(1))\mathbf{I}$ and $\|\boldsymbol{B}^{QQ}\|_{\mathrm{op}} = O_{d,\mathbb{P}}(d_q/r_q^2)$.

Furthermore, for $q < Q$,

$$\boldsymbol{B}^{qQ} = -\sum_{q'\in[Q-1]} (\boldsymbol{A}^{-1})_{qq'}\boldsymbol{C}_{q'}\boldsymbol{B}^{QQ}$$

Hence

$$\left\|\boldsymbol{B}^{qQ}\right\|_{\mathrm{op}} \leq \sum_{q'\in[Q-1]} \left\|(\boldsymbol{A}^{-1})_{qq'}\boldsymbol{D}^{q'Q}\boldsymbol{B}^{QQ}\right\|_{\mathrm{op}}$$

$$= \sum_{q,q'\in[Q-1]} O_{d,\mathbb{P}}\left(\frac{\sqrt{d_q d_{q'}}}{r_q r_{q'}}\right) \cdot o_{d,\mathbb{P}}\left(\frac{r_q r_Q}{\sqrt{d_{q'} d_Q}}\right) \cdot O_{d,\mathbb{P}}\left(\frac{d_Q}{r_Q^2}\right) = o_{d,\mathbb{P}}\left(\frac{\sqrt{d_q d_Q}}{r_q r_Q}\right),$$

which finishes the proof. $\qquad\qquad\qquad\qquad\qquad\qquad\qquad\qquad\qquad\qquad\qquad\qquad\square$

## L.1 Useful lemmas from [4]

For completeness, we reproduce in this section lemmas proven in [4].

**Lemma 25.** *The number $B(d,k)$ of independent degree-$k$ spherical harmonics on $\mathbb{S}^{d-1}$ is non-decreasing in $k$ for any fixed $d \geq 2$.*

**Lemma 26.** *For any fixed $k$, let $Q_k^{(d)}(x)$ be the $k$-th Gegenbauer polynomial. We expand*

$$Q_k^{(d)}(x) = \sum_{s=0}^{k} p_{k,s}^{(d)} x^s.$$

*Then we have*

$$p_{k,s}^{(d)} = O_d(d^{-k/2-s/2}).$$

**Lemma 27.** *Let $N = o_d(d^{\ell+1})$ for a fixed integer $\ell$. Let $(\boldsymbol{w}_i)_{i \in [N]} \sim \mathrm{Unif}(\mathbb{S}^{d-1})$ independently. Then as $d \to \infty$, we have*

$$\max_{i \neq j} |\langle \boldsymbol{w}_i, \boldsymbol{w}_j \rangle| = O_{d,\mathbb{P}}((\log d)^{k/2} d^{-k/2}).$$

**Proposition 6** (Bound on the Gram matrix). *Let $N \leq d^k / e^{A_d \sqrt{\log d}}$ for a fixed integer $k$ and any $A_d \to \infty$. Let $(\boldsymbol{\theta}_i)_{i \in [N]} \sim \mathrm{Unif}(\mathbb{S}^{d-1}(\sqrt{d}))$ independently, and $Q_k^{(d)}$ be the $k$'th Gegenbauer polynomial with domain $[-d, d]$. Consider the random matrix $\boldsymbol{W} = (\boldsymbol{W}_{ij})_{i,j \in [N]} \in \mathbb{R}^{N \times N}$, with $\boldsymbol{W}_{ij} = Q_k^{(d)}(\langle \boldsymbol{\theta}_i, \boldsymbol{\theta}_j \rangle)$. Then we have*

$$\lim_{d,N \to \infty} \mathbb{E}[\|\boldsymbol{W} - \mathbf{I}_d\|_{\mathrm{op}}] = 0.$$

## Footnotes

[5]Note that all models are trained with $\ell_2$ regularization.

[6]Strictly speaking, the model outlined in the main text requires $\boldsymbol{z}_i$ to be generated from the hyper-sphere of radius $\sqrt{d - d_0}$. In order to work with round numbers, in our experiments we use $\sqrt{d}$ instead of $\sqrt{d - d_0}$. The numerical difference between these two choices is negligible.

[7]Due to the large size of the test set, choosing these hyper-parameter based on the test set performance has a negligible over-fitting effect. In addition, in studying the approximation error overfitting is not relevant.