[Reviews · NeurIPS 2020]

Review 1

Summary and Contributions: This paper mainly studies the approximation error of random feature model, neural tangent kernel model, and two-layer neural network model, for non-uniform data distribution. Specifically, an "effective dimension" is defined to characterize the informative dimension of the data, which depend on both the dimension used to generate the target (d0) and the noise level on other dimensions. For RF and NT model, the approximation error bounds depends on the effective dimension. While for NN model, the bounds only depends on d0. This difference between two types of models can help understand the performance different between kernel methods and two-layer neural networks. If the effective dimension is much larger than d0, then neural networks work better than kernel methods. Intuitively, in this case kernel methods are harmed by the noise in the input data, but neural networks can choose the features to adapt the data distribution (if I am right). The theoretical results are asymptotic as N and d tend to infinity. Theoretical understandings are supported by numerical experiments, based on the assumption that image data are distributed on a low-dimension manifold. Discussions are made on finite-data case, but no rigorous results are provided in the finite width case. Thanks to the authors for their response. Though I'm still interested to see a sharper bound for neural networks, I agree with the intuition provided in the rebuttal. I'm not fully convinced about the ReLU issue. Since ReLU is nonsmooth, from the numerical analysis point of view, the approximation error can decay slower than models with smooth activation functions. Although ReLU can be approximated by smooth functions, as the activation functions tends to ReLU, the constant factors in the bounds may tend to infinity (e.g. depends on second-order derivatives), making the argument breaks finally at ReLU. So the results for ReLU may lose some orders on the polynomials. Nevertheless, I think the paper makes enough contribution to be accepted.

Strengths: 1. A concrete example and rigorous theoretical analysis are provided to explain the reason behind the performance difference between RF, NT models and neural network model. 2. Theoretical conclusions are supported by numerical evidence.

Weaknesses: 1. The analysis on neural networks are direct results induced from the kernel method results. It may not be tight, and can even suffer from the curse of dimensionality. Hence, the different between two types of models showed in the theorems may not fully characterize the actual performance difference. Neural networks can possibly perform much better than the bound. 2. While the theorems requires the activation function to be smooth, in the numerical experiments ReLU are used. Maybe it is more illustrative if smooth activation function can be used. Is there a reason for not using tanh, sigmoid, etc?

Correctness: The results seem to be correct.

Clarity: The paper is well organized and well written.

Relation to Prior Work: Yes.

Reproducibility: Yes

Additional Feedback:


Review 2

Summary and Contributions: - Various relationships between neural networks and kernel methods have been drawing attention in the context of recent literature that aims to better understand the superior performance of neural networks in many empirical tasks. One such line of work constitutes understanding when linear approximations of wide neural networks - Random features models or the Neural tangent models, both of which are also approximations to kernel-based approaches - can (or cannot) be used to replace neural networks without much drops in performance. - Informally speaking, the paper hypothesizes that in a supervised learning setting, when the target function depends on a low dimensional projection of the input data, then RKHS based approaches can suffer from a curse of dimensionality unless the input features are concentrated on the same low dimensional projection. Neural networks, on the other hand, do not suffer from this curse of dimensionality irrespective of the structure of the input features. - To support these hypotheses, the paper provides a data model under which the target function depends solely on a low dimensional projection of the data and show that: 1) For a fixed sample size, the test error of the neural tangent kernel and the kernel corresponding to the random features class is lower when the low dimensional structure of the input features matches that of the target function and vice versa. 2) The approximation error of the Neural tangent class and the random features class establish the same behavior for a fixed number of features (or width). 3) The approximation error of Neural networks only depends on the low dimensional projection of the input features on which the target function relies on and is independent of the effective input dimension. Empirical evidence is also provided to support their arguments. --------------------------------------------------------------------------------------------- --------------------------------------------------------------------------------------------- Thank you, authors, for providing clarification to points raised in the review. I maintain my pre-rebuttal evaluation.

Strengths: - The paper does a very good job of thoroughly investigating the validity of the two hypotheses presented in the overview. Even though the model is relatively simple, the theoretical results provided in the paper are quite instructive and provide some insightful results. The theory is rigorous and is well supported by empirical results. - The conceptual clarity of the work is high. The ideas are certainly relevant in the context of explaining some of the performance gaps between neural networks and kernel methods.

Weaknesses: - The main weakness of the paper is that of clarity and readability. The paper is quite dense and it appears that there is not enough emphasis on the clarity of the paper. As an example, in the overview of the paper (section 1.1), the authors discuss that adding high-frequency noise to images makes the distribution of the images more "isotropic" without clarifying what exactly this means. The discussion around anisotropy extends into the Model section (where it becomes a bit clearer). A simple, intuitive explanation could have helped improve the readability of the paper. - Certain, important, aspects of the Theorems are intuitively described. However, it would be helpful to explain the statement of the Theorems a bit further. For instance, the importance of some of the technical assumptions (the $$k^{\text{th}}$$ Hermite coefficient being non-zero or upper bounds on the derivatives) in the paper is not very clear. - Also, clarifying certain notation (e.g., $$\omega_d(\cdot)$$ or $$O_d(\cdot))$$ would also be helpful in improving the accessibility of the paper.

Correctness: To the extent that I could verify, the theoretical as well as the empirical analysis seems to be correct.

Clarity: Conceptually, the paper has very good clarity but it is a bit dense and is also a bit difficult to read if one is not very familiar with the notation and some of the technical language.

Relation to Prior Work: Yes

Reproducibility: Yes

Additional Feedback: - In Theorem 1, could the authors clarify the dependence of the test error on the regularization parameter $$\lambda$$? It could be interesting to see if the prediction error is minimised for a certain regimes in the space of the regularization parameter (of particular interest would be the case where $$\lambda$$ vanishes.) - In the section titled "Discussion", the authors mention "we provide a sharp characterization of the test error in the high-dimensional regime where both $$d$$ and $$n$$ diverge while being polynomially related." As far as I can tell, there are two regimes in which the various results of the paper are presented: 1) when $$ N \rightarrow \infty$$ while $$n$$ and $$d$$ are finite and polynomially related (Theorem 1) and 2) $$n \rightarrow \infty$$ while $$N$$ and $$d$$ are finite and polynomially related(Theorems 2, 3 ). Unless I am missing something here, it is unclear how the test error is characterized in the high-dimensional regime as claimed in the discussion. - In Figure 3, it is quite surprising to see that, at finite samples, the test error corresponding to the Neural network appears to be more sensitive to $$\kappa$$. Although, in the text, the authors explicitly state that it the other way around: kindly clarify. It would be insightful to consider the dependence of the estimation error of NN on $$\kappa.$$ Do the authors have some intuition for an explanation of this phenomenon?


Review 3

Summary and Contributions: The paper studies the approximations of two-layer neural networks (NN) and random feature (RF) models for target functions in the form g(Wx) with W be a d_0 x d matrix. It is shown that the learning with NN is only determined by the dimension d_0 (this part was already proved in previous works). However, for RF, the performance is determined by d, the dimension of the ambient space when x is distributed roughly uniformly. (This was also proved in the previous works. ) Combining them, we can see a clear separation of approximation abilities between NN and RF. The new observation of this paper is that when the input data x is concentrated in a low d_1-dimensional space. The learning of RF is only determined by max{d_0, d_1}. So when d_1 \approx d_0, the learning with RF is as efficient as NN, while when d_1 >> d_0, the separation of RF and NN becomes significant.

Strengths: From my understanding, this work is technically very sound and the claim that the performance of RF is determined by the intrinsic dimension of the input data is correct and intuitive.

Weaknesses: The contribution seems very incremental to me. I do not see any new insights from this paper. The theorems are sound but are also too complicated for understanding. I do not see any benefits to use the general setting to do theoretical analysis. Why don't the authors use the simplest setting that the target function is a single neuron g(w^Tx) and discuss how the distribution of x affects the approximability with random feature models. Moreover, compared to the simple case, the general setting does not bring any new insights, except making the analysis much involved and non-intuitive. Please correct me if I misunderstood something.

Correctness: Yes

Clarity: The paper is not easy to follow. The theorems are given in a form that is not easy to understand at least to me. Many notations are also used without definition, which seriously hurts the readability. For example, what are omega_d, O_d, o_{d,P}, tau^2 in the line 143-145? I know that some of them are defined in the different places of the appendix. One needs to waste a lot of time to dig out the real meaning of those notation. This is really annoying. In Theorem 1, the regularization parameter lambda is chosen in the order of O(1). Shouldn't we decrease the regularization parameter when we have more samples?

Relation to Prior Work: Yes

Reproducibility: Yes

Additional Feedback: === After rebuttal === Thanks authors for the response. Different from the trival situation that the data completely lie in a low dimensiontal space, the authors identify an effective dimension to consider the contribution of the extent of the concentration. From the understanding aspect, I still feel these results do not bring us any new insights. But the techniques developed here are nontrivial, and they may be helpful for studying the similar problems. So I raise my score to 6.


Review 4

Summary and Contributions: The paper proposes a theoretical study of the comparison of the representation capabilities of two-layer fully connected neural networks (NN), RKHS models, and random features (RF) (Rahimi and Recht, 2008) and neural tangent (NT) (Jacot et al. 2018) approximations of two-layer networks. The learning problem is studied in an infinite-sample setting, where the models are compared with respect to their approximation errors. The main conclusion of the work is that NN models have higher representation accuracy than the other models in comparison, when the geometric structure of the data deviates from a low-dimensional one, which is captured through the “effective dimension” concept defined in the paper.

Strengths: The presented bounds are mainly revisited versions of those in the previous work (Ghorbani et al. 2019), the main difference being that the learning problem in the current work is studied under the assumption that the input data is approximately low-dimensional and the function to be learnt depends only on the low-dimensional projection of the data. While the proposed results bear resemblance to those in (Ghorbani et al. 2019), the increment is justifiable in the sense that the particular attention given to the low-dimensional setting is of practical interest for many machine learning problems.

Weaknesses: The paper is well-written, and the presented results are discussed clearly with relevant experimentation. While the amount of contribution with respect to (Ghorbani et al. 2019) may be put into question, overall, I enjoyed reading this paper and found it useful. Some comments and questions for the authors: 1. The proposed results are obtained for a two-layer fully connected network architecture. I understand that this simplified model makes the theoretical analysis more tractable, but could the authors comment on to what extent they expect their results to be valid for deeper network topologies with more than two layers? 2. The proposed study assumes that the function to be learnt is of the form f_*(x) = phi(U^T x). Then, in this setting the NN models are shown to outperform, RKHS models and linear NN approximations in terms of representation power. I am wondering whether the particular function model “phi(U^T x)” assumed in the paper would tend to trivially favor NN models (in comparison to e.g. RKHS models), as it consists of a “linear projection followed by a nonlinear scalar function”, just like in neural network hypothesis classes. This being said, I also understand that the form f_*(x) = phi(U^T x) is the way that authors characterize the “low-dimensionality” of the function class to be learnt, to which I have no objection. Some further comments on these issues would be useful. 3. I guess an important future direction of this study would be the incorporation of finite sample effects into the bounds. I would encourage the authors to pursue their efforts in this direction towards a more mature extension of their work. 4. The current experimental results only compare the NN methods. How would the performance of RKHS methods be in these experiments? Minor comments: 1. The difference between the problem formulations (1) and (2) is not clear. Please explain their difference clearly, by modifying the notation if necessary. Are the coefficients (a, b) and the network weights W variable in both problems? Are some of them fixed in (2)? 2. In Theorem 1, please define what the notation “omega_d(.)” means.

Correctness: Seem to be so.

Clarity: Yes.

Relation to Prior Work: Yes.

Reproducibility: Yes

Additional Feedback: I have read the feedback from the authors. I would like to thank the authors for their answers to my questions.

[Author Response · NeurIPS 2020]

**Response to reviewer 1.** *Weakness 1: The result for neural network is an upper bound which may not be tight.* Indeed: Theorem 4 is just an upper bound on the approximation error, and a lower bound for NN is currently missing. The upper bound is sufficient to prove a separation result for $\kappa$ large enough. While the upper bound is not tight pointwise (per function $f_*$), we believe it is almost tight in a *minimax* sense. Namely, if $N \ll d_0^\ell$, then there is a degree-$(\ell+1)$ polynomial $f_*$ such that the corresponding approximation error $R_{\mathsf{NN},N}(f_*)$ is bounded below by a positive constant. This seems quite intuitive by parameter counting. We agree that this is an important question to be addressed. *Weakness 2: Our theory requires a smooth activation $\sigma$ while our experiments uses ReLU activation.* We believe that the smoothness $\sigma$ is a technical issue and can be relaxed using soft approximation arguments. We did not relax this requirement since the paper is already technically involved and this point is tangential to the main questions we address. Experimentally, it is also easy to check that replacing ReLU by a smoothed ReLU does not change the numerical results.

**Response to reviewer 2.** We will improve the readability, clarify the notations, and implications in the final version of the paper. *Additional comment 1: How does the result of Theorem 1 depend on $\lambda$?* Theorem 1 implies that —to the leading order– the test error is independent of $\lambda$ as long as $\lambda = \lambda(d) = O_d(1)$. In particular, $\lambda(d) = 0$ is optimal up to subleading terms. On the contrary, our proofs also imply that $\lambda(d) > d^\iota$ leads to suboptimal test error for any $\iota > 0$. $\lambda(d) = 0$ still leads to good generalization because, in high dimension, the non-vanishing high order derivatives of the kernel function induce regularization effects (hence the assumption that $h^{(k)}(0) > 0$ for some $k \geq \ell$). *Additional comment 2: What is the 'high dimensional regime'?* We considered two regimes: 1) $n = \infty$, $N$ and $d$ are polynomially related and goes to $\infty$ together. 2) $N = \infty$, $n$ and $d$ are polynomially related and go to $\infty$ together. *Additional comment 3: Figure 3 shows that at finite samples the test error corresponding to the neural network appears to be more sensitive to $\kappa$ which is puzzling.* We agree that the test error of the SGD trained NN deserves further investigation. However notice that the test error of NTK is substantially larger, for each given $\kappa$. Since the error is normalized to be between 0 and 1, it cannot grow much, whence the apparent lack of variability.

**Response to reviewer 3.** The reviewer's summary of our work is incomplete. We considered the following model: the feature vector $\boldsymbol{x} = \boldsymbol{U}(\boldsymbol{x}_1, \boldsymbol{x}_2) \in \mathbb{R}^d$ is composed of an isotropic signal feature $\boldsymbol{x}_1$ in $d_0$ dimensions and an isotropic junk feature $\boldsymbol{x}_2$ in $d - d_0$ dimensions; the target function only depends on $\boldsymbol{x}_1$. We characterize the risk of KRR, RF, NT in terms of the effective dimension $d_{\mathrm{eff}}$ which depends on $d$, $d_0$, and feature SNR (the variance of the signal feature over the variance of the junk feature). Crucially, the effective dimension $d_{\mathrm{eff}}$ does not depend uniquely on '$d_1$' (perhaps $d_0$ is meant here), the dimension of subspace of the feature vector 'concentrated in', as written by the reviewer.

Further, we do not agree with the reviewer that our results are incremental. We agree: it is natural to expect that the behaviors of RF, NT, KRR, only depend on some sort of 'intrinsic dimension'. The result would be trivial if the variance in the junk subspace was zero, and feature vectors lied exactly on a $d_0$-dimensional subspace. Our contribution is non-trivial in extending the result to the case when there is non-vanishing variance of junk features: we quantified how the effective dimension $d_{\mathrm{eff}}$ depends on the signal dimension $d_0$ and feature SNR. (Often $d_{\mathrm{eff}} \neq d_0$.) Let us also emphasize that a model in which the data lie exactly on a $d_0$-dimensional subspace would be extremely crude: in particular it would not explain the experiments in which we gradually add noise to the junk subspace.

*Weakness: Why do not just study a single neuron $g(\boldsymbol{w}^\mathsf{T}\boldsymbol{x})$?* First, limiting the analysis to the target function $g(\boldsymbol{w}^\mathsf{T}\boldsymbol{x})$ would not simplify the proofs: we would still need to develop the whole machinery and then apply them to this specific case. The real technical challenge arises in dealing with non-vanishing variance in the junk subspace. Second, the target function $g(\boldsymbol{w}^\mathsf{T}\boldsymbol{x})$ is too restrictive: we cannot consider this a remotely realistic model. On the other hand, functions of a low-dimensional subspace are a classical model in non-parametric theory.

*Clarity: why choosing $\lambda = O_d(1)$ instead of $\lambda \to 0$?* Notice that $\lambda \to 0$ is a special case of $\lambda = O_d(1)$. (See comments to 2nd reviewer.) Finally, we agree that there is some room to improve the clarity and readability of this paper. We will make these changes in the final version.

**Response to reviewer 4.** *Weakness 1: What do we expect for multi-layers fully-connected networks?* For KRR with multi-layers NT kernels with Gaussian weights (infinitely wide networks), the resulting kernel will still be an inner product kernel. Hence our Theorem 1 can be applied directly. We have also conducted experiments comparing 3-layers NN and 3-layers NT KRR in Figure 4. On the other hand, analyzing finite width NT KRR for 3-layers networks is beyond current techniques. *Weakness 2: Does the model $\varphi(\boldsymbol{U}^\mathsf{T}\boldsymbol{x})$ tend to trivially favor NN models?* As the reviewer writes, this is a natural way to model 'low-dimensionality' of the target function. In fact it is a standard model in the non-parametric regression literature and is directly related to other canonical functional classes. [Bac17] provides several pointer to the vast literature on this topic, and connections with other functional classes. *Weakness 4: How would the performance of RKHS methods be in these experiments?* We compared NN and KRR (RKHS methods) in Figure 1, 3, and 4 (in these figures, we use the shorthand NT KRR for the RKHS induced by the infinite-width NT kernel). We thank the reviewer for the additional remarks. We will make these changes in the final version.

[Bac17] Breaking the Curse of Dimensionality with Convex Neural Networks. F. Bach.

[Meta-Review · NeurIPS 2020]

Reviewers are positive about the solid theoretical results presented in the paper, which should be of interest to many researchers in the machine learning community. The writing, however, is quite dense and should be improved for better readability and clarity.